# MULTIPLICATIVE DIFFUSION MODELS: BEYOND GAUSSIAN LATENTS

**Robert Gruhlke**[1], **Valentin Resseguier**[2], **Merveille Talla**[2,3]
[1]Freie Universit at Berlin, Berlin, Germany
[2]INRAE, OPAALE, Rennes, France
[3]Inria centre at Rennes University, Rennes, France
r.gruhlke@fu-berlin.de,
{valentin.resseguier,merveille.talla}@inrae.fr

## ABSTRACT

We introduce a new class of generative models based on multiplicative score-driven diffusion. In contrast to classical diffusion models that rely on additive Gaussian noise, our construction is driven by skew-symmetric multiplicative noise. It yields conservative forward-backward dynamics inspired by the principles of physics. We prove that the forward process converges exponentially fast to a tractable non-Gaussian latent distribution, and we characterize this limit explicitly. A key property of our diffusion is that it preserves the distribution of data norms, resulting in a latent space that is inherently data-aware. Unlike the standard Gaussian prior, this structure better adapts to heavy-tailed and anisotropic data, providing a closer match between latent and observed distributions. On the algorithmic side, we derive the reverse-time stochastic differential equation and associated probability flow, and show that sliced score matching furnishes a consistent estimator for the backward dynamics. This estimation procedure is equivalent to maximizing an evidence lower bound (ELBO), bridging our framework with established variational principles. Empirically, we demonstrate the advantages of our model in challenging settings, including correlated Cauchy distributions and experimental fluid dynamics images ($d = 1024$). Across these tasks, our approach more accurately captures extreme events and tail behavior than classical diffusion models, particularly in the low-data regime. Our results suggest that multiplicative conservative diffusions open a principled alternative to current score-based generative models, with strong potential for domains where rare but critical events dominate.

## 1 INTRODUCTION

Mathematically equivalent (Song et al., 2021), diffusion models and score-based generative models demonstrate impressive skills and are among the current state-of-the-art for the generation of two- and three-dimensional images. Unconditioned sampling scores can be easily modified to conditioned sampling scores to address various inverse problems (Rybchuk et al., 2023; Rozet & Louppe, 2023; Daras et al., 2024; Bao et al., 2025). However, both learning and inference come with significant computational costs. In addition, even with large computational power, the generation of rare and extreme events remains a difficult task (Li et al., 2024; Stamatelopoulos & Sapsis, 2025). Those generative AI challenges may be more easily addressed by introducing physical-based inductive bias in the fully-data-driven approaches. In this paper, we take inspiration from physics and its conservative structure to build a multiplicative score-based generative model. It is inspired by transport noises in fluid dynamics (Kraichnan, 1968; Brzeźniak et al., 1991; Klyatskin, 1994; Piterbarg & Ostrovskii, 1997; Mikulevicius & Rozovskii, 2004; Mémin, 2014; Holm, 2015; Resseguier et al., 2021; Zhen et al., 2023) and, more generally, from slow-fast systems with multiplicative noise (Majda et al., 1999; Franzke et al., 2005; Gottwald & Melbourne, 2013; Gottwald & Harlim, 2013). Transport noise models may be understood as generative models based on stochastic fluid dynamics rather than fitted

neural networks. As other generative models, they suit particularly well to Bayesian inverse problems (Cotter et al., 2020b;a; Resseguier et al., 2022; Dufée et al., 2022).

Here, we might address problems outside the scope of fluid dynamics, though keeping the conservative structure of transport noise. The noising and denoising procedures that we propose maintain a part of the data information: the distribution of norm of the data point. The latent distribution is hence both tractable and close to the data distribution. More specifically, our contributions are the following.

**New generative model paradigm:** We introduce a new type of diffusion model where the noising process is multiplicative. We call it a Multiplicative Score-based Generative Model (MSGM). Involving random rotations around the origin, it greatly differs from previous diffusion models and opens a new research path. The key aspects are summarized in Figure 1.

**Deep theoretical analysis of MSGM:** Assuming a skew-symmetric structure and a rank condition for this noise, we proved several theoretical results, guiding the use of this new generative tool. The first theorem provides the Fokker-Planck equation of forward diffusion and its invariant measures. Then, we separately analyze the norm and direction of the diffusion. The norm is steady, whereas the direction follows a similar multiplicative stochastic differential equation (SDE). Two other theorems show that distributions of the direction and thus of the whole diffusion converge exponentially fast to a white noise in the weak sense. Asymptotically, the norm and direction are independent, and the latter is uniformly distributed over the $d$-sphere.

**General algorithm for MSGM:** We propose to estimate the scaled diffusion score by a neural network using sliced score matching, and our last theorem shows that it is equivalent to maximizing the ELBO. Sampling the non-Gaussian latent vectors reduces to a one-dimensional problem that we address with eCDF. For the denoising process, both SDE and ordinary differential equation (ODE) formulations are proposed.

**Application to extremes in moderate dimension:** We propose a numerical procedure to mimic the heavy-tail distribution with MSGM. We add a first layer to the neural network to perform a spherical decomposition with log-norm, and the latent distribution is characterized by the law of the data log-norm. Compared numerically with a standard diffusion model, MSGM better mimics multidimensional Cauchy distributions and measured fluid vorticity. The proximity between latent and data distributions facilitates the forward and the backward diffusions, and implicitly encompasses the correct distribution tail decays.

**Application in high dimension:** As a first step, we focus on MSGM scalability and design of sparse underlying tensors in the diffusion. While the latter is not covered completely by the theoretical analysis, our numerical experiments show promising image generation results.

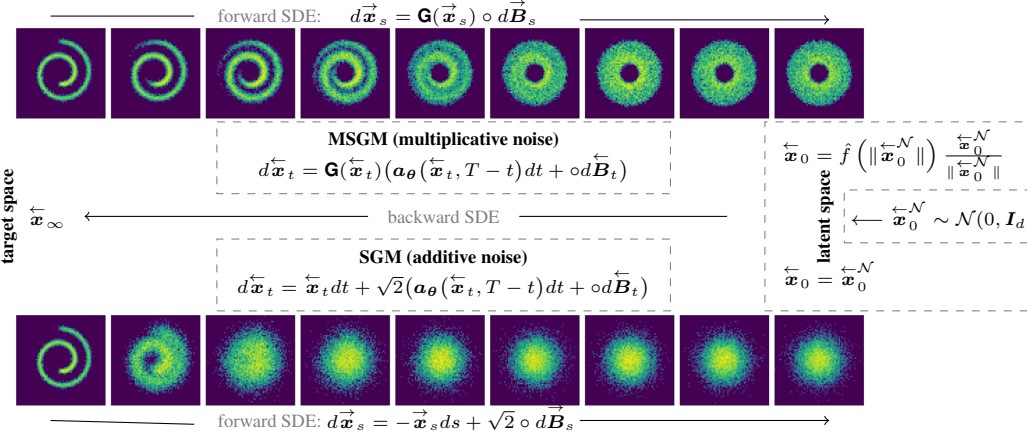

Figure 1: Illustration of multiplicative score-based generative modeling (**ours**) compared to additive score-based generative modeling.

## 2  ADDITIVE SCORE-BASED GENERATIVE MODEL

### 2.1  FORWARD AND BACKWARD SDEs

Diffusion models or score-based generative models (SGM) can be expressed in continuous time with stochastic differential equation (SDE) (Song et al., 2021). The forward SDE is

$$d\vec{\boldsymbol{x}}_s = -\vec{\boldsymbol{x}}_s ds + \sqrt{2}d\vec{\boldsymbol{B}}_s, \tag{2.1}$$

where $\vec{\boldsymbol{x}}_s \in \mathbb{R}^d$ is distributed according to some density $p_s$ for $s > 0$, $s \mapsto \vec{\boldsymbol{B}}_s$ is $d-$dimensional Brownian motion, and $\vec{\boldsymbol{x}}_0$ distributed according to the dataset of interest. It is an Ornstein-Uhlenbeck process: the continuous-time version of a first-order autoregressive (AR) model and the distribution $p_s$ convergences to a standard Gaussian density exponentially for $s \to \infty$, e.g. in total variation or Wasserstein distance. We can then define for $t \in [0, T]$ the backward equation

$$d\overleftarrow{\boldsymbol{x}}_t = \overleftarrow{\boldsymbol{x}}_t dt + 2\nabla \log p_{T-t}(\overleftarrow{\boldsymbol{x}}_t)dt + \sqrt{2}d\overleftarrow{\boldsymbol{B}}_t, \tag{2.2}$$

with $t \mapsto \overleftarrow{\boldsymbol{B}}_s$ another $d-$dimensional Brownian motion and $\overleftarrow{\boldsymbol{x}}_0 \sim p_T$ (identifying the density $p_T$ with its distribution). Then for any $s \in [0, T]$, $\overleftarrow{\boldsymbol{x}}_{T-s}$ and $\vec{\boldsymbol{x}}_s$ have the same law $p_s$. In practice, when an approximate score $\nabla \log p_{T-t}$ is available we initialize equation 2.2 with a standard Gaussian distribution $\overleftarrow{\boldsymbol{x}}_0 \sim \mathcal{N}(0, \boldsymbol{I}_d)$ and integrate the backward SDE from $t = 0$ to $t = T$ (i.e. from $s = T$ to $s = 0$), ideally letting $\overleftarrow{\boldsymbol{x}}_T$ become statistically similar to our dataset of interest.

### 2.2  A NEURAL NETWORK TO FIT THE SCORE

In practice, the score $\nabla \log p_{T-t}(\boldsymbol{x})$ is approximated by a surrogate model, $\boldsymbol{s_\theta}(\boldsymbol{x}, T-t)$, e.g., a fitted artificial neural network (ANN). Alternatively, one can work on $\boldsymbol{a_\theta}(\boldsymbol{x}, T-t) = \sqrt{2}\boldsymbol{s_\theta}(\boldsymbol{x}, T-t)$ (Huang et al., 2021). For large-dimensional problems, Song et al. (2020) proposes to learn this neural network by *Sliced Score Matching* (SSM). Here, $\boldsymbol{a_\theta}$ is obtained by minimizing the loss function

$$\mathcal{L}_{\text{SSM}}^{\text{SGM}}(\boldsymbol{\theta}) = \int_0^T \mathbb{E}_{\vec{\boldsymbol{x}}_s} \mathbb{E}_{\mathbf{v} \sim \text{Rad}(d)} \left[ \tfrac{1}{2}\|\boldsymbol{a_\theta}(\vec{\boldsymbol{x}}_s, s)\|^2 + (\mathbf{v} \cdot \nabla)((\sqrt{2}\boldsymbol{a_\theta}(\vec{\boldsymbol{x}}_s, s) - \vec{\boldsymbol{x}}_s) \cdot \mathbf{v}) \right] ds. \tag{2.3}$$

where $\|.\|$ is the Euclidean norm, $\text{Rad}(d)$ denotes the $d$-dimensional Rademacher distribution and $\mathbb{E}_{\vec{\boldsymbol{x}}_s}$ is the expectation along each path realization $\vec{\boldsymbol{x}}_s$. Section A details the most common score matching losses and their link to the concept of the Evidence Lower Bound (ELBO).

## 3  MULTIPLICATIVE SCORE-BASED GENERATIVE MODEL

Rather than relying on additive SDE equation 2.1, we propose a multiplicative SDE and the associated score-based generative model. Taking inspiration from physics, this approach introduces physical-based inductive bias and yields tractable latent distributions closer to the dataset distribution. In this section, we introduce our forward SDE based on skew-symmetric multiplicative noise, its corresponding latents, and backward SDE and analyze the limit properties of the process distribution. To share didactic similarities of the forward and backward processes as in the additive noise case, we will keep the same notation for the forward process $\vec{\boldsymbol{x}}_s$ and the backward process $\overleftarrow{\boldsymbol{x}}_s$, respectively.

### 3.1  FORWARD SDE

Instead of considering a forward SDE with additive noise, we rely on *multiplicative noise model*

$$d\vec{\boldsymbol{x}}_s = \mathbf{G}(\vec{\boldsymbol{x}}_s) \circ d\vec{\boldsymbol{B}}_s, \tag{3.1}$$

where $d \geqslant 2$, $\mathbf{G} : \mathbb{R}^d \to \mathbb{R}^{d \times d}$ is linear and $\circ$ stands for the *Stratonovich* notation. The readers unfamiliar with this notation may interpret the Stratonovich noise $s \mapsto \circ d\vec{\boldsymbol{B}}_s$ as a process with short correlation time but respecting the usual rules of differential calculus – says the chain rule. The discretized version of equation 3.1 – with an infinitely small time step $ds$ – may also provide insight:

$$\tfrac{1}{2}(\vec{\boldsymbol{x}}_{s+ds} - \vec{\boldsymbol{x}}_{s-ds}) = \mathbf{G}(\vec{\boldsymbol{x}}_s)\tfrac{1}{2}(\vec{\boldsymbol{B}}_{s+ds} - \vec{\boldsymbol{B}}_{s-ds}). \tag{3.2}$$

For deeper understanding, Section B recalls some important notions of stochastic calculus, including the Stratonovich notation and the relationship to Itô calculus. Let $\mathbf{G}$ be represented by a third-order tensor $[G_{i,j}^k] \in \mathbb{R}^{d,d,d}$ and define the random matrix $\boldsymbol{Z}_s = \sum_{k=1}^{d} \boldsymbol{G}^k (\vec{B}_s)_k$. Then, equation 3.1 can be written more explicitly as:

$$d\vec{\boldsymbol{x}}_s = \sum_{k=1}^{d} (\boldsymbol{G}^k \vec{\boldsymbol{x}}_s)(\circ d\vec{B}_s)_k = \sum_{k=1}^{d} \boldsymbol{G}^k (\circ d\vec{B}_s)_k \vec{\boldsymbol{x}}_s = \circ d\boldsymbol{Z}_s \vec{\boldsymbol{x}}_s \approx \tfrac{1}{2}(\boldsymbol{Z}_{s+ds} - \boldsymbol{Z}_{s-ds})\vec{\boldsymbol{x}}_s, \quad (3.3)$$

where the time increments of the random matrix, $\circ d\boldsymbol{Z}_s \approx \tfrac{1}{2}(\boldsymbol{Z}_{s+ds} - \boldsymbol{Z}_{s-ds})$, are uncorrelated.

Additionally, we will impose two assumptions valid throughout the paper:

Skew-symmetry : For any $k \in \{1, \dots, d\}$, the matrix $\boldsymbol{G}^k = (G_{i,j}^k)_{i,j}$ is skew-symmetric.  (A1)

Rank condition : For any $\boldsymbol{x} \in \mathbb{R}^d \backslash \{0\}$, $\mathrm{rank}(\mathbf{G}(\boldsymbol{x})) = d - 1$.  (A2)

In particular equation 3.1 does not describe a geometric Brownian motion, as the noise term $\boldsymbol{Z}_s$ is not diagonal. It includes zeros on the diagonal due to the skew-symmetry of $\boldsymbol{G}^k$. A geometric Brownian motion would necessitates $\boldsymbol{Z}_s = \mathrm{diag}(\vec{B}_s)$. A strategy to obtain a tensor $\mathbf{G}$ that matches assumptions A1 and A2 will be discussed in Section 6. By linearity, the skew-symmetry of all $\boldsymbol{G}^k$ (assumption A1) implies the skew-symmetry of the whole multiplicative noise matrix $\circ d\boldsymbol{Z}_s$. This structure is inspired by transport noises in fluid dynamics (Kraichnan, 1968; Piterbarg & Ostrovskii, 1997; Resseguier et al., 2021). In this analogy, $\vec{\boldsymbol{x}}_s$ would represent an image of temperature, advected by an incompressible fluid flow. Incompressibility leads to the skew-symmetry of the advection operator, and eventually to energy conservation of $\vec{\boldsymbol{x}}_s$. Here, we might address problems outside the scope of fluid dynamics, though maintaining the noise skew-symmetry (assumption A1) and thus the energy conservation, as discussed in Section 3.2.

With assumption A2, the noise spreads in a large space: $\mathrm{Im}(\mathbf{G}(\vec{\boldsymbol{x}}_s)) = \vec{\boldsymbol{x}}_s^{\perp}$. It ensures sufficient variability in the noising process and, in turn, a tractable distribution for $\vec{\boldsymbol{x}}_T$ when $T$ becomes large.

**Theorem 3.1.1.** *Let the assumptions A1 and A2 hold. Then, the Fokker-Planck equation of equation 3.1 reads*

$$\frac{\partial}{\partial s} p_s(\boldsymbol{x}) \;\; = \;\; \nabla_{\perp} \cdot \left( \tfrac{1}{2} \boldsymbol{\Sigma}(\boldsymbol{x}) \nabla_{\perp} p_s(\boldsymbol{x}) \right), \quad \boldsymbol{x} \in \mathbb{R}^d, \quad (3.4)$$

*with conditional noise covariance $\boldsymbol{\Sigma}(\boldsymbol{x}) := \mathbf{G}(\boldsymbol{x})\mathbf{G}(\boldsymbol{x})^{\mathsf{T}}$ and $\nabla_{\perp}$ denoting the orthogonal projection of nabla $\nabla$ on the tangent plane $\boldsymbol{x}^{\perp}$, i.e.*

$$\nabla_{\perp} \;\; := \;\; (\boldsymbol{I}_d - \boldsymbol{x}^n (\boldsymbol{x}^n)^{\mathsf{T}})\nabla, \quad (3.5)$$

*for $\boldsymbol{x}^n := \boldsymbol{x}/\|\boldsymbol{x}\|$ with $\boldsymbol{x} \in D := \mathbb{R}^d \setminus \{0\}$ and 0 otherwise, the unit vector orthogonal to the d-sphere. Moreover, any stationary density $p_{\infty}$ of equation 3.4 is rotation-invariant on $\mathbb{R}^d$.*

The proof is detailed in Section D.2. In order to highlight the connection to diffusion models on Riemannian manifolds, we note that $\nabla_{\perp}$ is the Riemannian gradient on the unit $d$-sphere: $\mathbb{S}^{d-1} = \{\boldsymbol{x} \in \mathbb{R}^d | \|\boldsymbol{x}\| = 1\}$, see Section H.1. For a possible extension of the considered diffusion equation 3.1 to the case of non-zero drift, we refer to section D.6.

## 3.2 Dynamics of norm and direction

We now consider for any $s \geq 0$ and $\vec{\boldsymbol{x}}_s \neq 0$, the spherical decomposition

$$\vec{\boldsymbol{x}}_s = \|\vec{\boldsymbol{x}}_s\| \vec{\boldsymbol{x}}_s^n \quad \text{with} \quad \vec{\boldsymbol{x}}_s^n := \vec{\boldsymbol{x}}_s / \|\vec{\boldsymbol{x}}_s\| \in \mathbb{S}^{d-1}. \quad (3.6)$$

First, we note that the norm, $\|\vec{\boldsymbol{x}}_s\|$, remains constant throughout the noising process. Indeed, the skew-symmetry of $\circ d\boldsymbol{Z}_s$ implies that $d\vec{\boldsymbol{x}}_s = \circ d\boldsymbol{Z}_s \vec{\boldsymbol{x}}_s$ is orthogonal to $\vec{\boldsymbol{x}}_s$ and hence:

$$d\|\vec{\boldsymbol{x}}_s\|^2 = 2\vec{\boldsymbol{x}}_s \cdot \circ d\vec{\boldsymbol{x}}_s = 0, \quad \forall s \geq 0. \quad (3.7)$$

Consequently, $\|\vec{\boldsymbol{x}}_s\| \equiv \|\vec{\boldsymbol{x}}_0\|$. The vector $\vec{\boldsymbol{x}}_s$ moves randomly on $\|\vec{\boldsymbol{x}}_0\|\mathbb{S}^{d-1}$, the $d$-sphere of radius $\|\vec{\boldsymbol{x}}_0\|$. Therefore, the distribution of the norms of the latent variable is exactly the distribution of

the norms of the points of the dataset. We refer to Section D.3 for more details. This property will have important consequences for our learning procedure and on the possibility of generating extreme events. Indeed, the property of likely large norm events will be conserved along the diffusion. In particular, the norm distribution is one-dimensional and we can rely on advanced techniques available for this setup, without worrying about the curse of dimensionality. In practice, we will fit the distribution of the log-norm, $F_{\log |.|_\epsilon}$, with eCDF. We refer to Section C for details and an overview of sampling from one-dimensional distributions.

We now focus on $p_s^n$, the distribution of the direction, $\overrightarrow{\boldsymbol{x}}_s^n$, in particular as $s \to \infty$. For better readability, we postpone the full discussion and the main theorems to Section D.4. Lemma D.4.1 introduces the Fokker-Planck equation of the direction on $\mathbb{S}^{d-1}$ and its unique invariant measure, $p_\infty^n$. Then, Theorem D.4.2 shows the exponential convergence of the initial distribution $p_0^n$ to $p_\infty^n$, the uniform distribution on the unit sphere $\mathbb{S}^{d-1}$. Consequently, since $\mathbb{S}^{d-1}$ is compact, this implies convergence in total variation of $p_s^n$ to $p_\infty^n$ and convergence in distribution of $\overrightarrow{\boldsymbol{x}}_s^n$ to $\overrightarrow{\boldsymbol{x}}_\infty^n \sim \mathcal{U}(\mathbb{S}^{d-1})$.

### 3.3 Non-Gaussian latent space

In this section, we characterize the generally non-Gaussian latent distribution. Although this sounds intractable at first glance, it will turn out that we can easily sample it.

In general, the latent space of MSGM is not Gaussian. It becomes Gaussian if and only if the distribution of the squared norms of the dataset points has $\chi^2$ distributions with $d$ degrees of freedom (see Section E.2). This property differs from the usual SGM. SGM latent variables are Gaussian, leading to $\chi^2$ distributions for the norms of the latent variables regardless of the data set. According to Lafon et al. (2023), without a heavy tail distribution for the latent variables, it is unlikely that the final samples will be generated with a heavy tail distribution, at least with variational autoencoders (VAE). With our approach, the distribution of the norms of the latent variables has heavy tails if and only if the distribution of the norms of dataset points has heavy tails. Therefore, we expect a significant improvement from our method in generating extreme events. In fact, for heavy-tailed data the KL divergence to the SGM latent distribution is infinite, whereas MSGM yields a finite value, see Section E.6. More generally, Section E.5 shows that the KL divergence from data to the latent distribution is always smaller under MSGM than SGM. So, only few time steps may be sufficient to integrate the forward and the backward MSGM diffusions. In any case, the MSGM latent vectors are still white noise in the weak sense (see Section E.1). Moreover, the norm and direction are independent from each other, which will drastically facilitate the sampling procedure. These results hold for latent vectors $\boldsymbol{x}_\infty \sim p_\infty$. In practice, integrations of forward and backward diffusions are only possible over a finite time $T$. However, the following theorem states that the law of the solution, $\overrightarrow{\boldsymbol{x}}_T$, will become close to $p_\infty$ exponentially as fast as $T \to +\infty$. So, we can confidently rely on finite-time integration.

**Theorem 3.3.1.** *Let assumptions A1 and A2 hold. Let $\overrightarrow{\boldsymbol{x}}_0 \sim p_0 \in \mathcal{C}^2(D)$ and let $p_{|.|}$ be the (radial) density of $\|\overrightarrow{\boldsymbol{x}}_0\|$. Then, the Fokker-Planck equation 3.4 has a unique solution $p_s \in \mathcal{C}^2(D) \cap L^2(D)$ for all $s > 0$. Moreover, the Fokker-Planck equation has the stationary distribution*

$$p_\infty(\boldsymbol{x}) = \frac{p_{|.|}\left(\|\boldsymbol{x}\|\right)}{\|\boldsymbol{x}\|^{d-1}} \frac{1}{|\mathbb{S}^{d-1}|}. \tag{3.8}$$

*In particular, $\|\overrightarrow{\boldsymbol{x}}_s\|$ and $\overrightarrow{\boldsymbol{x}}_s^n$ are asymptotically independent for $s \to +\infty$. Moreover, there exists $\alpha = \alpha(\mathbf{G}, d) > 0$ such that*

$$\|p_s - p_\infty\|_{L^2(\mathbb{R}^d)}^2 \leq \exp(-\alpha s)\|p_0 - p_\infty\|_{L^2(\mathbb{R}^d)}^2. \tag{3.9}$$

The proof and details on $\alpha$ are given in Section D.5 and a specific case is discussed in Section J.3. The factor $\|\boldsymbol{x}\|^{1-d}$ in equation 3.8 is expected. Indeed, $\frac{|\mathbb{S}^{d-1}|}{\|\boldsymbol{x}\|^{1-d}}$ is the volume of the scaled $d$-sphere $\|\boldsymbol{x}\|\mathbb{S}^{d-1}$, i.e. it corresponds to the uniform distribution on the scaled $d$-sphere $\|\boldsymbol{x}\|\mathbb{S}^{d-1}$.

We will now consider the practical question on how to draw samples from the latent distribution with density $\rho_\infty$ from equation 3.8. It is of product structure between the radial and the directional component. So, we can sample the norm $R_\infty$ and the direction $\overrightarrow{\boldsymbol{x}}_\infty^n$ separately and multiply them. The

norm $R_\infty$ can be sampled from an one-dimensional approximation of the data norm (see Section C) and the direction $\vec{\boldsymbol{x}}_\infty^n$ is uniform. So, we can sample a $\vec{\boldsymbol{x}}_\infty^{\mathcal{N}} \sim \mathcal{N}(0, \boldsymbol{I}_d)$ and set

$$\vec{\boldsymbol{x}}_\infty = R_\infty \vec{\boldsymbol{x}}_\infty^n \quad \text{with} \quad \vec{\boldsymbol{x}}_\infty^n = \vec{\boldsymbol{x}}_\infty^{\mathcal{N}}/\|\vec{\boldsymbol{x}}_\infty^{\mathcal{N}}\|, \quad R_\infty = \hat{f}(\|\vec{\boldsymbol{x}}_\infty^{\mathcal{N}}\|), \tag{3.10}$$

$$\text{and} \quad \hat{f}(r) := \exp\left(\hat{F}^{-1}_{\log|.|_\epsilon}(F_{\chi^2(d)}(r^2))\right) - \epsilon, \quad \forall r > 0. \tag{3.11}$$

If $\vec{\boldsymbol{x}}_\infty^{\mathcal{N}} = 0$, we set $\vec{\boldsymbol{x}}_\infty = 0$. Proposition E.3.2 shows that this procedure leads to samples with the correct distribution, up to the approximation of the log-norm CDF $\hat{F}_{\log|.|_\epsilon} \approx F_{\log|.|_\epsilon}$. Moreover, the direct map $\hat{F}_{\log|.|_\epsilon}$ can transform a latent vector, $\vec{\boldsymbol{x}}_T$, into a Gaussian one (see Section E.4). This transformation may be useful for future applications like inverse problems or time evolution fittings.

## 3.4 REVERSE ODE/SDE AND SCORE MATCHING

From the Itô forward SDE (see Lemma D.1.2), we know that the Stratonovich reverse SDE writes

$$d\overleftarrow{\boldsymbol{x}}_t = \mathbf{G}(\overleftarrow{\boldsymbol{x}}_t)\left(\mathbf{G}(\overleftarrow{\boldsymbol{x}}_t)^\intercal \nabla \log p_{T-t}(\overleftarrow{\boldsymbol{x}}_t)dt + \circ d\overleftarrow{\boldsymbol{B}}_t\right), \tag{3.12}$$

and the reverse probability flow ODE is given as

$$\frac{d\overleftarrow{\boldsymbol{x}}_t}{dt} = \tfrac{1}{2}\mathbf{G}(\overleftarrow{\boldsymbol{x}}_t)\left(\mathbf{G}(\overleftarrow{\boldsymbol{x}}_t)^\intercal \nabla \log p_{T-t}(\overleftarrow{\boldsymbol{x}}_t)\right). \tag{3.13}$$

The corresponding derivations are formulated in Proposition F.1 and Proposition F.2 and are proven in the appendix using Anderson (1982); Song et al. (2021). Following Huang et al. (2021), we directly model $\mathbf{G}(\overleftarrow{\boldsymbol{x}}_t)^\intercal \nabla \log p_{T-t}(\overleftarrow{\boldsymbol{x}}_t)$ by a neural network $\boldsymbol{a}_{\boldsymbol{\theta}}(\overleftarrow{\boldsymbol{x}}_t, T-t)$. Additionally, we incorporate a spherical input layer, see Section L.4.1. We fit the parameters $\boldsymbol{\theta}$ by sliced score matching (SSM) (Song et al., 2020), because in the multiplicative case we do not have an analytic formula for the conditional score $\nabla \log p_s(\vec{\boldsymbol{x}}_s|\vec{\boldsymbol{x}}_0)$ and because of the better scalability of SSM to high-dimensional problems that we would like to address in the future. To this end, we minimize the loss function:

$$\mathcal{L}_{\text{SSM}}(\boldsymbol{\theta}) = \mathbb{E}_{s\sim\mathcal{U}[0,T]}\mathbb{E}_{\vec{\boldsymbol{x}}_s}\mathbb{E}_{\mathbf{v}\sim\text{Rad}(d)}\left[\tfrac{1}{2}\|\boldsymbol{a}_{\boldsymbol{\theta}}(\vec{\boldsymbol{x}}_s, s)\|^2 + (\mathbf{v}\cdot\nabla)(\mathbf{G}(\vec{\boldsymbol{x}}_s)\boldsymbol{a}_{\boldsymbol{\theta}}(\vec{\boldsymbol{x}}_s, s))\cdot\mathbf{v}\right], \tag{3.14}$$

where $\text{Rad}(d)$ denotes the Rademacher distribution in $\mathbb{R}^d$. The following theorem states that even in our multiplicative case, score matching is equivalent to maximize the ELBO, $\mathcal{E}_\infty$. In line with Benton et al. (2024); Ren et al. (2025), this theorem generalizes the result of Huang et al. (2021) and gives a theoretical justification for our score-matching loss equation 3.14. The derivation of this loss from the ELBO below is detailed in Section G.7.

**Theorem 3.4.1.** *Let assumption A1 holds. Then, there exists a constant $C$ such that*

$$p_0(\boldsymbol{x}) \geqslant \mathcal{E}_\infty(\boldsymbol{x}) := C - \int_0^T \mathbb{E}_{\vec{\boldsymbol{x}}_s}\left[\tfrac{1}{2}\|\boldsymbol{a}_{\boldsymbol{\theta}}(\vec{\boldsymbol{x}}_s, s)\|^2 + \nabla\cdot(\mathbf{G}(\vec{\boldsymbol{x}}_s)\boldsymbol{a}_{\boldsymbol{\theta}}(\vec{\boldsymbol{x}}_s, s))\Big|\vec{\boldsymbol{x}}_0 = \boldsymbol{x}\right]ds. \tag{3.15}$$

We proof this theorem in Section G. The first term $C := \mathbb{E}\left[\log p_T(\vec{\boldsymbol{x}}_T)\big|\vec{\boldsymbol{x}}_0 = \boldsymbol{x}\right]$ is a constant w.r.t. to $\boldsymbol{\theta}$. So, it has no effect on the optimization procedure. Therefore, even with our multiplicative noise, the minimization of the ELBO corresponds precisely to Implicit Score Matching (ISM), which is itself equivalent to explicit score matching (ESM), denoising score matching (DSM), and SSM (Huang et al., 2021). Note that formally replacing $\mathbf{G}$ by $\sqrt{2}$, we get the SGM SSM loss. For an easier numerical comparison in Section 6, we will also rely on SSM to train our baseline SGM.

## 4 WORKFLOW

Algorithm 1 summarizes the proposed MSGM procedure. Here we make use of color to highlight the differences compared to SGM. For more details, we refer to Section L.

---

**Algorithm 1:** MSGM (Multiplicative Score-Based Generative Model).

---

**Input:** tensor $\mathbf{G}$, one-dimensional distribution model $\hat{f}_\gamma$, data $\{\overrightarrow{\boldsymbol{x}}_0^m\}_{m=1}^M$, $t_\epsilon$, time horizon $T$, time steps $N_T^f$ and $N_T^b$, time scheduler $g$, score model $\boldsymbol{a}_\theta$, initial ANN parameter $\boldsymbol{\theta}_0$, iterations $N_{\text{iter}}$

— **Training stage** —

1: $\gamma^* \leftarrow \texttt{fit\_distribution}(\hat{f}_\gamma, \{\|\overrightarrow{\boldsymbol{x}}_0^m\|\}_{m=1}^M)$           {Fitting of $\hat{f}_\gamma$, see Section C}
2: **for** $n = 0$ to $N_{\text{iter}} - 1$ **do**
3:     $\overrightarrow{\boldsymbol{x}}_0 \sim \frac{1}{M}\sum_{m=1}^M \delta_{\overrightarrow{\boldsymbol{x}}_0^m}$                 {Random mini-batch from dataset}
4:     $s \sim \mathcal{U}(I(t_\epsilon, T))$                     {Sample uniform gridded time}
5:     $\overrightarrow{\boldsymbol{x}}_s \leftarrow \texttt{SRK4}(s, \lfloor \frac{s}{T} N_T^f \rfloor, \overrightarrow{\boldsymbol{x}}_0, 0, g\mathbf{G})$     {Forward diffusion integration via Algorithm 2}
6:     $\mathbf{v} \sim \text{Rad}(d)$                           {Slicing directions}
7:     $\ell(\boldsymbol{\theta}_n) \leftarrow \mathcal{L}_{\text{SSM}}^{\text{MSGM}}(s, \overrightarrow{\boldsymbol{x}}_s, \mathbf{G}, g\boldsymbol{a}_{\boldsymbol{\theta}_n}, \mathbf{v})$     {Score-matching loss, from equation 3.14}
8:     $\boldsymbol{\theta}_{n+1} \leftarrow \texttt{optimizer\_update}(\boldsymbol{\theta}_n, \ell(\boldsymbol{\theta}_n))$           {e.g. via ADAM}
9: **end for**
10: $\boldsymbol{\theta}^\star \leftarrow \boldsymbol{\theta}_{N_{\text{iter}}}$                       {Set final ANN parameter}

— **Generative sampling stage** —

11: $\overleftarrow{\boldsymbol{x}}_0^{\mathcal{N}} \sim \mathcal{N}(0, \boldsymbol{I}_d),$                   {Sample strong white noise}
12: $\overleftarrow{\boldsymbol{x}}_0 = \hat{f}_{\gamma^*}\left(\|\overleftarrow{\boldsymbol{x}}_0^{\mathcal{N}}\|\right) \frac{\overleftarrow{\boldsymbol{x}}_0^{\mathcal{N}}}{\|\overleftarrow{\boldsymbol{x}}_0^{\mathcal{N}}\|}$       {Sample weak white noise, see equation 3.11}
13: $\overleftarrow{\boldsymbol{x}}_T \leftarrow \texttt{SRK4}(T, N_T^b, \overleftarrow{\boldsymbol{x}}_0, g\mathbf{G}\boldsymbol{a}_{\theta^*}, g\mathbf{G})$     {Reverse diffusion integration via Algorithm 2}
14: **return** $\gamma^*, \boldsymbol{\theta}^*, \overleftarrow{\boldsymbol{x}}_T$

---

## 5 RELATED WORKS

Combining machine learning and mechanistic approaches is now a common approach. We may cite physics-informed neural networks (PINNs) (Raissi et al., 2019; Lu & Xu, 2024), physics-based prior covariance (Beauchamp et al., 2025; Clarotto et al., 2024), deep augmentation (Holzschuh et al., 2023; Fan et al., 2025), neural Galerkin (Lee & Carlberg, 2020; Chen et al., 2021; Romor et al., 2023; Finzi et al., 2023; Bruna et al., 2024; Kim et al., 2022), and chaos from energy-based models (Fournier & Pierfrancesco, 2025) among others. Here, we shall focus on score-based generative models. Bastek et al. (2024) add the physical equations inside their score matching loss. Holzschuh et al. (2023) fit a score to correct a backward physical equation but does not propose any generative model. To denoise corrupted images, several authors (e.g. Zhou et al., 2014; Shan et al., 2022; Guha & Acton, 2023) encode the multiplicative structure of speckle noise. Since this noise is not correlated between pixels, this approach strongly differs from ours. Most of these works do not deal with score or generative models. Guha & Acton (2023); Ren et al. (2025); Shetty et al. (2025) do, but their framework simplifies to SGM by considering the pixel-wise logarithm of images. Guth et al. (2022); Lempereur & Mallat (2024) encode a target multiscale structure (e.g. turbulence) by a hierarchy of normalized wavelets conditioned by the larger scales. Chen & Vanden-Eijnden (2025) adapt the noise to that multiscale structure in a stochastic interpolant context. Lobbe et al. (2023; 2025) replaced the Gaussian process involved in the transport-noise equations (Kraichnan, 1968; Piterbarg & Ostrovskii, 1997; Resseguier et al., 2021) by a Shrodinger bridge (De Bortoli et al., 2021). They inserted a SGM inside a transport noise dynamics, whereas we inserted a dynamics similar to transport noise inside a SGM. Following general Bayesian approaches, some of the literature on transport-noise relies on the Girsanov theorem to fit a drift modification or evaluate a likelihood (Cotter et al., 2020a; Singh et al., 2025)(see Section G.9). By extending the ELBO of (Huang et al., 2021) to SDEs inspired by transport noise, Theorem 3.4.1 justifies our fit of the backward SDE drift.

Several authors have recently proposed Langevin equations (Arnaudon et al., 2019; Luesink & Street, 2025; Ayala et al., 2025) and SGM (De Bortoli et al., 2022; Huang et al., 2022; Lou et al., 2023; Benton et al., 2024) on Riemannian manifolds in order to generate data lying on a particular manifold. Clearly different, our goal is more classical: generating data in $\mathbb{R}^d$. In our work, neither data nor their noisy versions are restricted to a single manifold. However, each solution path of our forward and backward SDE lies on its particular Riemannian manifold, the scaled $d$-sphere $\|\overrightarrow{\boldsymbol{x}}_0\|\mathbb{S}^{d-1}$. De Bortoli et al. (2022) describes diffusions and SGM in the $d$-sphere $\mathbb{S}^{d-1}$. A detailed comparison is given in Section H.

Regarding extremes generation, using variational autoencoders (VAE), Lafon et al. (2023) argue that Gaussian latent restricts the generated samples to light-tail distributions. Accordingly, they propose to use fat-tailed latent distributions (see also Jaini et al. (2020); Huster et al. (2021) for normalizing flows (NF) and generative adversarial models (GAN) respectively). Yoon et al. (2023); Shariatian et al.; Pandey et al. (2024); Ren et al. (2025) proposed SGMs with ad hoc heavy-tailed (Lévy and Student-$t$) latent distributions. Our approach automatically makes the tails of the latent distribution fat when necessary. It learns it from the distribution of the data norm, $p_{|.|}$. Similarly, the diffusion proposed by Dharmakeerthi et al. (2025) adapt to data but through a nonlinear drift and an additive noise. Li et al. (2024); Price et al. (2025); Stamatelopoulos & Sapsis (2025) and references therein show that the usual SGM may correctly represent extremes, especially in "interpolation mode", that is, when extremes lie on the interior of the dataset but have difficulties with extremes lying on the dataset boundaries. Our numerical experiments in Section 6 suggest that our method probably overcomes this limitation. To represent the directionality of extremes, many authors decompose norms and directions of extreme events (Engelke et al., 2019; Palacios-Rodríguez et al., 2020; Lafon et al., 2023; Naveau & Segers, 2024). Large-amplitude criterion (e.g. exceeding a high threshold) or fat-tail model can be applied on the norm. Extreme directions may or may not become asymptotically independent of their magnitude (Engelke et al., 2019; Lafon et al., 2023). Build on random rotations, MSGM naturally suggests such a polar decomposition. The extreme direction of the MSGM latent vector is asymptotically independent of its magnitude. However, the direction of the reverse process does depend on the magnitude (see Section H.2).

## 6 EXPERIMENTS

For our numerical experiments, we choose to define a tensor $\boldsymbol{G}^k$ in a simple way. We sample $d$ random matrices, keep only their skew-symmetric parts, and normalize:

$$\mathbf{G} = \frac{\sqrt{d}}{\|\tilde{\mathbf{G}}\|_2}\,\tilde{\mathbf{G}} \quad \text{with} \quad \tilde{G}_{i,j}^k = \tfrac{1}{2}(M_{i,j}^k - M_{j,i}^k) \quad \text{and} \quad M_{i,j}^k \stackrel{iid}{\sim} \mathcal{N}(0,1). \tag{6.1}$$

In Section J we show that this random tensor $\mathbf{G}$ respects conditions A1 and A2 almost surely. Section K proposes alternative tensor definitions with sparse structures that allow high-dimensional applications. Following Section K.2.2, MSGM can generate images as in Section M.7.2. For the test cases below, we also checked in Section M.6.2 and Section M.7.1 that the MSGM generation skills are equivalent with these sparse and dense tensors. However, these sparse tensors do not match the framework of Section 3.1 so we postpone the associated numerical evaluations to future works.

### 6.1 MULTIVARIATE CAUCHY DISTRIBUTION

We first illustrate our method with a vector of Cauchy variables, $\boldsymbol{x}_{\text{Ca}}$, with scale parameter $\gamma$:

$$(x_{\text{Ca}})_i \stackrel{iid}{\sim} p_{\text{Ca}} \quad \text{with} \quad p_{\text{Ca}}(x) := \frac{\gamma/\pi}{x^2 + \gamma^2}. \tag{6.2}$$

It is an extreme case of fat-tailed distributions with a power-law tail: $p_{\text{Ca}}(x) \propto |x|^{-2}$ for large $|x|$. Real problems often involve both correlation and dimensionality $d > 2$. So, we correlate Cauchy variables, as $\boldsymbol{x}_0 = \boldsymbol{A}\boldsymbol{x}_{\text{Ca}}$, with a fixed matrix, $\boldsymbol{A}$, initialized with i.i.d. coefficients $\boldsymbol{A}_{i,j} \sim \mathcal{N}(0,1)$. Figure 2 confirms that, for $d = 4$, SGM hardly reproduces fat tails and extreme directionality, unlike MSGM. An explanation is the strong dissimilarity between the data distribution and the latent SGM distribution; see Section E.5 and Section E.6. For a larger number of ADAMS iterations, MSGM becomes more accurate, whereas SGM diverges (see Figure 4a). More plots, numerical comparisons, and experiments with variants of the state-of-the-art SGM can be found in Section M.6.1.

### 6.2 MEASURED VORTICITY FIELDS

We also test our algorithm on fluid dynamics experimental data: small images of vorticity fields. These fields are two-dimensional curl of fluid velocity measured by Particle Image Velocimetry (PIV) in wind tunnels (Georgeault & Heitz, 2026). Vorticity quantifies the local rotation speed of fluid and is known to have point-wise distributions with tails fatter than Gaussian ones (Wilczek & Friedrich, 2009). We focus on a benchmark fluid flow: a wake flow at Reynolds number 3900 created by a circular cylinder embedded in a mean stream (Parnaudeau et al., 2008). Each vorticity sample

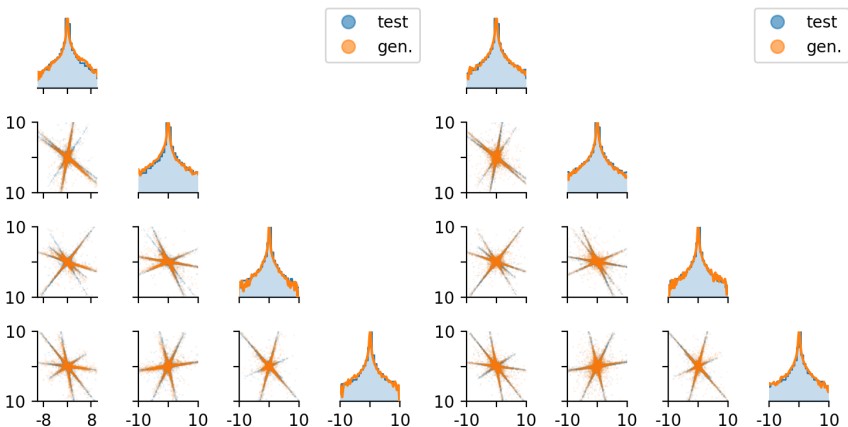

Figure 2: Pair plots of generated data (orange lines and dots) compared to ground truth data (blue lines and dots) with the SGM (left) and MSGM (right) for a vector of 4 correlated Cauchy variables. On the diagonal, log-histogram and logarithm of the pdf KDE estimation are superimposed.

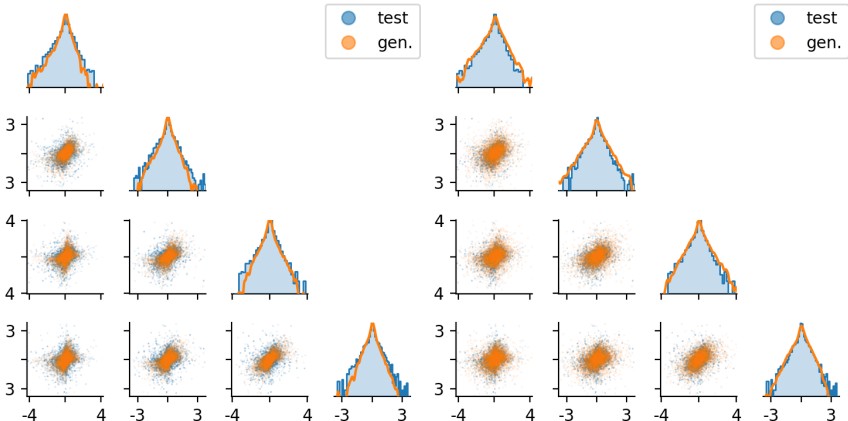

Figure 3: Pair plots of generated data (orange lines and dots) compared to ground truth data (blue lines and dots) with the SGM (left) and MSGM (right) trained on 1024 16-dimensional measured vorticity fields. On the diagonal, log-histogram and logarithm of the pdf KDE estimation are superimposed.

is evaluated at $d = 16$ spatial points to ensure low dimensionality. We use limited training data (1024 data points) to make rare events even more rare and learning more challenging. Section M.7 provides a deeper description of this experimental dataset. Figure 3 highlights a larger concentration of points generated by SGM in the center of the ground truth distribution. Accordingly, the tails of the marginals – i.e. the tails of the vorticity point-wise distributions – are underestimated : SGM underestimates rare large vorticity events. MSGM performs better since the MSGM latent distribution – easy to learn – is much closer to the data distribution than SGM latent distribution, as theoretically suggested by Section E.5 and experimentally verified in Section M.7. In particular, the MSGM latent distribution seems to have Laplace tails and to be more accurate in the low-data regime (see Figure 4). Additional we carried out high-dimensional experiments with $d = 1024$ in Section M.7.2 based on sparse tensor $G$ developed in Section K. More details on data, preprocessing, illustrations, and other numerical experiments are given in Section M.

## 7 CONCLUSION AND DISCUSSION

We have proposed a new type of diffusion model with multiplicative noise. After a theoretical analysis of this ansatz, an algorithm is specified to mimic fat-tailed distributions, surpassing SGM in this task.

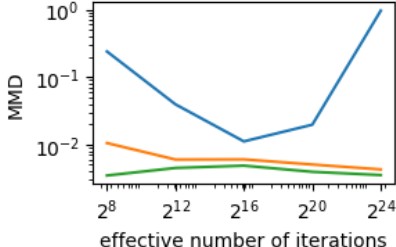 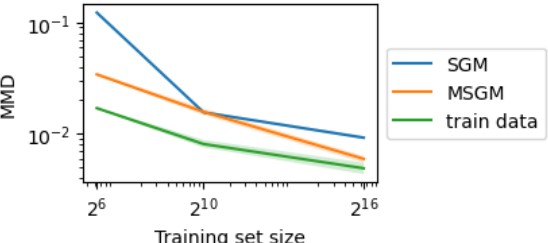

(a) Single MMD value for the correlated Cauchy distribution ($d = 4$) as a function of the number of effective ADAMS iterations.

(b) Mean and 80% CI of MMD for the vorticity measurements distribution ($d = 16$) as a function of the number of training samples.

Figure 4: Convergence behaviors of the Maximum Mean Discrepancy (MMD) for two different test cases. $10^4$ samples are used for each MMD evaluation.

At this point, limitations of the general MSGM framework may be difficult to know. We rather discuss the limitations of the first numerical procedure applied in Section 6. First, the forward SDE has to be integrated numerically since we do not know an analytic solution for large-rank tensors (see Section I for the solution with low-rank tensors). It implies either a slower training or a reduced number of iterations compared to SGM. Moreover, we do not know of any analytic solution for the score in finite time. This prevents the use of DSM and force us to use ISM or SSM, a less stable approach. In the next future, we can hope that the active communities of generative models on symmetric Riemannian manifolds and, more generally, of stochastic differential geometry could come up with more efficient sampling algorithms and score evaluation procedures for our diffusions on $d$-spheres. In addition, random matrix theory and free probabilities (Biane, 1997; Delyon & Yao, 2006; Demni, 2008; Lévy, 2008; Delyon, 2010; Demni & Hmidi, 2012; Cébron, 2014) may provide alternative sampling methods and helpful results for large-dimension cases. Indeed, for some choices of **G**, the semigroup of our forward SDE may be expressed as a unitary Brownian matrix, converging for large dimensions to a free multiplicative Brownian matrix. Both theories could facilitate the sampling and the score evaluation of the MSGM forward diffusion. Moreover, a dense third-order tensor **G** prevents image processing and other large-dimensional applications, related to, say, turbulent fluid dynamics. In fact, dimensions $d$ of such problems are very large – typically $d = O(10^5)$ or more. A dense tensor **G** as we use in our numerical experiments has $d^3$ coefficients, and the memory and computational costs would become prohibitive in these cases. To address this issue, Section K proposes several sparse tensors and alternative to assumptions A1 and A2. Section M.7.2 shows first MSGM generated images in dimension $d = 1024$. Furthermore, we are currently developing physics-based sparse tensors **G**. Here, MSGM forward SDE is the spatial discretization of a stochastic partial differential equation involving transport noises (Kraichnan, 1968; Piterbarg & Ostrovskii, 1997; Resseguier et al., 2021). We expect that the physical inductive bias will facilitate both inference and learning, especially in low-data mode. Alternatively, the rank assumption A2 may be expressed more simply with the algebraic properties of **G**, eventually producing simple examples of sparse and efficient tensors.

In addition to the improvements discussed above, many paths remain to be explored. First, our theoretical results could be generalized to other multiplicative diffusions. We have considered dense linear maps $x \mapsto \mathbf{G}(x)$ with $\text{Im}(\mathbf{G}(x)) = x^\perp$ for any $x \neq 0$. We believe that sparse linear maps of Section K and non-linear Lipchitz-continuous maps can yield similar theoretical results as long as that image condition is fulfilled for almost every $x \in \mathbb{R}^d$ (see Section K.1). The non-linear case would include in particular sphere-wise diffusions of De Bortoli et al. (2022) (see Section H.2). Second, we could address dynamical system forecasting. With the Gaussianization of MSGM latent vectors (see Section E.4) complex nonlinear dynamics could simplify to uncoupled one-dimensional linear dynamics as in Arbabi & Sapsis (2022). A third path to explore involves Bayesian inverse problems and data assimilation (Rozet & Louppe, 2023; Bao et al., 2025). Finally, our analytic solution issue could be bypassed by a normalizing flow approach: spherical decomposition, stochastic interpolants and flow along scaled $d$-spheres, taking inspiration from normalizing flow along Riemannian manifolds (e.g., Gemici et al., 2016; Mathieu & Nickel, 2020; Wu et al., 2025).

ACKNOWLEDGMENTS

We thank Nizar Demni for proofreading, Erwin Luesink, Gilles Carron, and Erwan Brugallé for comparison with Brownian motions on hypersheres, Thomas Optiz, Philippe Naveau, Etienne Mémin, Dominique Heitz, and Théo Voillemin for discussions, Jocelyn De-Goer-De-Herve and colab.IA for sharing GPUs. This work has been funded by the joint INRAE-Inria PhD grant, the ERC project 856408-STUOD, and by DFG under Germany´s Excellence Strategy (EXC-2046/2, project ID 390685689).

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

# Appendix

## Table of Contents

## A   LOSSES FOR SCORE MATCHING

This section presents some classical score-matching losses. In SGM backward SDE, the score $\nabla \log p_{T-t}(\boldsymbol{x})$ is replaced by a fitted neural network $\boldsymbol{s}_{\boldsymbol{\theta}}(\boldsymbol{x}, T-t)$. This fitting is performed by minimizing some losses, like denoising, implicit, or slicing score-matching losses. Alternatively, one can work on $\boldsymbol{a}_{\boldsymbol{\theta}}(\boldsymbol{x}, T-t) = \sqrt{2}\boldsymbol{s}_{\boldsymbol{\theta}}(\boldsymbol{x}, T-t)$ (Huang et al., 2021). It leads to this SGM backward SDE:

$$d\overleftarrow{\boldsymbol{x}}_t = \overleftarrow{\boldsymbol{x}}_t dt + \sqrt{2}(\boldsymbol{a}_{\boldsymbol{\theta}}(\boldsymbol{x}, T-t)dt + d\overleftarrow{\boldsymbol{B}}_t), \tag{A.1}$$

A typical loss to learn this neural network is denoising score matching (DSM)

$$\mathcal{L}_{\text{DSM}} = \int_0^T \tfrac{1}{2}\mathbb{E}_{\overrightarrow{\boldsymbol{x}}_s}\|\boldsymbol{a}_{\boldsymbol{\theta}}(\overrightarrow{\boldsymbol{x}}_s, s) - \sqrt{2}\nabla \log p_s(\overrightarrow{\boldsymbol{x}}_s|\overrightarrow{\boldsymbol{x}}_0)\|^2 ds. \tag{A.2}$$

where $\|.\|$ is the Euclidian norm. By integration by part, we can show that DSM is equivalent to Implicit Score Matching (ISM) (Hyvärinen & Dayan, 2005)

$$\mathcal{L}_{\text{ISM}} = \int_0^T \mathbb{E}_{\overrightarrow{\boldsymbol{x}}_s}\left(\tfrac{1}{2}\|\boldsymbol{a}_{\boldsymbol{\theta}}(\overrightarrow{\boldsymbol{x}}_s, s)\|^2 + \nabla \cdot (\sqrt{2}\boldsymbol{a}_{\boldsymbol{\theta}}(\overrightarrow{\boldsymbol{x}}_s, s))\right) ds. \tag{A.3}$$

A reference score $\nabla \log p_s$ is not needed anymore. However, the divergence term may be untractable for large-dimensional problems. Using the Hutchingson trick, Song et al. (2020) shows that this loss is equivalent to a trackable version : the Sliced Score Matching (SSM)

$$\mathcal{L}_{\text{SSM}} = \int_0^T \mathbb{E}_{\overrightarrow{\boldsymbol{x}}_s}\mathbb{E}_{\mathbf{v}\sim\mathcal{N}(0,\boldsymbol{I}_d)}\left(\tfrac{1}{2}\|\boldsymbol{a}_{\boldsymbol{\theta}}(\overrightarrow{\boldsymbol{x}}_s, s)\|^2 + (\mathbf{v}\cdot\nabla)(\sqrt{2}\boldsymbol{a}_{\boldsymbol{\theta}}(\overrightarrow{\boldsymbol{x}}_s, s)\cdot\mathbf{v})\right) ds. \tag{A.4}$$

Score matching is equivalent to maximizing the Evidence Lower Bound (ELBO) both in discrete time (Luo, 2022) and in continuous time (Huang et al., 2021). Indeed, denoting $\mathcal{E}_\infty$ the ELBO, Huang et al. (2021) shows that:

$$
\begin{aligned}
\mathcal{E}_\infty(\boldsymbol{x}) &= \mathbb{E}\left[\log p_0(\vec{\boldsymbol{x}}_T)\middle|\vec{\boldsymbol{x}}_0 = \boldsymbol{x}\right] \\
&\quad - \int_0^T \mathbb{E}_{\vec{\boldsymbol{x}}_s}\left[\tfrac{1}{2}\|\boldsymbol{a_\theta}(\vec{\boldsymbol{x}}_s, s)\|^2 + \nabla \cdot (\sqrt{2}\boldsymbol{a_\theta}(\vec{\boldsymbol{x}}_s, s))\middle|\vec{\boldsymbol{x}}_0 = \boldsymbol{x}\right] ds.
\end{aligned}
\tag{A.5}
$$

The first term does not depends on the neural network parameters $\boldsymbol{\theta}$. The expectation of the second term over $\boldsymbol{x}$ following the dataset distribution is $\mathcal{L}_{\text{ISM}}$. So, maximizing the ELBO is equivalent to minimize the ISM. Table 1 of Huang et al. (2021) summarizes the classical score matching losses.

# B  STOCHASTIC CALCULUS AND STRATONOVICH INTEGRALS

This appendix provides a concise overview of essential stochastic calculus concepts from Oksendal (1998); Kunita (1997) relevant to our work, especially the Stratonovich interpretation of stochastic differential equations (SDEs).

## B.1  ITÔ INTEGRALS AND SDES

Let $(\Omega, \mathcal{F}, \mathbb{P})$ be a probability space equipped with a filtration $(\mathcal{F}_t)_{t\geq 0}$ satisfying the usual conditions, and let $(\boldsymbol{B}_t)_{t\geq 0}$ be a standard $m$-dimensional Brownian motion. Given an adapted process $\boldsymbol{X}_t \in \mathbb{R}^{d\times m}$ satisfying appropriate integrability conditions, the *Itô integral* of $\boldsymbol{X}$ with respect to $\boldsymbol{B}$ is defined as the mean-square limit:

$$
\int_0^T \boldsymbol{X}_s \, d\boldsymbol{B}_s := \lim_{|\Pi|\to 0} \sum_{[t_i, t_{i+1}]\in\Pi} \boldsymbol{X}_{t_i}(\boldsymbol{B}_{t_{i+1}} - \boldsymbol{B}_{t_i}),
\tag{B.1}
$$

where the sum is taken over a partition $\Pi$ of $[0, T]$.

An SDE interpreted in the Itô sense reads:

$$
d\boldsymbol{X}_t = \boldsymbol{f}(\boldsymbol{X}_t, t) \, dt + \mathbf{G}(\boldsymbol{X}_t, t) \, d\boldsymbol{B}_t,
\tag{B.2}
$$

where $\boldsymbol{f} : \mathbb{R}^d \times \mathbb{R}_+ \to \mathbb{R}^d$ is the drift, and $\mathbf{G} : \mathbb{R}^d \times \mathbb{R}_+ \to \mathbb{R}^{d\times m}$ is the diffusion coefficient.

## B.2  STRATONOVICH INTEGRALS AND CHAIN RULE

Unlike the Itô integral, the *Stratonovich integral* is defined using a symmetric discretization:

$$
\int_0^T \boldsymbol{X}_s \circ d\boldsymbol{B}_s := \lim_{|\Pi|\to 0} \sum_{[t_i, t_{i+1}]\in\Pi} \frac{\boldsymbol{X}_{t_i} + \boldsymbol{X}_{t_{i+1}}}{2}(\boldsymbol{B}_{t_{i+1}} - \boldsymbol{B}_{t_i}).
\tag{B.3}
$$

A Stratonovich SDE is written as:

$$
d\boldsymbol{X}_t = \boldsymbol{f}_S(\boldsymbol{X}_t, t) \, dt + \mathbf{G}(\boldsymbol{X}_t, t) \circ d\boldsymbol{B}_t.
\tag{B.4}
$$

A key advantage of the Stratonovich formulation is that it satisfies the classical chain rule. For any smooth function $\phi : \mathbb{R}^d \to \mathbb{R}$, we have:

$$
d\phi(\boldsymbol{X}_t) = \nabla\phi(\boldsymbol{X}_t)^\top \boldsymbol{f}_S(\boldsymbol{X}_t, t) \, dt + \nabla\phi(\boldsymbol{X}_t)^\top \mathbf{G}(\boldsymbol{X}_t, t) \circ d\boldsymbol{B}_t.
\tag{B.5}
$$

Moreover, in multiscale deterministic or stochastic equations, if the fast component is a continuous process with infinitesimal correlation time, the slow component generally converges to the solution of another SDE. In this other SDE, the fast component is often replaced by a Stratonovich integral (Arnold, 1974). Note that it is not always true for nonlinear dynamics (Gottwald & Melbourne, 2013; Gottwald et al., 2015). Accordingly, the readers may interpret the Stratonovich noise $s \mapsto \circ d\boldsymbol{B}_s$ as a formal representation of a process with a short correlation time that nevertheless respects the classical rules of differential calculus, in particular, the chain rule.

### B.3 CONVERSION BETWEEN ITÔ AND STRATONOVICH FORMS

Given a Stratonovich SDE, it is always possible to convert it to the equivalent Itô form:

$$d\boldsymbol{X}_t = \left( \boldsymbol{f}_S(\boldsymbol{X}_t, t) + \frac{1}{2} \sum_{j=1}^{m} G_j(\boldsymbol{X}_t, t) \cdot \nabla G_j(\boldsymbol{X}_t, t) \right) dt + \mathbf{G}(\boldsymbol{X}_t, t) \, d\boldsymbol{B}_t, \tag{B.6}$$

where $G_j(\boldsymbol{x}, t)$ is the $j$-th column of the diffusion matrix $\mathbf{G}(\boldsymbol{x}, t)$. The additional drift term arises from the correction due to the non-zero quadratic variation of the noise.

### B.4 FOKKER–PLANCK EQUATION

An Itô SDE of the form

$$d\boldsymbol{X}_t = \boldsymbol{f}(\boldsymbol{X}_t, t) \, dt + \mathbf{G}(\boldsymbol{X}_t, t) \, d\boldsymbol{B}_t, \tag{B.7}$$

induces a time-evolution equation for the probability density $p(\boldsymbol{x}, t)$ of $\boldsymbol{X}_t$. This is known as the *Fokker–Planck equation*, given by:

$$\partial_t p(\boldsymbol{x}, t) = -\nabla \cdot (\boldsymbol{f}(\boldsymbol{x}, t) \, p(\boldsymbol{x}, t)) + \frac{1}{2} \nabla \cdot \left( \nabla \cdot (\boldsymbol{\Sigma}(\boldsymbol{x}, t) \, p(\boldsymbol{x}, t))^\top \right), \tag{B.8}$$

where $\boldsymbol{\Sigma}(\boldsymbol{x}, t) := \mathbf{G}(\boldsymbol{x}, t)\mathbf{G}(\boldsymbol{x}, t)^\top$ is the diffusion tensor. The Fokker–Planck equation describes the deterministic evolution of the probability density associated with the stochastic process.

## C SAMPLING FROM 1D DISTRIBUTIONS

Let us denote by $p_{|.|}$ the distribution of the norms, $\|\vec{\boldsymbol{x}}_T\|$. In MSGM, it is also the distribution of $\|\vec{\boldsymbol{x}}_0\|$ (see Proposition D.3.2). This distribution is arbitrary, but is a one-dimensional distribution. So, it is straightforward to learn and sample from, e.g., using an empirical cumulative distribution function (eCDF) (Cantelli, 1933; Glivenko, 1933; Tucker, 1959). Norms are positive and might be close to zero, so in practice we work with a regularized log-norm: $\log \|\boldsymbol{x}\|_\epsilon := \log(\|\boldsymbol{x}\| + \epsilon)$ with $\epsilon$ small, typically $\epsilon = 10^{-6}$. From a data set of the log-norms of $M$ training samples, $(\log \|\vec{\boldsymbol{x}}_T^{(i)}\|_\epsilon)_i = (\log \|\vec{\boldsymbol{x}}_0^{(i)}\|_\epsilon)_i$, we define eCDF $\hat{F}_{\log|.|_\epsilon}$ as

$$\hat{F}_{\log|.|_\epsilon} : \mathbb{R} \to [0, 1], \tag{C.1}$$

$$R \mapsto \frac{1}{M} \sum_{i=1}^{M} \mathbf{1}_{\{R \geqslant \log \|\vec{\boldsymbol{x}}_T^{(i)}\|_\epsilon\}}. \tag{C.2}$$

Then, we approximate the distribution of the norms, $p_{\log|.|_\epsilon}(R)\mathrm{d}R$, by the empirical one, $\hat{p}_{\log|.|_\epsilon}(R)\mathrm{d}R := \hat{F}_{\log|.|_\epsilon}(\mathrm{d}R)$. In particular, we can sample a new norm of latent variables, $\|\vec{\boldsymbol{x}}_T\|$, from a uniform one-dimensional variable $u \sim \mathcal{U}(0, 1)$ as follows

$$\|\vec{\boldsymbol{x}}_T\| = \|\vec{\boldsymbol{x}}_0\| = \exp(\log \|\vec{\boldsymbol{x}}_0\|_\epsilon) - \epsilon = \exp\left( \hat{F}_{\log|.|_\epsilon}^{-1}(u) \right) - \epsilon. \tag{C.3}$$

eCDF is an efficient tool, but it cannot generalize the distribution $\hat{p}_{|.|}$ outside the training set $(\|\vec{\boldsymbol{x}}_T^{(i)}\|)_i$. For better generalization, instead, one could use a one-dimensional kernel density estimation or fitting of parametric distributions. In the case of one-dimensional distributions with fat tails, classical kernel density estimation (KDE) suffers from bias in the tail estimation or peaks due to sparse data in the tails. In that case, one could consider more robust approaches that, in general, do not require the existence of moments of the target distribution Tokdar et al. (2024).

In this paper, we rely on the eCDF.

## D THE FOKKER-PLANCK EQUATION AND ITS INVARIANT MEASURES

### D.1 ITÔ FORM OF THE FORWARD SDE

Define the conditional noise covariance $\boldsymbol{\Sigma}(\boldsymbol{x})$ as

$$\boldsymbol{\Sigma}(\boldsymbol{x}) := \mathbf{G}(\boldsymbol{x})\mathbf{G}(\boldsymbol{x})^\intercal = \mathbb{E}[(d\vec{\boldsymbol{x}}_s)(d\vec{\boldsymbol{x}}_s)^\intercal | \vec{\boldsymbol{x}}_s = \boldsymbol{x}]. \tag{D.1}$$

We begin with a lemma.

**Lemma D.1.1.** *Let the skew-symmetry assumption A1 hold. Then,*

$$\frac{1}{2}\nabla \cdot (\boldsymbol{\Sigma}(\boldsymbol{x})) = \frac{1}{2}\sum_{k=1}^{d}(\boldsymbol{G}^k)^2\boldsymbol{x}. \tag{D.2}$$

*Proof.* Let us explicitly state the matrix divergence. For $k = 1, \ldots, d$ define $\boldsymbol{\Sigma}^k(\boldsymbol{x}) = [\Sigma_{ij}^k(\boldsymbol{x})] := \boldsymbol{G}^k\boldsymbol{x}\boldsymbol{x}^\intercal(\boldsymbol{G}^k)^\intercal$, then we decompose $\boldsymbol{\Sigma}$ as

$$\boldsymbol{\Sigma}(\boldsymbol{x}) := \mathbf{G}(\boldsymbol{x})\mathbf{G}^\intercal(\boldsymbol{x}) = \sum_{k=1}^{d}\boldsymbol{\Sigma}^k(\boldsymbol{x}). \tag{D.3}$$

Then, taking the divergence of the $j$-th column of $\boldsymbol{\Sigma}^k(\boldsymbol{x})$, we obtain

$$\left(\nabla \cdot \left(\boldsymbol{\Sigma}^k(\boldsymbol{x})\right)\right)_j = \nabla \cdot [\boldsymbol{\Sigma}^k(\boldsymbol{x})]_{:,j} = \sum_{i=1}^{d}\frac{\partial}{\partial x_i}\Sigma_{ij}^k(\boldsymbol{x}), \tag{D.4}$$

$$= \sum_{i=1}^{d}\frac{\partial}{\partial x_i}\sum_{p,q=1}^{d}G_{ip}^k x_p x_q G_{jq}^k, \tag{D.5}$$

$$= \sum_{i,p,q=1}^{d}G_{ip}^k(\delta_{ip}x_q + x_p\delta_{iq})G_{jq}^k, \tag{D.6}$$

$$= \sum_{p,q=1}^{d}G_{pp}^k x_q G_{jq}^k + G_{qp}^k x_p G_{jq}^k, \tag{D.7}$$

$$= [\mathrm{tr}(\boldsymbol{G}^k)\boldsymbol{G}^k x + \boldsymbol{G}^k(\boldsymbol{G}^k x)]_j. \tag{D.8}$$

By skew-symmetry $\mathrm{trace}(\boldsymbol{G}^k) = 0$ and consequently

$$\nabla \cdot (\boldsymbol{\Sigma}(\boldsymbol{x})) = \sum_{k=1}^{d}\left(\nabla \cdot \left(\boldsymbol{\Sigma}^k(\boldsymbol{x})\right)\right) = \sum_{k=1}^{d}\boldsymbol{G}^k(\boldsymbol{G}^k x).$$

$\square$

**Lemma D.1.2.** *(**Forward SDE - Itô**) Let the skew-symmetry assumption equation A1 hold. Then, the Itô form of the forward SDE equation 3.1 of $\vec{\boldsymbol{x}}_s$ is given by*

$$d\vec{\boldsymbol{x}}_s = \tfrac{1}{2}(\nabla \cdot \boldsymbol{\Sigma})(\vec{\boldsymbol{x}}_s)ds + \mathbf{G}(\vec{\boldsymbol{x}}_s)d\vec{\boldsymbol{B}}_s. \tag{D.9}$$

*Proof.* Using the standard Stratonovich-to-Itô formula (e.g. Kunita, 1997), it holds

$$d\vec{\boldsymbol{x}}_s = \tfrac{1}{2}d\langle\mathbf{G}(\vec{\boldsymbol{x}}_s), \vec{\boldsymbol{B}}_s\rangle_s + \mathbf{G}(\vec{\boldsymbol{x}}_s)d\vec{\boldsymbol{B}}_s, \tag{D.10}$$

$$= \tfrac{1}{2}\sum_{k=1}^{d}d\langle\boldsymbol{G}^k\vec{\boldsymbol{x}}_s, (\vec{B}_s)_k\rangle_s + \mathbf{G}(\vec{\boldsymbol{x}}_s)d\vec{\boldsymbol{B}}_s, \tag{D.11}$$

$$= \tfrac{1}{2}\sum_{k=1}^{d}(\boldsymbol{G}^k)^2\vec{\boldsymbol{x}}_s ds + \mathbf{G}(\vec{\boldsymbol{x}}_s)d\vec{\boldsymbol{B}}_s, \tag{D.12}$$

$$= \tfrac{1}{2}(\nabla \cdot \boldsymbol{\Sigma})(\vec{\boldsymbol{x}}_s)ds + \mathbf{G}(\vec{\boldsymbol{x}}_s)d\vec{\boldsymbol{B}}_s, \tag{D.13}$$

where the last equality comes from Lemma D.1.1. $\square$

## D.2 FOKKER-PLANCK EQUATION AND THEOREM 3.1.1

Let $\vec{\boldsymbol{x}}_0 \sim p_{\vec{\boldsymbol{x}}_0}$ with $p_{\vec{\boldsymbol{x}}_0} \in \mathcal{C}^2(\mathbb{R}^d)$ be twice continuously differentiable. Let $p_s$ denote the density of the distribution of $\vec{\boldsymbol{x}}_s$. For each $\boldsymbol{x} \in \mathbb{R}^d$ we define the normalized vector $\boldsymbol{x}^n := \frac{\boldsymbol{x}}{\|\boldsymbol{x}\|}$ for $\boldsymbol{x} \neq 0$ and 0 otherwise, which is orthogonal to the $d$-sphere $\mathbb{S}^{d-1}$. Furthermore, let $\nabla_\perp$ be the orthogonal projection of the gradient $\nabla$ in the tangent plane $\boldsymbol{x}^\perp = \mathcal{T}_{\boldsymbol{x}} \mathbb{S}^{d-1}$, the tangent space on the Riemannian manifold $\mathbb{S}^{d-1}$ at the point $\boldsymbol{x}$, defined for $f \in \mathcal{C}^2(\mathbb{R}^d)$ as

$$\nabla_\perp f \quad := \quad \nabla f - (\boldsymbol{x}^n \cdot \nabla f) \, \boldsymbol{x}^n. \tag{D.14}$$

**Lemma D.2.1.** *It holds for any smooth vector field $f$ that*

$$\nabla \cdot f = (\boldsymbol{x}^n \cdot \nabla)(\boldsymbol{x}^n \cdot f) + \nabla_\perp \cdot f. \tag{D.15}$$

*Proof.* Let $f$ be a smooth vector field, then

$$(\boldsymbol{x}^n (\boldsymbol{x}^n)^{\mathsf{T}} \nabla) \cdot f = \sum_{i=1}^d \sum_{j=1}^d x_i^n x_j^n \frac{\partial}{\partial x_j} f_i = \sum_{i=1}^d (\boldsymbol{x}^n)_i \langle \boldsymbol{x}^n, \nabla \rangle f_i = \langle \langle \boldsymbol{x}^n, \nabla \rangle f, \boldsymbol{x}^n \rangle. \tag{D.16}$$

It holds for each $j = 1, \dots, d$ that

$$[(\boldsymbol{x}^n \cdot \nabla)\boldsymbol{x}^n]_j = \left[ \left( \sum_{i=1}^d (\boldsymbol{x}^n)_i \frac{\partial}{\partial x_i} \right) \frac{\boldsymbol{x}}{\|\boldsymbol{x}\|} \right]_j = (\boldsymbol{x}^n)_j \frac{1}{\|\boldsymbol{x}\|} - \sum_{i=1}^d (\boldsymbol{x}^n)_i \frac{x_i}{\|\boldsymbol{x}\|^3} x_j, \tag{D.17}$$

$$= x_j / \|\boldsymbol{x}\|^2 - \left( \sum_{i=1}^d x_i^2 / \|\boldsymbol{x}\|^4 \right) x_j, \tag{D.18}$$

$$= 0. \tag{D.19}$$

Consequently,

$$(\boldsymbol{x}^n \cdot \nabla)(\boldsymbol{x}^n \cdot f) = (\boldsymbol{x}^n \cdot \nabla)\boldsymbol{x}^n \cdot f + (\boldsymbol{x}^n \cdot \nabla)f \cdot \boldsymbol{x}^n = (\boldsymbol{x}^n \cdot \nabla)f \cdot \boldsymbol{x}^n. \tag{D.20}$$

Using the decomposition $\nabla = \boldsymbol{x}^n (\boldsymbol{x}^n)^{\mathsf{T}} \nabla + (I - \boldsymbol{x}^n (\boldsymbol{x}^n)^{\mathsf{T}})\nabla = \boldsymbol{x}^n (\boldsymbol{x}^n)^{\mathsf{T}} \nabla + \nabla_\perp$, equation D.16 and equation D.20 we conclude

$$\nabla \cdot f = (\boldsymbol{x}^n (\boldsymbol{x}^n)^{\mathsf{T}} \nabla + \nabla_\perp) \cdot f = (\boldsymbol{x}^n \cdot \nabla)f \cdot \boldsymbol{x}^n + \nabla_\perp \cdot f = (\boldsymbol{x}^n \cdot \nabla)(\boldsymbol{x}^n \cdot f) + \nabla_\perp \cdot f.$$

$$\square$$

Define the conditional noise covariance $\boldsymbol{\Sigma}(\boldsymbol{x})$ as

$$\boldsymbol{\Sigma}(\boldsymbol{x}) := \mathbf{G}(\boldsymbol{x})\mathbf{G}(\boldsymbol{x})^{\mathsf{T}} = \mathbb{E}[(d\vec{\boldsymbol{x}}_s)(d\vec{\boldsymbol{x}}_s)^{\mathsf{T}} | \vec{\boldsymbol{x}}_s = \boldsymbol{x}]. \tag{D.21}$$

We can now state and proof Theorem 3.1.1.

**Theorem D.2.1.** *Let the assumptions A1 and A2 hold. Then, the Fokker-Planck equation of equation 3.1 reads*

$$\begin{cases} \frac{\partial}{\partial s} p_s(\boldsymbol{x}) &= \nabla_\perp \cdot \left( \frac{1}{2} \boldsymbol{\Sigma}(\boldsymbol{x}) \nabla_\perp p_s(\boldsymbol{x}) \right), \qquad \boldsymbol{x} \in \mathbb{R}^d, \\ p_0 &= p_{\vec{\boldsymbol{x}}_0}. \end{cases} \tag{D.22}$$

*Moreover, any stationary density $p_\infty$ of equation D.22 is rotation-invariant on $\mathbb{R}^d$.*

*Proof.* From the Itô SDE equation D.9, the Fokker-Planck equation describing the evolution of $(p_s)_{s \geq 0}$ is given as

$$\frac{\partial}{\partial s} p_s = \nabla \cdot \left( -\frac{1}{2} \nabla \cdot (\boldsymbol{\Sigma}(\boldsymbol{x})) \, p_s(\boldsymbol{x}) + \frac{1}{2} \nabla \cdot (\boldsymbol{\Sigma}(\boldsymbol{x}) p_s(\boldsymbol{x})) \right), \tag{D.23}$$

$$= \nabla \cdot \left( -\frac{1}{2} \nabla \cdot (\boldsymbol{\Sigma}(\boldsymbol{x})) \, p_s(\boldsymbol{x}) + \frac{1}{2} \nabla \cdot (\boldsymbol{\Sigma}(\boldsymbol{x})) \, p_s(\boldsymbol{x}) + \frac{1}{2} \boldsymbol{\Sigma}(\boldsymbol{x}) \nabla p_s(\boldsymbol{x}) \right), \tag{D.24}$$

$$= \nabla \cdot \left( \frac{1}{2} \boldsymbol{\Sigma}(\boldsymbol{x}) \nabla p_s(\boldsymbol{x}) \right). \tag{D.25}$$

The skew-symmetry condition in assumption A1 implies for any $\boldsymbol{x} \in \mathbb{R}^d$ that

$$((\boldsymbol{x}^n)^\mathsf{T} \boldsymbol{\Sigma}(\boldsymbol{x}))^\mathsf{T} = \boldsymbol{\Sigma}(\boldsymbol{x})\boldsymbol{x}^n = \sum_{k=1}^d G_k \boldsymbol{x} \|\boldsymbol{x}\| \underbrace{(\boldsymbol{x}^n)^\mathsf{T} G_k^\mathsf{T} \boldsymbol{x}^n}_{=0} = 0. \tag{D.26}$$

Combining this with the result of equation D.16, it holds that

$$\nabla \cdot (\boldsymbol{\Sigma}(\boldsymbol{x})\nabla p_s(\boldsymbol{x})) \quad = \quad (\boldsymbol{x}^n \cdot \nabla)(\boldsymbol{x}^n \cdot \boldsymbol{\Sigma}(\boldsymbol{x})\nabla p_s(\boldsymbol{x})) + \nabla_\perp \cdot (\boldsymbol{\Sigma}(\boldsymbol{x})\nabla p_s(\boldsymbol{x})). \tag{D.27}$$

The decomposition $\nabla = \boldsymbol{x}^n (\boldsymbol{x}^n)^\mathsf{T} \nabla + (I - \boldsymbol{x}^n (\boldsymbol{x}^n)^\mathsf{T})\nabla = \boldsymbol{x}^n (\boldsymbol{x}^n)^\mathsf{T} \nabla + \nabla_\perp$ and equation D.26 yields

$$\begin{align} \nabla_\perp \cdot (\boldsymbol{\Sigma}(\boldsymbol{x})\nabla p_s(\boldsymbol{x})) \quad &= \quad \nabla_\perp \cdot (\boldsymbol{\Sigma}(\boldsymbol{x})\nabla_\perp p_s(\boldsymbol{x})) + \nabla_\perp \cdot (\boldsymbol{\Sigma}(\boldsymbol{x})\boldsymbol{x}^n (\boldsymbol{x}^n)^\mathsf{T} \nabla p_s(\boldsymbol{x})), \tag{D.28} \\ &= \quad \nabla_\perp \cdot (\boldsymbol{\Sigma}(\boldsymbol{x})\nabla_\perp p_s(\boldsymbol{x})). \tag{D.29} \end{align}$$

Hence, by linearity

$$\frac{\partial}{\partial s} p_s = \nabla_\perp \cdot (\tfrac{1}{2} \boldsymbol{\Sigma}(\boldsymbol{x})\nabla_\perp p_s(\boldsymbol{x})). \tag{D.30}$$

We shall now explore the set of possible invariant densities $\rho_\infty$ of the Fokker-Planck equation D.22. We will show that $\rho_\infty$ is stationary if and only if it is rotation-invariant.

Let $p_\infty$ be rotation-invariant, i.e. $\nabla_\perp p_\infty = 0$ then it is a stationary solution of the Fokker-Planck equation. The set of rotation-invariant measures is not empty, e.g. containing the isotropic normal distributions $\mathcal{N}(0, \boldsymbol{I}_d)$.

Conversely, let $p_\infty$ be a stationary solution of the Fokker-Planck equation, in particular

$$\nabla_\perp \cdot \left(\tfrac{1}{2} \boldsymbol{\Sigma}(\boldsymbol{x})\nabla_\perp p_\infty(\boldsymbol{x})\right) = 0. \tag{D.31}$$

Integrating over the test function $\phi = p_\infty$ gives a necessary condition for $p_\infty$ to be an invariant measure:

$$\begin{align} 0 \quad &= \quad -\int_{\mathbb{R}^d} p_\infty(\boldsymbol{x})\nabla_\perp \cdot (\boldsymbol{\Sigma}(\boldsymbol{x})\nabla_\perp p_\infty(\boldsymbol{x}))\, d\boldsymbol{x}, \tag{D.32} \\ &= \quad \int_{\mathbb{R}^d} \nabla_\perp p_\infty(\boldsymbol{x})^\mathsf{T} \boldsymbol{\Sigma}(\boldsymbol{x})\nabla_\perp p_\infty(\boldsymbol{x})\, d\boldsymbol{x}, \tag{D.33} \\ &= \quad \int_{\mathbb{R}^d} \underbrace{\|\mathbf{G}^\mathsf{T}(\boldsymbol{x})\nabla_\perp p_\infty(\boldsymbol{x})\|^2}_{\geqslant 0}\, d\boldsymbol{x}. \tag{D.34} \end{align}$$

Hence, for a.e. $\boldsymbol{x} \in \mathbb{R}^d$ it holds that $\nabla_\perp p_\infty(\boldsymbol{x}) \in \ker(\mathbf{G}^\mathsf{T}(\boldsymbol{x}))$. By assumption A2, this kernel has dimension 1. Moreover, $\mathbf{G}^\mathsf{T}(\boldsymbol{x})\boldsymbol{x} = 0$ and by definition of $\nabla_\perp$ we have that $\nabla_\perp p_\infty(\boldsymbol{x}) \perp \boldsymbol{x}$. That means

$$\nabla_\perp p_\infty(\boldsymbol{x}) \in \ker(\mathbf{G}^\mathsf{T}(\boldsymbol{x})) \cap \boldsymbol{x}^\perp = \operatorname{span}\{\boldsymbol{x}\} \cap \boldsymbol{x}^\perp = \{0\}. \tag{D.35}$$

We conclude that $\nabla_\perp p_\infty(\boldsymbol{x}) = 0$ almost everywhere on $\mathbb{R}^d$, i.e. the measure is rotation-invariant. $\qquad \square$

## D.3 Distribution of the Norms

In this section, we see more precisely that the norm of the MSGM SDE solution remains constant along the noising process whereas in SGM the norm dynamics is random with a mean going to $\sqrt{d}$ asymptotically.

### D.3.1 Norm dynamics in SGM

The dynamics of the SGM diffusion norm is stochastic. The following proposition states that the norm of the SGM latent concentrates around its mean, $\sqrt{d}$, for large dimension $d$.

**Proposition D.3.1.** *If $\overrightarrow{\boldsymbol{x}}_s$ is an Ornstein Uhlenbeck process then*

$$\mathbb{E}\left[\|\overrightarrow{\boldsymbol{x}}_s\|^2 \big| \overrightarrow{\boldsymbol{x}}_0\right] = e^{-2s}\|\overrightarrow{\boldsymbol{x}}_0\|^2 + (1 - e^{-2s})d \xrightarrow[s \to \infty]{} d. \tag{D.36}$$

*and*

$$\|\overrightarrow{\boldsymbol{x}}_s\|^2 \quad = \quad \mathbb{E}\left[\|\overrightarrow{\boldsymbol{x}}_s\|^2 \big| \overrightarrow{\boldsymbol{x}}_0\right] + \sqrt{d}\, I_s + e^{-s} K_s = d\left(1 + \frac{1}{\sqrt{d}} \underset{s \to \infty}{O}(1)\right), \tag{D.37}$$

*with both $\mathbb{E}K_s^2$ and $\mathbb{E}I_s^2$ bounded for large time $s$ and $\mathbb{E}I_s^2$ independent of the dimension $d$.*

*Proof.* To get the dynamics of the squared norm mean in SGM, we can take the expectation of the following Itô equation

$$d\|\vec{\boldsymbol{x}}_s\|^2 = 2\vec{\boldsymbol{x}}_s \cdot d\vec{\boldsymbol{x}}_s + d < \vec{\boldsymbol{x}}^\intercal, \vec{\boldsymbol{x}} >_s = 2(\|\vec{\boldsymbol{x}}_s\|^2 - d)ds + 2\sqrt{2}\vec{\boldsymbol{x}}_s \cdot d\vec{\boldsymbol{B}}_s, \quad \forall s \geq 0. \quad \text{(D.38)}$$

Thus,

$$\mathbb{E}\left[\|\vec{\boldsymbol{x}}_s\|^2\big|\vec{\boldsymbol{x}}_0\right] = e^{-2s}\|\vec{\boldsymbol{x}}_0\|^2 + (1-e^{-2s})d \xrightarrow[s\to\infty]{} d. \quad \text{(D.39)}$$

To obtain the full nom dynamics from equation D.38, we note that $t \to e^{-2s}$ has finite variations. Accordingly,

$$d(e^{-2s}(\|\vec{\boldsymbol{x}}_s\|^2 - d)) = 2\sqrt{2}e^{-2s}\vec{\boldsymbol{x}}_s \cdot d\vec{\boldsymbol{B}}_s, \quad \text{(D.40)}$$

and a temporal integration and the analytic expression of the Ornstein Uhlenbeck process yields:

$$\|\vec{\boldsymbol{x}}_s\|^2 = \mathbb{E}\left[\|\vec{\boldsymbol{x}}_s\|^2\big|\vec{\boldsymbol{x}}_0\right] + 2\sqrt{2}\int_0^s e^{-2(s-s')}\vec{\boldsymbol{x}}_{s'} \cdot d\vec{\boldsymbol{B}}_{s'}, \quad \text{(D.41)}$$

$$= \mathbb{E}\left[\|\vec{\boldsymbol{x}}_s\|^2\big|\vec{\boldsymbol{x}}_0\right] + 2\sqrt{2}e^{-s}\vec{\boldsymbol{x}}_0 \cdot \int_0^s e^{-(s-s')}d\vec{\boldsymbol{B}}_{s'}$$

$$+ 2\sqrt{2}\int_0^s\int_0^{s'} e^{-(2s-s'-s'')}d\vec{\boldsymbol{B}}_{s'} \cdot d\vec{\boldsymbol{B}}_{s''}, \quad \text{(D.42)}$$

$$= \mathbb{E}\left[\|\vec{\boldsymbol{x}}_s\|^2\big|\vec{\boldsymbol{x}}_0\right] + \sqrt{d}\, I_s + e^{-s}K_s, \quad \text{(D.43)}$$

with the martingales

$$I_s = \sqrt{\frac{8}{d}}\int_0^s\int_0^{s'} e^{-(2s-s'-s'')}d\vec{\boldsymbol{B}}_{s'} \cdot d\vec{\boldsymbol{B}}_{s''}, \quad \text{(D.44)}$$

$$K_s = \sqrt{8}\,\vec{\boldsymbol{x}}_0 \cdot \int_0^s e^{-(s-s')}d\vec{\boldsymbol{B}}_{s'}. \quad \text{(D.45)}$$

$K_s$ corresponds to the martingale part of the Ornstein Uhlenbeck solution projected on $\vec{\boldsymbol{x}}_0$. It is well known that $\mathbb{E}K_s^2$ is bounded for large time $s$. $\mathbb{E}I_s^2$ may be less known and we shall evaluate it below:

$$\mathbb{E}I_s^2 = \frac{8}{d}e^{-4s}\mathbb{E}\left(\sum_{p=1}^d\int_0^s\int_0^{s'} e^{s'+s''}d(\vec{B}_{s'})_p \cdot d(\vec{B}_{s''})_p\right)^2, \quad \text{(D.46)}$$

$$= \frac{8}{d}e^{-4s}\mathbb{E}\sum_{p_1,p_2=1}^d\int_0^s\int_0^{s'_1}\int_0^s\int_0^{s'_2} e^{s'_1+s''_1 s'_2+s''_2}d(\vec{B}_{s'_1})_{p_1} \cdot d(\vec{B}_{s''_1})_{p_1} d(\vec{B}_{s'_2})_{p_2} \cdot d(\vec{B}_{s''_2})_{p_2},$$

$$\text{(D.47)}$$

$$= \frac{8}{d}e^{-4s}\mathbb{E}\sum_{p_1,p_2=1}^d\int_0^s\int_0^{s'_1}\int_0^s\int_0^{s'_2} e^{s'_1+s''_1+s'_2+s''_2}\delta_{p_1,p_2}\delta(s'_1-s'_2)\delta_{p_1,p_2}\delta(s''_1-s''_2)ds'_1 ds''_1 ds'_2 ds''_2,$$

$$\text{(D.48)}$$

$$= 8e^{-4s}\int_0^s\int_0^{s'} e^{2(s'+s'')}ds'ds'', \quad \text{(D.49)}$$

$$= 4e^{-4s}\int_0^s e^{2s'}(e^{2s'}-1)ds', \quad \text{(D.50)}$$

$$= e^{-4s}((e^{4s}-1)+2(e^{2s}-1)), \quad \text{(D.51)}$$

$$\xrightarrow[s\to\infty]{} 1. \quad \text{(D.52)}$$

$\square$

### D.3.2 NORM DYNAMICS IN MSGM

For MSGM, the norm follows totally different dynamics. We recall that the skew-symmetry of $\circ d\boldsymbol{Z}_s$ implies that $d\vec{\boldsymbol{x}}_s = \circ d\boldsymbol{Z}_s \vec{\boldsymbol{x}}_s$ is orthogonal to $\vec{\boldsymbol{x}}_s$ and hence:

$$d\|\vec{\boldsymbol{x}}_s\|^2 = 2\vec{\boldsymbol{x}}_s \cdot \circ d\vec{\boldsymbol{x}}_s = 0, \quad \forall s \geq 0. \tag{D.53}$$

Consequently, $\vec{\boldsymbol{x}}_s$ moves randomly on $\|\vec{\boldsymbol{x}}_0\|\mathbb{S}^{d-1}$, the $d$-sphere of radius $\|\vec{\boldsymbol{x}}_0\|$, and the increments $d\vec{\boldsymbol{x}}_s$ are tangent to the $d$-sphere. In particular, we obtain the following result.

**Proposition D.3.2.** *Let the skew-symmetry assumption A1 hold. Let $\vec{\boldsymbol{x}}_0$ be a random variable. Then, for all $s \geq 0$ the distribution of $\|\vec{\boldsymbol{x}}_s\|$ equals the distribution of $\|\vec{\boldsymbol{x}}_0\|$.*

Therefore, the distribution of the norms of the latent variable is exactly the distribution of the norms of the points of the dataset. Moreover, $\vec{\boldsymbol{x}}_s \equiv 0$ if and only if $\vec{\boldsymbol{x}}_0 = 0$. As a consequence, we can exclude all points exactly equal to zero from a dataset, treat them aside, and hence consider, without loss of generality, that $\vec{\boldsymbol{x}}_T \neq 0$ almost surely.

## D.4 FOKKER-PLANCK EQUATION OF THE DIRECTION

This subsection is devoted to the analysis of the Fokker-Planck equation on the unit sphere $\mathbb{S}^{d-1}$, i.e. the distribution of $\vec{\boldsymbol{x}}_s^n$, in particular as $s \to \infty$.

### D.4.1 MAIN RESULTS ON THE DISTRIBUTION OF DIRECTIONS

We saw in Section D.3 that $\vec{\boldsymbol{x}}_s$ moves randomly on the $d$-sphere of radius $\|\vec{\boldsymbol{x}}_0\|$ and that the increments, $d\vec{\boldsymbol{x}}_s = \mathbf{G}(\vec{\boldsymbol{x}}_s) \circ d\vec{\boldsymbol{B}}_s$, are tangent to the $d$-sphere. If the rank condition, assumptionA2 is verified, then the support of the noise distribution $\mathbf{G}(\vec{\boldsymbol{x}}_s) \circ d\vec{\boldsymbol{B}}_s$ coincides with the $d-1$-dimensional tangent space, i.e. it will likely explore all local directions around $\vec{\boldsymbol{x}}_s$. With time, the support of the solution distribution will gradually cover the whole $d$-sphere, i.e. every direction $\vec{\boldsymbol{x}}_s^n$ will become equiprobable. Lemma D.4.1 illustrates and precises this claim.

**Lemma D.4.1.** *Let assumptions A1 and A2 hold. Let a initial density $p_0^n \in \mathcal{C}^2(\mathbb{S}^{d-1})$ and $\boldsymbol{\Sigma}(\boldsymbol{x}^n) := \mathbf{G}(\boldsymbol{x}^n)\mathbf{G}(\boldsymbol{x}^n)^\intercal$. Then, the Fokker-Planck equation*

$$\frac{\partial}{\partial s}p_s^n(\boldsymbol{x}^n) = \nabla_\perp \cdot \left(\tfrac{1}{2}\boldsymbol{\Sigma}(\boldsymbol{x}^n)\nabla_\perp p_s^n(\boldsymbol{x}^n)\right), \quad \boldsymbol{x}^n \in \mathbb{S}^{d-1}, \tag{D.54}$$

*has a unique density solution $p_s^n \in \mathcal{C}^2(\mathbb{S}^{d-1})$ for all $s > 0$. Moreover, there is a unique invariant measure $p_\infty^n$ of that Fokker-Planck equation, i.e. the uniform distribution on the $d$-sphere $\mathcal{U}(\mathbb{S}^{d-1})$, with density*

$$p_\infty^n(\boldsymbol{x}^n) := \frac{1}{|\mathbb{S}^{d-1}|}, \quad \forall \boldsymbol{x}^n \in \mathbb{S}^{d-1}, \tag{D.55}$$

*with $|\mathbb{S}^{d-1}| = 2\pi^{d/2}/\Gamma\left(\frac{d}{2}\right)$ the volume of the $d$-sphere $\mathbb{S}^{d-1}$ and $\Gamma$ the gamma function.*

Lemma D.4.1 is a consequence of Theorem 3.1.1 as shown in Section D.4.2. Note that in this case $\nabla_\perp = \nabla_{\mathbb{S}^{d-1}}$ is the Riemannian gradient on $\mathbb{S}^{d-1}$, see Section H.1.

Given the unique invariant measure of Fokker-Planck equation formulated on $\mathbb{S}^{d-1}$, we can also show that we have exponential convergence of the initial distribution $p_0^n$ to $p_\infty^n$, the uniform distribution on the unit sphere $\mathbb{S}^{d-1}$.

**Theorem D.4.1.** *Let assumptions A1 and A2 hold. Then, there exists $\alpha = \alpha(\mathbf{G}, d) > 0$ with*

$$\|p_s^n - p_\infty^n\|_{L^2(\mathbb{S}^{d-1})}^2 \leq \exp(-\alpha s)\|p_0^n - p_\infty^n\|_{L^2(\mathbb{S}^{d-1})}^2. \tag{D.56}$$

*The convergence rate $\alpha$ is given as*

$$\alpha(\mathbf{G}, d) = (d-1)\min_{(\boldsymbol{x},\boldsymbol{y})\in S}\|\mathbf{G}^\intercal(\boldsymbol{x})\boldsymbol{y}\|^2, \quad S = \{(\boldsymbol{x},\boldsymbol{y}) \in \mathbb{S}^{d-1} \times \mathbb{S}^{d-1} | \boldsymbol{x} \perp \boldsymbol{y}\}. \tag{D.57}$$

Consequently, since $\mathbb{S}^{d-1}$ is compact this implies convergence in total variation of $p_s^n$ to $p_\infty^n$ and convergence in distribution of $\vec{\boldsymbol{x}}_s^n$ with $\vec{\boldsymbol{x}}_\infty^n \sim \mathcal{U}(\mathbb{S}^{d-1})$. The full proof is detailed in Section D.4.3.

### D.4.2 PROOF OF LEMMA D.4.1

*Proof. Existence and Uniqueness*:
Consider $L(p_s^n) = \nabla_\perp \cdot \left(\frac{1}{2}\boldsymbol{\Sigma}(\boldsymbol{x})\nabla_\perp p_s^n(\boldsymbol{x})\right) - \frac{\partial}{\partial s}p_s^n(\boldsymbol{x})$. $L$ is a parabolic type operator according to Friedman (1964) since $\boldsymbol{x} \mapsto \boldsymbol{\Sigma}(\boldsymbol{x})$ is positive definite by assumption A2 on $\mathbb{S}^{d-1}$. Indeed, for any $\boldsymbol{y} \in T_x\mathbb{S}^{d-1}$ the tangential (linear) space of $\mathbb{S}^{d-1}$ at $\boldsymbol{x}$,

$$\boldsymbol{y}^\intercal \boldsymbol{\Sigma}(\boldsymbol{x})\boldsymbol{y} = \|\mathbf{G}^\intercal(\boldsymbol{x})\boldsymbol{y}\|^2 \geq 0. \tag{D.58}$$

with equality if and only if $\mathbf{G}^\intercal(\boldsymbol{x})\boldsymbol{y} = 0$. Then, the rank condition A2 implies $\boldsymbol{y} = 0$ as previously in equation D.35. Consequently, the associated spatial operator $L_0$ defined by

$$L_0 p_s^n = \nabla_\perp \cdot \left(\frac{1}{2}\boldsymbol{\Sigma}(\boldsymbol{x})\nabla_\perp p_s^n(\boldsymbol{x})\right) \tag{D.59}$$

is an elliptic operator on $\mathbb{S}^{d-1}$, a compact manifold without boundary such that the semi-group $e^{sL_0}$ is strongly continuous on $\mathcal{C}^2(\mathbb{S}^{d-1})$, $s \geq 0$. As $p_0^n \in \mathcal{C}^2(\mathbb{S}^{d-1})$, according to chapter 1, proposition 1.1 in Taylor (2011) , there exists a unique solution $p_s^n \in \mathcal{C}^2(\mathbb{S}^{d-1})$, for $s \in [0, T[$ of equation D.54. As the semigroup is well-defined for all $s > 0$, this extends the uniqueness of the solution to all $s > 0$.

*Invariant measure*: Repeating the lines in the proof of Theorem 3.1.1 given in Section D.2 it follows that $p_\infty^n$ is rotation-invariant. The only rotation-invariant distribution on the $d$-sphere is the uniform distribution. $\qquad\square$

### D.4.3 PROOF OF THEOREM D.4.1 : LIMIT BEHAVIOR OF FOKKER-PLANCK EQUATION OF THE DIRECTION

**Theorem D.4.2.** *Let assumptions A1 and A2 hold. Then, there exists $\alpha = \alpha(\mathbf{G}, d) > 0$ with*

$$\|p_s^n - p_\infty^n\|_{L^2(\mathbb{S}^{d-1})}^2 \leq \exp(-\alpha s)\|p_0^n - p_\infty^n\|_{L^2(\mathbb{S}^{d-1})}^2. \tag{D.60}$$

*The convergence rate $\alpha$ is given as*

$$\alpha(\mathbf{G}, d) = (d-1)\min_{(\boldsymbol{x},\boldsymbol{y}) \in S}\|\mathbf{G}^\intercal(\boldsymbol{x})\boldsymbol{y}\|^2, \quad S = \{(\boldsymbol{x},\boldsymbol{y}) \in \mathbb{S}^{d-1} \times \mathbb{S}^{d-1}|\boldsymbol{x}\perp\boldsymbol{y}\}. \tag{D.61}$$

*Proof.* Let $p_s^n$ denoting the density of $\overrightarrow{\boldsymbol{x}}_s^n$. Define $e_s^n = p_s^n - p_\infty^n$ with $p_\infty^n \equiv |\mathbb{S}^{d-1}|^{-1}$ being the uniform distribution on $\mathbb{S}^{d-1}$. Then, by linearity of the Fokker-Planck equation, $e_t^n$ satisfies

$$\partial_t e_t^n = \nabla_\perp \cdot \left(\frac{1}{2}\boldsymbol{\Sigma}(\boldsymbol{x})\nabla_\perp e_t^n(\boldsymbol{x})\right). \tag{D.62}$$

Since $p_s^n$ and $p_\infty^n$ are densities on $\mathbb{S}^{d-1}$, we have $\int_{\mathbb{S}^{d-1}} e_s^n d\boldsymbol{x} = 0$ for all $s \geq 0$. Consequently, since $\mathbb{S}^{d-1}$ is a compact manifold without boundary, Poincaré inequality holds, i.e.

$$\|e_t^n\|_{L^2(\mathbb{S}^{d-1})}^2 \leq \frac{1}{d-1}\|\nabla_{\mathbb{S}^{d-1}}e_t^n\|_{L^2(\mathbb{S}^{d-1})}^2, \tag{D.63}$$

with

$$\nabla_{\mathbb{S}^{d-1}}e_t^n(\boldsymbol{y})\big|_{\boldsymbol{y}=\boldsymbol{x}} = \text{Proj}_{\mathcal{T}_{\boldsymbol{x},\mathbb{S}^{d-1}}} \nabla e_t^n(\boldsymbol{y})\big|_{\boldsymbol{y}=\boldsymbol{x}} = \nabla_\perp e_t^n(\boldsymbol{x}). \tag{D.64}$$

Consequently, integration by part on $\mathbb{S}^{d-1}$ leads to

$$\frac{1}{2}\frac{d}{dt}\|e_t^n\|_{L^2(\mathbb{S}^{d-1})}^2 = \int_{\mathbb{S}^{d-1}} e_t^n(\boldsymbol{x})\nabla_\perp \cdot \left(\frac{1}{2}\boldsymbol{\Sigma}(\boldsymbol{x})\nabla_\perp e_t^n(\boldsymbol{x})\right) d\boldsymbol{x}, \tag{D.65}$$

$$= -\int_{\mathbb{S}^{d-1}} \nabla_\perp e_t^n(\boldsymbol{x})^\intercal \boldsymbol{\Sigma}(\boldsymbol{x})\nabla_\perp e_t^n(\boldsymbol{x})d\boldsymbol{x}. \tag{D.66}$$

We will now bound $\boldsymbol{y}^\intercal\boldsymbol{\Sigma}(\boldsymbol{x})\boldsymbol{y}^\intercal$ from below for any $\boldsymbol{y} \in \boldsymbol{x}^\perp$ and $\boldsymbol{x} \in \mathbb{S}^{d-1}$, in particular with a bound independent of $\boldsymbol{x}$. Since $\boldsymbol{\Sigma}(\boldsymbol{x})$ is symmetric, it is real diagonalizable with eigen-basis denoted as $\boldsymbol{v}_1(\boldsymbol{x}), \ldots, \boldsymbol{v}_d(\boldsymbol{x}) \in \mathbb{R}^d$ and eigenvalues $\lambda_1(\boldsymbol{x}), \ldots, \lambda_d(\boldsymbol{x})$. By construction $\boldsymbol{\Sigma}(\boldsymbol{x})\boldsymbol{x} = 0$, hence we can set $\boldsymbol{v}_d(\boldsymbol{x}) := \boldsymbol{x}/\|\boldsymbol{x}\|$ and $\lambda_d(\boldsymbol{x}) \equiv 0$. Moreover, by the rank condition A2, $\lambda_i \neq 0$ for $i \neq d$. By orthonormality of the eigenvectors $\boldsymbol{v}_1(\boldsymbol{x}), \ldots, \boldsymbol{v}_{d-1}(\boldsymbol{x})$ then span the tangent plane $\boldsymbol{x}^\perp$ at $\boldsymbol{x}$ on $\mathbb{S}^{d-1}$. For any $i = 1, \ldots, d-1$, we have that

$$\lambda_i(\boldsymbol{x}) = \boldsymbol{v}_i(\boldsymbol{x})^\intercal \boldsymbol{\Sigma}(\boldsymbol{x})\boldsymbol{v}_i(\boldsymbol{x}) = \|\mathbf{G}^\intercal(\boldsymbol{x})\boldsymbol{v}_j(\boldsymbol{x})\|^2 \geq \min_{\boldsymbol{y}\in\boldsymbol{x}^\perp} \frac{\|\mathbf{G}^\intercal(\boldsymbol{x})\boldsymbol{y}\|^2}{\|\boldsymbol{y}\|^2}. \tag{D.67}$$

The polynomial $(\boldsymbol{x}, \boldsymbol{y}) \mapsto P(\boldsymbol{x}, \boldsymbol{y}) = \|\mathbf{G}^\mathsf{T}(\boldsymbol{x})\boldsymbol{y}\|^2$ on the compact $S = \{(\boldsymbol{x}, \boldsymbol{y}) \in \mathbb{S}^{d-1} \times \mathbb{S}^{d-1} | \boldsymbol{x} \perp \boldsymbol{y}\}$ attains its minimum $P^*$, which from the rank condition satisfies $P(\boldsymbol{x}, \boldsymbol{y}) \geq P^* > 0$ for all $(\boldsymbol{x}, \boldsymbol{y}) \in S$. As a consequence $\lambda_i(\boldsymbol{x}) \geq P^*$ for $i = 1, \ldots, d-1$ and $\boldsymbol{x} \in \mathbb{S}^{d-1}$, which implies for all $\boldsymbol{y} \in \boldsymbol{x}^\perp$ that

$$\boldsymbol{y}^\mathsf{T} \boldsymbol{\Sigma}(\boldsymbol{x})\boldsymbol{y} = \|\mathbf{G}^\mathsf{T}(\boldsymbol{x})\boldsymbol{y}\|^2 \geq P^*\|\boldsymbol{y}\|^2. \tag{D.68}$$

Therefore, combining equation D.63, equation D.64 and equation D.66, we obtain

$$\frac{1}{2}\frac{d}{dt}\|e_t^n\|_{L^2(\mathbb{S}^{d-1})}^2 \leq -P^*\|\nabla_\perp e_s^n\|_{L^2(\mathbb{S}^{d-1})}^2 \leq -P^*(d-1)\|e_s^n\|_{L^2(\mathbb{S}^{d-1})}^2. \tag{D.69}$$

Then, by Gronwall for $\alpha = P^*(d-1) > 0$, we conclude that

$$\|p_s^n - p_\infty^n\|_{L^2(\mathbb{S}^{d-1})}^2 = \|e_s^n\|_{L^2(\mathbb{S}^{d-1})}^2 \leq \|e_0^n\|_{L^2(\mathbb{S}^{d-1})}^2 \exp(-\alpha s). \tag{D.70}$$

$\square$

## D.5 PROOF OF THEOREM 3.3.1 : CONVERGENCE OF FOKKER-PLANCK EQUATION

This section is devoted to the analysis of the Fokker-Planck equation in the whole domain $\mathbb{R}^d$. Due to the fact that the norm of a point does not change in the SDE process as shown in equation 3.7 and the fact that $\boldsymbol{\Sigma}(\boldsymbol{x}) = 0$ for $\boldsymbol{x} = 0$, we exclude the origin in the analysis.

**Theorem D.5.1.** *Let $D = \mathbb{R}^d \backslash \{0\}$ for $d > 1$. Let assumptions A1 and A2 hold. Let $\vec{\boldsymbol{x}}_0 \sim p_0 \in \mathcal{C}^2(D)$ and let $p_{|.|}$ be the (radial) density of $\|\vec{\boldsymbol{x}}_0\|$. Then, the Fokker-Planck equation*

$$\frac{\partial}{\partial s}p_s(\boldsymbol{x}) = \nabla_\perp \cdot \left(\tfrac{1}{2}\boldsymbol{\Sigma}(\boldsymbol{x})\nabla_\perp p_s(\boldsymbol{x})\right), \quad \boldsymbol{x} \in D, \tag{D.71}$$

*has a unique solution $p_s \in \mathcal{C}^2(D) \cap L^2(D)$ for all $s > 0$. Moreover, the Fokker-Planck equation has the stationary distribution*

$$p_\infty(\boldsymbol{x}) = \frac{p_{|.|}\left(\|\boldsymbol{x}\|\right)}{\|\boldsymbol{x}\|^{d-1}}\frac{1}{|\mathbb{S}^{d-1}|}. \tag{D.72}$$

*In particular, $\|\vec{\boldsymbol{x}}_s\|$ and $\vec{\boldsymbol{x}}_s^n$ are asymptotically independent for $s \to +\infty$. Moreover, there exists $\alpha = \alpha(\mathbf{G}, d) > 0$ such that*

$$\|p_s - p_\infty\|_{L^2(\mathbb{R}^d)}^2 \leq \exp(-\alpha s)\|p_0 - p_\infty\|_{L^2(\mathbb{R}^d)}^2.$$

*The convergence rate $\alpha$ is given as*

$$\alpha(\mathbf{G}, d) = (d-1)\min_{(\boldsymbol{x}, \boldsymbol{y}) \in S}\|\mathbf{G}^\mathsf{T}(\boldsymbol{x})\boldsymbol{y}\|^2, \quad S = \{(\boldsymbol{x}, \boldsymbol{y}) \in \mathbb{S}^{d-1} \times \mathbb{S}^{d-1} | \boldsymbol{x} \perp \boldsymbol{y}\}. \tag{D.73}$$

*Proof.* In the following, we will frequently use the change of variables in polar coordinates in $\mathbb{R}^n$. More precisely, writing $\boldsymbol{x} = r\boldsymbol{x}^n$ with $r > 0$ and $\boldsymbol{x}^n \in \mathbb{S}^{n-1}$, the Lebesgue measure decomposes as

$$d\boldsymbol{x} = r^{n-1}\,dr\,d\sigma(\boldsymbol{x}^n), \tag{D.74}$$

where $d\sigma(\boldsymbol{x}^n) = d\boldsymbol{x}^n$ denotes the rotation-invariant surface measure on $\mathbb{S}^{n-1}$. This change of variables is justified by the change-of-variables theorem; see Folland (1999, Theorem 2.49) and the discussion on polar coordinates.

We will proof existence, uniqueness, regularity, invariant property and convergence separately.

*Existence*: Let $p_0(\boldsymbol{x}^n\|\|\vec{\boldsymbol{x}}_0\| = r)$ be the start value of the FP equation D.54 on $\mathbb{S}^{d-1}$ of Lemma D.4.1. This gives rise to a smooth unique density solution $p_s^n(\boldsymbol{x}^n\|\|\vec{\boldsymbol{x}}_0\| = r)$ for $s > 0$ and any $r > 0$. Moreover, $p_s^n(\boldsymbol{x}^n\|\|\vec{\boldsymbol{x}}_0\| = r) = p_s^n(\boldsymbol{x}^n\|\|\vec{\boldsymbol{x}}_s\| = r)$ since $d\|\vec{\boldsymbol{x}}_s\| = 0$. Now define

$$\rho_s(\boldsymbol{x}) = p_s^n(\boldsymbol{x}^n\|\|\vec{\boldsymbol{x}}_s\| = \|\boldsymbol{x}\|)p_{|.|}(\|\boldsymbol{x}\|)\|\boldsymbol{x}\|^{1-d},$$

where we denote $\boldsymbol{x}^n = \frac{\boldsymbol{x}}{\|\boldsymbol{x}\|}$. $\rho_s$ is a density since

$$
\begin{aligned}
\int_{\mathbb{R}^d} \rho_s(\boldsymbol{x})\, \mathrm{d}\boldsymbol{x} &= \int_{\mathbb{R}^d} p_s^n(\boldsymbol{x}^n | |\vec{\boldsymbol{x}}_s| = \|\boldsymbol{x}\|) p_{|.|}(\|\boldsymbol{x}\|) \|\boldsymbol{x}\|^{1-d}\, \mathrm{d}\boldsymbol{x}, && \text{(D.75)} \\
&= \int_{\mathbb{R}_+} \int_{\mathbb{S}^{d-1}} p_s^n(\boldsymbol{x}^n | \|\vec{\boldsymbol{x}}_s\| = r) p_{|.|}(r) r^{1-d} r^{d-1}\, \mathrm{d}r\, \mathrm{d}\boldsymbol{x}^n, && \text{(D.76)} \\
&= \int_{\mathbb{R}_+} \int_{\mathbb{S}^{d-1}} p_s^n(\boldsymbol{x}^n | \|\vec{\boldsymbol{x}}_s\| = r) p_{|.|}(r)\, \mathrm{d}r\, \mathrm{d}\boldsymbol{x}^n, && \text{(D.77)} \\
&= \int_{\mathbb{R}_+} p_{|.|}(r) \left( \int_{\mathbb{S}^{d-1}} p_s^n(\boldsymbol{x}^n | \|\vec{\boldsymbol{x}}_s\| = r)\, \mathrm{d}\boldsymbol{x}^n \right)\, \mathrm{d}r, && \text{(D.78)} \\
&= 1. && \text{(D.79)}
\end{aligned}
$$

We have $\nabla_\perp = \frac{1}{\|\boldsymbol{x}\|}\nabla_{\mathbb{S}^{d-1}}$ and $\nabla_\perp$ does not act on radial functions. Besides, $\boldsymbol{\Sigma}(\boldsymbol{x}) := \mathbf{G}(\boldsymbol{x})\mathbf{G}(\boldsymbol{x})^\mathsf{T}$ with $\mathbf{G}$ linear so $\boldsymbol{\Sigma}(\boldsymbol{x}) = \boldsymbol{\Sigma}(\|\boldsymbol{x}\| \frac{\boldsymbol{x}}{\|\boldsymbol{x}\|}) = \|\boldsymbol{x}\|^2 \boldsymbol{\Sigma}(\boldsymbol{x}^n)$. Hence

$$
\begin{aligned}
\nabla_\perp \rho_s(\boldsymbol{x}) &= \nabla_\perp p_s^n(\boldsymbol{x}^n | \|\vec{\boldsymbol{x}}_s\| = \|\boldsymbol{x}\|) p_{|.|}(\|\boldsymbol{x}\|) \|\boldsymbol{x}\|^{1-d}, && \text{(D.80)} \\
&= p_{|.|}(\|\boldsymbol{x}\|) \|\boldsymbol{x}\|^{1-d} \nabla_\perp p_s^n(\boldsymbol{x}^n | \|\vec{\boldsymbol{x}}_s\| = \|\boldsymbol{x}\|), && \text{(D.81)} \\
&= p_{|.|}(\|\boldsymbol{x}\|) \|\boldsymbol{x}\|^{1-d} \left( \frac{1}{\|\boldsymbol{x}\|} \nabla_{\mathbb{S}^{d-1}} \right) p_s^n(\boldsymbol{x}^n | \|\vec{\boldsymbol{x}}_s\| = \|\boldsymbol{x}\|), && \text{(D.82)}
\end{aligned}
$$

and

$$
\begin{aligned}
\boldsymbol{\Sigma}(\boldsymbol{x})\nabla_\perp \rho_s(\boldsymbol{x}) &= \boldsymbol{\Sigma}(\boldsymbol{x}) p_{|.|}(\|\boldsymbol{x}\|) \|\boldsymbol{x}\|^{1-d} \frac{1}{\|\boldsymbol{x}\|} \nabla_{\mathbb{S}^{d-1}} p_s^n(\boldsymbol{x}^n | \|\vec{\boldsymbol{x}}_s\| = \|\boldsymbol{x}\|), && \text{(D.83)} \\
&= \|\boldsymbol{x}\|^2 \boldsymbol{\Sigma}(\boldsymbol{x}^n) p_{|.|}(\|\boldsymbol{x}\|) \|\boldsymbol{x}\|^{1-d} \frac{1}{\|\boldsymbol{x}\|} \nabla_{\mathbb{S}^{d-1}} p_s^n(\boldsymbol{x}^n | \|\vec{\boldsymbol{x}}_s\| = \|\boldsymbol{x}\|), && \text{(D.84)} \\
&= \|\boldsymbol{x}\|^2 p_{|.|}(\|\boldsymbol{x}\|) \|\boldsymbol{x}\|^{1-d} \frac{1}{\|\boldsymbol{x}\|} \boldsymbol{\Sigma}(\boldsymbol{x}^n) \nabla_{\mathbb{S}^{d-1}} p_s^n(\boldsymbol{x}^n | \|\vec{\boldsymbol{x}}_s\| = \|\boldsymbol{x}\|) && \text{(D.85)}
\end{aligned}
$$

$$
\begin{aligned}
\nabla_\perp \cdot (\boldsymbol{\Sigma}(\boldsymbol{x})\nabla_\perp \rho_s(\boldsymbol{x})) &= \frac{1}{\|\boldsymbol{x}\|} \nabla_{\mathbb{S}^{d-1}} \cdot \left( \|\boldsymbol{x}\| p_{|.|}(\|\boldsymbol{x}\|) \|\boldsymbol{x}\|^{1-d} \boldsymbol{\Sigma}(\boldsymbol{x}^n) \nabla_{\mathbb{S}^{d-1}} p_s^n(\boldsymbol{x}^n | \|\vec{\boldsymbol{x}}_s\| = \|\boldsymbol{x}\|) \right), \\
& && \text{(D.86)} \\
&= p_{|.|}(\|\boldsymbol{x}\|) \|\boldsymbol{x}\|^{1-d} \nabla_{\mathbb{S}^{d-1}} \cdot \left( \boldsymbol{\Sigma}(\boldsymbol{x}^n) \nabla_{\mathbb{S}^{d-1}} p_s^n(\boldsymbol{x}^n | \|\vec{\boldsymbol{x}}_s\| = \|\boldsymbol{x}\|) \right), && \text{(D.87)} \\
&= p_{|.|}(\|\boldsymbol{x}\|) \|\boldsymbol{x}\|^{1-d} 2 \frac{\partial}{\partial s} p_s^n(\boldsymbol{x}^n | \|\vec{\boldsymbol{x}}_s\| = \|\boldsymbol{x}\|), && \text{(D.88)} \\
&= 2 \frac{\partial}{\partial s} \left( p_{|.|}(\|\boldsymbol{x}\|) \|\boldsymbol{x}\|^{1-d} p_s^n(\boldsymbol{x}^n | \|\vec{\boldsymbol{x}}_s\| = \|\boldsymbol{x}\|) \right), && \text{(D.89)} \\
&= 2 \frac{\partial}{\partial s} \rho_s(\boldsymbol{x}), && \text{(D.90)}
\end{aligned}
$$

i.e. $\frac{\partial}{\partial s}\rho_s(\boldsymbol{x}) = \frac{1}{2}\nabla_\perp \cdot (\boldsymbol{\Sigma}(\boldsymbol{x})\nabla_\perp \rho_s(\boldsymbol{x}))$. Then, $\rho_s$ solves the Fokker-Planck equation on $\mathbb{R}^d$.

*Uniqueness*: Assume there exists another $\tilde{\rho}$ solving the FP on $\mathbb{R}^d$ verify

$$
\tilde{\rho}_s(\boldsymbol{x}) = \rho_{1,s}(\boldsymbol{x}^n | \|\boldsymbol{x}\|) \rho_{2,s}(\|\boldsymbol{x}\|).
$$

Since $d\|\vec{\boldsymbol{x}}_s\| = 0$, by marginalizing $\tilde{\rho}_s$ (integrating on $\mathbb{S}^{d-1}$), we have the uniqueness of the radial density $\rho_{2,s}(r) = \int_{\mathbb{S}^{d-1}} \tilde{\rho}_s(r\boldsymbol{x}^n) d\boldsymbol{x}^n = p_{\|\vec{\boldsymbol{x}}_s\|}(r) r^{1-d} = p_{|.|}(r) r^{1-d}$.

Since $d\|\vec{\boldsymbol{x}}_s\| = 0$, we have $\frac{\partial}{\partial s}\rho_{2,s}(\|\boldsymbol{x}\|) = 0$. Therefore,

$$
\frac{\partial}{\partial s}\tilde{\rho}_s(\boldsymbol{x}) = \frac{\partial}{\partial s}\left(\rho_{1,s}(\boldsymbol{x}^n\|\boldsymbol{x}\|)\rho_{2,s}(\|\boldsymbol{x}\|)\right), \tag{D.91}
$$

$$
= \rho_{2,s}(\|\boldsymbol{x}\|)\frac{\partial}{\partial s}\rho_{1,s}(\boldsymbol{x}^n\|\boldsymbol{x}\|) + \rho_{1,s}(\boldsymbol{x}^n\|\boldsymbol{x}\|)\frac{\partial}{\partial s}\rho_{2,s}(\|\boldsymbol{x}\|), \tag{D.92}
$$

$$
= \rho_{2,s}(\|\boldsymbol{x}\|)\frac{\partial}{\partial s}\rho_{1,s}(\boldsymbol{x}^n\|\boldsymbol{x}\|). \tag{D.93}
$$

In addition,

$$
2\frac{\partial}{\partial s}\tilde{\rho}_s(\boldsymbol{x}) = \nabla_\perp \cdot \left(\boldsymbol{\Sigma}(\boldsymbol{x})\nabla_\perp\tilde{\rho}_s(\boldsymbol{x})\right), \tag{D.94}
$$

$$
= \nabla_\perp \cdot \left(\boldsymbol{\Sigma}(\boldsymbol{x})\nabla_\perp\rho_{1,s}(\boldsymbol{x}^n\|\boldsymbol{x}\|)\rho_{2,s}(\|\boldsymbol{x}\|)\right), \tag{D.95}
$$

$$
= \rho_{2,s}(\|\boldsymbol{x}\|)\|\boldsymbol{x}\|^2\nabla_\perp \cdot \left(\boldsymbol{\Sigma}(\boldsymbol{x}^n)\nabla_\perp\rho_{1,s}(\boldsymbol{x}^n\|\boldsymbol{x}\|)\right), \tag{D.96}
$$

$$
= \rho_{2,s}(\|\boldsymbol{x}\|)\nabla_{\mathbb{S}^{d-1}} \cdot \left(\boldsymbol{\Sigma}(\boldsymbol{x}^n)\nabla_{\mathbb{S}^{d-1}}\rho_{1,s}(\boldsymbol{x}^n\|\boldsymbol{x}\|)\right), \tag{D.97}
$$

and finally,

$$
\rho_{2,s}(\|\boldsymbol{x}\|)\left(2\frac{\partial}{\partial s}\rho_{1,s}(\boldsymbol{x}^n\|\boldsymbol{x}\|) - \nabla_{\mathbb{S}^{d-1}} \cdot \left(\boldsymbol{\Sigma}(\boldsymbol{x}^n)\nabla_{\mathbb{S}^{d-1}}\rho_{1,s}(\boldsymbol{x}^n\|\boldsymbol{x}\|)\right)\right) = 0. \tag{D.98}
$$

Then, $\rho_{1,s}(\boldsymbol{x}^n\|\boldsymbol{x}\|)$ is a solution of the de Fokker-Planck equation on the sphere, for any $\boldsymbol{x}$ such that $\|\boldsymbol{x}\| \in A := \{r \in \mathbb{R}^+ | \rho_{2,s}(r) > 0\}$. If $\|\boldsymbol{x}\| \notin A$, then $\rho_{2,s}(\|\boldsymbol{x}\|) = 0$ and

$$
\rho_s(\boldsymbol{x}) = p_s^n(\boldsymbol{x}^n\|\vec{\boldsymbol{x}}_s\| = \|\boldsymbol{x}\|)p_{|\cdot|}(\|\boldsymbol{x}\|)\|\boldsymbol{x}\|^{1-d}.
$$

For $\|\boldsymbol{x}\| \in A$, then $\rho_{2,s}(\|\boldsymbol{x}\|) \neq 0$ and $\rho_{1,s}(.\|\boldsymbol{x}\|)$ is solution of the Fokker Planck equation D.54. According to lemma D.4.1, the Fokker Planck equation D.54 has a unique density solution $p_s^n(\boldsymbol{x}^n\|\vec{\boldsymbol{x}}_0\| = \|\boldsymbol{x}\|)$. Hence, for any $\boldsymbol{x}$ such that $p_{|\cdot|}(\|\boldsymbol{x}\|) > 0$, we have

$$
\rho_s(\boldsymbol{x}) = p_s^n(\boldsymbol{x}^n\|\vec{\boldsymbol{x}}_s\| = \|\boldsymbol{x}\|)p_{|\cdot|}(\|\boldsymbol{x}\|)\|\boldsymbol{x}\|^{1-d}.
$$

It is true for any $\boldsymbol{x}$. So, $\rho_s(\boldsymbol{x})$ is the unique solution of the Fokker Planck in $\mathbb{R}^d$.

*Regularity*: By definition of the marginal density, we have

$$
p_{|\cdot|}(r) := \int_{\mathbb{S}^{d-1}} p_0(\Phi(r,\boldsymbol{\theta}))r^{d-1}\,\mathrm{d}\boldsymbol{\theta} = r^{d-1}\int_{\mathbb{S}^{d-1}} p_0(\Phi(r,\boldsymbol{\theta}))\,\mathrm{d}\boldsymbol{\theta}.
$$

with $\Phi(r,\boldsymbol{\theta}) = r\boldsymbol{\theta}$. According to the assumption $\vec{\boldsymbol{x}}_0 \sim p_0 \in \mathcal{C}^2(\mathbb{R}^d \setminus \{0\})$ and compactness of $\mathbb{S}^{d-1}$, one can conclude that $p_{|\cdot|} \in \mathcal{C}^2([0,\infty[)$.

Since $p_0(\boldsymbol{x}^n\|\vec{\boldsymbol{x}}_0\| = r)$ is $\mathcal{C}^2$, we have that $p_s^n(\boldsymbol{x}^n \mid |\vec{\boldsymbol{x}}_s| = r)$ is smooth by Lemma D.4.1 for any $s > 0$. Consequently, $\rho_s(\boldsymbol{x})$ is smooth on $D$ for any $s > 0$.

*Invariant distribution*: The distribution

$$
p_\infty(\boldsymbol{x}) = \frac{p_{|\cdot|}(\|\boldsymbol{x}\|)}{\|\boldsymbol{x}\|^{d-1}}\frac{1}{|\mathbb{S}^{d-1}|}.
$$

is radial function in $\|\boldsymbol{x}\|$. The operator $\nabla_\perp$ does not act on radial functions and $\frac{1}{|\mathbb{S}^{d-1}|}$ is in the kernel of $\nabla_\perp$ such that $\nabla_\perp(\frac{1}{|\mathbb{S}^{d-1}|}) = 0$. Hence

$$
\frac{\partial}{\partial s}p_\infty(\boldsymbol{x}) = \nabla_\perp \cdot \left(\tfrac{1}{2}\boldsymbol{\Sigma}(\boldsymbol{x})\nabla_\perp p_\infty(\boldsymbol{x})\right), \quad \boldsymbol{x} \in D, \tag{D.99}
$$

$$
= \nabla_\perp \cdot \left(\tfrac{1}{2}\boldsymbol{\Sigma}(\boldsymbol{x})\nabla_\perp\left(\frac{p_{|\cdot|}(\|\boldsymbol{x}\|)}{\|\boldsymbol{x}\|^{d-1}}\frac{1}{|\mathbb{S}^{d-1}|}\right)\right), \tag{D.100}
$$

$$
= \nabla_\perp \cdot \left(\tfrac{1}{2}\frac{p_{|\cdot|}(\|\boldsymbol{x}\|)}{\|\boldsymbol{x}\|^{d-1}}\boldsymbol{\Sigma}(\boldsymbol{x})\nabla_\perp\left(\frac{1}{|\mathbb{S}^{d-1}|}\right)\right), \tag{D.101}
$$

$$
= 0. \tag{D.102}
$$

Therefore, the Fokker-Planck distribution $p_\infty$ is the stationary . In addition , $p_\infty$ is a density since

$$\int_{\mathbb{R}^d} p_\infty(\boldsymbol{x})\,\mathrm{d}\boldsymbol{x} = \int_{\mathbb{R}^d} \frac{p_{|\cdot|}\left(\|\boldsymbol{x}\|\right)}{\|\boldsymbol{x}\|^{d-1}} \frac{1}{|\mathbb{S}^{d-1}|}\,\mathrm{d}\boldsymbol{x}, \tag{D.103}$$

$$= \int_{\mathbb{R}_+}\int_{\mathbb{S}^{d-1}} \frac{p_{|\cdot|}\left(r\right)}{r^{d-1}} \frac{1}{|\mathbb{S}^{d-1}|} r^{d-1}\,\mathrm{d}r\,\mathrm{d}\boldsymbol{x}^n, \tag{D.104}$$

$$= \int_{\mathbb{R}_+}\int_{\mathbb{S}^{d-1}} p_{|\cdot|}\left(r\right) \frac{1}{|\mathbb{S}^{d-1}|}\,\mathrm{d}r\,\mathrm{d}\boldsymbol{x}^n, \tag{D.105}$$

$$= \int_{\mathbb{R}_+} p_{|\cdot|}(r)\,\mathrm{d}r \int_{\mathbb{S}^{d-1}} \frac{1}{|\mathbb{S}^{d-1}|}\,\mathrm{d}\boldsymbol{x}^n, \tag{D.106}$$

$$= 1. \tag{D.107}$$

*Convergence*: Hence, we obtain for $p_s = \rho_s$ that, we can bound the speed of convergence

$$\|p_s - p_\infty\|_{L^2(\mathbb{R}^d)}^2 = \int_{\mathbb{R}^d} \left| p_s^n(\boldsymbol{x}^n \mid |\vec{\boldsymbol{x}}_s| = \|\boldsymbol{x}\|) p_{|\cdot|}(\|\boldsymbol{x}\|)\|\boldsymbol{x}\|^{1-d} - \frac{p_{|\cdot|}\left(\|\boldsymbol{x}\|\right)}{\|\boldsymbol{x}\|^{d-1}} \frac{1}{|\mathbb{S}^{d-1}|} \right|^2 \,\mathrm{d}\boldsymbol{x},$$

$$\tag{D.108}$$

$$= \int_{\mathbb{R}^d} \left| p_s^n(\boldsymbol{x}^n \mid |\vec{\boldsymbol{x}}_s| = \|\boldsymbol{x}\|) - \frac{1}{|\mathbb{S}^{d-1}|} \right|^2 \frac{p_{|\cdot|}(\|\boldsymbol{x}\|)^2}{\|\boldsymbol{x}\|^{2d-2}} \,\mathrm{d}\boldsymbol{x}, \tag{D.109}$$

$$= \int_{\mathbb{R}_+} \left( \int_{\mathbb{S}^{d-1}} \left| p_s(\boldsymbol{\theta} \mid |\vec{\boldsymbol{x}}_s| = r) - \frac{1}{|\mathbb{S}^{d-1}|} \right|^2 \,\mathrm{d}\boldsymbol{\theta} \right) \frac{p_{|\cdot|}(r)^2}{r^{d-1}}\,\mathrm{d}r, \tag{D.110}$$

$$= \int_{\mathbb{R}_+} \left( \left\| p_s(\cdot \mid |\vec{\boldsymbol{x}}_s| = r) - \frac{1}{|\mathbb{S}^{d-1}|} \right\|_{L^2(\mathbb{S}^{d-1})}^2 \right) \frac{p_{|\cdot|}(r)^2}{r^{d-1}}\,\mathrm{d}r, \tag{D.111}$$

$$\leq \exp(-\alpha s)\int_{\mathbb{R}_+} \left( \left\| p_0(\boldsymbol{\theta} \mid \|\vec{\boldsymbol{x}}_0\| = r) - \frac{1}{|\mathbb{S}^{d-1}|} \right\|_{L^2(\mathbb{S}^{d-1})}^2 \right) \frac{p_{|\cdot|}(r)^2}{r^{d-1}}\,\mathrm{d}r,$$

$$\tag{D.112}$$

$$= \exp(-\alpha s)\int_{\mathbb{R}^d} \left| p_0(\boldsymbol{x}^n \mid \|\vec{\boldsymbol{x}}_0\| = \|\boldsymbol{x}\|) - \frac{1}{|\mathbb{S}^{d-1}|} \right|^2 \frac{p_{|\cdot|}(\|\boldsymbol{x}\|)^2}{\|\boldsymbol{x}\|^{2d-2}}\,\mathrm{d}\boldsymbol{x}, \tag{D.113}$$

$$= \exp(-\alpha s)\int_{\mathbb{R}^d} \left| p_0(\boldsymbol{x}^n \mid \|\vec{\boldsymbol{x}}_0\| = \|\boldsymbol{x}\|) p_{|\cdot|}(\|\boldsymbol{x}\|)\|\boldsymbol{x}\|^{1-d} - \frac{p_{|\cdot|}\left(\|\boldsymbol{x}\|\right)}{\|\boldsymbol{x}\|^{d-1}} \frac{1}{|\mathbb{S}^{d-1}|} \right|^2 \,\mathrm{d}\boldsymbol{x},$$

$$\tag{D.114}$$

$$= \exp(-\alpha s)\|p_0 - p_\infty\|_{L^2(\mathbb{R}^d)}^2, \tag{D.115}$$

where in the inequality we used Theorem D.4.2. The upper bound is finite since $p_\infty \in L^2(\mathbb{R}^d)$. In order to see this, we will show that the function $p_{|\cdot|}(r)r^{\frac{1-d}{2}}$ is in $L^2(0, \infty)$. Recall that

$$p_{|\cdot|}(r) := \int_{\mathbb{S}^{d-1}} p_0(\Phi(r, \boldsymbol{\theta}))r^{d-1}\,\mathrm{d}\boldsymbol{\theta} = r^{d-1}\int_{\mathbb{S}^{d-1}} p_0(\Phi(r, \boldsymbol{\theta}))\,\mathrm{d}\boldsymbol{\theta}.$$

Then, application of Jensen inequality leads

$$\int_{\mathbb{R}_+} p_{|\cdot|}(r)^2 r^{1-d}\,\mathrm{d}r = \int_{\mathbb{R}_+} \left( r^{d-1} \int_{\mathbb{S}^{d-1}} p_0(\Phi(r,\boldsymbol{\theta}))\,\mathrm{d}\boldsymbol{\theta} \right)^2 r^{1-d}\,\mathrm{d}r, \tag{D.116}$$

$$= \int_{\mathbb{R}_+} \left( \int_{\mathbb{S}^{d-1}} p_0(\Phi(r,\boldsymbol{\theta}))\,\mathrm{d}\boldsymbol{\theta} \right)^2 r^{d-1}\,\mathrm{d}r, \tag{D.117}$$

$$\leq |\mathbb{S}^{d-1}| \int_{\mathbb{R}_+} \int_{\mathbb{S}^{d-1}} p_0(\Phi(r,\boldsymbol{\theta}))^2 r^{d-1}\,\mathrm{d}\boldsymbol{\theta}\,\mathrm{d}r, \tag{D.118}$$

$$= |\mathbb{S}^{d-1}| \|p_0\|_{L^2(\mathbb{R}^d)}^2. \tag{D.119}$$

Consequently,

$$\|p_\infty\|_{L^2(\mathbb{R}^d)}^2 = \left\| \frac{p_{|\cdot|}(\|\boldsymbol{x}\|)}{\|\boldsymbol{x}\|^{d-1}} \frac{1}{|\mathbb{S}^{d-1}|} \right\|_{L^2(\mathbb{R}^d)}^2, \tag{D.120}$$

$$= \frac{1}{|\mathbb{S}^{d-1}|^2} \int_{\mathbb{S}^{d-1}} \int_{R_+} \frac{p_{|\cdot|}(r)^2}{r^{d-1}}\mathrm{d}r\mathrm{d}\boldsymbol{\theta} \leq \frac{1}{|\mathbb{S}^{d-1}|} \|p_0\|_{L^2(\mathbb{R}^d)}^2 < \infty. \tag{D.121}$$

$\square$

## D.6 Beyond pure Stratonovich noise

A possible extension to the described diffusion in equation 3.1 would be to add a drift term, i.e. considering

$$d\vec{\boldsymbol{x}}_s = \boldsymbol{A}\vec{\boldsymbol{x}}_s ds + \mathsf{G}(\vec{\boldsymbol{x}}_s) \circ d\vec{\boldsymbol{B}}_s,$$

with a skew-symmetric matrix $\boldsymbol{A} \in \mathbb{R}^{d,d}$. Then, the associated Fokker-Planck equations will additional involve an advection term $(\boldsymbol{A}\boldsymbol{x}) \cdot \nabla_\perp p_s$, which can be used to improve the speed of convergence of the dynamics.

## E  Latent distribution

The latent vectors $\vec{\boldsymbol{x}}_\infty \sim p_\infty$ of additive SGM are Gaussian white noises. This is not the case for MSGM in general. This appendix will elaborate on this point. First, we will show that MSGM latent vectors are white noise in the weak sense. Then, we will discuss the conditions for these latent vectors to be Gaussian, how to sample them, and how to transform map them to another latent space which is Gaussian. We also show that the MSGM latent distribution is always closer than the SGM latent distribution to the data distribution. Finally, we focus on the case of Cauchy data distribution, where SGM leads to singularity, unlike MSGM.

### E.1  The invariant measures define white noises in the weak sense

In additive SGM, latent vectors $\vec{\boldsymbol{x}}_\infty \sim p_\infty$ are Gaussian white noise in the strong sense, i.e. for any $i \neq j$, the coordinates $(\vec{\boldsymbol{x}}_\infty)_i$ and $(\vec{\boldsymbol{x}}_\infty)_j$ are centered, independent, and identically distributed. In contrast, the latent vectors of MSGM are white noises in the weak sense, as stated by the following proposition. For any $i \neq j$, the coordinates $(\vec{\boldsymbol{x}}_\infty)_i$ and $(\vec{\boldsymbol{x}}_\infty)_j$ are uncorrelated but neither Gaussian nor independent, in general.

**Proposition E.1.1.** *Let the assumptions A1 and A2 hold, $\mathbb{E}\|\vec{\boldsymbol{x}}_\infty\|^2 < +\infty$, and $p_\infty$ be a stationary density of the Fokker-Planck equation D.22. Then, $\vec{\boldsymbol{x}}_\infty \sim p_\infty$ is a white noise in the weak sense, i.e. $\mathbb{E}\vec{\boldsymbol{x}}_\infty = 0$, $\mathbb{E}(\vec{\boldsymbol{x}}_\infty)_i^2 < +\infty$ independent of $i$, and $\mathbb{E}(\vec{\boldsymbol{x}}_\infty)_i(\vec{\boldsymbol{x}}_\infty)_j = 0, \forall i,j \in \{1,\dots,d\}$ with $i \neq j$.*

*Proof.* From Theorem D.2.1, $p_\infty$ is rotation-variant. So there exist a function $h : \mathbb{R}^+ \to \mathbb{R}^+$ such that for any $\boldsymbol{x} \in \mathbb{R}^d, p_\infty(\boldsymbol{x}) = h(\|\boldsymbol{x}\|)$ . Then,

$$\mathbb{E}(\vec{\boldsymbol{x}}_\infty)_i = \int_{\mathbb{R}^d} x_i h(\|\boldsymbol{x}\|) d\boldsymbol{x} = \int_{\mathbb{R}^{d-1}} \left( \int_{\mathbb{R}} x_i h(\|\boldsymbol{x}\|) dx_i \right) \Pi_{k \neq i} dx_k = 0, \tag{E.1}$$

since the function $x_i \to x_i h(\|\boldsymbol{x}\|)$ is even.

Similarly, for $i \neq j$ in $\{1, \dots, d\}$

$$\mathbb{E}(\vec{\boldsymbol{x}}_\infty)_i (\vec{\boldsymbol{x}}_\infty)_j = \int_{\mathbb{R}^d} x_i x_j h(\|\boldsymbol{x}\|) d\boldsymbol{x}, \tag{E.2}$$

$$= \int_{\mathbb{R}^{d-1}} x_j \underbrace{\left( \int_{\mathbb{R}} x_i h(\|\boldsymbol{x}\|) dx_i \right)}_{=0} \Pi_{k \neq i} dx_k, \tag{E.3}$$

$$= 0. \tag{E.4}$$

Moreover, for $i$ in $\{1, \dots, d\}$, we have

$$+\infty > \mathbb{E}\|\vec{\boldsymbol{x}}_\infty\|^2, \tag{E.5}$$

$$= \mathbb{E} \sum_{i=1}^{d} (\vec{\boldsymbol{x}}_\infty)_i^2, \tag{E.6}$$

$$\geqslant \mathbb{E}(\vec{\boldsymbol{x}}_\infty)_i^2, \tag{E.7}$$

$$= \int_{\mathbb{R}^{d-1}} \left( \int_{\mathbb{R}} x_i^2 h(\|\boldsymbol{x}\|) dx_i \right) \Pi_{k \neq i} dx_k, \tag{E.8}$$

which does not depends of $i$. $\qquad\square$

**Remark 1.** *Since $\mathbb{E}(\vec{\boldsymbol{x}}_\infty)_i^2$ does not depend on $i$, we can easily evaluate it from Theorem 3.3.1 and Proposition D.3.2*

$$\mathbb{E}(\vec{\boldsymbol{x}}_\infty)_i^2 = \frac{1}{d}\mathbb{E}\|\vec{\boldsymbol{x}}_\infty\|^2 = \lim_{s \to +\infty} \frac{1}{d}\mathbb{E}\|\vec{\boldsymbol{x}}_s\|^2 = \frac{1}{d}\mathbb{E}\|\vec{\boldsymbol{x}}_0\|^2, \tag{E.9}$$

*and thus*

$$\mathbb{E}(\vec{\boldsymbol{x}}_\infty \vec{\boldsymbol{x}}_\infty^\intercal) = \frac{1}{d}\mathbb{E}\|\vec{\boldsymbol{x}}_0\|^2 \boldsymbol{I}_d. \tag{E.10}$$

*Therefore, monitoring the covariance of $\vec{\boldsymbol{x}}_T$ and its distance to $\frac{1}{d}\mathbb{E}\|\vec{\boldsymbol{x}}_0\|^2 \boldsymbol{I}_d$ are convenient proxies of the forward SDE convergence.*

### E.2 CONDITION OF GAUSSIANITY FOR THE LATENT VECTOR

**Proposition E.2.1.** *Let assumptions A1 and A2 hold and $p_{|.|^2}$ be the density of $\|\vec{\boldsymbol{x}}_0\|^2$. Then, the latent distribution $p_\infty$ is Gaussian if and only if $p_{|.|^2}$ is a scaled $\chi^2$ distribution with $d$ degrees of freedom, denoted $\alpha^2 \chi_d^2$ with $\alpha \geqslant 0$.*

*Proof.* From Theorem D.2.1, we know that $p_\infty$ is rotation invariant, i.e. it is a function of $\|\boldsymbol{x}\|$. If this distribution is Gaussian, it has to be of the form $\mathcal{N}(0, \alpha^2 \boldsymbol{I}_d)$ with $\alpha \geqslant 0$. Then, $\|\vec{\boldsymbol{x}}_0\|^2 = \|x_\infty\|^2 \sim \alpha^2 \chi_d^2$. Reciprocally, if there exists $\alpha \geqslant 0$ such that $p_{|.|^2} = \alpha^2 \chi_d^2$ then $p_{|.|} = \alpha \chi_d$, where we denote by $\alpha \chi_d$ the distribution of a positive random variable $X = \sqrt{\alpha^2 R}$ such that $R \sim \chi_d^2$. From Theorem 3.3.1, we know that

$$p_\infty(\boldsymbol{x}) = p_{|.|}(\|\boldsymbol{x}\|) \frac{\|\boldsymbol{x}\|^{1-d}}{|\mathbb{S}^{d-1}|} = p_{\alpha \chi_d}(\|\boldsymbol{x}\|) \frac{\|\boldsymbol{x}\|^{1-d}}{|\mathbb{S}^{d-1}|}. \tag{E.11}$$

It is the distribution $\mathcal{N}(0, \alpha^2 \boldsymbol{I}_d)$ written in spherical form. So, the latent distribution $p_\infty$ is Gaussian. $\qquad\square$

**Remark 2.** *Isotropic Gaussian data $\vec{x}_0 \sim \mathcal{N}(0, \alpha^2 \mathbf{I}_d)$ will hence leads to Gaussian latent space. But the contrapositive is not true. To see this, let us consider a general spherical decomposition of the data distribution $p_0$ :*

$$p_0(\boldsymbol{x}) = p^{\otimes}\left(\|\boldsymbol{x}\|, \frac{\boldsymbol{x}}{\|\boldsymbol{x}\|}\right) \|\boldsymbol{x}\|^{1-d} = p_{|.|}(\|\boldsymbol{x}\|) p_0^n\left(\frac{\boldsymbol{x}}{\|\boldsymbol{x}\|} \bigg| \|\boldsymbol{x}\|\right) \|\boldsymbol{x}\|^{1-d}. \tag{E.12}$$

*The latent distribution would be Gaussian as long as the distribution of norm is $p_{|.|} = p_{\alpha\chi_d}$. But the conditional distribution of direction can be any valid conditional distribution on the $d$-sphere. For instance,*

$$p_0^n\left(\frac{\boldsymbol{x}}{\|\boldsymbol{x}\|} \bigg| \|\boldsymbol{x}\|\right) = \delta\left(\frac{\boldsymbol{x}}{\|\boldsymbol{x}\|} - \boldsymbol{e}^{(1)}\right), \quad \text{with} \quad \boldsymbol{e}^{(1)} = (1, 0, \ldots, 0), \tag{E.13}$$

*is a valid candidate even though $p_0$ is not Gaussian (since its support is $\mathbb{R}^+ \times \{0\}^{d-1}$).*

### E.3 A TRACTABLE ALGORITHM TO SAMPLE LATENT VECTORS

With the following proposition, if we know the distribution of norms, $p_{|.|}$, we can sample latent vectors from $p_\infty$.

**Proposition E.3.1.** *Let $\overleftarrow{\boldsymbol{x}}_0^{\mathcal{N}} \sim \mathcal{N}(0, \mathbf{I}_d)$ and $\overleftarrow{\boldsymbol{x}}_0 = F^{-1}\left(\overleftarrow{\boldsymbol{x}}_0^{\mathcal{N}}\right)$ with $F^{-1}(\boldsymbol{x}) := f(\|\boldsymbol{x}\|) \frac{\boldsymbol{x}}{\|\boldsymbol{x}\|}$ if $\boldsymbol{x} \neq 0$ and $0$ otherwise,*

$$f(r) := F_{|.|}^{-1}(F_{\chi^2(d)}(r^2)), \quad \forall r > 0, \tag{E.14}$$

*for the generalized inverse CDF $F_{|.|}^{-1}$ of $p_{|.|}$ and $F_{\chi^2}$ is the CDF of the $\chi^2$ distribution with $d$ degrees of freedom. Then $\overleftarrow{\boldsymbol{x}}_0 \sim p_\infty$.*

*Proof.* Since $\overleftarrow{\boldsymbol{x}}_0^{\mathcal{N}} \sim \mathcal{N}(0, \mathbf{I}_d)$, we know that $\|\overleftarrow{\boldsymbol{x}}_0^{\mathcal{N}}\|^2 \sim \chi_{d-1}^2$, i.e. $F_{\chi^2}(\|\overleftarrow{\boldsymbol{x}}_0^{\mathcal{N}}\|^2) \sim \mathcal{U}(0,1)$ and then $R := f(\|\overleftarrow{\boldsymbol{x}}_0^{\mathcal{N}}\|) = F_{|.|}^{-1}\left(F_{\chi^2}\left(\|\overleftarrow{\boldsymbol{x}}_0^{\mathcal{N}}\|^2\right)\right) \sim p_{|.|}$. In addition, the normalized vector is $\frac{\overleftarrow{\boldsymbol{x}}_0^{\mathcal{N}}}{\|\overleftarrow{\boldsymbol{x}}_0^{\mathcal{N}}\|} \sim \mathcal{U}(\mathbb{S}^{d-1})$. The norm $\|\overleftarrow{\boldsymbol{x}}_0^{\mathcal{N}}\|$ and the direction $\frac{\overleftarrow{\boldsymbol{x}}_0^{\mathcal{N}}}{\|\overleftarrow{\boldsymbol{x}}_0^{\mathcal{N}}\|}$ are independent. Therefore, $R$ and $\frac{\overleftarrow{\boldsymbol{x}}_0^{\mathcal{N}}}{\|\overleftarrow{\boldsymbol{x}}_0^{\mathcal{N}}\|}$ are also independent. We can conclude that $\overleftarrow{\boldsymbol{x}}_0 = R \frac{\overleftarrow{\boldsymbol{x}}_0^{\mathcal{N}}}{\|\overleftarrow{\boldsymbol{x}}_0^{\mathcal{N}}\|}$ follows the correct distribution. $\square$

In practice, we do not exactly know the distribution of the data norm $p_{|.|}$. So, we do not have access to $F_{|.|}$ or $f$. Instead, we approximate the distribution of $\log \|\vec{\boldsymbol{x}}_0\|_\epsilon$ with $\|\boldsymbol{x}\|_\epsilon := \|\boldsymbol{x}\| + \epsilon$, denoted $p_{\log|.|_\epsilon}$, by a model $\hat{p}_{\log|.|_\epsilon}$, or equivalently $F_{\log|.|_\epsilon}$ by a model $\hat{F}_{\log|.|_\epsilon}$ (see Section C). We perform a similar sampling procedure for the latent vectors, replacing $F$ by our approximation. We obtain samples of an approximate latent distribution $\hat{p}_\infty$, as stated by Proposition E.3.2.

**Proposition E.3.2.** *Let $\overleftarrow{\boldsymbol{x}}_0^{\mathcal{N}} \sim \mathcal{N}(0, \mathbf{I}_d)$ and $\overleftarrow{\boldsymbol{x}}_0 = \hat{F}^{-1}\left(\overleftarrow{\boldsymbol{x}}_0^{\mathcal{N}}\right)$ with $\hat{F}^{-1}(\boldsymbol{x}) := \hat{f}(\|\boldsymbol{x}\|) \frac{\boldsymbol{x}}{\|\boldsymbol{x}\|}$ if $\boldsymbol{x} \neq 0$ and $0$ otherwise,*

$$\hat{f}(r) := \exp\left(\hat{F}_{\log|.|_\epsilon}^{-1}(F_{\chi^2(d)}(r^2))\right) - \epsilon, \quad \forall r > 0, \tag{E.15}$$

*for the generalized inverse of the approximated CDF $\hat{F}_{\log|.|_\epsilon}^{-1}$ associated to the approximated PDF $\hat{p}_{\log|.|_\epsilon}$, and $F_{\chi^2}$ is the CDF of the $\chi^2$ distribution with $d$ degrees of freedom. Then $\overleftarrow{\boldsymbol{x}}_0 \sim \hat{p}_\infty$, where $\hat{p}_\infty$ is the empirical approximation of $p_\infty$, that is $\hat{p}_\infty(\boldsymbol{x}) := \hat{p}_{\log|.|_\epsilon}(\log\|\boldsymbol{x}\|_\epsilon) \frac{\|\boldsymbol{x}\|^{1-d}}{|\mathbb{S}^{d-1}|}, \forall \boldsymbol{x} \in \mathbb{R}^d$.*

*Proof.* We can follow the same proof that for Proposition E.3.1 replacing $F_{|.|}$, $f$, $p_{|.|}$, and $p_\infty$ by $\hat{F}_{\log|.|_\epsilon}$, $\hat{f}$, $\hat{p}_{\log|.|_\epsilon}$, and $\hat{p}_\infty$ respectively. $\square$

### E.4 GAUSSIANIZATION OF THE LATENT VECTORS

If needed, we can easily build a second latent space with standard Gaussian vectors. As stated by the following proposition, for any (non-Gaussian) latent vector $\vec{\boldsymbol{x}}_T$, we can create a Gaussian vector $\vec{\boldsymbol{x}}_T^{\mathcal{N}}$

$$\vec{\boldsymbol{x}}_T^{\mathcal{N}} = R_T \vec{\boldsymbol{x}}_T^n, \quad \text{with} \quad \vec{\boldsymbol{x}}_T^n = \vec{\boldsymbol{x}}_T / \|\vec{\boldsymbol{x}}_T\|, \quad \text{and} \quad R_T = \hat{f}^{-1}(\|\vec{\boldsymbol{x}}_T\|). \tag{E.16}$$

If $\vec{\boldsymbol{x}}_T$ is zero, we just set $\vec{\boldsymbol{x}}_T^{\mathcal{N}}$ to zero.

**Proposition E.4.1.** *Let $\vec{\boldsymbol{x}}_T \sim \hat{p}_\infty$, where $\hat{p}_\infty$ is the empirical approximation of $p_\infty$, that is $\hat{p}_\infty(\boldsymbol{x}) := \hat{p}_{\log|.|_\epsilon}(\log\|\boldsymbol{x}\|_\epsilon)\frac{\|\boldsymbol{x}\|^{1-d}}{|\mathbb{S}^{d-1}|}, \forall \boldsymbol{x} \in \mathbb{R}^d$, and $\vec{\boldsymbol{x}}_T^{\mathcal{N}} = \hat{F}\left(\vec{\boldsymbol{x}}_T\right)$ with $\hat{F}(\boldsymbol{x}) := \hat{f}^{-1}(\|\boldsymbol{x}\|)\frac{\boldsymbol{x}}{\|\boldsymbol{x}\|}$ if $\boldsymbol{x} \neq 0$ and $0$ otherwise,*

$$\hat{f}^{-1}(r) = \sqrt{(F_{\chi^2}^{-1}(\hat{F}_{\log|.|_\epsilon}(r))}, \quad \forall r > 0, \tag{E.17}$$

*for the approximated CDF $\hat{F}_{\log|.|_\epsilon}$ associated to the approximated PDF $\hat{p}_{\log|.|_\epsilon}$, and $F_{\chi^2}$ is the CDF of the $\chi^2$ distribution with $d$ degrees of freedom. Then $\vec{\boldsymbol{x}}_T^{\mathcal{N}} \sim \mathcal{N}(0, \boldsymbol{I}_d)$.*

*Proof.* We can follow the same proof that for Proposition E.3.1 replacing $F_{|.|}^{-1}$, $f$, $\chi^2$, $\mathcal{N}(0, \boldsymbol{I}_d)$, $p_{|.|}$, and $p_\infty$ by $\hat{F}_{\log|.|_\epsilon}$, $\hat{f}^{-1}$, $\hat{p}_{\log|.|_\epsilon}$, $\hat{p}_\infty$, $\chi^2$, and $\mathcal{N}(0, \boldsymbol{I}_d)$ respectively. □

### E.5 A SHORTER DISTANCE BETWEEN LATENT AND DATA DISTRIBUTION

The following result states, that the latent space of MSGM is closer to the data distribution compared to the SGM latent distribution in KL-divergence.

**Proposition E.5.1.** *Let the assumptions A1 and A2 hold, $p_{|.|^2}$ be the density of $\|\vec{\boldsymbol{x}}_0\|^2$, $p_\infty$ and $p_\infty^{\mathcal{N}} = \mathcal{N}(0, \boldsymbol{I}_d)$ be the MSGM and the SGM latent distributions respectively, then*

$$D_{KL}(p_\infty \| p_0) \leqslant D_{KL}(p_\infty^{\mathcal{N}} \| p_0), \tag{E.18}$$

*with equality if and only if $p_{|.|^2}$ is a $\chi^2$ distribution with $d$ degrees of freedom.*

*Proof.* We recall that the MSGM latent pdf is

$$p_\infty(\boldsymbol{x}) = \frac{p_{|.|}(\|\boldsymbol{x}\|)}{\|\boldsymbol{x}\|^{d-1}}\frac{1}{|\mathbb{S}^{d-1}|}. \tag{E.19}$$

and the data distribution reads

$$p_0(\boldsymbol{x}) = \frac{p_{|.|}(\|\boldsymbol{x}\|)}{\|\boldsymbol{x}\|^{d-1}} p_0(\boldsymbol{x}^n | \|\vec{\boldsymbol{x}}_0\| = \|\boldsymbol{x}\|). \tag{E.20}$$

Let denotes $p_{\chi_d^2}$ the $\chi^2$ distribution with $d$ degrees of freedom

$$p_0^{\mathcal{LN}}(\boldsymbol{x}) = \frac{p_{\chi_d^2}(\|\boldsymbol{x}\|)}{\|\boldsymbol{x}\|^{d-1}} p_0(\boldsymbol{x}^n | \|\vec{\boldsymbol{x}}_0\| = \|\boldsymbol{x}\|). \tag{E.21}$$

It is the distribution of $\vec{\boldsymbol{x}}_0^{\mathcal{LN}} = F\left(\vec{\boldsymbol{x}}_0\right)$ with $F(\boldsymbol{x}) := f^{-1}(\|\boldsymbol{x}\|)\frac{\boldsymbol{x}}{\|\boldsymbol{x}\|}$ if $\boldsymbol{x} \neq 0$ and $0$ otherwise, and

$$f^{-1}(r) = \sqrt{(F_{\chi^2}^{-1}(F_{|.|}(r))}, \quad \forall r > 0, \tag{E.22}$$

and $F_{|.|}(R) = \int_0^R p_{|.|}(r)dr$ the cdf associated to $p_{|.|}$.

We have

$$
\begin{align}
0 \;\leqslant\;& D_{KL}(p_0 \| p_0^{\mathcal{LN}}), \tag{E.23}\\
=\;& \int p_0(\boldsymbol{x}) \log \frac{p_0}{p_0^{\mathcal{LN}}} d\boldsymbol{x}, \tag{E.24}\\
=\;& \int p_0(\boldsymbol{x}) \log \frac{p_{|.|}(\|\boldsymbol{x}\|)}{p_{\chi_d^2}(\|\boldsymbol{x}\|)} d\boldsymbol{x}, \tag{E.25}\\
=\;& \int p_0(\boldsymbol{x}) \log \frac{p_{|.|}(\|\boldsymbol{x}\|)}{\|\boldsymbol{x}\|^{d-1}|\mathbb{S}^{d-1}|} \frac{\|\boldsymbol{x}\|^{d-1}|\mathbb{S}^{d-1}|}{p_{\chi_d^2}(\|\boldsymbol{x}\|)} d\boldsymbol{x}, \tag{E.26}\\
=\;& \int p_0(\boldsymbol{x}) \log \frac{p_\infty(\boldsymbol{x})}{p_\infty^{\mathcal{N}}(\boldsymbol{x})} d\boldsymbol{x}, \tag{E.27}\\
=\;& \int p_0(\boldsymbol{x}) \log \frac{p_0(\boldsymbol{x})}{p_\infty^{\mathcal{N}}(\boldsymbol{x})} \frac{p_\infty(\boldsymbol{x})}{p_0(\boldsymbol{x})} d\boldsymbol{x}, \tag{E.28}\\
=\;& \int p_0(\boldsymbol{x}) \log \frac{p_0(\boldsymbol{x})}{p_\infty^{\mathcal{N}}(\boldsymbol{x})} d\boldsymbol{x} - \int p_0(\boldsymbol{x}) \log \frac{p_0(\boldsymbol{x})}{p_\infty(\boldsymbol{x})} d\boldsymbol{x}, \tag{E.29}\\
=\;& D_{KL}(p_0 \| p_\infty^{\mathcal{N}}) - D_{KL}(p_0 \| p_\infty), \tag{E.30}
\end{align}
$$

with equality if and only if $p_0 = p_0^{\mathcal{LN}}$ i.e. $p_{|.|} = p_{\chi_d^2}$. $\qquad\square$

### E.6 Relevance of MSGM latent space for heavy-tail distributions.

This appendix provides an analysis of why the latent space of MSGM is better suited to heavy-tailed data distribution as compared to the latent space of SGM. This subsection can be viewed as an extension of Proposition E.5.1. In particular the derived inequality in Proposition E.5.1 becomes meaning less if both sides are not finite. However, as we will see for example of heavy tail distribution such as the (product) Cauchy distribution, this is not the case. To this end we will show in Section E.6.1 that we KL divergence of data distribution and SGM latent space is not finite and that it is finite for the data distribution and the MSGM latent space in Section E.6.2.

We note that the analysis can be extended to a broader class of heavy tailed distributions and more general SGM latent spaces such as general Gaussian distributions.

#### E.6.1 Infinite KL Divergence between Cauchy distribution and standard Gaussian

Let $\phi(\boldsymbol{x}) = \frac{1}{\sqrt{2\pi}} e^{-\boldsymbol{x}^2/2}$ be the density of the standard Gaussian $\mathcal{N}(0, I)$, and let $p_0(\boldsymbol{x}) = \prod_{i=1}^{d} \frac{1}{\pi(1+x_i^2)}$ be the product density of univariate Cauchy distributions. Then, the following holds.

**Lemma E.6.1.**
$$
D_{\mathrm{KL}}(p_0 \| \phi) = \infty.
$$

*Proof.* Let $L > 1$ and define the set
$$
M = \{\boldsymbol{x} \in \mathbb{R}^d \mid x_1 \geq L, |x_j| \leq 1, \quad j = 2, \dots, d\}.
$$
Then for $\boldsymbol{x} \in M$ and $C := \frac{1}{\pi^d 2^{d-1}}$
$$
p_0(\boldsymbol{x}) = \prod_{i=1}^{d} \frac{1}{\pi(1+x_i^2)} \geq \frac{1}{\pi^d} \cdot \frac{1}{1+x_1^2} \cdot \prod_{i=2}^{d} \frac{1}{1+1} = \frac{1}{\pi^d 2^{d-1}} \cdot \frac{1}{1+x_1^2} = C \frac{1}{1+x_1^2}.
$$
Moreover, for any $\boldsymbol{x} \in \mathbb{R}^d$, it holds that
$$
\phi(\boldsymbol{x}) = (2\pi)^{-d/2} e^{-(x_1^2 + \sum_{i=2}^{d} x_i^2)/2} \leq (2\pi)^{-d/2} e^{-x_1^2/2}.
$$
Consequently, for $L$ large enough,
$$
p_0(\boldsymbol{x}) \log \frac{p_0(\boldsymbol{x})}{\phi(\boldsymbol{x})} \geq \frac{x_1^2}{2} + \mathcal{O}(\log x_1),
$$

where $\mathcal{O}$ refers to Landau-symbol of big-O notation. Together, for $\boldsymbol{x} \in M$ and $L$ large enough, there exists $\underline{C} > 0$ such that

$$p_0(\boldsymbol{x}) \log \frac{p_0(\boldsymbol{x})}{\phi(\boldsymbol{x})} \geq \frac{C}{1 + x_1^2} \frac{x_1^2}{4} \geq \underline{C} > 0.$$

Consequently,

$$\int_M p_0(\boldsymbol{x}) \log \frac{p_0(\boldsymbol{x})}{\phi(\boldsymbol{x})} \mathrm{d}\boldsymbol{x} \geq \int_T^\infty \left( \int \cdots \int_{|x_j| \leq 1, j \geq 2} \underline{C} \mathrm{d}x_2 \cdots \mathrm{d}x_d \right) \mathrm{d}x_1 = \infty.$$

By Lebesgue decomposition, we conclude $D_{\mathrm{KL}}(p_0 \| \phi) = \infty$. $\qquad\square$

### E.6.2 Finite KL divergence between Cauchy distribution and its related $\rho_\infty$

Let $d \geq 2$ and again consider the product of Cauchy densities

$$p_0(\boldsymbol{x}) = \prod_{i=1}^d \frac{1}{\pi(1 + x_i^2)}.$$

Let $\boldsymbol{x}_0 \sim p_0$ and let $p_R$ be the density of $R = \|\vec{\boldsymbol{x}}_0\|$. Then, motivated by our latent space distribution equation 3.8, consider the density

$$p_\infty(\boldsymbol{x}) = \frac{p_R(\|\boldsymbol{x}\|)}{\|\boldsymbol{x}\|^{d-1} |\mathbb{S}^{d-1}|}, \quad \boldsymbol{x} \in \mathbb{R}^d \setminus \{0\}, \tag{E.31}$$

Then, the following holds

**Lemma E.6.2.**

$$D_{\mathrm{KL}}(p_0 \| p_\infty) < \infty.$$

*Proof.* It holds that

$$\log \frac{p_0(\boldsymbol{x})}{p_\infty(\boldsymbol{x})} = \log p_0(\boldsymbol{x}) - \log p_R(\|\boldsymbol{x}\|) + (d - 1) \log \|\boldsymbol{x}\| + \log |\mathbb{S}^{d-1}|.$$

Hence,

$$D_{\mathrm{KL}}(p_0 \| p_\infty) = \mathbb{E}_p[\log p_0(\vec{\boldsymbol{x}}_0)] - \mathbb{E}_p[\log p_R(\|\vec{\boldsymbol{x}}_0\|)] + (d - 1)\mathbb{E}_p[\log \|\vec{\boldsymbol{x}}_0\|] + \log |\mathbb{S}^{d-1}|,$$

where $\mathbb{E}_p$ denotes the expectation with respect to the probability measure $p_0 \mathrm{d}\boldsymbol{x}$. We will show, that each term separately is finite. We start with the first term, followed by the the third. The finiteness of the second term turns out to be a consequence of the finiteness of the second term.

- *First term*: It holds that

$$\log p_0(\vec{\boldsymbol{x}}_0) = -d \log \pi - \sum_{i=1}^d \log(1 + (\vec{x}_0)_i^2)$$

for $\vec{\boldsymbol{x}}_0 \sim p_0$. Since coordinates of $(\vec{x}_0)_i$ are iid it is enough to check the marginal integrals for finiteness. In particular it holds that

$$\int_{-\infty}^\infty \frac{\log(1 + x^2)}{\pi(1 + x^2)} \mathrm{d}x = \log(4) < \infty.$$

Consequently

$$|\mathbb{E}_p[\log p_0(\vec{\boldsymbol{x}}_0)]| < \infty. \tag{E.32}$$

- *Third term*: For the second term, let $R = \|\vec{x}_0\|$ and $M = \max\limits_{i=1,\ldots,d} |(\vec{x}_0)_i|$. Then, almost surely

$$M \leq R \leq \sqrt{d} M \quad \Rightarrow \quad \log M \leq \log R \leq \log M + \frac{1}{2} \log d.$$

Consequently,

$$|\mathbb{E}_p[\log R] - \mathbb{E}_p[\log M]| \leq \frac{1}{2} \log d.$$

Thus if we show $\mathbb{E}_p[\log M] < \infty$, then $\mathbb{E}_p[\log R] < \infty$ as well, since both expectation only differ up to a finite factor. Using the CDF $F$ of $(\vec{x}_0)_1$ e.g. for $(\vec{x}_0)_1$ it holds that

$$\mathbb{P}(\|(\vec{x}_0)_1\| \leq t) = F(t) - F(-t) = \frac{2}{\pi} \arctan t, \quad t \geq 0.$$

Consequently, since $(\vec{x}_0)_1, \ldots, (\vec{x}_0)_d$ are iid, the CDF $F_M$ of $M$ satisfies

$$F_M(t) = \mathbb{P}(M \leq t) = \left(\frac{2}{\pi} \arctan t\right)^d, \quad t \geq 0.$$

Hence, the density $f_M$ of $M$ is given (for $d \geq 2$) as

$$f_M(t) = \frac{\partial}{\partial t} F_M(t) = d \left(\frac{2}{\pi}\right)^d (\arctan t)^{d-1} \frac{1}{1+t^2}.$$

Now, by a integral splitting we find that

$$\mathbb{E}_p[\log M] = \int_0^\infty \log t f_M(t) \mathrm{d}t = \int_0^1 \log t f_M(t) \mathrm{d}t + \int_1^\infty \log(t) f_M(t) \mathrm{d}t. \qquad \text{(E.33)}$$

By noting that for $0 \leq t$, $f_M(t) \leq Ct^{d-1}$ for some $C > 0$ and

$$\int_0^1 f_M(t)(-\log(t)) \mathrm{d}t \leqslant \int_0^1 t^{a-1}(-\log(t)) \mathrm{d}t = \frac{1}{a^2}, \quad a > 0,$$

the first integrant of equation E.33 is finite using $a = d$. For $t \geq 1$ $\arctan(t) \leq \pi/2$ and hence $f_M(t) \leq C' \frac{1}{1+t^2}$ for some $C' > 0$ and the second integral of equation E.33 is finite since

$$\int_1^\infty \frac{\log(t)}{1+t^2} \mathrm{d}t = 1 < \infty.$$

It follows that $\mathbb{E}_p[\log M]$ is finite.

- *Second term*: Recall that

$$p_R(r) = \int_{\mathbb{S}^{d-1}} p_0(r\boldsymbol{\theta}) r^{d-1} \mathrm{d}\sigma(\boldsymbol{\theta}) = r^{d-1} \int_{\mathbb{S}^{d-1}} p_0(r\boldsymbol{\theta}) \mathrm{d}\sigma(\boldsymbol{\theta}).$$

Since for $\boldsymbol{x} = r\boldsymbol{\theta}$ with $\boldsymbol{\theta} = (\theta_1, \ldots, \theta_d)$, with $\theta_i^2 \leq 1$ and using the fact that

$$p_0(r\boldsymbol{\theta}) = \prod_{i=1}^d \frac{1}{\pi(1 + r^2\theta_i^2)}$$

we conclude

$$p_0(r\boldsymbol{\theta}) \geq \prod_{i=1}^d \frac{1}{\pi(1 + r^2)} = \frac{1}{\pi^d(1 + r^2)^d}.$$

Therefore,

$$p_R(r) \geq r^{d-1} \frac{1}{\pi^d(1 + r^2)^d} \cdot |\mathbb{S}^{d-1}| =: C_d \frac{r^{d-1}}{(1 + r^2)^d}.$$

Hence,

$$\log p_R(r) \geq \log C_d + (d-1)\log r - d\log(1+r^2),$$

which yields

$$\mathbb{E}_p[\log p_R(\|\overrightarrow{\boldsymbol{x}}_0\|)] \geq \log C_d + (d-1)\mathbb{E}_p[\log R] - d\mathbb{E}_p\log(1+R^2)]. \qquad \text{(E.34)}$$

For the third term in equation E.34 it holds that $R^2 = \sum_{i=1}^{d}(\overrightarrow{x}_0)_i^2$. Now for $M \leq 1$ we have since $R \leq \sqrt{d}M \leq \sqrt{d}$ that $\log(1+R^2) \leq \log(1+\sqrt{d})$ is independent of $R$. For $M \geq 1$, $\log(1+R^2) \leq \log(1+dM^2) \leq \log(dM^2+dM^2) = \log(2d) + 2\log(M)$. Since we already showed that $\mathbb{E}_p[\log R]$ is finite, we conclude that the lower bound in equation E.34 is finite. For the upper bound, note that

$$p_0(r\boldsymbol{\theta}) \leq \frac{1}{\pi^d}, \quad \forall r > 0.$$

Thus,

$$p_R(r) \leq r^{d-1}\frac{1}{\pi^d}|\mathbb{S}^{d-1}| =: C_d r^{d-1}$$

for some $C_d > 0$. So

$$\log p_R(r) \leq \log C_d + (d-1)\log r.$$

And finally,

$$\mathbb{E}_p[\log_R(R)] \leq \log C_d + (d-1)\mathbb{E}_p[\log R] < \infty$$

since $\mathbb{E}_p[\log R] < \infty$

- *Fourth term*: Finite since volume of the finite dimensional hypersphere.

$\square$

## F  BACKWARD DIFFUSION

This section is devoted to the derivation of the reverse SDE and ODE of our proposed MSGM in Itô and Stratonovich form.

**Proposition F.1.** *(**Backward SDE**) Let the skew-symmetry assumption A1 hold. Then, the Itô form of the reverse SDE associated to the forward SDE 3.1 is given by the SDE*

$$d\overleftarrow{\boldsymbol{x}}_t = \tfrac{1}{2}(\nabla\cdot\boldsymbol{\Sigma})(\overleftarrow{\boldsymbol{x}}_t)dt + \mathbf{G}(\overleftarrow{\boldsymbol{x}}_t)\mathbf{G}(\overleftarrow{\boldsymbol{x}}_t)^{\mathsf{T}}\nabla\log p_{T-t}(\overleftarrow{\boldsymbol{x}}_t)dt + \mathbf{G}(\overleftarrow{\boldsymbol{x}}_t)d\overleftarrow{\boldsymbol{B}}_t. \qquad \text{(F.1)}$$

*In the Stratonovich form, it reads:*

$$d\overleftarrow{\boldsymbol{x}}_t = \mathbf{G}(\overleftarrow{\boldsymbol{x}}_t)\left(\mathbf{G}(\overleftarrow{\boldsymbol{x}}_t)^{\mathsf{T}}\nabla\log p_{T-t}(\overleftarrow{\boldsymbol{x}}_t)dt + \circ d\overleftarrow{\boldsymbol{B}}_t\right). \qquad \text{(F.2)}$$

*Proof.* From Anderson (1982); Song et al. (2021) and the Itô forward SDE (see Lemma D.1.2), we know that the Itô reverse SDE with negative $ds$ writes

$$d\overleftarrow{\boldsymbol{x}}_s = \tfrac{1}{2}(\nabla\cdot\boldsymbol{\Sigma})(\overleftarrow{\boldsymbol{x}}_s)ds - (\nabla\cdot\boldsymbol{\Sigma})(\overleftarrow{\boldsymbol{x}}_s)ds - \mathbf{G}(\overleftarrow{\boldsymbol{x}}_s)\mathbf{G}(\overleftarrow{\boldsymbol{x}}_s)^{\mathsf{T}}\nabla\log p_s(\overleftarrow{\boldsymbol{x}}_s)ds$$
$$\qquad + \mathbf{G}(\overleftarrow{\boldsymbol{x}}_s)d\overleftarrow{\boldsymbol{B}}'_s, \qquad \text{(F.3)}$$
$$= -\tfrac{1}{2}(\nabla\cdot\boldsymbol{\Sigma})(\overleftarrow{\boldsymbol{x}}_s)ds - \mathbf{G}(\overleftarrow{\boldsymbol{x}}_s)\mathbf{G}(\overleftarrow{\boldsymbol{x}}_s)^{\mathsf{T}}\nabla\log p_s(\overleftarrow{\boldsymbol{x}}_s)ds + \mathbf{G}(\overleftarrow{\boldsymbol{x}}_s)d\overleftarrow{\boldsymbol{B}}'_s. \qquad \text{(F.4)}$$

Replacing the decreasing $s \in [0,T]$ by $s = T - t$ with increasing $t \in [0,T]$ and using another Brownian motion $\overleftarrow{\boldsymbol{B}}$, we obtain the Itô backward SDE with positive $dt$

$$d\overleftarrow{\boldsymbol{x}}_t = \tfrac{1}{2}(\nabla\cdot\boldsymbol{\Sigma})(\overleftarrow{\boldsymbol{x}}_t)dt + \mathbf{G}(\overleftarrow{\boldsymbol{x}}_t)\mathbf{G}(\overleftarrow{\boldsymbol{x}}_t)^{\mathsf{T}}\nabla\log p_{T-t}(\overleftarrow{\boldsymbol{x}}_t)dt + \mathbf{G}(\overleftarrow{\boldsymbol{x}}_t)d\overleftarrow{\boldsymbol{B}}_t. \qquad \text{(F.5)}$$

Then, Lemma D.1.1 and the standard Stratonovich-to-Itô formula (e.g. Kunita, 1997) yields the Stratonovich form of the backward SDE:

$$d\overleftarrow{\boldsymbol{x}}_t = -\tfrac{1}{2}d\langle \mathbf{G}(\overleftarrow{\boldsymbol{x}}_t), \overleftarrow{\boldsymbol{B}}_t\rangle_t + \tfrac{1}{2}(\nabla \cdot \boldsymbol{\Sigma})(\overleftarrow{\boldsymbol{x}}_t)dt + \mathbf{G}(\overleftarrow{\boldsymbol{x}}_t)\mathbf{G}(\overleftarrow{\boldsymbol{x}}_t)^{\intercal}\nabla \log p_{T-t}(\overleftarrow{\boldsymbol{x}}_t)dt$$

$$+ \mathbf{G}(\overleftarrow{\boldsymbol{x}}_t)\circ d\overleftarrow{\boldsymbol{B}}_t, \tag{F.6}$$

$$= -\tfrac{1}{2}(\nabla \cdot \boldsymbol{\Sigma})(\overleftarrow{\boldsymbol{x}}_t)dt + \tfrac{1}{2}(\nabla \cdot \boldsymbol{\Sigma})(\overleftarrow{\boldsymbol{x}}_t)dt$$

$$+ \mathbf{G}(\overleftarrow{\boldsymbol{x}}_t)\left(\mathbf{G}(\overleftarrow{\boldsymbol{x}}_t)^{\intercal}\nabla \log p_{T-t}(\overleftarrow{\boldsymbol{x}}_t)dt + \circ d\overleftarrow{\boldsymbol{B}}_t\right), \tag{F.7}$$

$$= \mathbf{G}(\overleftarrow{\boldsymbol{x}}_t)\left(\mathbf{G}(\overleftarrow{\boldsymbol{x}}_t)^{\intercal}\nabla \log p_{T-t}(\overleftarrow{\boldsymbol{x}}_t)dt + \circ d\overleftarrow{\boldsymbol{B}}_t\right). \tag{F.8}$$

$\square$

**Proposition F.2.** *(Backward probability flow ODE) Let the skew-symmetry assumption A1 hold. Then, the reverse probability flow associated to the forward SDE 3.1 is given by the ODE*

$$\frac{d\overleftarrow{\boldsymbol{x}}_t}{dt} = \tfrac{1}{2}\mathbf{G}(\overleftarrow{\boldsymbol{x}}_t)\mathbf{G}(\overleftarrow{\boldsymbol{x}}_t)^{\intercal}\nabla \log p_{T-t}(\overleftarrow{\boldsymbol{x}}_t). \tag{F.9}$$

*Proof.* From Song et al. (2021) and the Itô forward SDE (see Lemma D.1.2), we know that the reverse probability flow writes with negative $ds$

$$d\overleftarrow{\boldsymbol{x}}_s = \tfrac{1}{2}(\nabla \cdot \boldsymbol{\Sigma})(\overleftarrow{\boldsymbol{x}}_s)ds - \tfrac{1}{2}(\nabla \cdot \boldsymbol{\Sigma})(\overleftarrow{\boldsymbol{x}}_s)ds - \tfrac{1}{2}\mathbf{G}(\overleftarrow{\boldsymbol{x}}_s)\mathbf{G}(\overleftarrow{\boldsymbol{x}}_s)^{\intercal}\nabla \log p_s(\overleftarrow{\boldsymbol{x}}_s)dt, \tag{F.10}$$

$$= -\tfrac{1}{2}\mathbf{G}(\overleftarrow{\boldsymbol{x}}_s)\mathbf{G}(\overleftarrow{\boldsymbol{x}}_s)^{\intercal}\nabla \log p_s(\overleftarrow{\boldsymbol{x}}_s)dt. \tag{F.11}$$

Replacing the decreasing $s \in [0, T]$ by $s = T - t$ with increasing $t \in [0, T]$ and using another Brownian motion $\overleftarrow{\boldsymbol{B}}$, we obtain the Itô backward SDE with positive $dt$

$$\frac{d\overleftarrow{\boldsymbol{x}}_t}{dt} = \tfrac{1}{2}\mathbf{G}(\overleftarrow{\boldsymbol{x}}_t)\mathbf{G}(\overleftarrow{\boldsymbol{x}}_t)^{\intercal}\nabla \log p_{T-t}(\overleftarrow{\boldsymbol{x}}_t). \tag{F.12}$$

$\square$

# G  PROOF OF THEOREM 3.4.1: EQUIVALENCE BETWEEN ELBO AND SCORE MATCHING

This appendix derives a score-matching-based ELBO for MSGM training. In this work, we focus on the simple forward multiplicative SDE equation 3.1. Nevertheless, we here derive a slightly more general theorem, where we include a possibly non-zero Stratonovich drift $\boldsymbol{f}_S$.

## G.1  STATEMENT OF THE THEOREM

**Theorem G.1.1.** *Let us consider the forward SDE*

$$d\overrightarrow{\boldsymbol{x}}_s = \boldsymbol{f}_S(\overrightarrow{\boldsymbol{x}}_s)ds + \mathbf{G}(\overrightarrow{\boldsymbol{x}}_s)\circ d\overrightarrow{\boldsymbol{B}}_s, \tag{G.1}$$

*where assumption A1 holds. Then, we have*

$$p_0(\boldsymbol{x}|\boldsymbol{\theta}) \geqslant \mathcal{E}_\infty(\boldsymbol{x}|\boldsymbol{\theta}), \tag{G.2}$$

*with the following ELBO*

$$\mathcal{E}_\infty(\boldsymbol{x}|\boldsymbol{\theta}) := \mathbb{E}\left[\log p_T(\overrightarrow{\boldsymbol{x}}_T)\big|\overrightarrow{\boldsymbol{x}}_0 = \boldsymbol{x}\right] \tag{G.3}$$

$$- \int_0^T \mathbb{E}_{\overrightarrow{\boldsymbol{x}}_s}\left[\tfrac{1}{2}\|\boldsymbol{a}_{\boldsymbol{\theta}}(\overrightarrow{\boldsymbol{x}}_s, s)\|^2 + \nabla \cdot (\mathbf{G}(\overrightarrow{\boldsymbol{x}}_s)\boldsymbol{a}_{\boldsymbol{\theta}}(\overrightarrow{\boldsymbol{x}}_s, s)) - f_S(\overrightarrow{\boldsymbol{x}}_s)\big|\overrightarrow{\boldsymbol{x}}_0 = \boldsymbol{x}\right] ds.$$

*Proof.* Here, we review the work of Huang et al. (2021) on SGM and generalize some of their results to derive an ELBO and justify score matching for MSGM. Note that Benton et al. (2024); Ren et al. (2025) proposes a very general SGM framework with associated ELBO and score matching losses. The MSGM ELBO and thus the above theorem can be understood as a particular case of their work. The explicit dependence in $\boldsymbol{\theta}$ is omitted for readability.

## G.2 NOTATIONS CORRESPONDENCE

The forward and backward processes are denote $Y_s$ and $X_t$ in Huang et al. (2021) and $\overrightarrow{\boldsymbol{x}}_s$ and $\overleftarrow{\boldsymbol{x}}_t$ in this paper. The forward Itō equation of Huang et al. (2021) is denoted:

$$dY_s = f(Y_s, s)ds + g(Y_s, s)d\hat{B}_s. \tag{G.4}$$

Lemma D.1.1 gives the forward Itō equation of MSGM. It yields the following notation correspondence:

$$\begin{align}
g(\boldsymbol{x}, s) &= \mathbf{G}(\boldsymbol{x}), \tag{G.5}\\
D(\boldsymbol{x}) &= \tfrac{1}{2}g(\boldsymbol{x})g(\boldsymbol{x})^{\mathsf{T}} = \tfrac{1}{2}\boldsymbol{\Sigma}(\boldsymbol{x}), \tag{G.6}\\
f(\boldsymbol{x}) &= \tfrac{1}{2}(\nabla \cdot \boldsymbol{\Sigma})^{\mathsf{T}}(\boldsymbol{x}) + \boldsymbol{f}_S(\boldsymbol{x}). \tag{G.7}
\end{align}$$

And the backward equation is :

$$d\overleftarrow{\boldsymbol{x}}_t = \boldsymbol{\mu}(\overleftarrow{\boldsymbol{x}}_t, t)dt + \mathbf{G}(\overleftarrow{\boldsymbol{x}}_t, t)d\overleftarrow{\boldsymbol{B}}_t, \tag{G.8}$$

with a drift

$$\begin{align}
\boldsymbol{\mu}(\boldsymbol{x}, t) &= -f(\boldsymbol{x}) + 2(\nabla \cdot D)^{\mathsf{T}}(\boldsymbol{x}) + 2D(\boldsymbol{x})\nabla \log p_{T-t}(\boldsymbol{x}), \tag{G.9}\\
&= -\boldsymbol{f}_S(\boldsymbol{x}) + \tfrac{1}{2}(\nabla \cdot \boldsymbol{\Sigma})^{\mathsf{T}}(\boldsymbol{x}) + \boldsymbol{\Sigma}(\boldsymbol{x})\nabla \log p_{T-t}(\boldsymbol{x}), \tag{G.10}
\end{align}$$

where we would arrive at the approximate backward SDE of Figure 1 if we replace $\nabla \log p_{T-t}(\overleftarrow{\boldsymbol{x}}_t)$ by $\boldsymbol{s_\theta}(\overleftarrow{\boldsymbol{x}}_t, T-t)$ also parametrized as $\boldsymbol{a_\theta} = \mathbf{G}^{\mathsf{T}}\boldsymbol{s_\theta}$. We note that in our case, $\boldsymbol{f}_S = 0$, the drift reads $\boldsymbol{\mu} = \tfrac{1}{2}(\nabla \cdot \boldsymbol{\Sigma})^{\mathsf{T}} + \boldsymbol{\Sigma}\nabla \log p_{T-t}$, and the SDE simplifies with Stratonovich notations equation 3.12.

## G.3 MARGINAL DENSITY FROM FEYNMAN-KAC REPRESENTATION

The Appendix D of Huang et al. (2021) treats the general case of multiplicative noise. It states that

$$p_0(\boldsymbol{x}) = \mathbb{E}\left[p_T(\overrightarrow{\boldsymbol{x}}_T)\exp\left(\int_0^T (-\nabla \cdot \boldsymbol{\mu}(\overrightarrow{\boldsymbol{x}}_s, T-s) + \nabla \cdot \tfrac{1}{2}(\nabla \cdot \boldsymbol{\Sigma})^{\mathsf{T}}(\overrightarrow{\boldsymbol{x}}_s, T-s))ds\right)\Big|\overrightarrow{\boldsymbol{x}}_0 = \boldsymbol{x}\right], \tag{G.11}$$

$$= \mathbb{E}\left[p_T(\overrightarrow{\boldsymbol{x}}_T)\exp\left(-\int_0^T \nabla \cdot (\boldsymbol{\mu} - \tfrac{1}{2}(\nabla \cdot \boldsymbol{\Sigma})^{\mathsf{T}})(\overrightarrow{\boldsymbol{x}}_s, T-s)ds\right)\Big|\overrightarrow{\boldsymbol{x}}_0 = \boldsymbol{x}\right], \tag{G.12}$$

where

$$\begin{align}
d\overrightarrow{\boldsymbol{x}}_s &= -\tilde{\boldsymbol{\mu}}(\overrightarrow{\boldsymbol{x}}_s, T-s)ds + \mathbf{G}(\overrightarrow{\boldsymbol{x}}_s, T-s)d\boldsymbol{B}_s', \tag{G.13}\\
\tilde{\boldsymbol{\mu}}(\boldsymbol{x}, t) &= \boldsymbol{\mu}(\boldsymbol{x}, t) - (\nabla \cdot \boldsymbol{\Sigma})^{\mathsf{T}}(\boldsymbol{x}), \tag{G.14}
\end{align}$$

and $\boldsymbol{B}_s'$ is a Brownian motion.

**Remark 3.** *In our case, $\tilde{\boldsymbol{\mu}}(\boldsymbol{x}, t) = \boldsymbol{\mu}(\boldsymbol{x}, t) - (\nabla \cdot \boldsymbol{\Sigma})^{\mathsf{T}}(\boldsymbol{x}) = \boldsymbol{\Sigma}(\boldsymbol{x})\nabla \log p_{T-t}(\boldsymbol{x}) - \tfrac{1}{2}(\nabla \cdot \boldsymbol{\Sigma})^{\mathsf{T}}(\boldsymbol{x})$.*

**Remark 4.** *Note that $\tilde{\boldsymbol{\mu}} - \boldsymbol{\mu} = -(\nabla \cdot \boldsymbol{\Sigma})^{\mathsf{T}} = -(\nabla \cdot \boldsymbol{\Sigma})^{\mathsf{T}}$ is twice the Itō to Stratonovich correction of the backward SDE equation G.8 (see Lemma D.1.2). It is expected since this SDE can be reversed in time once written with Stratonovich notations equation 3.12 (Kunita, 1997). Then, changing back from Stratonovich to Itō notations but with a different sign in front of the drift, we obtain the forward SDE equation G.14 verified by $\overrightarrow{\boldsymbol{x}}_s$ including twice the Itō to Stratonovich correction.*

## G.4 CHANGE OF MEASURE AND JENSEN'S INEQUALITY

From the Feynman-Kac representation equation G.12 and Jensen's inequality, we obtain an ELBO as in Huang et al. (2021).

Let $(\Omega, \mathcal{F}, \mathbb{P})$ be the underlying probability space for which $\boldsymbol{B}'$ is a Brownian motion. Suppose $\mathbb{Q}$ is another probability measure on $(\Omega, \mathcal{F})$ equivalent to $\mathbb{P}$ (i.e., they have the same measure zero sets). We can hence apply the change-of-measure

$$p_0(\boldsymbol{x}) = \mathbb{E}\left[\frac{d\mathbb{P}}{d\mathbb{Q}}p_T(\overrightarrow{\boldsymbol{x}}_T)\exp\left(-\int_0^T \nabla \cdot (\boldsymbol{\mu} - \tfrac{1}{2}(\nabla \cdot \boldsymbol{\Sigma})^{\mathsf{T}})(\overrightarrow{\boldsymbol{x}}_s, T-s)ds\right)\Big|\overrightarrow{\boldsymbol{x}}_0 = \boldsymbol{x}\right] \tag{G.15}$$

Then, we apply Jensen's inequality:

$$\log p_0(\boldsymbol{x}) \geqslant \underbrace{\mathbb{E}\left[\log\frac{d\mathbb{P}}{d\mathbb{Q}} + \log p_T(\vec{\boldsymbol{x}}_T) - \int_0^T \nabla\cdot(\boldsymbol{\mu} - \tfrac{1}{2}(\nabla\cdot\boldsymbol{\Sigma})^\intercal)(\vec{\boldsymbol{x}}_s, T-s)ds \Big| \vec{\boldsymbol{x}}_0 = \boldsymbol{x}\right]}_{=\mathcal{E}^\infty} .. \tag{G.16}$$

Compared to Huang et al. (2021), we have the additional term $-\tfrac{1}{2}(\nabla\cdot\boldsymbol{\Sigma})^\intercal$, that is, $-\tfrac{1}{2}(\nabla\cdot\boldsymbol{\Sigma})^\intercal$.

## G.5 GIRSANOV THEOREM

Huang et al. (2021) apply the Girsanov theorem to the following forward SDE equation (17) of Huang et al. (2021)):

$$d\vec{\boldsymbol{x}}_s = (-\boldsymbol{\mu} + \mathbf{G}\boldsymbol{a})ds + \mathbf{G}d\hat{\boldsymbol{B}}_s, \tag{G.17}$$

since the Itō to Stratonovich correction $\tfrac{1}{2}(\boldsymbol{\mu} - \tilde{\boldsymbol{\mu}}) = \tfrac{1}{2}(\nabla\cdot\boldsymbol{\Sigma})^\intercal$ is zero in Huang et al. (2021). However, it is not the case in MSGM and here we use the Girsanov theorem to this forward SDE instead:

$$d\vec{\boldsymbol{x}}_s = (-\tilde{\boldsymbol{\mu}} + \mathbf{G}\boldsymbol{a})ds + \mathbf{G}d\hat{\boldsymbol{B}}_s. \tag{G.18}$$

The Girsanov theorem (Oksendal, 1998, Theorem 8.6.3) states the following. Let $\hat{B}$ be an Itō process solving

$$d\hat{\boldsymbol{B}}_s = -\boldsymbol{a}(\omega, s)ds + d\boldsymbol{B}'_s, \tag{G.19}$$

for $\omega \in \Omega$ and $\hat{\boldsymbol{B}}_0 = 0$ where $\boldsymbol{a}$ satisfies the Novikov's condition. Then $\hat{\boldsymbol{B}}$ is a Brownian motion with respect to $\mathbb{Q}$ and :

$$\mathbb{E}\left[\log\frac{d\mathbb{P}}{d\mathbb{Q}}\Big|\vec{\boldsymbol{x}}_0 = \boldsymbol{x}\right] = \mathbb{E}\left[\int_0^T \boldsymbol{a}(\omega, s)\cdot d\boldsymbol{B}'_s - \tfrac{1}{2}\int_0^T \|\boldsymbol{a}(\omega, s)\|_2^2 ds\Big|\vec{\boldsymbol{x}}_0 = \boldsymbol{x}\right], \tag{G.20}$$

$$= -\tfrac{1}{2}\int_0^T \mathbb{E}_{\vec{\boldsymbol{x}}_s}\left[\|\boldsymbol{a}(\omega, s)\|_2^2\Big|\vec{\boldsymbol{x}}_0 = \boldsymbol{x}\right]ds, \tag{G.21}$$

since $T \mapsto \int_0^T \boldsymbol{a}(\omega, s)\cdot d\boldsymbol{B}'_s$ is a martingale and thus $\mathbb{E}\left[\int_0^T \boldsymbol{a}(\omega, s)\cdot d\boldsymbol{B}'_s\right] = 0$ (Oksendal, 1998, Theorem 3.2.1).

## G.6 ELBO EVALUATION

Equation G.21 enable us to evaluate the ELBO $\mathcal{E}^\infty$ given by equation G.16. To evaluate the divergence term, we note that:

$$(\boldsymbol{\mu} - \tfrac{1}{2}(\nabla\cdot\boldsymbol{\Sigma})^\intercal)(\boldsymbol{x}, T-s) = -\boldsymbol{f}_S(\boldsymbol{x}) + \tfrac{1}{2}(\nabla\cdot\boldsymbol{\Sigma})^\intercal(\boldsymbol{x}) + \boldsymbol{\Sigma}(\boldsymbol{x})s_{\boldsymbol{\theta}}(\boldsymbol{x}, s) - \tfrac{1}{2}(\nabla\cdot\boldsymbol{\Sigma})^\intercal(\boldsymbol{x}), \tag{G.22}$$

$$= -\boldsymbol{f}_S(\boldsymbol{x}) + \mathbf{G}(\boldsymbol{x})\boldsymbol{a}_{\boldsymbol{\theta}}(\boldsymbol{x}, s). \tag{G.23}$$

Then, the ELBO simplifies to:

$$\mathcal{E}^\infty(\boldsymbol{x}) = \mathbb{E}\left[\log\frac{d\mathbb{P}}{d\mathbb{Q}}\Big|\vec{\boldsymbol{x}}_0 = \boldsymbol{x}\right] + \mathbb{E}\left[\log p_T(\vec{\boldsymbol{x}}_T)\Big|\vec{\boldsymbol{x}}_0 = \boldsymbol{x}\right] \tag{G.24}$$

$$+ \int_0^T \mathbb{E}_{\vec{\boldsymbol{x}}_s}\left[-\nabla\cdot(\boldsymbol{\mu} - \tfrac{1}{2}(\nabla\cdot\boldsymbol{\Sigma})^\intercal)\Big|\vec{\boldsymbol{x}}_0 = \boldsymbol{x}\right]ds,$$

$$= \mathbb{E}\left[\log p_T(\vec{\boldsymbol{x}}_T)\Big|\vec{\boldsymbol{x}}_0 = \boldsymbol{x}\right] \tag{G.25}$$

$$- \int_0^T \mathbb{E}_{\vec{\boldsymbol{x}}_s}\left[\tfrac{1}{2}\|\boldsymbol{a}_{\boldsymbol{\theta}}(\vec{\boldsymbol{x}}_s, s)\|_2^2 + \nabla\cdot(\mathbf{G}(\vec{\boldsymbol{x}}_s)\boldsymbol{a}_{\boldsymbol{\theta}}(\vec{\boldsymbol{x}}_s, s) - \boldsymbol{f}_S(\vec{\boldsymbol{x}}_s))\Big|\vec{\boldsymbol{x}}_0 = \boldsymbol{x}\right]ds.$$

$$\square$$

We recall that in our case, $\boldsymbol{f}_S$ cancels out. The first term $\mathbb{E}\left[\log p_T(\vec{\boldsymbol{x}}_T)\big|\vec{\boldsymbol{x}}_0 = \boldsymbol{x}\right]$ is a constant w.r.t. to $\boldsymbol{\theta}$. So, if when maximizing the ELBDO, this term has no effect on the optimization procedure. Therefore, even with our multiplicative noise, the minimization of the ELBO corresponds precisely to Implicit Score Matching (ISM), which is itself equivalent to Explicit Score Matching (ESM), Sliced Score Matching (SSM) and Denoising Score Matching (DSM) (Huang et al., 2021).

## G.7 FROM ELBO TO OUR SSM LOSS

Here we show how to derive our practical SSM loss equation 3.14 from Theorem 3.4.1. We assume the skew-symmetry condition A1 and zero Stratonovich drift, i.e. $f_s = 0$. The theorem states the

$$p_0(\boldsymbol{x}_0|\boldsymbol{\theta}) \geqslant \mathcal{E}_\infty(\boldsymbol{x}_0|\boldsymbol{\theta}) := C(\boldsymbol{x}_0) - \mathcal{L}_\infty(\boldsymbol{x}_0|\boldsymbol{\theta}) \tag{G.26}$$

with $C$ being a constant with respect to the parameters $\boldsymbol{\theta}$ to be learned. More precisely,

$$C(\boldsymbol{x}_0) = \mathbb{E}\left[\log p_T(\vec{\boldsymbol{x}}_T)\big|\vec{\boldsymbol{x}}_0 = \boldsymbol{x}_0\right], \tag{G.27}$$

$$\mathcal{L}_\infty(\boldsymbol{x}_0|\boldsymbol{\theta}) = \int_0^T \mathbb{E}_{\vec{\boldsymbol{x}}_s}\left[\tfrac{1}{2}\|\boldsymbol{a}_\theta(\vec{\boldsymbol{x}}_s, s)\|^2 + \nabla \cdot (\mathbf{G}(\vec{\boldsymbol{x}}_s)\boldsymbol{a}_\theta(\vec{\boldsymbol{x}}_s, s) - f_S(\vec{\boldsymbol{x}}_s))\Big|\vec{\boldsymbol{x}}_0 = \boldsymbol{x}_0\right]ds. \tag{G.28}$$

Then, we average over the data $\boldsymbol{x}_0$ to obtain the following lower bound for the likelihood of the dataset:

$$\mathbb{E}_{\boldsymbol{x}_0}p_0(\boldsymbol{x}_0|\boldsymbol{\theta}) \geqslant \mathbb{E}_{\boldsymbol{x}_0}\mathcal{E}_\infty(\boldsymbol{x}_0|\boldsymbol{\theta}) = \mathbb{E}_{\boldsymbol{x}_0}C(\boldsymbol{x}_0) - \mathbb{E}_{\boldsymbol{x}_0}\mathcal{L}_\infty(\boldsymbol{x}_0|\boldsymbol{\theta}). \tag{G.29}$$

Our objective is to find the neural network parameters $\boldsymbol{\theta}$, that try to maximize the likelihood of the data set, $\mathbb{E}_{\boldsymbol{x}_0}p_0(\boldsymbol{x}_0|\boldsymbol{\theta})$. Since $\mathbb{E}_{\boldsymbol{x}_0}C(\boldsymbol{x}_0)$ is a constant with respect to $\boldsymbol{\theta}$, we maximize $-\mathbb{E}_{\boldsymbol{x}_0}\mathcal{L}_\infty(\boldsymbol{x}_0|\boldsymbol{\theta})$. Let us explicit the two terms above with the Hutchinson trick, $\mathbb{E}_{\mathbf{v}\sim\mathrm{Rad}(d)}[\mathbf{v}\mathbf{v}^\intercal] = \boldsymbol{I}_d$ (Song et al., 2020)

$$\mathbb{E}_{\boldsymbol{x}_0}C(\boldsymbol{x}_0) = \mathbb{E}_{\boldsymbol{x}_T}\left[\log p_T(\vec{\boldsymbol{x}}_T)\right], \tag{G.30}$$

$$\mathbb{E}_{\boldsymbol{x}_0}\mathcal{L}_\infty(\boldsymbol{x}_0|\boldsymbol{\theta}) = \int_0^T \mathbb{E}_{\vec{\boldsymbol{x}}_s}\left[\tfrac{1}{2}\|\boldsymbol{a}_\theta(\vec{\boldsymbol{x}}_s, s)\|^2 + \nabla \cdot (\mathbf{G}(\vec{\boldsymbol{x}}_s)\boldsymbol{a}_\theta(\vec{\boldsymbol{x}}_s, s) - f_S(\vec{\boldsymbol{x}}_s))\right]ds, \tag{G.31}$$

$$= T\int_0^T \mathbb{E}_{\vec{\boldsymbol{x}}_s}\left[\tfrac{1}{2}\|\boldsymbol{a}_\theta(\vec{\boldsymbol{x}}_s, s)\|^2 + \nabla \cdot (\mathbf{G}(\vec{\boldsymbol{x}}_s)\boldsymbol{a}_\theta(\vec{\boldsymbol{x}}_s, s))\right]\tfrac{1}{T}ds, \tag{G.32}$$

$$= T\mathbb{E}_{s\sim\mathcal{U}[0,T]}\mathbb{E}_{\vec{\boldsymbol{x}}_s}\left[\tfrac{1}{2}\|\boldsymbol{a}_\theta(\vec{\boldsymbol{x}}_s, s)\|^2 + \nabla \cdot (\mathbf{G}(\vec{\boldsymbol{x}}_s)\boldsymbol{a}_\theta(\vec{\boldsymbol{x}}_s, s))\right], \tag{G.33}$$

$$= T\mathbb{E}_{s\sim\mathcal{U}[0,T]}\mathbb{E}_{\vec{\boldsymbol{x}}_s}\left[\tfrac{1}{2}\|\boldsymbol{a}_\theta(\vec{\boldsymbol{x}}_s, s)\|^2 + \nabla \cdot (\mathbb{E}_{\mathbf{v}\sim\mathrm{Rad}(d)}[\mathbf{v}\mathbf{v}^\intercal]\mathbf{G}(\vec{\boldsymbol{x}}_s)\boldsymbol{a}_\theta(\vec{\boldsymbol{x}}_s, s))\right], \tag{G.34}$$

$$= T\mathbb{E}_{s\sim\mathcal{U}[0,T]}\mathbb{E}_{\vec{\boldsymbol{x}}_s}\mathbb{E}_{\mathbf{v}\sim\mathrm{Rad}(d)}\left[\tfrac{1}{2}\|\boldsymbol{a}_\theta(\vec{\boldsymbol{x}}_s, s)\|^2 + (\mathbf{v} \cdot \nabla)(\mathbf{G}(\vec{\boldsymbol{x}}_s)\boldsymbol{a}_\theta(\vec{\boldsymbol{x}}_s, s)) \cdot \mathbf{v}\right]. \tag{G.35}$$

$$= T\mathcal{L}_{\mathrm{SSM}}(\boldsymbol{\theta}). \tag{G.36}$$

Therefore, maximizing the ELBO, $\mathbb{E}_{\boldsymbol{x}_0}\mathcal{E}_\infty(\boldsymbol{x}_0|\boldsymbol{\theta}) = \mathbb{E}_{\boldsymbol{x}_0}C(\boldsymbol{x}_0) - \mathbb{E}_{\boldsymbol{x}_0}\mathcal{L}_\infty(\boldsymbol{x}_0|\boldsymbol{\theta})$, is equivalent to minimizing our practical score-matching loss, $\mathcal{L}_{\mathrm{SSM}}(\boldsymbol{\theta})$.

## G.8 REMARK ON THE SCORE PARAMETRIZATION

Following Huang et al. (2021), we directly model $\mathbf{G}(\overleftarrow{\boldsymbol{x}}_t)^\intercal \nabla \log p_s(\boldsymbol{x})$ by a neural network $\boldsymbol{a}_\theta(\boldsymbol{x}, s)$. If needed, the projected score, $\nabla_\perp \log p_s$, can be retrieved directly from $\boldsymbol{a}_\theta$ as shown below. Note that the full score,

$$\nabla \log p_s = \nabla_\perp \log p_s + (\boldsymbol{x}^n \cdot \nabla)\log p_s, \tag{G.37}$$

involves a radial term, $(\boldsymbol{x}^n \cdot \nabla)\log p_s$ that cannot be directly estimated in MSGM.

**Proposition G.1.** *We assume that assumptions A1 and A2 hold, and that we have an approximation,* $a_{\theta}$, *of the scaled score and an orthonormal basis* $u_2(x), \ldots, u_d(x)$ *of* $x^{\perp}$, *that we concatenate in* $U(x) = [u_2(x), \ldots, u_d(x)] \in \mathbb{R}^{d \times (d-1)}$. *Then,*

$$[U^{\intercal}(x)\Sigma(x)U(x)]^{-1}U^{\intercal}(x)\mathbf{G}(x)a_{\theta}(x,s). \tag{G.38}$$

*approximates the projected score*

$$U^{\intercal}(x)\nabla_{\perp}\log p_s(x). \tag{G.39}$$

*Proof.* Since $\mathbb{R}^d = \mathbb{R}x^n \overset{\perp}{\oplus} x^{\perp}$, we have $I_d = x^n(x^n)^{\intercal} + U(x)U^{\intercal}(x)$. Using $\Sigma(x)x^n = 0$, we obtain

$$
\begin{aligned}
U^{\intercal}(x)\mathbf{G}(x)a_{\theta}(x,s) &\approx U^{\intercal}(x)\mathbf{G}(x)\mathbf{G}(x)^{\intercal}\nabla\log p_s(x), &&\text{(G.40)}\\
&= U^{\intercal}(x)\Sigma(x)[x^n(x^n)^{\intercal} + U(x)U^{\intercal}(x)]\nabla\log p_s(x), &&\text{(G.41)}\\
&= U^{\intercal}(x)\Sigma(x)U(x)U^{\intercal}(x)\nabla\log p_s(x), &&\text{(G.42)}\\
&= U^{\intercal}(x)\Sigma(x)U(x)U^{\intercal}(x)\nabla_{\perp}\log p_s(x). &&\text{(G.43)}
\end{aligned}
$$

$U^{\intercal}(x)\Sigma(x)U(x) \in \mathbb{R}^{(d-1)\times(d-1)}$ is full rank, so

$$U^{\intercal}(x)\nabla_{\perp}\log p_s(x) \approx [U^{\intercal}(x)\Sigma(x)U(x)]^{-1}U^{\intercal}(x)\mathbf{G}(x)a_{\theta}(x,s). \tag{G.44}$$

$\square$

It is also possible to model the score, $\nabla\log p_s(x)$ directly by a neural network, $s_{\theta}(x,s)$ using the following score-matching loss:

$$
\mathcal{L}_{\text{SSM}}(\theta) = \mathbb{E}_{s\sim\mathcal{U}[0,T]}\mathbb{E}_{\overrightarrow{x}_s}\mathbb{E}_{\mathbf{v}\sim\text{Rad}(d)}\left[\tfrac{1}{2}\|a_{\theta}(\overrightarrow{x}_s,s)\|^2 + (\mathbf{v}\cdot\nabla)(\mathbf{G}(\overrightarrow{x}_s)a_{\theta}(\overrightarrow{x}_s,s) - f_S(\overrightarrow{x}_s))\cdot\mathbf{v}\right],
\tag{G.45}
$$

$$
= \mathbb{E}_{s\sim\mathcal{U}[0,T]}\mathbb{E}_{\overrightarrow{x}_s}\mathbb{E}_{\mathbf{v}\sim\text{Rad}(d)}\left[\tfrac{1}{2}\|s_{\theta}(\overrightarrow{x}_s,s)\|^2_{\Sigma(\overrightarrow{x}_s)} + (\mathbf{v}\cdot\nabla)(\Sigma(\overrightarrow{x}_s)s_{\theta}(\overrightarrow{x}_s,s) - f_S(\overrightarrow{x}_s))\cdot\mathbf{v}\right]ds.
\tag{G.46}
$$

However, for any $\alpha\in\mathbb{R}$, $\mathcal{L}_{\text{SSM}}(\theta)(s_{\theta}) = \mathcal{L}_{\text{SSM}}(\theta)(s_{\theta} + \alpha x^n)$, i.e. our loss function is insensible to the radial component of the score $(x^n\cdot\nabla)\log p_s$. Therefore, our MSGM framework does not provide estimation for the radial score $(x^n\cdot\nabla)\log p_s$. Moreover, the optimization problem parametrized by $s_{\theta}$ is ill-defined, and the loss should probably be regularized as follows:

$$
\mathcal{L}^{\text{reg}}_{\text{SSM}}(\theta) = \mathcal{L}_{\text{SSM}}(\theta) + \gamma\mathbb{E}_{s\sim\mathcal{U}[0,T]}\mathbb{E}_{\overrightarrow{x}_s}\left[(x\cdot s_{\theta})^2\right], \tag{G.47}
$$

with $\gamma > 0$ large, says $\gamma = 10^6$.

### G.9 GIRSANOV THEOREM IN THE TRANSPORT NOISE LITERATURE

Following the work done by Huang et al. (2021) for additive noise, we have relied on the Girsanov theorem (Oksendal, 1998) to prove the equivalence between score matching and ELBO maximization for MSGM. Girsanov theorem is widely used, we may cite here its recent uses in the transport noise literature. In a Bayesian context, Cotter et al. (2020a; 2023); González et al. (2025); Singh et al. (2025) introduce nudging in their particle filter. Also used with other type of noises, nudging biases the noise to make the solution closer to the observations. Similarly, in our case, the weighted score, $a_{\theta}(\overleftarrow{x}_t, T-t)$, biases the noise, $d\overleftarrow{B}_t/dt$, in our backward SDE to make its solution closer to the forward SDE solution (see equation 3.12). This noise change is the core of Girsanov theorem (see equation G.19). Resseguier (2023) also proposed to fit a parametric model for the transport noise by maximum likelihood estimation.

## H COMPARISON WITH DIFFUSIONS ON RIEMANNIAN MANIFOLDS

This appendix describes the similarities between MSGM on $\mathbb{R}^d$ and SGMs on manifolds. To introduce the subject, we first recall some theoretical elements related to Riemannian manifolds. The link with SGMs on manifolds also suggests a particular neural network architecture that we exploit in this work.

## H.1 RIEMANNIAN MANIFOLDS AND DIFFERENTIATION

This section is devoted to a brief introduction to Riemannian manifolds and the associated differential calculus. For a more comprehensive discussion, we refer to Lee (2018). Let $\mathcal{M}$ be a smooth $n$-dimensional embedded submanifold of $\mathbb{R}^d$, where $n \leq d$. For any $\boldsymbol{x} \in \mathcal{M}$ we denote by $T_x\mathcal{M}$ the tangential (linear) space of $\mathcal{M}$ at $\boldsymbol{x}$. We denote by $g$ a Riemannian metric on $\mathcal{M}$, which assigns to each $\boldsymbol{x} \in \mathcal{M}$ an inner product

$$g_x : T_x\mathcal{M} \times T_x\mathcal{M} \to \mathbb{R}.$$

In the case of a smooth embedded manifold in the Euclidean space, the induced metric is given by

$$g_x(u, v) = \langle u, v \rangle_{\mathbb{R}^n}, \quad \forall u, v \in T_x\mathcal{M}.$$

This makes $(\mathcal{M}, g)$ a Riemannian manifold. Let $\{\boldsymbol{e}^{(1)}, \ldots, \boldsymbol{e}^{(n)}\}$ be an orthonormal basis of $T_x\mathcal{M}$. Then, the orthogonal projection onto $T_x\mathcal{M}$ is the linear operator $P_x : \mathbb{R}^d \to T_x\mathcal{M}$ that satisfies

$$P_x(v) = \arg \min_{w \in T_x\mathcal{M}} \|v - w\|_{\mathbb{R}^d} = \sum_{i=1}^{n} \langle v, \boldsymbol{e}^{(i)} \rangle \boldsymbol{e}^{(i)}.$$

While the concept of Riemannian gradients can be derived for general manifolds, here we limit ourselves to the simpler presentation of embedded manifolds in the Euclidean space. In this setup, the Riemannian manifold can be defined as the classical gradient projected to the tangential space. In particular, for $f : \mathbb{R}^d \to \mathbb{R}$ smooth, its *Riemannian gradient* can be computed as

$$\nabla_{\mathcal{M}} f(\boldsymbol{x}) = P_x(\nabla f(\boldsymbol{x})),$$

where $\nabla f(\boldsymbol{x})$ is the Euclidean gradient. Furthermore, we want to define the Riemannian divergence in this framework. For a tangent vector field $\mathbf{f} : \mathcal{M} \to \mathbb{R}^d$ with $\mathbf{f}(\boldsymbol{x}) \in T_x\mathcal{M}$, the *Riemannian divergence* is given as

$$\mathrm{div}_{\mathcal{M}} \mathbf{f}(\boldsymbol{x}) = \sum_{i=1}^{n} \langle \partial_{\boldsymbol{e}^{(i)}} \mathbf{f}(\boldsymbol{x}), \boldsymbol{e}^{(i)} \rangle,$$

where $\partial_{\boldsymbol{e}^{(i)}} \mathbf{f}$ is the Euclidean directional derivative. Finally, the *Laplace-Beltrami operator* $\Delta_{\mathcal{M}}$ can be defined as

$$\Delta_{\mathcal{M}} f = \mathrm{div}_{\mathcal{M}}(\nabla_{\mathcal{M}} f),$$

which generalizes the Laplacian to $\mathcal{M}$.

In the special case that $\mathcal{M} = r\mathbb{S}^{d-1}$, for a radius $r > 0$ then $n = d - 1$ and

$$T_x\mathcal{M} = T_x r\mathbb{S}^{d-1} = \{v \in \mathbb{R}^d \mid \langle v, \boldsymbol{x} \rangle = 0\},$$

and $P_x(v) = v - \frac{1}{r^2}\langle v, \boldsymbol{x} \rangle \boldsymbol{x}$. Since $\boldsymbol{x}^n = \frac{\boldsymbol{x}}{r}$ we obtain $P_x(v) = (I - \boldsymbol{x}^n(\boldsymbol{x}^n)^{\mathsf{T}})v$ and as a result

$$\nabla_{\mathcal{M}} f(\boldsymbol{x}) = P_x(\nabla f(\boldsymbol{x})) = (I - \boldsymbol{x}^n(\boldsymbol{x}^n)^{\mathsf{T}})\nabla f(\boldsymbol{x}) = \nabla_{\perp} f(\boldsymbol{x}). \tag{H.1}$$

Regarding the Riemannian divergence, we note that $\boldsymbol{x}^n, \boldsymbol{e}^{(1)}, \ldots, \boldsymbol{e}^{(n)}$ defines an orthonormal basis of $\mathbb{R}^d$. By Lemma D.2.1

$$\nabla \cdot f(\boldsymbol{x}) = (\boldsymbol{x}^n \cdot \nabla)(\boldsymbol{x}^n \cdot f(\boldsymbol{x})) + \nabla_{\perp} \cdot f(\boldsymbol{x}).$$

For $\mathbf{f}(\boldsymbol{x}) \in T_x\mathcal{M}$, we have $\mathbf{f}(\boldsymbol{x}) \cdot \boldsymbol{x}^n = 0$. Thus:

$$\nabla_{\perp} \cdot \mathbf{f}(\boldsymbol{x}) = \nabla \cdot \mathbf{f}(\boldsymbol{x}) - \underbrace{(\boldsymbol{x}^n \cdot \nabla)(\mathbf{f}(\boldsymbol{x}) \cdot \boldsymbol{x}^n)}_{=0} = \nabla \cdot \mathbf{f}(\boldsymbol{x}).$$

Differentiating the tangency condition $\mathbf{f}(\boldsymbol{x}) \cdot \boldsymbol{x}^n = 0$ along $\boldsymbol{x}^n$ leads

$$0 = \partial_{\boldsymbol{x}^n}(\mathbf{f}(\boldsymbol{x}) \cdot \boldsymbol{x}^n) = \langle \partial_{\boldsymbol{x}^n} \mathbf{f}(\boldsymbol{x}), \boldsymbol{x}^n \rangle + \langle \mathbf{f}(\boldsymbol{x}), \partial_{\boldsymbol{x}^n} \boldsymbol{x}^n \rangle.$$

Since $\partial_{\boldsymbol{x}^n} \boldsymbol{x}^n = 0$, we conclude that $\langle \partial_{\boldsymbol{x}^n} \mathbf{f}(\boldsymbol{x}), \boldsymbol{x}^n \rangle = 0$. Finally, expanding $\nabla \cdot \mathbf{f}(\boldsymbol{x})$ in $\boldsymbol{x}^n, \boldsymbol{e}^{(1)}, \ldots, \boldsymbol{e}^{(n)}$ leads to

$$\nabla_{\perp} \cdot \mathbf{f}(\boldsymbol{x}) = \nabla \cdot \mathbf{f}(\boldsymbol{x}) = \sum_{i=1}^{n} \langle \partial_{\boldsymbol{e}^{(i)}} \mathbf{f}(\boldsymbol{x}), \boldsymbol{e}^{(i)} \rangle + \underbrace{\langle \partial_{\boldsymbol{x}^n} \mathbf{f}(\boldsymbol{x}), \boldsymbol{x}^n \rangle}_{=0} = \mathrm{div}_{\mathcal{M}} \mathbf{f}(\boldsymbol{x}). \tag{H.2}$$

In our setting $\mathrm{Im}(\boldsymbol{\Sigma}(\boldsymbol{x})) = \boldsymbol{x}^{\perp} = T_x\mathcal{M}$. Hence, the right-hand side of the Fokker-Planck equation 3.4

$$\mathrm{div}_{\mathcal{M}}(\boldsymbol{\Sigma}(\boldsymbol{x})\nabla_{\mathcal{M}} f(\boldsymbol{x})) = \nabla_{\perp} \cdot (\boldsymbol{\Sigma}(\boldsymbol{x})\nabla_{\perp} f(\boldsymbol{x})), \tag{H.3}$$

generalizes the notion of a divergence-form operator to the manifold setup.

## H.2 CONDITIONAL DIFFUSIONS ON SCALED $d$-SPHERES

Several authors have recently developed SGM on Riemannian manifolds (De Bortoli et al., 2022; Huang et al., 2022; Benton et al., 2024) in order to generate data lying on a particular manifold. Clearly different, our goal is more classical: generating data in $\mathbb{R}^d$. However, each solution path of our forward and backward SDE lies on its scaled $d$-sphere $\|\vec{\boldsymbol{x}}_0\|\mathbb{S}^{d-1}$. Clearly, $d$-spheres are particular cases of Riemannian manifolds and possibly the most studied. De Bortoli et al. (2022) describes diffusions of $\vec{\boldsymbol{x}}^n$ in the $d$-sphere $\mathbb{S}^{d-1}$. The simplest one involves a Brownian motion on the $d$-sphere that converges to the uniform distribution on the $d$-sphere, $p_\infty^n$. Unfortunately, this appealing proposal does not directly apply to our framework: the Brownian motion on the $d$-sphere is not a solution of our forward SDE of $\vec{\boldsymbol{x}}^n$. Indeed, in general, there exists $\vec{\boldsymbol{x}}^n \in \mathbb{S}^{d-1}$ such that $\boldsymbol{\Sigma}(\boldsymbol{x}^n) = \sum_{k=1}^{d}(\boldsymbol{G}^k\boldsymbol{x}^n)(\boldsymbol{G}^k\boldsymbol{x}^n)^\intercal \neq I_{\mathbb{S}^{d-1}}$. So, the Fokker-Planck equation of De Bortoli et al. (2022),

$$\frac{\partial}{\partial s}p^n(\boldsymbol{x}^n) \;=\; \mathrm{div}_{\mathbb{S}^{d-1}}(\nabla_{\mathbb{S}^{d-1}}p^n(\boldsymbol{x}^n)), \quad \forall \boldsymbol{x}^n \in \mathbb{S}^{d-1}, \tag{H.4}$$

and our Fokker-Planck equation for the direction,

$$\frac{\partial}{\partial s}p^n(\boldsymbol{x}^n) \;=\; \mathrm{div}_{\mathbb{S}^{d-1}}(\boldsymbol{\Sigma}(\boldsymbol{x}^n)\nabla_{\mathbb{S}^{d-1}}p^n(\boldsymbol{x}^n)), \quad \forall \boldsymbol{x}^n \in \mathbb{S}^{d-1}, \tag{H.5}$$

do not coincide. However, the analyses from the SGM-on-manifold community on the finite-time distribution, its score, approximations, and score-matching losses choices could certainly facilities the MSGM training process in the future.

In our case, the norm of solution being constant along path, we can write both the forward and the backward equations of the direction on the unit $d$-sphere from equation 3.1 and equation 3.12:

$$d\vec{\boldsymbol{x}}_t^n \;=\; \mathbf{G}(\vec{\boldsymbol{x}}_t^n) \circ d\vec{\boldsymbol{B}}_t, \tag{H.6}$$

$$d\overleftarrow{\boldsymbol{x}}_t^n \;=\; \frac{1}{\|\overleftarrow{\boldsymbol{x}}_t\|}\mathbf{G}(\overleftarrow{\boldsymbol{x}}_t)\left(\mathbf{G}(\overleftarrow{\boldsymbol{x}}_t)^\intercal \nabla \log p_{T-t}\left(\|\overleftarrow{\boldsymbol{x}}_t\|\overleftarrow{\boldsymbol{x}}_t^n\right)dt + \circ d\overleftarrow{\boldsymbol{B}}_t\right), \tag{H.7}$$

$$\;=\; \mathbf{G}(\overleftarrow{\boldsymbol{x}}_t^n)\left(\mathbf{G}(\overleftarrow{\boldsymbol{x}}_t^n)^\intercal\left(\|\overleftarrow{\boldsymbol{x}}_t\|\nabla_\perp \log p_{T-t}\left(\|\overleftarrow{\boldsymbol{x}}_t\|\overleftarrow{\boldsymbol{x}}_t^n\right)\right)dt + \circ d\overleftarrow{\boldsymbol{B}}_t\right). \tag{H.8}$$

We note that $\|\boldsymbol{x}\|\nabla_\perp = \nabla_{\mathbb{S}^{d-1}} = \partial_{\boldsymbol{x}^n}$ is the Riemannian gradient on the scaled $d$-sphere $\|\boldsymbol{x}\|\mathbb{S}^{d-1}$. Therefore, using $p_s^\otimes$, the density of the couple of variables $(\|\overleftarrow{\boldsymbol{x}}_s\|, \overleftarrow{\boldsymbol{x}}_s^n) \in \mathbb{R}^+ \times \mathbb{S}^{d-1}$,

$$\|\overleftarrow{\boldsymbol{x}}_t\|\nabla_\perp \log p_{T-t}\left(\|\overleftarrow{\boldsymbol{x}}_t\|\overleftarrow{\boldsymbol{x}}_t^n\right) \;=\; \frac{\partial}{\partial \boldsymbol{x}^n}\log p_{T-t}\left(\|\overleftarrow{\boldsymbol{x}}_t\|\overleftarrow{\boldsymbol{x}}_t^n\right) \tag{H.9}$$

$$\;=\; \frac{\partial}{\partial \boldsymbol{x}^n}\log\left(p_{T-t}^\otimes\left(\|\overleftarrow{\boldsymbol{x}}_t^n\|, \overleftarrow{\boldsymbol{x}}_t^n\right)\|\overleftarrow{\boldsymbol{x}}_t\|^{1-d}\right) \tag{H.10}$$

$$\;=\; \frac{\partial}{\partial \boldsymbol{x}^n}\log\left(p_{T-t}^n\left(\overleftarrow{\boldsymbol{x}}_t^n \mid \|\overleftarrow{\boldsymbol{x}}_t^n\|\right)p_{|.|}\left(\|\overleftarrow{\boldsymbol{x}}_t\|\right)\|\overleftarrow{\boldsymbol{x}}_t\|^{1-d}\right), \tag{H.11}$$

$$\;=\; \frac{\partial}{\partial \boldsymbol{x}^n}\log p_{T-t}^n\left(\overleftarrow{\boldsymbol{x}}_t^n \mid \|\overleftarrow{\boldsymbol{x}}_t\|\right) \tag{H.12}$$

$$\;=\; \frac{\partial}{\partial \boldsymbol{x}^n}\log p_{T-t}^n\left(\overleftarrow{\boldsymbol{x}}_t^n \mid \|\overleftarrow{\boldsymbol{x}}_0\|\right) \tag{H.13}$$

$$\;=\; \nabla_{\mathbb{S}^{d-1}}\log p_{T-t}^n\left(\overleftarrow{\boldsymbol{x}}_t^n \mid \|\overleftarrow{\boldsymbol{x}}_0\|\right) \tag{H.14}$$

and finally

$$d\overleftarrow{\boldsymbol{x}}_t^n \;=\; \mathbf{G}(\overleftarrow{\boldsymbol{x}}_t^n)\left(\mathbf{G}(\overleftarrow{\boldsymbol{x}}_t^n)^\intercal \nabla_{\mathbb{S}^{d-1}}\log p_{T-t}^n\left(\overleftarrow{\boldsymbol{x}}_t^n \mid \|\overleftarrow{\boldsymbol{x}}_0\|\right)dt + \circ d\overleftarrow{\boldsymbol{B}}_t\right). \tag{H.15}$$

In contrast, forward and backward SDEs of De Bortoli et al. (2022) read

$$d\vec{\boldsymbol{x}}_t^n \;=\; d\vec{\boldsymbol{B}}_t^{\mathbb{S}^{d-1}}, \tag{H.16}$$

$$d\overleftarrow{\boldsymbol{x}}_t^n \;=\; \nabla_{\mathbb{S}^{d-1}}\log p_{T-t}^n\left(\overleftarrow{\boldsymbol{x}}_t^n\right)dt + d\overleftarrow{\boldsymbol{B}}_t^{\mathbb{S}^{d-1}}, \tag{H.17}$$

where $\overset{\rightarrow}{\boldsymbol{B}}_t^{\mathbb{S}^{d-1}}$ and $\overset{\leftarrow}{\boldsymbol{B}}_t^{\mathbb{S}^{d-1}}$ are Brownian motions on the $d$-sphere. They can be defined from Stroock's representation (Hsu, 2002, Example 3.3.2) as

$$d\overset{\rightarrow}{\boldsymbol{B}}_t^{\mathbb{S}^{d-1}} = (\boldsymbol{I}_d - (\overset{\rightarrow}{\boldsymbol{x}}_t^n)(\overset{\rightarrow}{\boldsymbol{x}}_t^n)^\mathsf{T}) \circ d\overset{\rightarrow}{\boldsymbol{B}}_t, \tag{H.18}$$

$$d\overset{\leftarrow}{\boldsymbol{B}}_t^{\mathbb{S}^{d-1}} = (\boldsymbol{I}_d - (\overset{\leftarrow}{\boldsymbol{x}}_t^n)(\overset{\leftarrow}{\boldsymbol{x}}_t^n)^\mathsf{T}) \circ d\overset{\leftarrow}{\boldsymbol{B}}_t. \tag{H.19}$$

The first main difference with MSGM is that the projection on the tangent plane, $(\boldsymbol{I}_d - (\boldsymbol{x}^n)(\boldsymbol{x}^n)^\mathsf{T})$, (quadratic in $\boldsymbol{x}^n$) is replaced in our approach by $\mathbf{G}(\overset{\leftarrow}{\boldsymbol{x}}_t^n)$ (linear in $\boldsymbol{x}^n$). Accordingly the noise (conditional) covariance, $(\boldsymbol{I}_d - (\boldsymbol{x}^n)(\boldsymbol{x}^n)^\mathsf{T})^2 = (\boldsymbol{I}_d - (\boldsymbol{x}^n)(\boldsymbol{x}^n)^\mathsf{T})$ (projection property), is replaced by $\mathbf{G}(\overset{\leftarrow}{\boldsymbol{x}}_t^n)\mathbf{G}(\overset{\leftarrow}{\boldsymbol{x}}_t^n)^\mathsf{T} = \boldsymbol{\Sigma}(\overset{\leftarrow}{\boldsymbol{x}}_t^n)$. To make our diffusion coincide with equation H.16, we would have to consider

$$\mathbf{G}(\boldsymbol{x}) := \|\boldsymbol{x}\|(\boldsymbol{I}_d - \boldsymbol{x}^n(\boldsymbol{x}^n)^\mathsf{T}), \tag{H.20}$$

which is Lipchitz continuous but nonlinear. As such, the noise covariance would be

$$\boldsymbol{\Sigma}(\boldsymbol{x}) = \|\boldsymbol{x}\|^2 \boldsymbol{I}_d - \boldsymbol{x}\boldsymbol{x}^\mathsf{T}. \tag{H.21}$$

In general, we can hardly expect such a simple form from MSGM noise covariance. However, for the random tensor equation 6.1, we can show (see equation J.10) that:

$$2\mathbb{E}\boldsymbol{\Sigma}(\boldsymbol{x}) = \|\boldsymbol{x}\|^2 \boldsymbol{I}_d - \boldsymbol{x}\boldsymbol{x}^\mathsf{T}. \tag{H.22}$$

In addition, our score involved in the backward SDE equation H.15 depends on the norm $\|\overset{\leftarrow}{\boldsymbol{x}}_t\|$. The norm $\|\overset{\leftarrow}{\boldsymbol{x}}_t\| = \|\overset{\leftarrow}{\boldsymbol{x}}_0\|$ appears as a covariable – with prior distribution $p_{|.|}$ – for the diffusion on the unit $d$-sphere. This is another major difference of our approach compared to SGM on manifolds. Besides, from this point of view, we can better understand how the direction and magnitude are re-coupled during MSGM generation. Along the reverse diffusion, the conditional score direction H.14 will focus along some orientations, counterbalancing the direction equiprobability of the latent space, i.e. reversing the "whitening" of the forward process. On different scaled $d$-sphere $\|x_0\|\mathbb{S}^{d-1}$, the conditional score direction will be oriented differently, pushing along some orientations on some spheres and along other directions on spheres of larger radius. Accordingly, along the backward diffusion, the directions tend to align differently on different hypershperes. The distribution of direction become more and more radius-dependent.

If data samples $\overset{\rightarrow}{\boldsymbol{x}}_0$ are snapshots of a conservative dynamical system, all data points probably have the similar energy $E = \|\overset{\rightarrow}{\boldsymbol{x}}_0\|^2$, i.e. $\mathrm{Var}(E)/\mathbb{E}[E]^2$ is small. All data points are on closed scaled $d$-spheres $\sqrt{E}\,\mathbb{S}^{d-1}$ and our approach becomes even closer to De Bortoli et al. (2022).

### H.3 LINK WITH NEURAL NETWORK ARCHITECTURE

Form equation H.14, we also note that

$$\mathbf{G}(\overset{\leftarrow}{\boldsymbol{x}}_t)^\mathsf{T} \nabla \log p_{T-t}\left(\overset{\leftarrow}{\boldsymbol{x}}_t\right) = \mathbf{G}(\overset{\leftarrow}{\boldsymbol{x}}_t^n)^\mathsf{T} \nabla_{\mathbb{S}^{d-1}} \log p_{T-t}^n\left(\overset{\leftarrow}{\boldsymbol{x}}_t^n \mid \|\overset{\leftarrow}{\boldsymbol{x}}_0\|\right), \tag{H.23}$$

justifying our neural network spherical architecture equation L.33

$$\mathbf{G}(\overset{\leftarrow}{\boldsymbol{x}}_t)^\mathsf{T} \nabla \log p_{T-t}(\overset{\leftarrow}{\boldsymbol{x}}_t) \approx \boldsymbol{a_\theta}(\overset{\leftarrow}{\boldsymbol{x}}_t, T-t) = \tilde{\boldsymbol{a}}_{\boldsymbol{\theta}}\left(\frac{\|\overset{\leftarrow}{\boldsymbol{x}}_t\|}{\|\overset{\leftarrow}{\boldsymbol{x}}_t\|_\epsilon}\overset{\leftarrow}{\boldsymbol{x}}_t^n, \log\|\overset{\leftarrow}{\boldsymbol{x}}_t\|_\epsilon, T-t\right). \tag{H.24}$$

## I ANALYTIC ILLUSTRATIONS ON SIMPLIFIED CASES

### I.1 THE TWO-DIMENSIONAL CASE

We note here that for $d = 2$, we can find an analytic solution for our multiplicative forward SDE. Moreover, it corresponds to the Brownian motion on the circle.

Let us recall this forward SDE:

$$\mathrm{d}\vec{\boldsymbol{x}}(t) = \mathbf{G}(\vec{\boldsymbol{x}}(t)) \circ d\vec{\boldsymbol{B}}_t = \sum_{k=1}^{K} \boldsymbol{G}^k \vec{\boldsymbol{x}}(t) \circ d\vec{B}_t^k = \left(\sum_{k=1}^{K} \boldsymbol{G}^k \circ d\vec{B}_t^k\right) \vec{\boldsymbol{x}}(t), \tag{I.1}$$

In dimension 2,

$$\mathrm{d}\vec{\boldsymbol{x}}(t) = \alpha J \vec{\boldsymbol{x}}(t) \circ d\vec{B}_t^1, \tag{I.2}$$

where $\vec{\boldsymbol{x}} = \begin{pmatrix} \vec{x}_1 \\ \vec{x}_2 \end{pmatrix} \in \mathbb{R}^2$, $J = \begin{pmatrix} 0 & -1 \\ 1 & 0 \end{pmatrix}$ is the $\frac{\pi}{2}$-rotation.

$$\mathrm{d}\begin{pmatrix} \vec{x}_1 \\ \vec{x}_2 \end{pmatrix} = \alpha \begin{pmatrix} -\vec{x}_2 \\ \vec{x}_1 \end{pmatrix} \circ d\vec{B}_t^1, \tag{I.3}$$

Then, in the complex plane, $\vec{x}^C = \vec{x}_1 + i\vec{x}_2 \in \mathbb{C}$, with $i = \sqrt{-1}$ and we have:

$$\mathrm{d}\vec{x}^C(t) = \alpha i \vec{x}^C \circ d\vec{B}_t^1, \tag{I.4}$$

since

$$\mathrm{d}\vec{x}_1 + i\mathrm{d}\vec{x}_2 = \alpha i(\vec{x}_1 + i\vec{x}_2) \circ d\vec{B}_t^1 = \alpha(-\vec{x}_2 + i\vec{x}_1) \circ d\vec{B}_t^1. \tag{I.5}$$

The solution is the Brownian motion on the circle:

$$\vec{x}^C(t) = \vec{x}^C(0) \exp(\alpha i \vec{B}_t^1), \tag{I.6}$$

i.e.

$$\vec{\boldsymbol{x}}(t) = R(\alpha \vec{B}_t^1)\vec{\boldsymbol{x}}(0) = \begin{pmatrix} \cos(\alpha \vec{B}_t^1) & -\sin(\alpha \vec{B}_t^1) \\ \sin(\alpha \vec{B}_t^1) & \cos(\alpha \vec{B}_t^1) \end{pmatrix} \vec{\boldsymbol{x}}(0), \tag{I.7}$$

i.e.

$$\vec{x}_1(t) = \vec{x}_1(0)\cos(\alpha \vec{B}_t^1) - \vec{x}_2(0)\sin(\alpha \vec{B}_t^1), \tag{I.8}$$

$$\vec{x}_2(t) = \vec{x}_1(0)\sin(\alpha \vec{B}_t^1) + \vec{x}_2(0)\cos(\alpha \vec{B}_t^1). \tag{I.9}$$

The key element of the proof was the possibility to write the forward diffusion with a single skew-symmetric matrix in equation I.2. Below we generalize this idea to larger dimension $d \geqslant 2$.

## I.2 TENSOR BUILT FROM A SINGLE SKEW-SYMMETRIC MATRIX

Here we assume that whole tensor $\mathbf{G}$ is built from the same dense skew-symmetric matrix $\boldsymbol{G}^1$ i.e.

$$\boldsymbol{G}^k = \boldsymbol{G}^1, \quad \forall k \in \{1, \dots, d\}, \tag{I.10}$$

with $\boldsymbol{G}^1$ a skew-symmetric matrix. As explained in Section K.2.1, this tensor respect the condition A1 but not the A2. Nevertheless, this case and its analytic solution may be insightful.

### I.2.1 MATRIX EXPONENTIAL

Here the full Brownian matrix $\boldsymbol{Z}$ can be simply factorized as

$$\boldsymbol{Z}_s = \sum_{k=1}^{d} \boldsymbol{G}^k (\vec{B}_s)_k = \boldsymbol{G}^1 \sum_{k=1}^{d} (\vec{B}_s)_k. \tag{I.11}$$

It has the same distribution than

$$\boldsymbol{Z}_s' = \sqrt{d}\,\boldsymbol{G}^1\vec{B}_s', \tag{I.12}$$

with $\vec{B}'$ another single Brownian motion. The forward diffusion simplify to

$$d\vec{\boldsymbol{x}}_s = \sqrt{d}\,\boldsymbol{G}^1\vec{\boldsymbol{x}}_s \circ d\vec{B}_s', \tag{I.13}$$

with solution

$$\vec{\boldsymbol{x}}_s = \exp(\boldsymbol{Z}_s')\vec{\boldsymbol{x}}_0 = \exp\left(\sqrt{d}\,\boldsymbol{G}^1\vec{B}_s'\right)\vec{\boldsymbol{x}}_0, \tag{I.14}$$

since $\boldsymbol{Z}_s'$ and $d\boldsymbol{Z}_s'$ commute.

### I.2.2 DIAGONALIZATION IN THE COMPLEX PLANE

$G^1$ has pure imaginary eigenvalues and can be diagonalized in $\mathbb{C}$ on an orthonormal basis

$$\boldsymbol{G}^1 = \boldsymbol{U}_{\mathbb{C}}(i\boldsymbol{\Lambda})\boldsymbol{U}_{\mathbb{C}}^H, \tag{I.15}$$

with $\boldsymbol{U}_{\mathbb{C}}$ a complex unitary matrix, $\boldsymbol{\Lambda}$ a real diagonal matrix, and the superscript $H$ denotes the conjugate transpose. Then, the solution can be easily evaluate as follow

$$\vec{\boldsymbol{x}}_s = \boldsymbol{U}_{\mathbb{C}} \exp\left(i\sqrt{d}\,\boldsymbol{\Lambda}\vec{B}'_s\right)\boldsymbol{U}_{\mathbb{C}}^H \vec{\boldsymbol{x}}_0, \tag{I.16}$$

For an even dimension $d$, and for all $j \in \{1, \ldots, d/2\}$, there exists $\lambda_j \in \mathbb{R}$ such that

$$(\boldsymbol{U}_{\mathbb{C}}^H \vec{\boldsymbol{x}}_s)_{2j-1} = \exp\left(i\sqrt{d}\,\lambda_j\vec{B}'_s\right)(\boldsymbol{U}_{\mathbb{C}}^H \vec{\boldsymbol{x}}_0)_{2j-1}, \tag{I.17}$$

$$(\boldsymbol{U}_{\mathbb{C}}^H \vec{\boldsymbol{x}}_s)_{2j} = \exp\left(-i\sqrt{d}\,\lambda_j\vec{B}'_s\right)(\boldsymbol{U}_{\mathbb{C}}^H \vec{\boldsymbol{x}}_0)_{2j}. \tag{I.18}$$

For an odd dimension $d$, $G^1$ has at least one zero eigenvalue. Without loss of generality, we consider $\boldsymbol{\Lambda}_{d,d} = 0$ and for all $j \in \{1, \ldots, (d-1)/2\}$, there exists $\lambda_j \in \mathbb{R}$ such that

$$(\boldsymbol{U}_{\mathbb{C}}^H \vec{\boldsymbol{x}}_s)_{2j-1} = \exp\left(i\sqrt{d}\,\lambda_j\vec{B}'_s\right)(\boldsymbol{U}_{\mathbb{C}}^H \vec{\boldsymbol{x}}_0)_{2j-1}, \tag{I.19}$$

$$(\boldsymbol{U}_{\mathbb{C}}^H \vec{\boldsymbol{x}}_s)_{2j} = \exp\left(-i\sqrt{d}\,\lambda_j\vec{B}'_s\right)(\boldsymbol{U}_{\mathbb{C}}^H \vec{\boldsymbol{x}}_0)_{2j}, \tag{I.20}$$

$$(\boldsymbol{U}_{\mathbb{C}}^H \vec{\boldsymbol{x}}_s)_d = (\boldsymbol{U}_{\mathbb{C}}^H \vec{\boldsymbol{x}}_0)_d. \tag{I.21}$$

### I.2.3 REAL SOLUTION WITH SINE AND COSINE

The diagonalization matrix, $\boldsymbol{U}$, is complex but we can find a real unitary matrix, $\boldsymbol{U}_{\mathbb{R}}$, to make $\boldsymbol{G}_1$ block diagonal, and then expressing the solution with cosinus and sinus as in equation I.8 and equation I.9:

$$(\boldsymbol{U}_{\mathbb{R}}^{\mathsf{T}} \vec{\boldsymbol{x}}_s)_{2j-1} = \cos\left(\sqrt{d}\,\lambda_j\vec{B}'_s\right)(\boldsymbol{U}_{\mathbb{R}}^{\mathsf{T}} \vec{\boldsymbol{x}}_0)_{2j-1} - \sin\left(\sqrt{d}\,\lambda_j\vec{B}'_s\right)(\boldsymbol{U}_{\mathbb{R}}^{\mathsf{T}} \vec{\boldsymbol{x}}_0)_{2j}, \tag{I.22}$$

$$(\boldsymbol{U}_{\mathbb{R}}^{\mathsf{T}} \vec{\boldsymbol{x}}_s)_{2j} = \sin\left(\sqrt{d}\,\lambda_j\vec{B}'_s\right)(\boldsymbol{U}_{\mathbb{R}}^{\mathsf{T}} \vec{\boldsymbol{x}}_0)_{2j-1} + \cos\left(\sqrt{d}\,\lambda_j\vec{B}'_s\right)(\boldsymbol{U}_{\mathbb{R}}^{\mathsf{T}} \vec{\boldsymbol{x}}_0)_{2j}. \tag{I.23}$$

For an odd dimension $d$, the real solution reads

$$(\boldsymbol{U}_{\mathbb{R}}^{\mathsf{T}} \vec{\boldsymbol{x}}_s)_{2j-1} = \cos\left(\sqrt{d}\,\lambda_j\vec{B}'_s\right)(\boldsymbol{U}_{\mathbb{R}}^{\mathsf{T}} \vec{\boldsymbol{x}}_0)_{2j-1} - \sin\left(\sqrt{d}\,\lambda_j\vec{B}'_s\right)(\boldsymbol{U}_{\mathbb{R}}^{\mathsf{T}} \vec{\boldsymbol{x}}_0)_{2j}, \tag{I.24}$$

$$(\boldsymbol{U}_{\mathbb{R}}^{\mathsf{T}} \vec{\boldsymbol{x}}_s)_{2j} = \sin\left(\sqrt{d}\,\lambda_j\vec{B}'_s\right)(\boldsymbol{U}_{\mathbb{R}}^{\mathsf{T}} \vec{\boldsymbol{x}}_0)_{2j-1} + \cos\left(\sqrt{d}\,\lambda_j\vec{B}'_s\right)(\boldsymbol{U}_{\mathbb{R}}^{\mathsf{T}} \vec{\boldsymbol{x}}_0)_{2j}, \tag{I.25}$$

$$(\boldsymbol{U}_{\mathbb{R}}^{\mathsf{T}} \vec{\boldsymbol{x}}_s)_d = (\boldsymbol{U}_{\mathbb{R}}^{\mathsf{T}} \vec{\boldsymbol{x}}_0)_d. \tag{I.26}$$

Figure 5 illustrates the solution for $d = 4$ with 20000 realizations of $\vec{\boldsymbol{x}}_T$ at large time $T = 100$, with $\lambda_1 = 1, \lambda_2 = 10, \vec{\boldsymbol{x}}_0 = (1, 1, 1, 1)$, and $\boldsymbol{U}_{\mathbb{R}} = \boldsymbol{I}_4$. A rotation-invariant distribution, $p_\infty$, would induce rotation-invariant marginals and hence point cloud projections appearing rotation-invariant. This is clearly not the case here. This counter example shows that low-rank tensors as defined in equation I.10 cannot guaranty rotation-invariant latent distribution, and thus prevent the use of our simple eCDF-based sampling procedure.

Figure 7 illustrates the latent vector support for a random initial condition $\vec{\boldsymbol{x}}_0 = \mathcal{N}((1, 1, 1, 1), 0.01\boldsymbol{I}_4)$. The supporting manifold is not one-dimensional anymore, but still depend on the initial direction distribution, $p_0^n$.

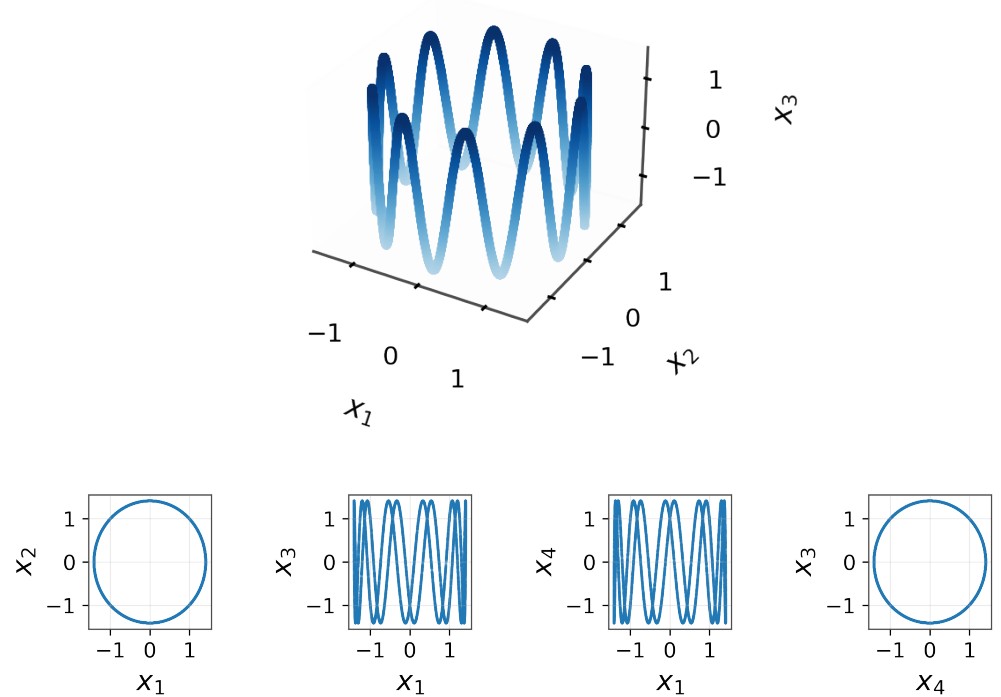

Figure 5: Projection of samples, $\vec{x}_T$, sketching the support of the invariant measure, $p_\infty$, for a low-rank tensor I.10, $d = 4$ and $\vec{x}_0 = (1, 1, 1, 1)$. The top plot is in space $(x_1, x_2, x_3)$, the bottom plots are, form left to right, in space $(x_1, x_2)$, $(x_1, x_3)$, $(x_1, x_4)$, and $(x_4, x_3)$.

Moreover, as expected from the expression above, the initial norm, $\|\vec{x}_0\|$, scales the one-dimensional manifold supporting the invariant measure (not shown) and the initial direction, $\vec{x}_0^n$, has an influence at large time. Figure 6 shows the same example with $\vec{x}_0 = (\sqrt{2}, \sqrt{2}, 0, 0)$. The initial norm is the same but the initial direction is different. Therefore, the limit distribution, $p_\infty$, if it exists does depend on the initial direction, $\vec{x}_0^n$, making the latent sampling intractable.

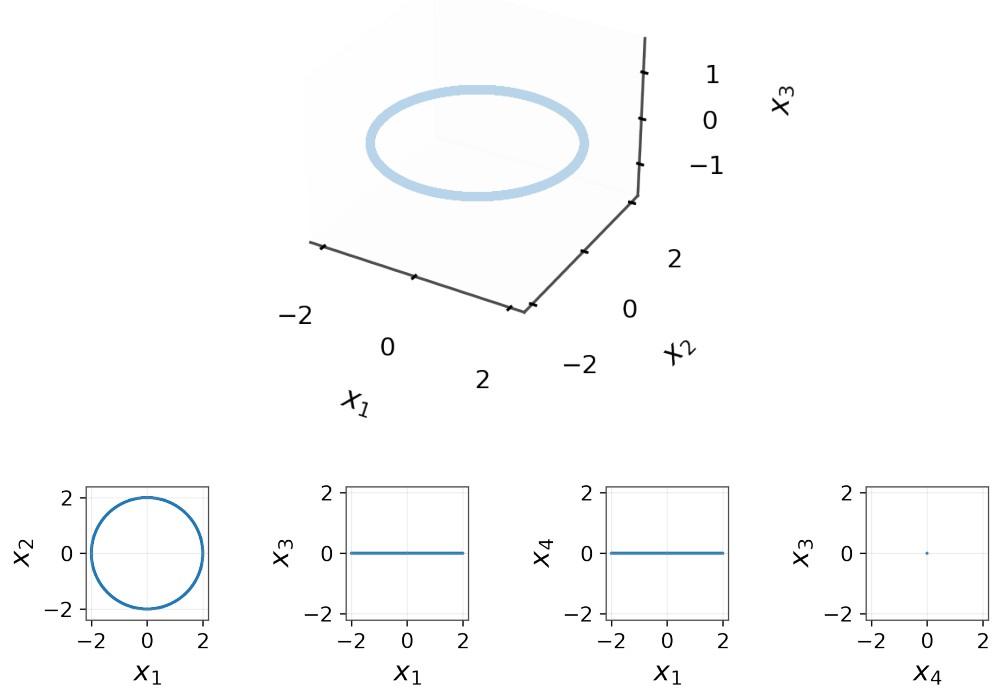

Figure 6: Projection of samples, $\vec{x}_T$, sketching the support of the invariant measure, $p_\infty$, for a low-rank tensor I.10, $d = 4$ and $\vec{x}_0 = (\sqrt{2}, \sqrt{2}, 0, 0)$. The top plot is in space $(x_1, x_2, x_3)$, the bottom plots are, form left to right, in space $(x_1, x_2)$, $(x_1, x_3)$, $(x_1, x_4)$, and $(x_4, x_3)$.

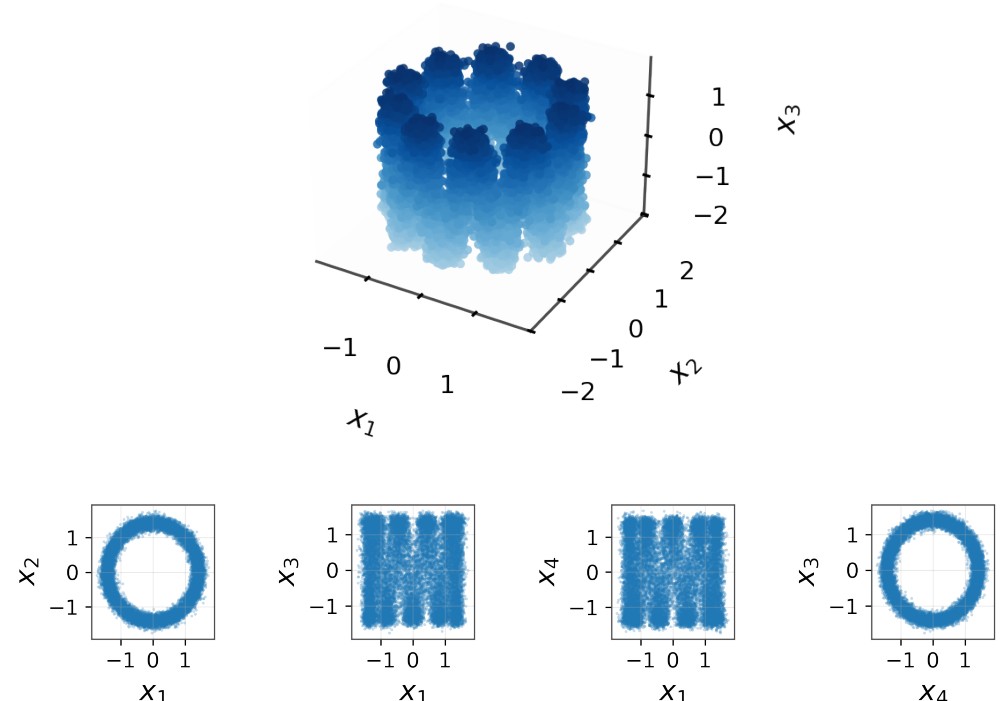

Figure 7: Projection of samples, $\vec{x}_T$, sketching the support of the invariant measure, $p_\infty$, for a low-rank tensor I.10, $d = 4$ and $\vec{x}_0 = \mathcal{N}((1, 1, 1, 1), 0.01\boldsymbol{I}_4)$. The top plot is in space $(x_1, x_2, x_3)$, the bottom plots are, form left to right, in space $(x_1, x_2)$, $(x_1, x_3)$, $(x_1, x_4)$, and $(x_4, x_3)$.

### I.3 Non-commutativity in the general case

For a general tensor $\mathbf{G}$ in dimension $d > 2$, it is temping to look for a solution $\vec{\boldsymbol{x}}_s$ of the forward SDE

$$d\vec{\boldsymbol{x}}_s = \circ d\boldsymbol{Z}_s \vec{\boldsymbol{x}}_s,$$

of the form $\exp(\boldsymbol{Z}_s)\vec{\boldsymbol{x}}_0$ with $\boldsymbol{Z} = \sum_{k=1}^d \boldsymbol{G}^k (\vec{B}_s)_k$. $\boldsymbol{Z}_s$ being skew-symmetric, $\exp(\boldsymbol{Z}_s)$ is unitary and such a solution would be reminiscent of the rotation form of equation I.6 and equation I.16 derived above. However, $\exp(\boldsymbol{Z}_s)\vec{\boldsymbol{x}}_0$ is not a solution of equation 3.1 in general, since $d\boldsymbol{Z}_s \boldsymbol{Z}_s \neq \boldsymbol{Z}_s d\boldsymbol{Z}_s$.

## J Rank and skew-symmetry conditions for random tensor $G$

In this appendix, we treat the case of random tensor $\mathbf{G}$ as defined by equation 6.1. We will show that this tensor respects both assumptions A1 and A2 almost surely. Then, we will discuss the speed of contraction of the Fokker-Planck equation with this tensor.

### J.1 Proof of the rank condition

**Proposition J.1.** *Let $M^k \in \mathbb{R}^{d,d}$ be iid random matrices with entries drawn independently from $\mathcal{N}(0,1)$. Define the skew-symmetric matrices $\boldsymbol{G}^k = \frac{1}{2}(\boldsymbol{M}^k - (\boldsymbol{M}^k)^\mathsf{T})$ and for $\boldsymbol{x} \in \mathbb{R}^d \setminus \{0\}$ define the (random) matrix*

$$\mathbf{G}(\boldsymbol{x}) := [\boldsymbol{G}^1 \boldsymbol{x}, \boldsymbol{G}^2 \boldsymbol{x}, \dots, \boldsymbol{G}^d \boldsymbol{x}] \in \mathbb{R}^{d,d}.$$

*Then, almost surely* $\mathrm{rank}(\mathbf{G}(\boldsymbol{x})) = d - 1$.

*Proof.* Let $\boldsymbol{x} \neq 0$. Let $M$ be a random standard Gaussian matrix. Then, let $\boldsymbol{D} = \boldsymbol{M} - \boldsymbol{M}^\mathsf{T}$. Then, $\boldsymbol{D}$ is Gaussian matrix with entries drawn from $\mathcal{N}(0,2)$, in particular

$$\mathbb{E}[D_{ij}D_{k\ell}] = \mathbb{E}[(M_{ij} - M_{ij})(M_{k\ell} - M_{\ell k})] = 2(\delta_{ik}\delta_{j\ell} - \delta_{i\ell}\delta_{jk}). \tag{J.1}$$

Consequently,

$$\mathbb{E}[(\boldsymbol{M} - \boldsymbol{M}^\mathsf{T})\boldsymbol{x}\boldsymbol{x}^\mathsf{T}(\boldsymbol{M}^\mathsf{T} - \boldsymbol{M})] = -\mathbb{E}[\boldsymbol{D}\boldsymbol{x}\boldsymbol{x}^\mathsf{T}\boldsymbol{D}]. \tag{J.2}$$

Now, for the covariance structure it holds

$$-(\mathbb{E}[\boldsymbol{D}\boldsymbol{x}\boldsymbol{x}^\mathsf{T}\boldsymbol{D}])_{ik} = \mathbb{E}[(\boldsymbol{D}\boldsymbol{x})_i(\boldsymbol{D}\boldsymbol{x})_k], \tag{J.3}$$

$$= \sum_{j=1}^d \sum_{\ell=1}^d \mathbb{E}[D_{ij}x_j D_{k\ell}x_\ell], \tag{J.4}$$

$$= \sum_{j=1}^d \sum_{\ell=1}^d x_j x_\ell \mathbb{E}[D_{ij}D_{k\ell}], \tag{J.5}$$

$$= 2\sum_{j=1}^d \sum_{\ell=1}^d x_j x_\ell (\delta_{ik}\delta_{j\ell} - \delta_{i\ell}\delta_{jk}), \tag{J.6}$$

$$= 2\delta_{ik}\sum_{j=1}^d \sum_{\ell=1}^d x_j x_\ell \delta_{j\ell} - 2\sum_{j=1}^d \sum_{\ell=1}^d x_j x_\ell \delta_{i\ell}\delta_{jk}, \tag{J.7}$$

$$= 2\delta_{ik}\|\boldsymbol{x}\|^2 - 2x_i x_k. \tag{J.8}$$

Hence $\mathbb{E}[\boldsymbol{D}\boldsymbol{x}\boldsymbol{x}^\mathsf{T}\boldsymbol{D}] = 2(\|\boldsymbol{x}\|\boldsymbol{I}_d - \boldsymbol{x}\boldsymbol{x}^\mathsf{T})$. Consequently for any $k = 1, \dots, d$ it holds that

$$\mathbb{E}[(\boldsymbol{G}^k x)(\boldsymbol{G}^k x)^\mathsf{T}] = \frac{1}{4}\mathbb{E}[(\boldsymbol{M}^k - (\boldsymbol{M}^k)^\mathsf{T})\boldsymbol{x}\boldsymbol{x}^\mathsf{T}((\boldsymbol{M}^k)^\mathsf{T} - \boldsymbol{M}^k)] = \frac{1}{2}(\|\boldsymbol{x}\|^2 \boldsymbol{I}_d - \boldsymbol{x}\boldsymbol{x}^\mathsf{T}). \tag{J.9}$$

As a result, the matrix $\mathbf{G}(\boldsymbol{x})$ has columns $\boldsymbol{G}^k x \overset{iid}{\sim} \mathcal{N}(0, \boldsymbol{V})$ with

$$\boldsymbol{V} = \boldsymbol{V}(\boldsymbol{x}) = \mathbb{E}\boldsymbol{\Sigma}(\boldsymbol{x}) = \frac{1}{2}(\|\boldsymbol{x}\|^2 \boldsymbol{I}_d - \boldsymbol{x}\boldsymbol{x}^\mathsf{T}), \tag{J.10}$$

of rank $d-1$. Therefore, $\boldsymbol{\Sigma}(\boldsymbol{x}) = \mathbf{G}(\boldsymbol{x})\mathbf{G}(\boldsymbol{x})^{\mathsf{T}} \sim W_d(\boldsymbol{V}(\boldsymbol{x}), d)$ is a Wishart matrix.

Let $\boldsymbol{K} = \boldsymbol{K}(\boldsymbol{x})$ be a matrix

$$\boldsymbol{K} = [\boldsymbol{K}_1, \ldots, \boldsymbol{K}_{d-1}], \tag{J.11}$$

with column vectors $\boldsymbol{K}_i$ forming an orthonormal basis of the hyperplane $\boldsymbol{x}^{\perp}$. Then by construction, we have

$$\frac{2}{\|\boldsymbol{x}\|^2} \boldsymbol{K}\boldsymbol{V}\boldsymbol{K}^{\mathsf{T}} = \boldsymbol{I}_{d-1}. \tag{J.12}$$

This means, that

$$\frac{\sqrt{2}}{\|\boldsymbol{x}\|} \boldsymbol{K}\mathbf{G}(\boldsymbol{x}) = (\frac{\sqrt{2}}{\|\boldsymbol{x}\|} \boldsymbol{K}G^1 x \ldots \frac{\sqrt{2}}{\|\boldsymbol{x}\|} \boldsymbol{K}G^d x), \quad \frac{\sqrt{2}}{\|\boldsymbol{x}\|} \boldsymbol{K}G^k x \overset{iid}{\sim} \mathcal{N}(0, \boldsymbol{I}_{d-1}). \tag{J.13}$$

Therefore,

$$\frac{\sqrt{2}}{\|\boldsymbol{x}\|} \boldsymbol{K}\boldsymbol{\Sigma}(\boldsymbol{x}) \left( \frac{\sqrt{2}}{\|\boldsymbol{x}\|} \boldsymbol{K} \right)^{\mathsf{T}} = \left( \frac{\sqrt{2}}{\|\boldsymbol{x}\|} \boldsymbol{K}\mathbf{G}(\boldsymbol{x}) \right) \left( \frac{\sqrt{2}}{\|\boldsymbol{x}\|} \boldsymbol{K}\mathbf{G}(\boldsymbol{x}) \right)^{\mathsf{T}} \sim W_{d-1}(\boldsymbol{I}_{d-1}, d), \tag{J.14}$$

is a Wishart matrix, in particular $W_p(C, n)$ denotes the Wishart distribution with $n$ degrees of freedom. In the case $n \geq p$, such matrix is invertible almost surely (Muirhead, 2009, Theorem 3.1.4). In our case $n = d > p = d - 1$ thus almost surely

$$\mathrm{rank}\left( \frac{\sqrt{2}}{\|\boldsymbol{x}\|} \boldsymbol{K}\mathbf{G}(\boldsymbol{x}) \right) = \mathrm{rank}\left( \left( \frac{\sqrt{2}}{\|\boldsymbol{x}\|} \boldsymbol{K}\mathbf{G}(\boldsymbol{x}) \right)^{\mathsf{T}} \right) = d - 1. \tag{J.15}$$

Now, since $\mathbf{G}(\boldsymbol{x})^{\mathsf{T}}\boldsymbol{x} = 0$ we obtain almost surely that

$$d - 1 = \mathrm{rank}\left( \frac{\sqrt{2}}{\|\boldsymbol{x}\|} \boldsymbol{K}\mathbf{G}(\boldsymbol{x}) \right) \leq \mathrm{rank}(\mathbf{G}(\boldsymbol{x})) \leq d - 1, \tag{J.16}$$

which yields the claim.

$\square$

## J.2 TENSOR RENORMALIZATION

In practice, we renormalize the tensor $\mathbf{G}$ as follows:

$$\mathbf{G} = \frac{\sqrt{d}}{\|\tilde{\mathbf{G}}\|_2} \tilde{\mathbf{G}} \quad \text{with} \quad \tilde{G}_{ij}^k = \tfrac{1}{2}(M_{i,j}^k - M_{j,i}^k). \tag{J.17}$$

The normalization ensures that the trace of matrix defining the Itô term of our forward SDE – i.e. the term driving the exponential decreases of $\mathbb{E}\vec{\boldsymbol{x}}_s$ (see the forward Itô SDE equation D.9) – is

$$\mathrm{tr}\left( \tfrac{1}{2}\sum_k \boldsymbol{G}^k\boldsymbol{G}^k \right) = -\mathrm{tr}\left( \tfrac{1}{2}\sum_k \boldsymbol{G}^k(\boldsymbol{G}^k)^{\mathsf{T}} \right) = -\tfrac{1}{2}\sum_k \|\boldsymbol{G}^k\|_2^2 = -\tfrac{1}{2}\|\mathbf{G}\|_2^2 = -\tfrac{1}{2}d, \tag{J.18}$$

similarly to the trace of the matrix defining the Itô term of classical Ornstein Uhlenbeck forward SDE :

$$\mathrm{tr}\left( -\boldsymbol{I}_d \right) = -d. \tag{J.19}$$

This normalization helps to better control the speed of convergence of the forward SDE without changing its skew-symmetry nor its rank.

## J.3 MEAN SPEED OF CONVERGENCE WITH RENORMALIZED TENSOR

Note that in this case, for $(\boldsymbol{x}, \boldsymbol{y}) \in S = \{(\boldsymbol{x}, \boldsymbol{y}) \in \mathbb{S}^{d-1} \times \mathbb{S}^{d-1} | \boldsymbol{x} \perp \boldsymbol{y}\}$,

$$\mathbb{E}\|\mathbf{G}(\boldsymbol{x})\boldsymbol{y}\|^2 = \boldsymbol{y}^{\mathsf{T}}\mathbb{E}\boldsymbol{\Sigma}(\boldsymbol{x})\boldsymbol{y} = \boldsymbol{y}^{\mathsf{T}}\mathbb{E}(\sum_{k=1}^d (\boldsymbol{G}^k\boldsymbol{x})(\boldsymbol{G}^k\boldsymbol{x})^{\mathsf{T}})\boldsymbol{y} = \boldsymbol{y}^{\mathsf{T}}\frac{d}{2}(\|\boldsymbol{x}\|^2\boldsymbol{I}_d - \boldsymbol{x}\boldsymbol{x}^{\mathsf{T}})\boldsymbol{y} = \frac{d}{2}. \tag{J.20}$$

So, we can expect exponential convergence of the Fokker-Planck equation with the speed

$$\mathbb{E}[\alpha(\mathbf{G}, d)] = (d-1)\mathbb{E} \min_{(\boldsymbol{x}, \boldsymbol{y}) \in S} \|\mathbf{G}^{\mathsf{T}}(\boldsymbol{x})\boldsymbol{y}\|^2 = (d-1)\mathbb{E}\|\mathbf{G}^{\mathsf{T}}(\vec{\boldsymbol{x}}_0)\boldsymbol{y}_0\|^2 = \frac{1}{2}d(d-1). \tag{J.21}$$

Therefore, the convergence gets faster when the dimension increases.

However, the tensor $\mathbf{G}$ is normalized (see equation J.17), so the evaluation of the convergence speed is modified. We note first that:

$$\mathbb{E}\|\tilde{\mathbf{G}}\|^2 \;=\; d\,\mathbb{E}\|\tilde{\boldsymbol{G}}^1\|^2 = \frac{d}{4}\mathbb{E}\|M^1 - (M^1)^{\mathsf{T}}\|^2 = \frac{d}{2}\sum_{ij}(\mathbb{E}(M_{ij}^1)^2 - \mathbb{E}M_{ij}^1 M_{ji}^1), \quad (\text{J}.22)$$

$$=\; \frac{d}{2}\sum_{ij}(1 - \delta_{ij}^2) = \frac{1}{2}d^2(d-1). \tag{J.23}$$

So, we obtain an estimate by Cauchy-Schwartz and Jensen's inequality

$$\mathbb{E}\|\mathbf{G}(\boldsymbol{x})\boldsymbol{y}\|^2 \;=\; \mathbb{E}\|\frac{\sqrt{d}}{\|\tilde{\mathbf{G}}\|}\,\tilde{\mathbf{G}}(\boldsymbol{x})\boldsymbol{y}\|^2 = \mathbb{E}\left[\frac{d}{\|\tilde{\mathbf{G}}\|^2}\|\tilde{\mathbf{G}}(\boldsymbol{x})\boldsymbol{y}\|^2\right], \tag{J.24}$$

$$\leqslant\; \mathbb{E}\left[\frac{d}{\|\tilde{\mathbf{G}}\|^2}\right]\mathbb{E}\|\tilde{\mathbf{G}}(\boldsymbol{x})\boldsymbol{y}\|^2, \tag{J.25}$$

$$\leqslant\; \frac{d}{\mathbb{E}\|\tilde{\mathbf{G}}\|^2}\mathbb{E}\|\tilde{\mathbf{G}}(\boldsymbol{x})\boldsymbol{y}\|^2, \tag{J.26}$$

$$=\; d\frac{d/2}{d^2(d-1)/2}, \tag{J.27}$$

$$=\; \frac{1}{d-1}, \tag{J.28}$$

and finally we obtain the following bound

$$\mathbb{E}\alpha(\mathbf{G}, d) \leqslant (d-1)\mathbb{E}\|\mathbf{G}^{\mathsf{T}}(\overrightarrow{\boldsymbol{x}}_0)\boldsymbol{y}_0\|^2 = 1. \tag{J.29}$$

## K  GOING BEYOND THE RANK CONDITION FOR MSGM SCALABILITY

The dense tensor of Section J imposes a computational complexity as $O(d^3)$. To scale up the method, we shall consider sparse tensor $\mathbf{G}$. However, the rank condition A2 makes it difficult to find sparse tensors. Therefore, we here open the discussions to a weaker set of assumptions.

### K.1  WEAKER ASSUMPTIONS

We recall here the two main assumptions of the paper

Skew-symmetry : For any $k \in \{1,\dots,d\}$, the matrix $\boldsymbol{G}^k = (G_{i,j}^k)_{i,j}$ is skew-symmetric.  (A1)

Rank condition : For any $\boldsymbol{x} \in \mathbb{R}^d\backslash\{0\}$, $\mathrm{rank}(\mathbf{G}(\boldsymbol{x})) = d-1$.  (A2)

Note that the Fokker-Planck equation 3.4, Proposition F.2, Proposition F.1, and Theorem 3.4.1 require only the assumption A1. So, the backward SDE, ODE and score-matching loss are general enough and do not prevent the use of sparse tensor $\mathbf{G}$. In contrast, our current proof of the asymptotic results Theorem 3.1.1, Theorem D.4.1, and Theorem 3.3.1 rely on the restrictive assumption A2, and unfortunately, it seems difficult to find a sparse tensor $\mathbf{G}$ matching this assumption.

### K.1.1  RANK CONDITION ALMOST EVERYWHERE

Therefore, we discuss here a weaker set of assumptions where the noise rank condition A2 is verified for almost all $\boldsymbol{x} \in \mathbb{R}^d$ only. This set of assumptions will yield a definition of a sparse tensor in Section K.2.2 providing satisfactory numerical results in practice.

Skew-symmetry : For any $k \in \{1,\dots,d\}$, the matrix $\boldsymbol{G}^k = (G_{i,j}^k)_{i,j}$ is skew-symmetric.  (A1)

Rank condition almost everywhere: For almost all $\boldsymbol{x} \in \mathbb{R}^d$, $\mathrm{rank}(\mathbf{G}(\boldsymbol{x})) = d-1$.  (A3)

The assumption A3 means the set $A_G = \{\boldsymbol{x} \in \mathbb{R}^d | \mathbf{G}(\boldsymbol{x})) < d-1\}$ has zero Lebesgue measure, i.e. $\int_{A_G} dx = 0$. Obviously, the assumption A2 implies the assumption A3.

The right-hand side of the Fokker-Planck equation 3.4 is a function of $\nabla_\perp p_s$ only. Hence, under the weaker assumptions A1 and A3, rotational invariant distributions are still invariant measures of the Fokker-Planck equation. Following the proof of Theorem D.2.1, we saw that the invariant densities, $p_\infty$, are characterized by $\|\mathbf{G}(\boldsymbol{x})^\intercal \nabla_\perp p_\infty(\boldsymbol{x})\| = 0$ almost surely. So, if $\mathbf{G}(\boldsymbol{x})$ has rank $d-1$ for almost all $\boldsymbol{x}$ with respect to the Lebesgue measure, then this requires $\nabla_\perp p_\infty(\boldsymbol{x}) = 0$ almost everywhere. Therefore, the invariant measures of Fokker-Planck equation 3.4 must be rotational invariant almost everywhere.

However, the existence and uniqueness of a classical solution (Lemma D.4.1) and the convergence guaranties to the invariant distribution (Theorem D.4.1) need more careful analysis. We only sketch some challenges involved, the full analysis will be carried out in a follow up work.

If the diffusion process enters the area of points $\boldsymbol{x}$, such that $\text{rank}(\mathbf{G}(\boldsymbol{x})) < d - 1$, one has to make sure that the diffusion process is not trapped in such an area, even if it has a measure zero. In particular, let $D \subset \mathbb{S}^{d-1}$ be defined as

$$D = \{\boldsymbol{x}^n \in \mathbb{S}^{d-1} \mid \text{rank}(G(\boldsymbol{x}^n)) < d - 1\}.$$

Then, we call $D$ a trap set. If the process $\overrightarrow{\boldsymbol{x}}_s^n$ once ever enters $D$ with positive probability, it cannot leave $D$ again, i.e.

$$\mathbb{P}(\overrightarrow{\boldsymbol{x}}_s^n \in D, \ \forall s \geq s_0 | \overrightarrow{\boldsymbol{x}}_{s_0}^n \in D) > 0.$$

Hence, in this case, convergence to the correct invariant measure has to ensure that the trap set is not invariant under the diffusion-controlled dynamic. Such a analysis then sufficiently can be implied by Hörmander / bracket-generating conditions, i.e. hypoellipticity analysis. Based on this, the asymptotic results Theorem 3.1.1, Theorem D.4.1, and Theorem 3.3.1 must be adapted. In this case we expect the convergence rate to the invariant distribution to be slower compared to the exponential convergence rate obtained in the case of strong rank condition, see also Section K.3 for a related discussion.

Although a detailed analysis of this research question is out of the scope of the current manuscript, we want to stress its relevance related to the *scalability* of the proposed method for the high-dimensional case. The standard construction via a random dense tensor $\boldsymbol{G}$ poses scalability problems. On the other side, sparse tensors provide a tool to enable such scalability provided that they satisfy the (weaker) rank conditions. While the non-local sparse tensor discussed in Section K.2.3, satisfy the strong rank condition, the local sparse tensors from Section K.2.2 only satisfy the weak rank conditions. Still, the latter have been applied in our numerical investigation for the high-dimensional test cases with Particle Image Velocimetry measurements as discussed in Section M.7 yielding first very promising results.

### K.1.2    Itô term rank condition

Now, we discuss another weaker set of assumptions where the noise rank condition A2 is replaced by an Itô drift rank condition. Although attractive, a detailed analysis in Section K.2.1 will lead us to consider this set of assumptions as insufficient for the MSGM sampling procedure.

$$\text{Skew-symmetry : For any } k \in \{1, \ldots, d\}, \text{ the matrix } \boldsymbol{G}^k = (\boldsymbol{G}^k_{i,j})_{i,j} \text{ is skew-symmetric.} \quad \text{(A1)}$$

$$\text{Itô term rank condition : the matrix } \boldsymbol{S} := \frac{1}{2}\sum_{k=1}^{d}(\boldsymbol{G}^k)(\boldsymbol{G}^k)^\intercal \text{ is full rank} \quad \text{(A4)}$$

From Lemma D.1.1, we note that $\boldsymbol{S} = -\nabla\nabla \cdot \boldsymbol{\Sigma}$. Lemma D.1.2 gives the Itô forward diffusion which can be expressed with $\boldsymbol{S}$. The assumption A2 is not needed for these lemmas. These results are true as long as the assumption A1 is verified. Taking the expectation of the Itô diffusion, we get:

$$\frac{d}{ds}\mathbb{E}\overrightarrow{\boldsymbol{x}}_s = -\boldsymbol{S}\mathbb{E}\overrightarrow{\boldsymbol{x}}_s. \quad \text{(K.1)}$$

Instead of controlling the convergence of the full distribution $p_s$, the assumption A4 controls the convergence of the mean only. It leads to the following property justifying our assumption choice.

**Proposition K.1.1.** *Let the assumption A1 holds. Then, the following assertions are equivalent*

- *The assumption A4 holds.*

- $\mathbb{E}[\vec{\boldsymbol{x}}_s | \vec{\boldsymbol{x}}_0] \underset{s \to \infty}{\longrightarrow} 0.$

- $\mathrm{Var}[\vec{\boldsymbol{x}}_s | \vec{\boldsymbol{x}}_0] \underset{s \to \infty}{\longrightarrow} \|\vec{\boldsymbol{x}}_0\|^2.$

*Proof.* $\boldsymbol{S}$ is positive semi-definite, so it is diagonalizable in an orthonormal basis, and from equation K.1, $\mathbb{E}\vec{\boldsymbol{x}}_s \underset{s \to \infty}{\longrightarrow} 0$ if and only if $\boldsymbol{S}$ is positive definite, i.e. the assumption A4 is verified.

Besides, by assumption A1, the norm $\|\vec{\boldsymbol{x}}_s\|$ is conserved along the diffusion, so

$$\mathrm{Var}[\vec{\boldsymbol{x}}_s | \vec{\boldsymbol{x}}_0] = \|\vec{\boldsymbol{x}}_s\|^2 - \|\mathbb{E}[\vec{\boldsymbol{x}}_s | \vec{\boldsymbol{x}}_0]\|^2 = \|\vec{\boldsymbol{x}}_0\|^2 - \|\mathbb{E}[\vec{\boldsymbol{x}}_s | \vec{\boldsymbol{x}}_0]\|^2 \tag{K.2}$$

which converges to $\|\vec{\boldsymbol{x}}_0\|^2$ if and only if $\mathbb{E}[\vec{\boldsymbol{x}}_s | \vec{\boldsymbol{x}}_0] \underset{s \to \infty}{\longrightarrow} 0.$ $\qquad\square$

We highlight the fact that the assumption A4 is weaker than the assumption A2 as stated by Proposition K.1.2. It is actually a strictly weaker assumption since the tensors defined in Section K.2.1 and Section K.2.2 respect assumption A4 but not assumption A2.

**Proposition K.1.2.** *Let the assumption A1 holds. Then, the assumption A2 implies the assumption A4.*

*Proof.* If the assumption A2 holds, then, Theorem 3.3.1 implies that $\vec{\boldsymbol{x}}_s \xrightarrow[s \to 0]{\mathcal{L}} \vec{\boldsymbol{x}}_\infty = \|\vec{\boldsymbol{x}}_\infty\| \vec{\boldsymbol{x}}_\infty^n.$ The asymptotic latent direction, $\vec{\boldsymbol{x}}_\infty^n$, is independent of the initial condition $\vec{\boldsymbol{x}}_0$ and has zero mean. Therefore,

$$\mathbb{E}[\vec{\boldsymbol{x}}_s | \vec{\boldsymbol{x}}_0] \underset{s \to \infty}{\longrightarrow} \mathbb{E}[\vec{\boldsymbol{x}}_\infty | \vec{\boldsymbol{x}}_0] = \mathbb{E}[\|\vec{\boldsymbol{x}}_\infty\| \vec{\boldsymbol{x}}_\infty^n | \vec{\boldsymbol{x}}_0] = \|\vec{\boldsymbol{x}}_0\| \mathbb{E}[\vec{\boldsymbol{x}}_\infty^n | \vec{\boldsymbol{x}}_0] = \|\vec{\boldsymbol{x}}_0\| \mathbb{E}[\vec{\boldsymbol{x}}_\infty^n] = 0, \tag{K.3}$$

and by Proposition K.1.1, the assumption A4 holds. $\qquad\square$

## K.2 SPARSE TENSORS

Here we propose several possible choices of sparse tensors.

First, we will consider a simple low-rank tensor in Section K.2.1 and show that it makes the latent distribution untractable. Then, we will introduce a sparse local tensor in Equation (K.8), which is adapted to MSGM and leads to good generative skills in practice. Finally, we propose a sparse nonlocal tensor in Equation (K.17) that involves more Brownian motions but meets the original assumptions A1 and A2 of our paper.

### K.2.1 LOW-RANK TENSOR

A simple choice of tensor with $d^2 = O(d^2)$ non-zero coefficients would be to take $d$ times the same dense random skew-symmetric matrix $\boldsymbol{G}^1$ i.e.

$$G_{i,j}^k = G_{i,j}^1 = \tfrac{1}{2}(M_{i,j}^1 - M_{j,i}^1), \tag{K.4}$$

$$M_{i,j}^1 \overset{iid}{\sim} \mathcal{N}(0,1). \tag{K.5}$$

Section I.2 provides an analytic solution for the forward diffusion in this case. Such a solution would be a strong advantage for our learning procedure, bypassing the need for numerical integration of the forward diffusion, and enabling denoising score matching instead of sliced score matching. However, Proposition K.2.1 below shows that there is a rank deficiency, probably inducing the existence of non-rotation-invariant latent distribution, $p_\infty$, preventing MSGM sampling in practice. Indeed, numerically illustrated in dimension $d = 4$, the analytic solution of Section I.2 shows a latent distribution intractable in practice. The latent distribution is not rotation-invariant and does depends on the initial direction distribution, $p_0^n$. It seems to be a direct consequence of the rank deficiency.

We conclude that low-rank tensors as in equation K.4 are not a suitable choice for MSGM. Moreover, it suggests that assumptions A1 and A4 as in Section K.1.2 are not sufficient for MSGM.

**Proposition K.2.1.** *If* $\mathsf{G}$ *is defined from equation K.4 and equation K.5, then, for any* $\boldsymbol{x} \in \mathbb{R}^d$, $\mathrm{rank}(\mathsf{G}(\boldsymbol{x})) \leqslant 1$. *Assumption A1 is verified, assumptions A2 and A3 are not for* $d \geqslant 3$, *and assumption A4 is verified almost surely if and only if the dimension* $d$ *if even. Moreover, we have* $\boldsymbol{S} = \frac{d}{2}\boldsymbol{G}^1(\boldsymbol{G}^1)^\intercal$ *and* $\mathbb{E}\boldsymbol{S} = \frac{d(d-1)}{4}I_d$.

*Proof.* The tensor defined by equation K.4 and equation K.5 obviously matches the skew-symmetric condition A1.

For odd dimension $d$, $\boldsymbol{G}^1$ – like all skew-symmetric matrix – is singular. Thus $\boldsymbol{S}$ is singular and even the weak condition A4 is not satisfied.

For even $d$ the polynomial $p \colon \mathbb{R}^{d\times d} \to \mathbb{R}$, $\boldsymbol{M} \mapsto \det(\frac{1}{2}(\boldsymbol{M} - \boldsymbol{M}^\top))$ is non-zero since there exists invertible skew-symmetric matrices. Therefore, the set $\{\boldsymbol{M} \in \mathbb{R}^{d,d} \mid \det(\boldsymbol{M} - \boldsymbol{M}^\top) = 0\}$ forms a proper algebraic variety with zero Lebesgue measure. Hence, since the Gaussian distribution is absolutely continuous w.r.t. to the Lebesgue measure, it holds

$$\mathbb{P}(\det(\boldsymbol{G}^1) = 0) = 0, \tag{K.6}$$

and so $\boldsymbol{G}^1$ is invertible with full rank with probability 1. Thus

$$\boldsymbol{S} = \frac{d}{2}\boldsymbol{G}^1(\boldsymbol{G}^1)^\intercal. \tag{K.7}$$

is positive definite. Therefore, A4 is verified for even dimension $d$.

However, for any $d \geqslant 3$, neither conditions A2 nor condition A3 is satisfied. Indeed, for any $\boldsymbol{x} \in \mathbb{R}^d\setminus\{0\}$, $\mathrm{rank}(\mathsf{G}(\boldsymbol{x})) = \mathrm{rank}[\boldsymbol{G}^1\boldsymbol{x}, \ldots, \boldsymbol{G}^1\boldsymbol{x}] \leqslant 1$. This is expected since the diffusion involves a single Brownian motion (see Section I.2).

Since the entries in $\boldsymbol{M}^1$ are independent standard normal Gaussian, we have $\mathbb{V}(G^1_{i,j}) = \frac{1}{4}(\mathbb{V}(M^1_{i,j}) + \mathbb{V}(M^1_{j,i}) = \frac{1}{2}$. Then, $[\boldsymbol{G}^1(\boldsymbol{G}^1)^\top]_{ik} = \sum\limits_{j=1}^{d} G^1_{ij}G^1_{kj}$. Hence for $i = k$

$$\mathbb{E}[[\boldsymbol{G}^1(\boldsymbol{G}^1)^\top]_{ik}] = \sum_{j=1}^{d}\mathbb{E}[(G^1_{ij})^2] = \sum_{j\neq i}\frac{1}{2} = \frac{d-1}{2},$$

since $G^1_{ii} = 0$. For $i \neq k$, $G^1_{ij}$ and $G^1_{kj}$ involve independent entries of $\boldsymbol{M}^1$, leading to $\mathbb{E}[G^1_{ij}G^1_{kj}] = 0$. As a consequence

$$\mathbb{E}[\boldsymbol{S}] = \frac{d}{2}\mathbb{E}[\boldsymbol{G}^1(\boldsymbol{G}^1)^\top] = \frac{d(d-1)}{4}I_d \in \mathbb{R}^{d,d}.$$

$\square$

### K.2.2 Local sparse tensor

Let us define a tensor with only $2d = O(d)$ non-zero coefficients.

$$G^k_{i,j} = \begin{cases} 1 & \text{if } i = j - 1[d] = k \\ -1 & \text{if } i - 1[d] = j = k \\ 0 & \text{otherwise.} \end{cases} , \quad 1 \leqslant i,j,k \leqslant d, \tag{K.8}$$

with $[d]$ stands for modulo $d$. It is built from a subset of the canonical basis of skew-symmetric matrices, keeping only $d$ matrices with most non-zero values close to the diagonal. It ensures a strong sparsity and a local structure for $\boldsymbol{x} \to \boldsymbol{G}^k\boldsymbol{x}$.

The skew-symmetry assumption A1 is obviously fulfilled from the definition K.8. However, the strict rank condition assumption A2 is not in general. Fortunately, the assumptions A3 and A4 still hold. In particular, the Itô term matrix simplifies as shown by the following proposition.

We implemented this version of sparse tensor. For small dimension applications in Section M.6.2 and Section M.7.1, it has been found to provide numerical results as good as the dense tensor implementation (see Figures 29 and 43). For large dimension applications, dense tensors can complicate or even prevent MSGM applications. There, we obtained satisfactory results with local sparse tensor (see Figure 49 in Section M.7.2).

**Proposition K.2.2.** *If* $\mathbf{G}$ *is defined from equation K.8, then, for any* $\boldsymbol{x} \in (\mathbb{R}\backslash\{0\})^d$, *rank*$(\mathbf{G}(\boldsymbol{x})) = d-1$ *Moreover, we have* $\boldsymbol{S} = \boldsymbol{I}_d$ *and the assumptions A1, A3, and A4 are verified.*

*Proof.* For any $\boldsymbol{x} \in (\mathbb{R}\backslash\{0\})^d$, we have

$$\mathbf{G}(\boldsymbol{x}) = [\boldsymbol{G}^1\boldsymbol{x}, \ldots, \boldsymbol{G}^d\boldsymbol{x}] = \begin{pmatrix} x_2 & 0 & \cdots & 0 & -x_d \\ -x_1 & x_3 & \cdots & 0 & 0 \\ 0 & -x_2 & \ddots & \vdots & \vdots \\ \vdots & \vdots & \ddots & x_d & 0 \\ 0 & 0 & \cdots & -x_{d-1} & x_1 \end{pmatrix}. \tag{K.9}$$

To simplify notations, all the indices in this proof will be defined modulo $d$. For instance, $x_{i+1}$ for $i = d$ stands for $x_1$.

For any $\boldsymbol{y} \in \mathbb{R}^d$, $\mathbf{G}(\boldsymbol{x})^T\boldsymbol{y} = 0$ $(\in \mathbb{R}^d)$ if and only if, for all $i \leqslant d$, $x_{i+1}y_i - x_iy_{i+1} = 0$ and $y_{i+1} = \frac{x_{i+1}}{x_i}y_i$. Finally,

$$y_i = \Pi_{j=1}^{i-1}\frac{x_{j+1}}{x_j}y_1 = \frac{x_i}{x_1}y_1. \tag{K.10}$$

Therefore, $\boldsymbol{y} \in \mathbb{R}\boldsymbol{x}$. Reciprocally, we can verify that $\mathbb{R}\boldsymbol{x} \subset \mathrm{Ker}(\mathbf{G}(\boldsymbol{x}))$. We conclude that rank$(\mathbf{G}(\boldsymbol{x})) = d - \dim(\mathrm{Ker}(\mathbf{G}(\boldsymbol{x}))) = d-1$.

To evaluate the matrix $\boldsymbol{S}$, we note that

$$\boldsymbol{G}^k = \boldsymbol{e}_k\boldsymbol{e}_{k+1}^{\intercal} - \boldsymbol{e}_{k+1}\boldsymbol{e}_k^{\intercal}, \tag{K.11}$$

with $(\boldsymbol{e}_k)_k$ the canonical basis of $\mathbb{R}^d$. Then,

$$\boldsymbol{S} = -\frac{1}{2}\sum_{k=1}^d (\boldsymbol{G}^k)^2, \tag{K.12}$$

$$= -\frac{1}{2}\sum_{k=1}^d (\boldsymbol{e}_k\boldsymbol{e}_{k+1}^{\intercal} - \boldsymbol{e}_{k+1}\boldsymbol{e}_k^{\intercal})^2, \tag{K.13}$$

$$= -\frac{1}{2}\sum_{k=1}^d (0 - \boldsymbol{e}_k\boldsymbol{e}_k^{\intercal} - \boldsymbol{e}_{k+1}\boldsymbol{e}_{k+1}^{\intercal} + 0), \tag{K.14}$$

$$= \frac{1}{2}(\boldsymbol{I}_d + \boldsymbol{I}_d), \tag{K.15}$$

$$= \boldsymbol{I}_d. \tag{K.16}$$

$\square$

### K.2.3 NON-LOCAL SPARSE TENSOR

We also propose another tensor with $d(d-1) = O(d^2)$ non-zero coefficients.

$$G_{i,j}^{k,k'} = \begin{cases} 1 & \text{if } i - k'[d] = j = k \\ -1 & \text{if } i = j - k'[d] = k \\ 0 & \text{otherwise.} \end{cases} \quad , \quad \begin{array}{l} 1 \leqslant i,j,k \leqslant d, \\ 1 \leqslant k' \leqslant \lceil\frac{d-1}{2}\rceil, \end{array} \tag{K.17}$$

where $\lceil\frac{d-1}{2}\rceil$ is the least integer greater than or equal to $\frac{d-1}{2}$. It is the canonical basis for skew-symmetric matrices. It ensures a relative sparsity and encodes a non-local structure for $\boldsymbol{x} \to \boldsymbol{G}^{k,k'}\boldsymbol{x}$.

Here, the sparse tensor $\mathbf{G}$ is of size $d \times d \times d(d-1)/2$ instead of $d \times d \times d$. Our theoretical framework differs slightly. The forward diffusion involves $d(d-1)/2$ one-dimensional Brownian motions. Consequently, the neural network, $\boldsymbol{a}_\theta(\boldsymbol{x},s)$, approximating the scaled score, $\mathbf{G}(\boldsymbol{x})\nabla\log p_s(\boldsymbol{x})$, has $d(d-1)/2$ coefficients. The size of the neural network parameters $\boldsymbol{\theta}$ can increase and may complicate the training procedure. An alternative could be to work with a neural network, $\boldsymbol{s}_\theta(\boldsymbol{x},s)$, which approximates the true score, $\nabla\log p_s(\boldsymbol{x})$, having $d$ coefficients only.

This choice of tensor meets all the assumptions, including A1 and A2 as proofed below. However, because of the additional implementation complexity mentioned above, we postpone its numerical evaluation to MSGM for future work.

**Proposition K.2.3.** *If* $\mathbf{G}$ *is defined from equation K.17, then, for any* $x \in \mathbb{R}^d \backslash \{0\}$, $rank(\mathbf{G}(x)) = d - 1$ *Moreover, we have* $S = I_d$ *and the assumptions A1, A2, A3, and A4 are verified.*

*Proof.* For any $x \in \mathbb{R}^d \backslash \{0\}$, we have

$$\mathbf{G}(x) = [G^{1,1}x, \dots, G^{d,\lceil \frac{d-1}{2} \rceil}x] \tag{K.18}$$

We already know that $\text{Im}(\mathbf{G}(x)) \subset x^\perp$ since $x^\intercal \mathbf{G}(x) = 0$. Now we assume that $y \in x^\perp$, and we define

$$Q = \frac{1}{\|x\|^2}(yx^\intercal - xy^\intercal). \tag{K.19}$$

Applying on $x$, we get:

$$Qx = \frac{1}{\|x\|^2}(y\|x\|^2 - xy \cdot x) = y. \tag{K.20}$$

Besides, $(G^{k,k'})_{k,k'}$ is the canonical basis of skew-symmetric matrices and $Q$ is skew-symmetric so there exists $\alpha \in \mathbb{R}^{\frac{d(d-1)}{2}}$ such that

$$Q = \sum_{k'=1}^{\lceil \frac{d-1}{2} \rceil} \sum_{k=1}^{d} \alpha_{k,k'} G^{k,k'} \tag{K.21}$$

and thus

$$y = Qx = \sum_{k'=1}^{\lceil \frac{d-1}{2} \rceil} \sum_{k=1}^{d} \alpha_{k,k'} G^{k,k'} x = \mathbf{G}(x)\alpha \in \text{Im}(\mathbf{G}(x)). \tag{K.22}$$

We conclude that $\text{Im}(\mathbf{G}(x)) = x^\perp$ and $rank(\mathbf{G}(x)) = d - 1$.

To evaluate the matrix $S$, we note that

$$G^{k,k'} = e_k e_{k+k'}^\intercal - e_{k+k'} e_k^\intercal, \tag{K.23}$$

with $(e_k)_k$ the canonical basis of $\mathbb{R}^d$, and defining again all the indices modulo $d$.

$$
\begin{aligned}
S &= -\frac{1}{2} \sum_{k'=1}^{\lceil \frac{d-1}{2} \rceil} \sum_{k=1}^{d} (G^{k,k'})^2, & \text{(K.24)} \\
&= -\frac{1}{2} \sum_{k'=1}^{\lceil \frac{d-1}{2} \rceil} \sum_{k=1}^{d} \left( e_k e_{k+k'}^\intercal - e_{k+k'} e_k^\intercal \right)^2, & \text{(K.25)} \\
&= -\frac{1}{2} \sum_{k'=1}^{\lceil \frac{d-1}{2} \rceil} \sum_{k=1}^{d} (0 - e_k e_k^\intercal - e_{k+k'} e_{k+k'}^\intercal + 0), & \text{(K.26)} \\
&= \frac{1}{2}(I_d + I_d), & \text{(K.27)} \\
&= I_d. & \text{(K.28)}
\end{aligned}
$$

$\square$

## K.3 DISCUSSION ABOUT LOCAL AND NON LOCAL STRUCTURE

The random tensor of Section J and the large sparse tensor of Section K.2.3 may be interpreted as non-local since $x \to G^k x$ changes coefficients $x_i$ of $x$ which are not sorted next to each other in $x$. For large dimension $d$, we believe that this can accelerate the convergence in comparison with local tensors, like the sparse tensor of Section K.2.2 or a discretized version of transport noise SPDEs. Indeed, for local dynamics the randomness may take time to spread by going from one coefficient to the next whereas in non-local dynamics the randomness can spread directly in the whole state space

at each time step. Our preliminary numerical results (not shown) seems to confirm this intuition. Moreover, the stronger theoretical properties of those non-local tensors – rank condition A2 and thus exponential convergence of the distribution – also tends to confirm our conjecture. However, diffusion models in large dimension strongly rely on the powerful skills of convolutional neural networks (CNN), which have – up to attention layers – an intrinsic local structures. Accordingly, it may be difficult for a CNN to learn how to denoise a non-local noising process. More theoretical and experimental works would be needed to confirm this intuition. This is out of the scope of this already lengthy paper and is currently under investigation by the authors.

## L    NUMERICAL SCHEME

### L.1    NUMERICAL INTEGRATION OF SDES

#### L.1.1    STOCHASTIC RUNGE-KUTTA METHOD FOR STRATONOVICH SDES

We consider the Stratonovich stochastic differential equation (SDE):

$$dx_t = f_S(t, x_t)\, dt + \tilde{\mathbf{G}}(t, x_t) \circ dB_t, \tag{L.1}$$

where $f_S : \mathbb{R} \times \mathbb{R}^d \to \mathbb{R}^d$ is the drift, $\tilde{\mathbf{G}} : \mathbb{R} \times \mathbb{R}^d \to \mathbb{R}^{d \times m}$ is the diffusion term, and $B_t$ is an $d$-dimensional Wiener process.

The following Runge-Kutta (RK) method (Kloeden et al., 1992) approximates the solution $x_{n+1} \approx x(t_{n+1})$ over the interval $[t_n, t_{n+1}]$, with time step $\Delta t = t_{n+1} - t_n$ and Wiener increment $\Delta B_n = B_{t_{n+1}} - B_{t_n}$:

$$K_1 = f_S(t_n, x_n)\, \Delta t + \tilde{\mathbf{G}}(t_n, x_n)\, \Delta B_n, \tag{L.2}$$

$$K_2 = f_S\left(t_n + \frac{\Delta t}{2}, x_n + \frac{K_1}{2}\right) \Delta t + \tilde{\mathbf{G}}\left(t_n + \frac{\Delta t}{2}, x_n + \frac{K_1}{2}\right) \Delta B_n, \tag{L.3}$$

$$K_3 = f_S\left(t_n + \frac{\Delta t}{2}, x_n + \frac{K_2}{2}\right) \Delta t + \tilde{\mathbf{G}}\left(t_n + \frac{\Delta t}{2}, x_n + \frac{K_2}{2}\right) \Delta B_n, \tag{L.4}$$

$$K_4 = f_S(t_n + \Delta t, x_n + K_3)\, \Delta t + \tilde{\mathbf{G}}(t_n + \Delta t, x_n + K_3)\, \Delta B_n, \tag{L.5}$$

$$x_{n+1} = x_n + \frac{1}{6}(K_1 + 2K_2 + 2K_3 + K_4). \tag{L.6}$$

This method leverages the structure of Stratonovich SDEs and their differential geometry properties. It is particularly well-suited to our SDE equation 3.1 with skew-symmetric noise and no Stratonovich drift.

#### L.1.2    RENORMALISATION

Both our forward SDE equation 3.1 and backward SDE equation 2.2 preserve the solution norm $\|x_t\|$. However, even the above Runge Kutta discretization can break this symmetry. To enforce it numerically, we normalize after each time step.

The final integration scheme is summarized in Algorithm 2. Here, we highlight the differences compared to the classical RK4 in color. Note that the optional of normalization in line 10 of the Algorithm is relevant only for MSGM but not for SGM.

### L.2    SCHEDULING

In order to enable both a sufficient statistical convergence of the forward SDE at time $s = T$ and a convenient time step, we implemented a time scheduling for both SGM and MSGM. We first recall the basic principle of scheduling in continuous time, then propose a method for MSGM, and finally discuss the theoretical consequences.

**Algorithm 2:** `SRK4` for conservative Stratonovich SDEs with renormalization.

---

**Input:** Integration time $T$, number of time step $N_T$, initial condition $\boldsymbol{x}_0$, drift $\boldsymbol{f}_S$, diffusion $\tilde{\mathsf{G}}$

1: $\Delta t \leftarrow \frac{T}{N_T}$; {Time step}
2: **for** $n = 0$ to $N_T - 1$ **do**
3: $\quad \Delta \boldsymbol{B}_n \sim \mathcal{N}(0, \Delta t\, \boldsymbol{I}_d)$ {Wiener increment}
4: $\quad t_n \leftarrow n\Delta t$
5: $\quad \boldsymbol{K}_1 \leftarrow \boldsymbol{f}_S(t_n, \boldsymbol{x}_n)\,\Delta t \;+\; \tilde{\mathsf{G}}(t_n, \boldsymbol{x}_n)\,\Delta \boldsymbol{B}_n$
6: $\quad \boldsymbol{K}_2 \leftarrow \boldsymbol{f}_S\big(t_n + \frac{\Delta t}{2},\, \boldsymbol{x}_n + \frac{\boldsymbol{K}_1}{2}\big)\,\Delta t \;+\; \tilde{\mathsf{G}}\big(t_n + \frac{\Delta t}{2},\, \boldsymbol{x}_n + \frac{\boldsymbol{K}_1}{2}\big)\,\Delta \boldsymbol{B}_n$
7: $\quad \boldsymbol{K}_3 \leftarrow \boldsymbol{f}_S\big(t_n + \frac{\Delta t}{2},\, \boldsymbol{x}_n + \frac{\boldsymbol{K}_2}{2}\big)\,\Delta t \;+\; \tilde{\mathsf{G}}\big(t_n + \frac{\Delta t}{2},\, \boldsymbol{x}_n + \frac{\boldsymbol{K}_2}{2}\big)\,\Delta \boldsymbol{B}_n$
8: $\quad \boldsymbol{K}_4 \leftarrow \boldsymbol{f}_S(t_n + \Delta t_n,\, \boldsymbol{x}_n + \boldsymbol{K}_3)\,\Delta t \;+\; \tilde{\mathsf{G}}(t_n + \Delta t_n,\, \boldsymbol{x}_n + \boldsymbol{K}_3)\,\Delta \boldsymbol{B}_n$
9: $\quad \tilde{\boldsymbol{x}}_{n+1} \leftarrow \boldsymbol{x}_n + \frac{1}{6}(\boldsymbol{K}_1 + 2\boldsymbol{K}_2 + 2\boldsymbol{K}_3 + \boldsymbol{K}_4)$ {Classical RK4 blend}
10: $\quad \boldsymbol{x}_{n+1} \leftarrow \dfrac{\|\boldsymbol{x}_0\|}{\|\tilde{\boldsymbol{x}}_{n+1}\|}\, \tilde{\boldsymbol{x}}_{n+1}$ {Optional step : Enforce $\|\boldsymbol{x}_{n+1}\| = \|\boldsymbol{x}_0\|$)}
11: **end for**
12: **return** $\boldsymbol{x}_{N_T}$ {Approximation of $\boldsymbol{x}_T$}

---

### L.2.1 USUAL SCHEDULING

In continuous time (Song et al., 2021), a convenient way is to make a change of variable, replacing the time $s$ by

$$z(s) \;=\; \int_0^s g^2(s')ds'. \tag{L.7}$$

with

$$g^2(s) = \frac{1}{2}\beta(s) = \frac{1}{2}\left(\beta_m + (\beta_M - \beta_m)\frac{s}{T}\right), \tag{L.8}$$

and $\beta_M > \beta_m > 0$. Note that other schedulings are also possible (Strasman et al., 2024) and can possibly be optimize to better adapt to the problem at hand. We first describe the hyperparameters values chosen in our numerical experiments and then explain how scheduling affects SGM and MSGM theories.

Since we built our code from an existing one (https://github.com/CW-Huang/sdeflow-light, Huang et al. (2021)), by default we choose the values provided there for SGM scheduling: $\beta_m = 0.1$ and $\beta_M = 20$. We expect these values to be already finely tuned and we have verified that this couple of values gives indeed better results than many other choices (not shown). We believe that these default values of the SGM hyperparameters enable a fair comparison to MSGM. For some test cases, we found another SGM scheduling that works better and we use it instead. All scheduling hyperparameters are provided in the tables summarizing test cases in Section M.

For small time $s$, the time remapping is linear : $g^2(s) \underset{s\to 0}{\sim} \frac{1}{2}\beta_m$ and $z(s) \underset{s\to 0}{\sim} \frac{1}{2}\beta_m s$ whereas for large time, $g^2(s) \underset{s\to T}{\sim} \frac{1}{2}\beta_M$ and using the Taylor expansion around $T$, yielding

$$z(s) = z(T) + z'(t)(s - T) + o(s - T),$$

we find that

$$z(s) \;=\; \frac{1}{2}\left(\beta_m + \frac{1}{2}(\beta_M - \beta_m)\frac{s}{T}\right)s, \tag{L.9}$$

$$= \frac{1}{2}\left(\frac{\beta_M + \beta_m}{2}T + \beta_M(s - T)\right) + \underset{s\to T}{o}(s - T), \tag{L.10}$$

$$\underset{s\to T}{\longrightarrow} \frac{\beta_M + \beta_m}{4}T. \tag{L.11}$$

As such, SGM forward and backward SDEs become:

$$d\overrightarrow{\boldsymbol{x}}_s = -g^2(s)\overrightarrow{\boldsymbol{x}}_s ds + \sqrt{2}g(s)d\overrightarrow{\boldsymbol{B}}_s, \tag{L.12}$$

$$d\overleftarrow{\boldsymbol{x}}_t = g^2(T-t)\overleftarrow{\boldsymbol{x}}_t dt + \sqrt{2}g(T-t)\left(\boldsymbol{a_\theta}(T-t,\overleftarrow{\boldsymbol{x}}_t)dt + \circ d\overleftarrow{\boldsymbol{B}}_t\right), \tag{L.13}$$

where $\boldsymbol{a_\theta}(T-t,\overleftarrow{\boldsymbol{x}}_t)$ approximates $\sqrt{2}g(T-t)\nabla\log p_{T-t}(\overleftarrow{\boldsymbol{x}}_t)$. The backward SDE can be integrated with the Stochastic Runge-Kutta Algorithm 2 where

$$\boldsymbol{f}_S(t,\overleftarrow{\boldsymbol{x}}_t) = g^2(T-t)\overleftarrow{\boldsymbol{x}}_t + \sqrt{2}g(T-t)\boldsymbol{a_\theta}(T-t,\overleftarrow{\boldsymbol{x}}_t) \quad\text{and}\quad \tilde{\mathbf{G}}(t,\overleftarrow{\boldsymbol{x}}_t) = \sqrt{2}g(T-t). \tag{L.14}$$

### L.2.2 SCHEDULING FOR MSGM

We propose a similar scheduling for MSGM. Scheduled forward and backward SDEs write:

$$d\overrightarrow{\boldsymbol{x}}_s = g(s)\mathbf{G}(\overrightarrow{\boldsymbol{x}}_s)\circ d\overrightarrow{\boldsymbol{B}}_s, \tag{L.15}$$

$$d\overleftarrow{\boldsymbol{x}}_t = g(T-t)\mathbf{G}(\overleftarrow{\boldsymbol{x}}_t)\left(\boldsymbol{a_\theta}(T-t,\overleftarrow{\boldsymbol{x}}_t)dt + \circ d\overleftarrow{\boldsymbol{B}}_t\right), \tag{L.16}$$

where $\boldsymbol{a_\theta}(T-t,\overleftarrow{\boldsymbol{x}}_t)$ approximates $g(T-t)\mathbf{G}(\overleftarrow{\boldsymbol{x}}_t)^{\intercal}\nabla\log p_{T-t}(\overleftarrow{\boldsymbol{x}}_t)$. Numerically, following Algorithm 2 we can integrate the forward SDE with

$$\boldsymbol{f}_S(s,\overrightarrow{\boldsymbol{x}}_s) = 0 \quad\text{and}\quad \tilde{\mathbf{G}}(s,\overrightarrow{\boldsymbol{x}}_s) = g(s)\mathbf{G}(\overrightarrow{\boldsymbol{x}}_s), \tag{L.17}$$

and the backward SDE with

$$\boldsymbol{f}_S(t,\overleftarrow{\boldsymbol{x}}_t) = g(T-t)\mathbf{G}(\overleftarrow{\boldsymbol{x}}_t)\boldsymbol{a_\theta}(T-t,\overleftarrow{\boldsymbol{x}}_t) \quad\text{and}\quad \tilde{\mathbf{G}}(t,\overleftarrow{\boldsymbol{x}}_t) = g(T-t)\mathbf{G}(\overleftarrow{\boldsymbol{x}}_t). \tag{L.18}$$

### L.2.3 THEORETICAL RESULTS

We can verify that our theoretical results remain under this time scheduling. The new Fokker-Planck equation is

$$\frac{\partial}{\partial s}p_s = \nabla_\perp \cdot \left(\tfrac{1}{2}g^2(s)\boldsymbol{\Sigma}(\boldsymbol{x})\nabla_\perp p_s(\boldsymbol{x})\right). \tag{L.19}$$

which can be rewritten as

$$\frac{\partial}{\partial s}p_s^g = \nabla_\perp \cdot \left(\tfrac{1}{2}\boldsymbol{\Sigma}(\boldsymbol{x})\nabla_\perp p_s^g(\boldsymbol{x})\right). \tag{L.20}$$

$$p_s^g = p_{z(s)}. \tag{L.21}$$

Besides, for $0 \le s' \le z(T)$ for $\beta_M > \beta_m$ Taylor expansion at $s'_0 = \frac{\beta_M+\beta_m}{4}T$ yields

$$z^{-1}(s') = \frac{-\beta_m T + \sqrt{\beta_m^2 T^2 + 4T(\beta_M-\beta_m)z}}{\beta_M-\beta_m} \tag{L.22}$$

$$= T + \frac{2}{\beta_M}(s'-s_0)\left(1 + \underset{s'\to s'_0}{o}(1)\right), \tag{L.23}$$

$$\underset{s'\to s'_0}{\longrightarrow} T. \tag{L.24}$$

Therefore, from the convergence of $p_s^g$ (already proofed) we have the convergence of $p_{s'} = p_{z^{-1}(s')}^g$. The rate of convergence is still exponential:

$$\|p_{s'} - p_\infty\|_{L^2(\mathbb{R}^d)}^2 = \|p_{z^{-1}(s')}^g - p_\infty\|_{L^2(\mathbb{R}^d)}^2, \tag{L.25}$$

$$\le \|p_{z^{-1}(0)}^g - p_\infty\|_{L^2(\mathbb{R}^d)}^2 \exp(-\alpha(\mathbf{G},d)z^{-1}(s')), \tag{L.26}$$

$$= \|p_0 - p_\infty\|_{L^2(\mathbb{R}^d)}^2 \exp\left(-\alpha(\mathbf{G},d)T\right) \tag{L.27}$$

$$\left(1 - \frac{\alpha(\mathbf{G},d)}{\beta_M/2}\left(s' - \frac{\beta_M+\beta_m}{4}T\right)\left(1 + \underset{s'\to\frac{\beta_M+\beta_m}{4}T}{o}(1)\right)\right).$$

Besides the ELBO remains valid :

$$p_0(\boldsymbol{x}) \geqslant \mathcal{E}_\infty(\boldsymbol{x}) \quad := \quad \mathbb{E}\left[\log p_0(\overrightarrow{\boldsymbol{x}}_T)\Big|\overrightarrow{\boldsymbol{x}}_0 = \boldsymbol{x}\right] \tag{L.28}$$
$$- \int_0^T \mathbb{E}_{\overrightarrow{\boldsymbol{x}}_s}\left[\tfrac{1}{2}\|\boldsymbol{a}_{\boldsymbol{\theta}}(\overrightarrow{\boldsymbol{x}}_s, s)\|^2 + \nabla \cdot (g(s)\mathbf{G}(\overrightarrow{\boldsymbol{x}}_s)\boldsymbol{a}_{\boldsymbol{\theta}}(\overrightarrow{\boldsymbol{x}}_s, s))\Big|\overrightarrow{\boldsymbol{x}}_0 = \boldsymbol{x}\right] ds.$$

The above results guaranty the applicability of a classical scheduling to MSGM. An analyze with a scheduling more adapted to MSGM has still to be done and is left to future work. For example, the work of Strasman et al. (2024) may be adapted to the multiplicative structure of our SDE.

## L.3 LOSS EVALUATION

Following the existing code (`https://github.com/CW-Huang/sdeflow-light`,Huang et al. (2021)), we sample final integration time $s$ of the forward SDEs uniformly on $[t_\epsilon, T]$ with $T = 1$ with $t_\epsilon$ small. According to Theorem 3.4.1, we consider the following SSM loss:

$$\mathcal{L}_{\text{SSM}}(\boldsymbol{\theta}) = \hat{\mathbb{E}}_{\overrightarrow{\boldsymbol{x}}_0}\hat{\mathbb{E}}_{s\sim\mathcal{U}[t_\epsilon,T]}\hat{\mathbb{E}}_{\overrightarrow{\boldsymbol{x}}_s|\overrightarrow{\boldsymbol{x}}_0}\hat{\mathbb{E}}_{v_s\sim\text{Rad}(d)}\mathcal{L}_{\text{SSM}}(s, \overrightarrow{\boldsymbol{x}}_s, g\mathbf{G}, \boldsymbol{a}_{\boldsymbol{\theta}_n}, v_s), \tag{L.29}$$

with

$$\mathcal{L}_{\text{SSM}}(s, \boldsymbol{x}, g\mathbf{G}, \boldsymbol{a}_{\boldsymbol{\theta}_n}, \mathbf{v}) = \tfrac{1}{2}\|\boldsymbol{a}_{\boldsymbol{\theta}}(\boldsymbol{x}, s)\|^2 + (\mathbf{v} \cdot \nabla)(g(s)\mathbf{G}(\boldsymbol{x})\boldsymbol{a}_{\boldsymbol{\theta}}(\boldsymbol{x}, s)) \cdot \mathbf{v}, \tag{L.30}$$

where $\hat{\mathbb{E}}$ is the averaged over the generated samples. Each training sample $\overrightarrow{\boldsymbol{x}}_0$ of a batch is chosen randomly among the train set. For each of them, we sample one time $s$, one solution $\overrightarrow{\boldsymbol{x}}_s$, and one slicing direction $v_s \sim \text{Rad}(d)$.

For SGM, we take $\mathbf{G} = \sqrt{2}$ in the above expressions and following Song et al. (2021), the solution $\overrightarrow{\boldsymbol{x}}_s$ of the SGM scheduled forward SDE equation L.12 is

$$\overrightarrow{\boldsymbol{x}}_s = \exp(-\tfrac{1}{2}z(s))\overrightarrow{\boldsymbol{x}}_0 + \sqrt{1 - \exp(-z(s))}\overrightarrow{\boldsymbol{x}}_\infty, \tag{L.31}$$

where $z(s) := \int_0^s g^2(s')ds'$ is given by equation L.9, $\overrightarrow{\boldsymbol{x}}_0$ is chosen randomly among the train set and $\overrightarrow{\boldsymbol{x}}_\infty \sim \mathcal{N}(0, \boldsymbol{I}_d)$.

Unfortunately, to evaluate the MSGM loss, we cannot apply the same methodology, since, for $d > 2$ we are not aware of an analytic expression for the solution of the MSGM forward SDE, neither with nor without scheduling (equation L.15 and equation 3.1 respectively). We integrate that SDE numerically with the stochastic Runge-Kutta method with renormalization (see Section L.1.1 and Section L.1.2). Through this integration, we have to compute the solution $\overrightarrow{\boldsymbol{x}}_{s_k}$ for many time steps $s_k := kT/N_T \in [0, T]$. Instead of sampling a random continuous time $s \sim \mathcal{U}([t_\epsilon, T])$, we choose a random discrete time as follow

$$s \sim \mathcal{U}(I(t_\epsilon, T)) \quad \text{with} \quad I(t_\epsilon, T) = \{s_k|s_k = k\tfrac{T}{N_T}, k \in \{1, \ldots, N_T\}, s_k \geqslant t_\epsilon\}. \tag{L.32}$$

The numerical integration of the forward SDE implies a larger computational cost compared to SGM. Therefore, as explained in Section M.3, for fair comparisons between SGM and MSGM, the number of ADAMS iterations will be smaller.

For two-dimensional test cases, we could have used the analytic example of Section I to integrate the forward MSGM SDE. However, we prefer to propose and analyze an algorithm that is not tied to the dimension 2. So, we perform all our numerical experiments with the same algorithm whatever the dimension. SGM forward equation is integrated analytically, whereas the MSGM is integrated numerically.

## L.4 NEURAL NETWORK ARCHITECTURE

### L.4.1 SPHERICAL DECOMPOSITION AS AN INPUT LAYER

In line with our spherical decomposition equation 3.6, we add a fixed input layer to the network used in MSGM:

$$\boldsymbol{a}_{\boldsymbol{\theta}}(\boldsymbol{x}, s) = \tilde{\boldsymbol{a}}_{\boldsymbol{\theta}}\left(\boldsymbol{x}/\|\boldsymbol{x}\|_\epsilon, \log\|\boldsymbol{x}\|_\epsilon, s\right), \quad \text{with} \quad \|\boldsymbol{x}\|_\epsilon := \|\boldsymbol{x}\| + \epsilon. \tag{L.33}$$

The geometrical interpretation of Section H.3 also suggests that form.

For SGM, if not stated otherwise, we use a default architecture:

$$\boldsymbol{a_\theta}(\boldsymbol{x}, s) = \tilde{\boldsymbol{a}}_{\boldsymbol{\theta}}(\boldsymbol{x}, s). \tag{L.34}$$

### L.4.2 NETWORK ARCHITECTURE FOR LOW-DIMENSIONAL TEST CASES (MLP)

Following the existing code (https://github.com/CW-Huang/sdeflow-light, Huang et al. (2021)), we parameterize the vector field $\tilde{\boldsymbol{a}}_{\boldsymbol{\theta}} : \mathbb{R}^{\tilde{d}} \times \mathbb{R} \to \mathbb{R}^d$ ($\tilde{d} = d$ or $d + 1$) with a 4-layer MLP conditioned on an index $t \in \mathbb{R}$ by concatenation. Let $H = 128$ be the hidden width. For input $\boldsymbol{x} \in \mathbb{R}^{\tilde{d}}$, we form $\boldsymbol{h}_0 = [\boldsymbol{x}; t] \in \mathbb{R}^{\tilde{d}+1}$ and compute

$$\begin{aligned}
\boldsymbol{h}_1 &= \mathrm{swish}(\boldsymbol{W}_1 \boldsymbol{h}_0 + \boldsymbol{b}_1), & \boldsymbol{W}_1 &\in \mathbb{R}^{H \times (\tilde{d}+1)}, \\
\boldsymbol{h}_2 &= \mathrm{swish}(\boldsymbol{W}_2 \boldsymbol{h}_1 + \boldsymbol{b}_2), & \boldsymbol{W}_2 &\in \mathbb{R}^{H \times H}, \\
\boldsymbol{h}_3 &= \mathrm{swish}(\boldsymbol{W}_3 \boldsymbol{h}_2 + \boldsymbol{b}_3), & \boldsymbol{W}_3 &\in \mathbb{R}^{H \times H}, \\
\boldsymbol{y} &= \boldsymbol{W}_4 \boldsymbol{h}_3 + \boldsymbol{b}_4, & \boldsymbol{W}_4 &\in \mathbb{R}^{d \times H},
\end{aligned}$$

with $(\mathrm{swish}(z))_i = z_i\, \sigma(z_i)$ and $\sigma$ the logistic sigmoid. We set $\tilde{\boldsymbol{a}}_{\boldsymbol{\theta}}(\boldsymbol{x}, t) = \boldsymbol{y} \in \mathbb{R}^d$. No residual connections, normalization, or dropout are used. Table 1 summarizes the hyperparameters of this default architecture.

Table 1: MLP architecture hyperparameters.

| Hyperparameter | Value |
|---|---|
| Input dimension | $\tilde{d} = d$ or $d + 1$ |
| Index dimension | 1 |
| Hidden width | 128 |
| Depth | 3 hidden layers |
| Activation | Swish ($x \mapsto x\sigma(x)$) |
| Output dimension | $d$ |
| Output layer | Linear |
| Residual connections | None |
| Normalization / Dropout | None |

### L.4.3 NETWORK ARCHITECTURE FOR HIGH-DIMENSIONAL TEST CASES (UNET FOR $32 \times 32$ VORTICITY FIELDS)

For high-dimensional experiments of Section M.7.2, we model the score field $\tilde{\boldsymbol{a}}_{\boldsymbol{\theta}}(\boldsymbol{x}, t)$ using a 2D UNet operating on images $\boldsymbol{x}'$ of size $H \times W$ representing vorticity snapshots ($H = W = 16$ or 32). Some part of our algorithm was built for vectors rather than images. So depending on the portion of the algorithm, images $\boldsymbol{x}' \in \mathbb{R}^{1 \times H \times W}$ are reshaped into vectors $\boldsymbol{x} \in \mathbb{R}^d$ with $d = HW$ or vectors are reshaped as a one–channel images $\boldsymbol{x}' \in \mathbb{R}^{1 \times H \times W}$.

**Optional spherical premodule.** When enabled, we apply the spherical decomposition of Section L.4.1:

$$(\boldsymbol{x}_\epsilon^n, \log \|\boldsymbol{x}\|_\varepsilon) = \mathrm{NormalizeLogRadius}(x), \qquad \boldsymbol{x}_\epsilon^n = \frac{x}{\|\boldsymbol{x}\|_\varepsilon}.$$

The normalized field $\boldsymbol{x}_\epsilon^n$ is passed to the UNet, while $\log \|\boldsymbol{x}\|_\varepsilon$ is embedded through a small MLP and added to the temporal embedding, giving a conditioning mechanism analogous to the time embedding of diffusion models.

**UNet backbone.** The core architecture follows the DDPM UNet of Dhariwal & Nichol (2021): a fully convolutional encoder–decoder with skip connections, residual blocks, and optional attention at intermediate resolutions. We use one input channel and one output channel (vorticity). Let $C_0$ denote the base width. The feature width at resolution level $k$ is $C_0 m_k$ where $C_0$ is the base channel width and $m_k$ is the channel multiplier.

The UNet receives $(\boldsymbol{x}, t)$ (and optionally $\log \|\boldsymbol{x}\|_\epsilon$) and computes:

$$\tilde{\boldsymbol{a}}_\theta(\boldsymbol{x}, t) = \text{UNet}_\theta(\text{reshape}(x), \text{Emb}(t) + \text{Emb}_{\log}(\log \|\boldsymbol{x}\|_\epsilon)),$$

followed by flattening back to dimension $d$ if needed.

Table 2: UNet architecture hyperparameters.

| Component | Setting |
|---|---|
| Input / output channels | 1 |
| Input resolution | $H = W \in \{16, 32\}$ |
| Base channels width $C_0$ | 32 |
| Channel multipliers | $(1, 2, 4)$ |
| Residual blocks per stage | 2 |
| Attention resolutions | $8 \times 8$ and $4 \times 4$ (for $16 \times 16$ input) |
| Activation | SiLU |
| Time embedding | sinusoidal + MLP |
| Log-norm conditioning | optional MLP added to time embedding |
| Dropout | 0 |
| Upsampling / downsampling | convolutional |
| Output | 1-channel vorticity field |

This UNet is used as a drop-in replacement for the small-dimensional MLP of Section L.4.2, enabling MSGM/SGM to scale to image-like vorticity fields up to dimension $d = 1024$.

## M  DETAILS ABOUT OUR NUMERICAL EXPERIMENTS

We will show that – for comparable training time – MSGM can generate distribution of better quality than SGM when data distribution tails are heavy or close to being heavy. For distributions with lighter tails such as Gaussian ones, SGM and MSGM produce similar results, except for a small number of backward time steps where SGM can become unstable. MSGM is more robust in this aspect.

Our code can be found here: `https://github.com/vressegu/sdeflow-light/tree/3rdSub` and the preprocessed vorticity data we used in Section M.7 can be found here: `https://github.com/vressegu/MSGM-data`. Original experimental data (Georgeault & Heitz, 2026) are freely available at `https://doi.org/10.57745/DHJXM6`, through the multidisciplinary repository Recherche Data Gouv (`https://entrepot.recherche.data.gouv.fr`).

### M.1  TEST CASES

We will illustrate MSGM and compare it to SGM through different test cases. We first consider four examples sampled from known distributions: the Swiss roll, a multidimensional Gaussian distribution, and the multidimensional Cauchy distribution with and without correlations. Then, we will address the experimental fluid dynamics data. For each test case, a table summarizes the nominal parameters used in the experiments (see tables 3, 4, 5, 6, and 7). All are performed on CPU. In addition, we additional cover a high-dimensional application with imagine processing, see section Section M.7 with a GPU A40 NVL with 48 Go of VRAM.

### M.2  DATA PREPROCESSING

The data set and distribution are centered before processing. For SGM, data sets are renormalized, component by component, by their estimated standard deviations. This preconditioning can significantly reduce the number of conditioning of the covariance of the data set, and therefore facilitate the SGM (Guth et al., 2022). Generated data are then re-scaled for plots and other post-processings. For MSGM, it is not necessary and may even be counterproductive for conservative dynamical systems. In fact, it changes the definition of energy $\|\vec{\boldsymbol{x}}_0\|^2$. The modified energy has no physical meaning. It may have a very different distribution, possibly much less relevant for the data structure. So, we do not renormalize the data set before training MSGM.

Table 3: Swiss roll test case: parameters of the nominal numerical experiments.

| Parameter | SGM | MSGM |
|---|---|---|
| Dimension $d$ | 2 | 2 |
| Number of used training data points ($M$) | $2^{20} \times 256$ | $2^{20} \times 256$ |
| Number of test data points | $10^4$ | $10^4$ |
| Reference number of ADAMS steps | $2^{20}$ | $2^{20}$ |
| Number of ADAMS steps ($N_{\text{iter}}$) | $2^{20}$ | $2^{20}$ |
| CPU time / ADAMS steps (in ms) | 4 | 3 |
| Batch size | 256 | 256 |
| Number of time steps (forward) $N_T^f$ | 1 | 16 |
| Number of time steps (backward) $N_T^b$ | 16 | 16 |
| $\beta_{\min}$ | 0.1 | 0.1 |
| $\beta_{\max}$ | 20 | 20 |
| $t_\varepsilon$ | $10^{-3}$ | $10^{-3}$ |
| Learning rate | $10^{-3}$ | $10^{-3}$ |
| Neural network architecture | default | spherical (equation L.33) |
| MMD (MMD(train)= $0.9 \times 10^{-2}$) | $1.9 \times 10^{-2}$ | $0.9 \times 10^{-2}$ |

## M.3 Comparison strategy

We will perform different qualitative visual comparisons with pairplots and quantitative assessment with Maximum Mean Discrepancy (MMD) (Gretton et al., 2012). Given two ensembles $\mathbb{X} = (x^{(i)}) \in (\mathbb{R})^N$ and $\mathbb{Y} = (y^{(i)}) \in (\mathbb{R})^N$ samples of random variables $X$ and $Y$ respectively, we define $\text{MMD}(x, y)$ as:

$$\text{MMD}^2(\mathbb{X}, \mathbb{Y}) \quad = \quad \frac{1}{N^2} \sum_{i,j=1}^{N} \left( k(x^{(i)}, x^{(j)}) - 2k(x^{(i)}, y^{(j)}) + k(y^{(i)}, y^{(j)}) \right), \qquad \text{(M.1)}$$

$$k(u, v) \quad = \quad \exp(-\|u - v\|^2). \qquad \text{(M.2)}$$

If $\mathbb{X}^{\text{test}}$ is the test set and $\mathbb{X}^{gen}$ our generated ensemble, $\text{MMD}(\mathbb{X}^{\text{test}}, \mathbb{X}^{gen})$ is a metric of the precision of our generated ensemble and hence our AI generative algorithm. A small MMD means close distributions. However, MMD is a relative metric. So we compare $\text{MMD}(\mathbb{X}^{\text{test}}, \mathbb{X}^{\text{gen}}_{\text{SGM}})$, $\text{MMD}(\mathbb{X}^{\text{test}}, \mathbb{X}^{\text{gen}}_{\text{MSGM}})$ and $\text{MMD}(\mathbb{X}^{\text{test}}, \mathbb{X}^{\text{train}})$ where $\mathbb{X}^{\text{gen}}_{\text{SGM}}$ and $\mathbb{X}^{\text{gen}}_{\text{MSGM}}$ are generated from SGM and MSGM respectively, and $\mathbb{X}^{\text{train}}$ is the train set. $\text{MMD}(\mathbb{X}^{\text{test}}, \mathbb{X}^{\text{train}})$ provides a reference MMD, encoding in particular possible distribution shifts between the train and the test sets.

The numerical integration of the MSGM forward SDE is an additional significant computational cost compared to SGM, and hence a slower training procedure. This cost scales linearly in $N_T$ due to the "for" loop in time. Empirically, it appears to scale as $\zeta = \sqrt{d} N_t / 2^4$ (not shown), probably due to the vectorized $d \times d \times d$ tensor products involved in each integration time step. In most of the numerical experiments below, $N_t = 2^4$ and thus $\zeta = \sqrt{d}$. The SGM iteration steps are $\zeta$ times faster than the MSGM iteration steps. Consequently, the number of iterations for the SGM is $\max(1, \lfloor \zeta \rfloor)$ times larger than the number of iterations for the MSGM. As such, we can compare the results of SGM and MSGM at a similar training cost. By convention, we take the number of iterations for SGM as a reference and refer to it as the reference number of iterations. Summary tables 3, 4, 5, 6, and 7) provide the values for the reference number of iterations, the true number of iterations, and the execution time per ADAMS step.

## M.4 Swiss roll

We first illustrate our method with the Swiss roll distribution. It is a simple two-dimensional distribution: `https://homepages.ecs.vuw.ac.nz/~marslast/Code/Ch6/lle.py`. Its curved shape makes it difficult to grasp by linear Gaussian approaches. Both MSGM and SGM mimic the Swiss roll distribution well, as illustrated by the pairplot 8. However, the diffusion distribution $p_s$ differs from Figure 9 to Figure 10. In particular, latent distributions are completely

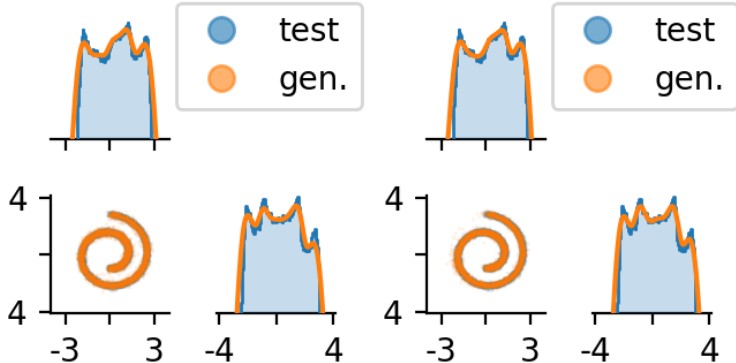

Figure 8: Pair plots of generated data (orange dots) compared to ground truth data (blue dots) with the SGM (left) and MSGM (right) for Swissroll data. On the diagonal, log-histogram of ground truth data (continuous blue line) and logarithm of the pdf KDE estimation of generated data (orange line) are superimposed.

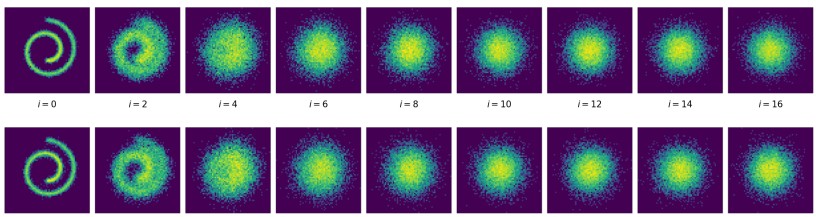

Figure 9: Evolution of the solution log-pdf $\log(p_s(x_1, x_2))$ of SGM forward SDE (top) and backward SDE (bottom) for Swiss roll data.

different. Figure 11 illustrates the convergence of the SGM and MSGM approaches as a function of the reference number of ADAMS iterations and as a function of number of time steps for integrating the backward SDE. The precision of each sampler is quantified through MMD and the confidence intervals of MMD are estimated from the samples of 10 MMD.

## M.5 ANISOTROPIC GAUSSIAN DISTRIBUTION

For a complete numerical analysis, we compare SGM and MSGM on correlated Gaussian data $x_0 \sim \mathcal{N}(0, AA^T)$, with a fixed matrix, $A$, initialized with i.i.d. coefficients $A_{i,j} \sim \mathcal{N}(0, 1)$. For 32 time steps backward, the pairplots in Figures 12, 13, and 14 present similar generative skills, but for 8 or 16 time steps backward, only MSGM gives good results. For 8 time steps backwards, MSGM still provides a good distribution, whereas the SGM backward SDE completely diverges. Figures 15,

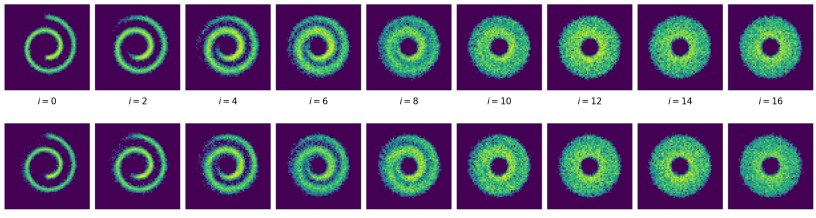

Figure 10: Evolution of the solution log-pdf $\log(p_s(x_1, x_2))$ of MSGM forward SDE (top) and backward SDE (bottom) for Swiss roll data.

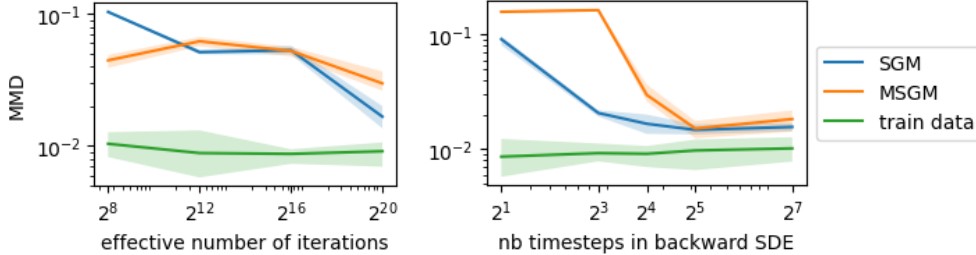

Figure 11: Convergence of MMD (mean and 80% confidence interval) for Swiss roll distribution as a function of reference number of ADAMS iterations (left) and as a function of number of time steps for integrating the backward SDE (right).

Table 4: Gaussian test case: parameters of the nominal numerical experiments.

| Parameter | SGM | MSGM |
|---|---|---|
| Dimension $d$ | 16 | 16 |
| Number of used training data points $(M)$ | $1048576 \times 256$ | $262144 \times 256$ |
| Number of test data points | $10^4$ | $10^4$ |
| Reference number of ADAMS steps | $2^{20}$ | $2^{20}$ |
| Number of ADAMS steps $(N_{\text{iter}})$ | $2^{20} = 1048576$ | $262144$ |
| CPU time / ADAMS steps (in ms) | 3 | 23 |
| Batch size | 256 | 256 |
| Number of time steps (forward) $N_T^f$ | 1 | 16 |
| Number of time steps (backward) $N_T^b$ | 16 | 16 |
| $\beta_{\min}$ | 0.1 | 0.1 |
| $\beta_{\max}$ | 20 | 20 |
| $t_\varepsilon$ | $10^{-3}$ | $10^{-3}$ |
| Learning rate | $10^{-3}$ | $10^{-3}$ |
| Neural network architecture | default | spherical (equation L.33) |
| MMD (MMD(train)=$1.5 \times 10^{-2}$) | $11 \times 10^{-2}$ | $2.5 \times 10^{-2}$ |

16, and 17 also highlight this and show that the converged dynamics of the pdf $p_s$ differs between SGM and MSGM. Figure 18 also confirms that MSGM converges faster with the number of time steps, and is generally more stable.

## M.6 MULTIVARIATE CAUCHY DISTRIBUTION

Cauchy distributions are worst-case heavy-tail distributions in the sense that they do not have finite moments. Still, they appear in applications of hydrology, e.g. annual maximum one-day rainfalls and river discharges. Consequently, we analyze the expressivity of MSGM in this extreme case. Note that due to the absence of finite moments, convergence in common metrics such as Wasserstein-$p$ or total variation is not well defined.

### M.6.1 VECTOR OF INDEPENDENT CAUCHY VARIABLES

We first illustrate our method with a vector of independent Cauchy variables: $\boldsymbol{x}_0 = \boldsymbol{x}_{\text{Ca}}$ with $\boldsymbol{x}_{\text{Ca}}$ defined by equation 6.2 with scale parameter $\gamma = 1/50$. As expected, Figure 19 and Figure 20 confirm that SGM does not reproduce fat tails unlike MSGM. Moreover, SGM misaligns the far data points that have the coordinate $x_1 < -3$. An explanation of the superior skills is the similarity between the data distribution and the latent distribution in MSGM: a property not shared by SGM, as illustrated in Figures 21 and 22.

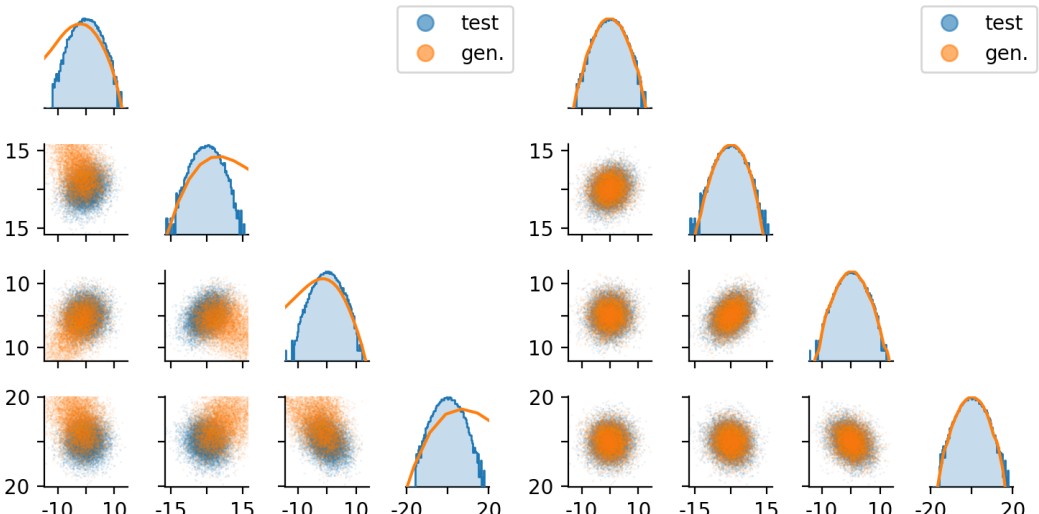

Figure 12: Pair plots of generated data (orange dots) compared to ground truth data (blue dots) with the SGM (left) and MSGM (right) with 8 time steps backward for a vector of 4 correlated Gaussian variables, among 16 correlated Gaussian variables used for training. On the diagonal, log-histogram of ground truth data (continuous blue line), and logarithm of the pdf KDE estimation of generated data (orange line) are superimposed.

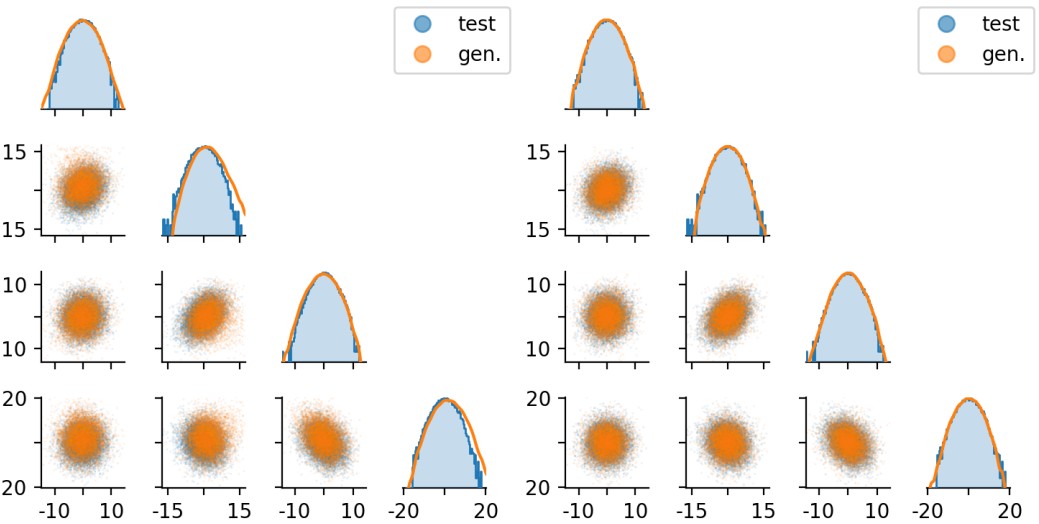

Figure 13: Pair plots of generated data (orange dots) compared to ground truth data (blue dots) with the SGM (left) and MSGM (right) with 16 time steps backward for a vector of 4 correlated Gaussian variables, among 16 correlated Gaussian variables used for training. On the diagonal, log-histogram of ground truth data (continuous blue line), and logarithm of the pdf KDE estimation of generated data (orange line) are superimposed.

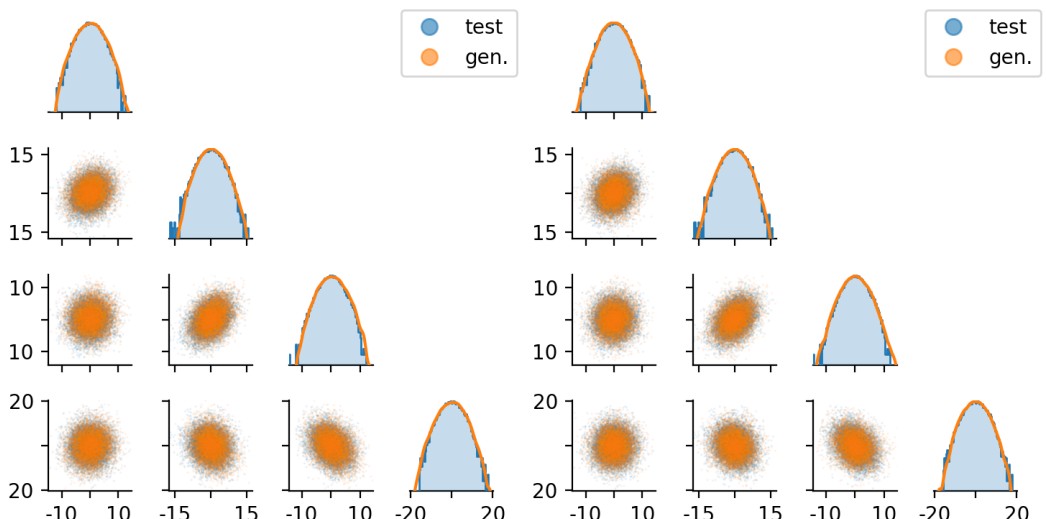

Figure 14: Pair plots of generated data (orange dots) compared to ground truth data (blue dots) with the SGM (left) and MSGM (right) with 32 time steps backward for a vector of 4 correlated Gaussian variables, among 16 correlated Gaussian variables used for training. On the diagonal, log-histogram of ground truth data (continuous blue line), and logarithm of the pdf KDE estimation of generated data (orange line) are superimposed.

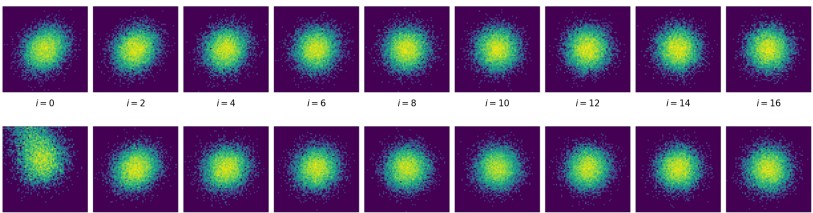

Figure 15: Evolution of the solution log-pdf $\log(p_s(x_1, x_2))$ of SGM forward SDE (top) and backward SDE (bottom, with 8 time steps) for Gaussian data.

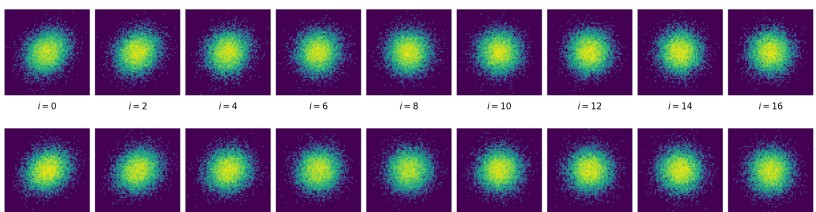

Figure 16: Evolution of the solution log-pdf $\log(p_s(x_1, x_2))$ of SGM forward SDE (top) and backward SDE (bottom, with 32 time steps) for Gaussian data.

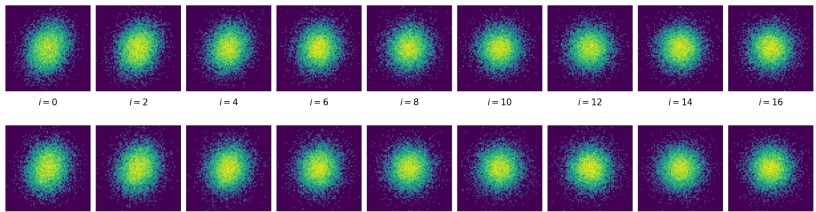

Figure 17: Evolution of the solution log-pdf $\log(p_s(x_1, x_2))$ of MSGM forward SDE (top) and backward SDE (bottom, with 16 time steps) for Gaussian data.

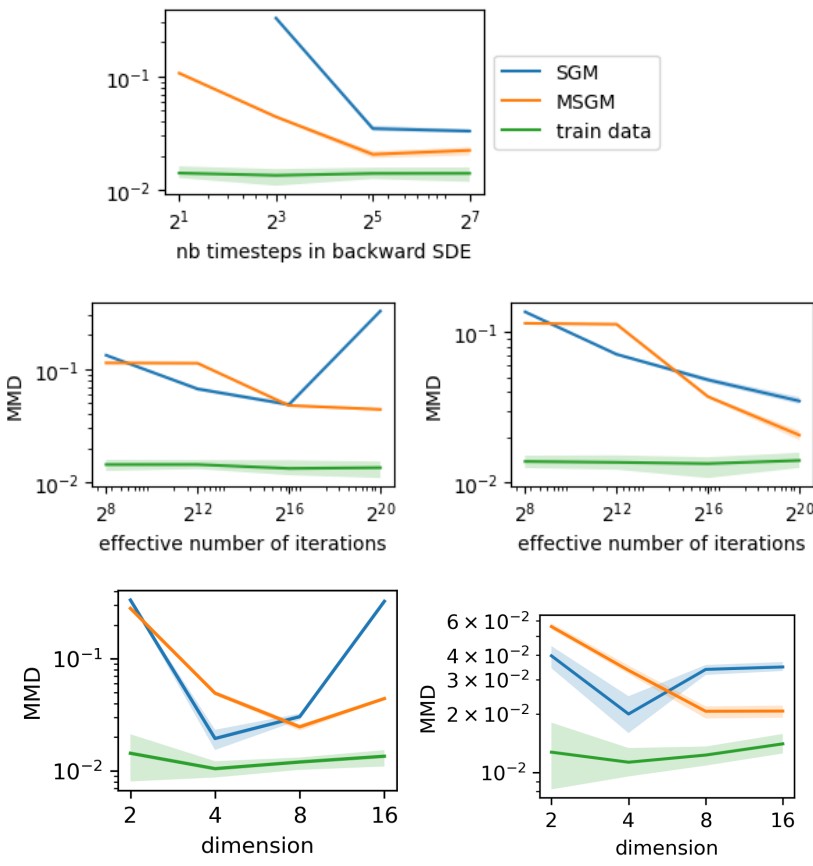

Figure 18: Convergence of MMD (mean and 80% confidence interval) for the Gaussian data as a function of number of time steps for integrating the backward SDE $N_T^b$ (top), as a function of reference number of ADAMS iterations (middle) for $N_T^b = 8$ (left) and $N_T^b = 32$ (right), and as a function of dimension (bottom) for $N_T^b = 8$ (left) and $N_T^b = 32$ (right).

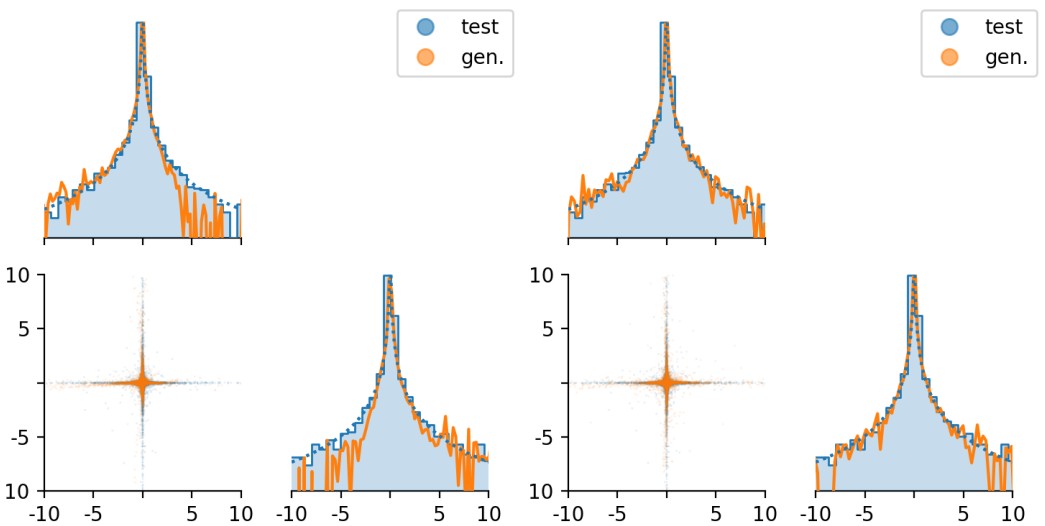

Figure 19: Pair plots of generated data (orange dots) compared to ground truth data (blue dots) with the SGM (left) and MSGM (right) for a vector of two independent Cauchy variables. On the diagonal, log-histogram of ground truth data (continuous blue line), theoretical log-pdf (dashed blue line), and logarithm of the pdf KDE estimation of generated data (orange line) are superimposed.

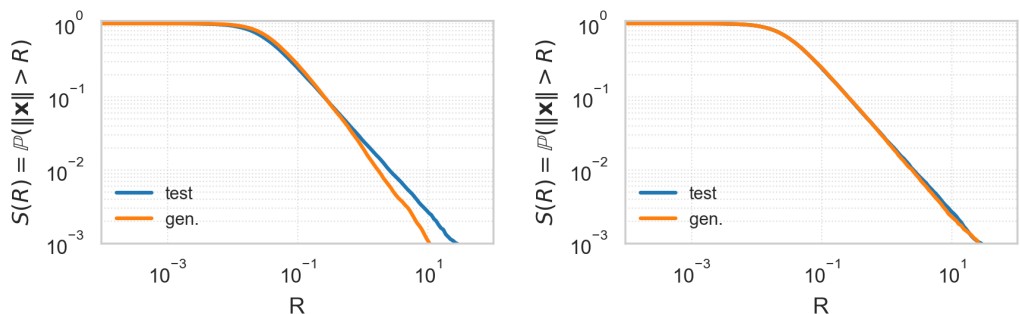

Figure 20: Survival function of generated data (orange line) compared to ground truth data (blue line) with the SGM (left) and MSGM (right) for a vector of two independent Cauchy variables.

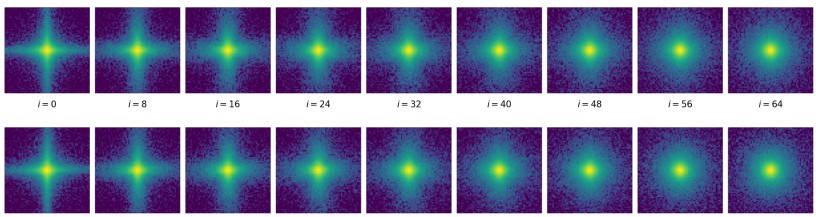

Figure 21: Evolution of the solution log-pdf $\log(p_s(x_1, x_2))$ of MSGM forward SDE (top) and backward SDE (bottom) for a vector of two independent Cauchy variables, with fast scheduling: $\beta_m = 0.1$, $\beta_M = 0.4$ and our neural network architecture based on spherical decomposition equation L.33.

Table 5: Vector of independent Cauchy variables: parameters of the nominal numerical experiments.

| Parameter | SGM | MSGM |
|---|---|---|
| Dimension $d$ | 2 | 2 |
| Number of used training data points ($M$) | $2^{20} \times 256$ | $209715 \times 256$ |
| Number of test data points | $10^5$ | $10^5$ |
| Reference number of ADAMS steps | $2^{20}$ | $2^{20}$ |
| Number of ADAMS steps ($N_{\text{iter}}$) | $2^{20} = 1048576$ | $209715$ |
| CPU time / ADAMS steps (in ms) | 3 | 27 |
| Batch size | 256 | 256 |
| Number of time steps (forward) $N_T^f$ | 1 | 64 |
| Number of time steps (backward) $N_T^b$ | 128 | 128 |
| $\beta_{\min}$ | 0.1 | 0.1 |
| $\beta_{\max}$ | 20 | 0.4 |
| $t_\varepsilon$ | $10^{-3}$ | $10^{-3}$ |
| Learning rate | $10^{-3}$ | $10^{-3}$ |
| Neural network architecture | default | spherical (equation L.33) |
| MMD (MMD(train)=$2.8 \times 10^{-3}$) | $7.5 \times 10^{-3}$ | $3.3 \times 10^{-3}$ |

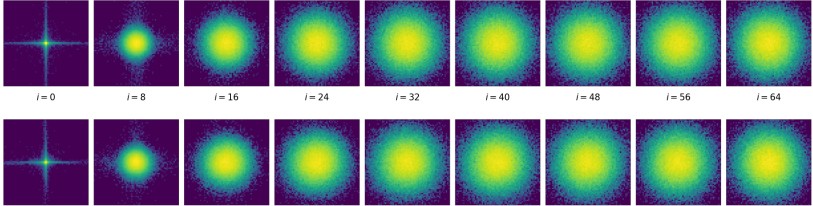

Figure 22: Evolution of the solution log-pdf $\log(p_s(x_1, x_2))$ of SGM forward SDE (top) and backward SDE (bottom) for a vector of two independent Cauchy variables, with default scheduling: $\beta_m = 0.1$, $\beta_M = 20$ and default neural network architecture.

Figure 19 compares MSGM with fast scheduling ($\beta_m = 0.1$, $\beta_M = 0.4$) and a neural network architecture based on spherical decomposition equation L.33 with SGM with default scheduling ($\beta_m = 0.1$, $\beta_M = 20$) and default neural network architecture. For a fair comparison of MSGM, we complement our numerical analysis with Figures 23-27: we test SGM with fast and default schedulings, and with both spherical-decomposition-based and default network architectures. This fast scheduling seems not adapted to SGM, making the sample generation highly inaccurate in the pairplot of Figure 24. In contrast, the network architecture with spherical decomposition does improve the SGM sampling procedure, especially for distribution tails. However, even with this architecture, SGM remains less efficient than MSGM. First, the estimated tail is less clean. Secondly, the samples generated far are not properly aligned with the test samples, especially for $x_2 < -3$. Third, outside the $x_1$ and $x_2$ axes, SGM generates too few samples close to the origin (say points $\boldsymbol{x}$ with $\|\boldsymbol{x}\|_{\frac{1}{2}} > 2$ and $\|\boldsymbol{x}\|_1 < 2$).

For Cauchy distributions, we still compare MMD values. However, it is not well defined mathematically and is hardly relevant numerically. Indeed, the Gaussian kernel structure of the MMD is probably not adapted to samples that are so far from each other. In our experiments, we used $10^4$ samples to compute an approximate MMD. Other quantities of interest can also be utilized, such as the survival function $t \mapsto \mathbb{P}(\|\boldsymbol{x}\| > R)$, illustrated in Figure 20. As expected MSGM clearly outperforms SGM on this metric. Indeed by construction our learning method is robust in terms of the radial distribution $\|\boldsymbol{x}\|$ obtained directly from the data and not after the noising process. This is valid since the norm distribution does not change in time due to equation 3.7.

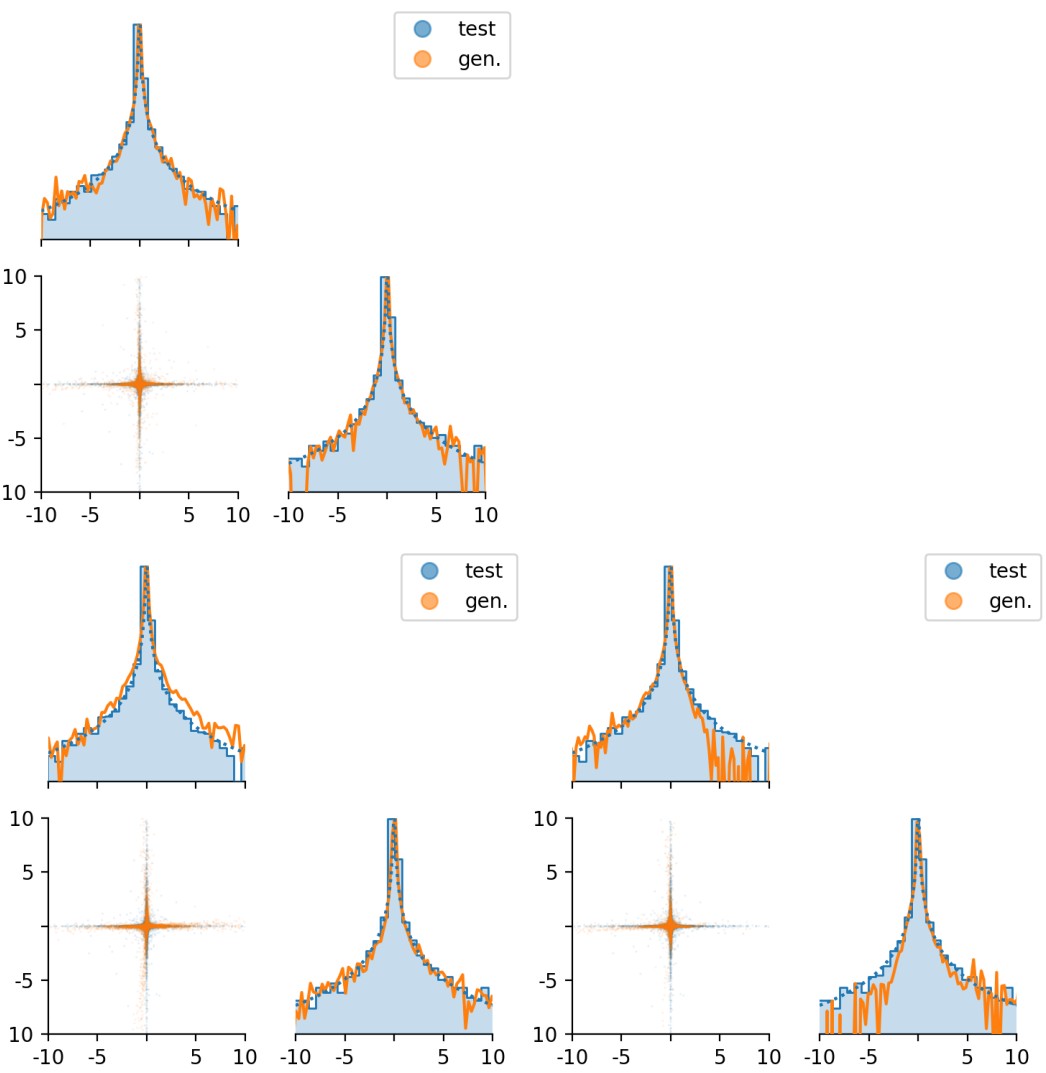

Figure 23: Generated data (orange lines and dots) compared to ground truth data (blue lines and dots) with the MSGM (top left corner) and the SGM (bottom) for two-dimensional Cauchy distribution. SGM plots correspond to a default scheduling: $\beta_m = 0.1$, $\beta_M = 20$. Left plots correspond to our neural network architecture based on spherical decomposition equation L.33 whereas the right plot correspond to default neural network architecture. On the diagonal, log-histogram of ground truth data (continuous blue line), theoretical log-pdf (dashed blue line), and logarithm of the pdf KDE estimation of generated data (orange line) are superimposed.

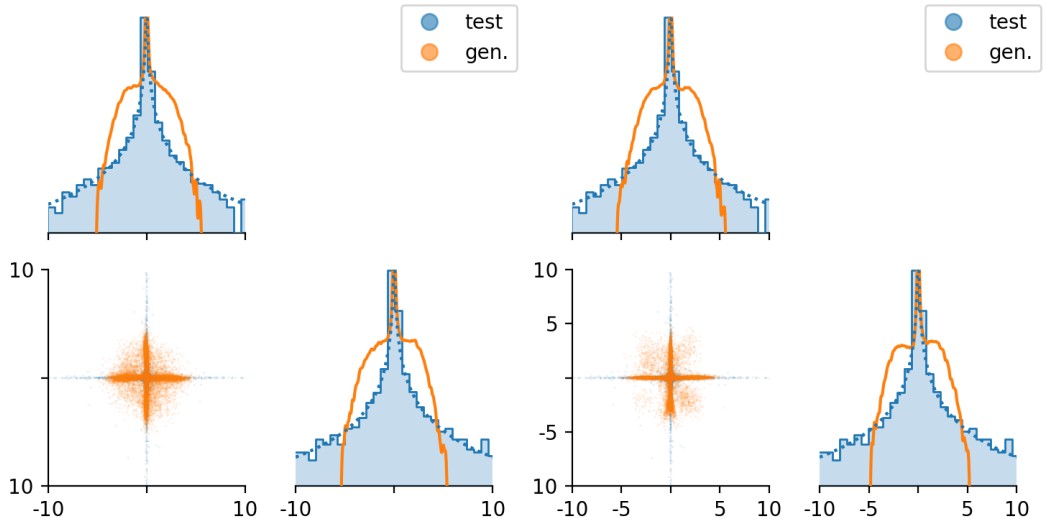

Figure 24: Generated data (orange lines and dots) compared to ground truth data (blue lines and dots) with the SGM for two-dimensional Cauchy distribution. Plots correspond to a fast scheduling: $\beta_m = 0.1$, $\beta_M = 0.4$. The left plot corresponds to our neural network architecture based on spherical decomposition equation L.33 whereas right plot corresponds to default neural network architecture. On the diagonal, log-histogram of ground truth data (continuous blue line), theoretical log-pdf (dashed blue line), and logarithm of the pdf KDE estimation of generated data (orange line) are superimposed.

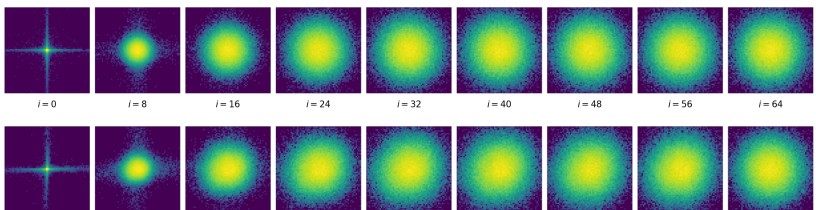

Figure 25: Evolution of the solution log-pdf $\log(p_s(x_1, x_2))$ of SGM forward SDE (top) and backward SDE (bottom) for a vector of two independent Cauchy variables, with default scheduling: $\beta_m = 0.1$, $\beta_M = 20$ and our neural network architecture based on spherical decomposition equation L.33.

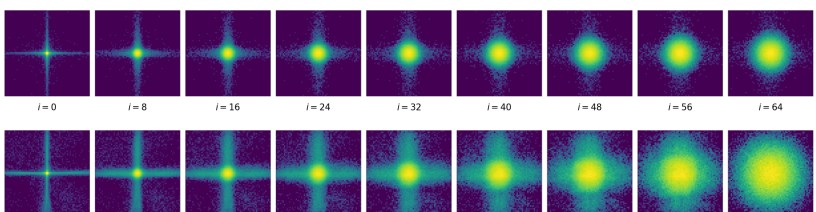

Figure 26: Evolution of the solution log-pdf $\log(p_s(x_1, x_2))$ of SGM forward SDE (top) and backward SDE (bottom) for a vector of two independent Cauchy variables, with fast scheduling: $\beta_m = 0.1$, $\beta_M = 0.4$ and default neural network architecture.

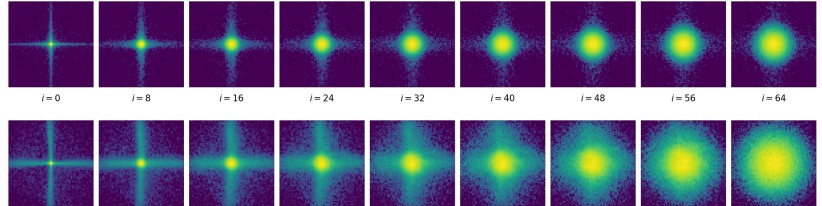

Figure 27: Evolution of the solution log-pdf $\log(p_s(x_1, x_2))$ of SGM forward SDE (top) and backward SDE (bottom) for a vector of two independent Cauchy variables, with fast scheduling: $\beta_m = 0.1$, $\beta_M = 0.4$ and our neural network architecture based on spherical decomposition equation L.33.

Table 6: Vector of correlated Cauchy variables: parameters of the nominal numerical experiments.

| Parameter | SGM | MSGM |
|---|---|---|
| Dimension $d$ | 4 | 4 |
| Number of used training data points ($M$) | $2^{20} \times 256$ | $2^{20} \times 256$ |
| Number of test data points | $10^5$ | $10^5$ |
| Reference number of ADAMS steps | $2^{20}$ | $2^{20}$ |
| Number of ADAMS steps ($N_{\text{iter}}$) | $2^{20}$ | $2^{20}$ |
| CPU time / ADAMS steps (in ms) | 3 | 45 |
| Batch size | 256 | 256 |
| Number of time steps (forward) $N_T^f$ | 1 | 128 |
| Number of time steps (backward) $N_T^b$ | 128 | 128 |
| $\beta_{\min}$ | 0.1 | 0.01 |
| $\beta_{\max}$ | 20 | 1 |
| $t_\varepsilon$ | $10^{-4}$ | $10^{-4}$ |
| Learning rate | $10^{-3}$ | $10^{-3}$ |
| Neural network architecture | default | spherical (equation L.33) |
| MMD (MMD(train)=$3.5 \times 10^{-3}$) | $11.2 \times 10^{-3}$ | $5.2 \times 10^{-3}$ |

### M.6.2   VECTOR OF CORRELATED CAUCHY VARIABLES

To address dimensionality issues, we consider the correlated Cauchy variables already presented in Section 6.1. In terms of survival function, MSGM is as expected more accurate than SGM (see Figure 28).

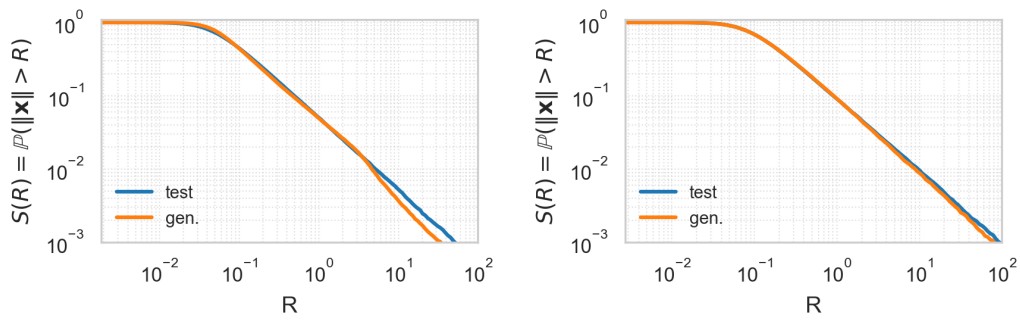

Figure 28: Survival function of generated data (orange line) compared to ground truth data (blue line) with the SGM (left) and MSGM (right) for a vector of $4$ correlated Cauchy variables.

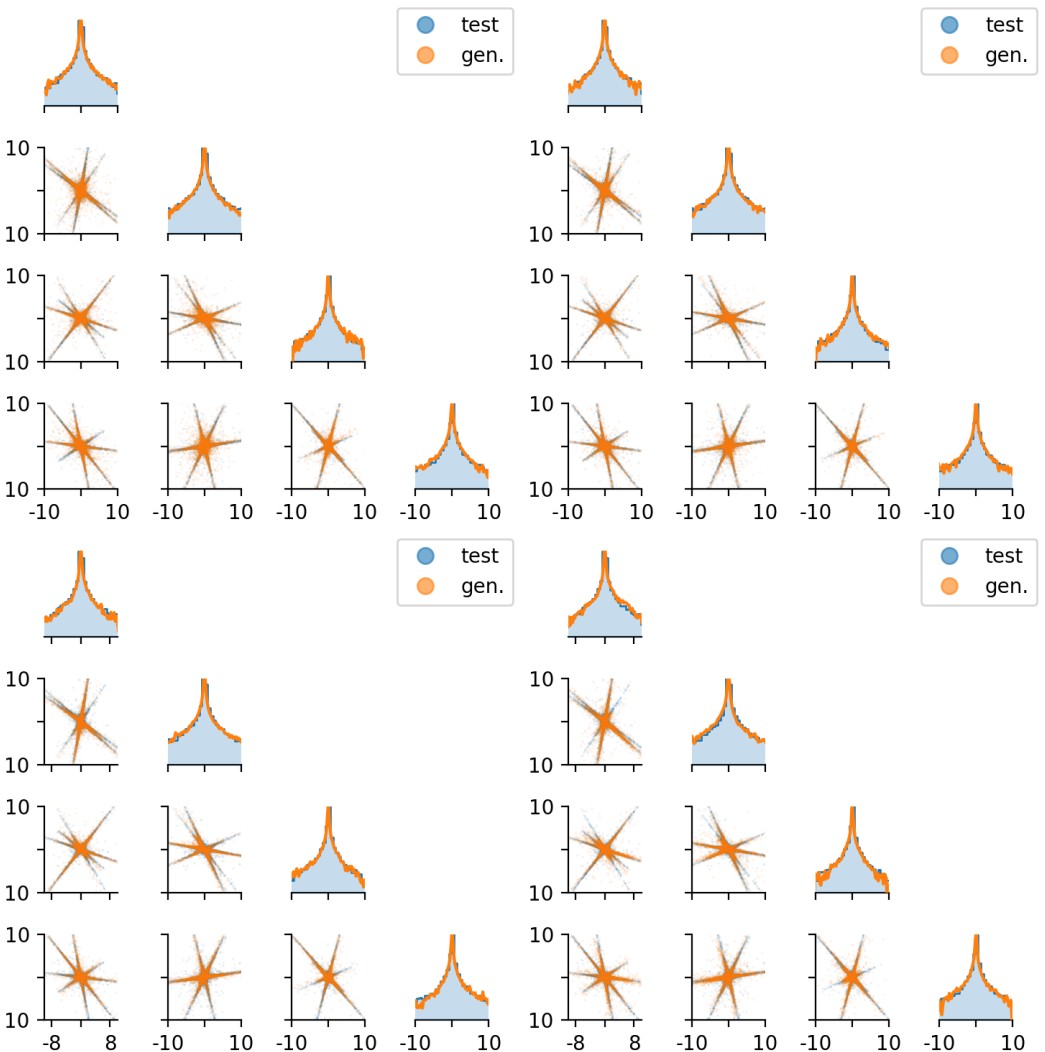

Figure 29: Generated data (orange lines and dots) compared to ground truth data (blue lines and dots) with the MSGM (top) with dense tensor **G** (top left corner), with sparse local tensor **G** (top right corner), and the SGM (bottom) for a vector of 4 correlated Cauchy variables. SGM correspond to a default scheduling: $\beta_m = 0.1$, $\beta_M = 20$. Left and top plots correspond to our neural network architecture based on spherical decomposition equation L.33 whereas the right bottom plot corresponds to default neural network architecture. On the diagonal, log-histogram of ground truth data (continuous blue line), and logarithm of the pdf KDE estimation of generated data (orange line) are superimposed.

As for independent Cauchy variables, we present complementary numerical experiments with different scheduling and different neural network architectures in Figures 29 and 30. Figures 31-35 unveil the corresponding diffusion dynamics from $s = 0$ to $s = T$ and from $t = 0$ to $t = T$. Again, the neural network architecture based on spherical decomposition significantly improves the SGM generative skills but MSGM remains a more efficient sampler. Not all the branches of the star-like pdf are well sampled and, outside the branches, the regions near the origin is not well sampled.

One can wonder if the poorer results of SGM would improve for a larger number of ADAMS iterations. To answer this question, we run longer experiments with $2^{24} = 16777216$ ADAMS iterations. Figures 36, 37, and 4a show that MSGM slightly improves with an increasing number of iterations, whereas SGM diverges. For a fair comparison, the MMD convergence Figure 4a is expressed in terms of effective number of ADAMS iterations, i.e. we proportionally reduce

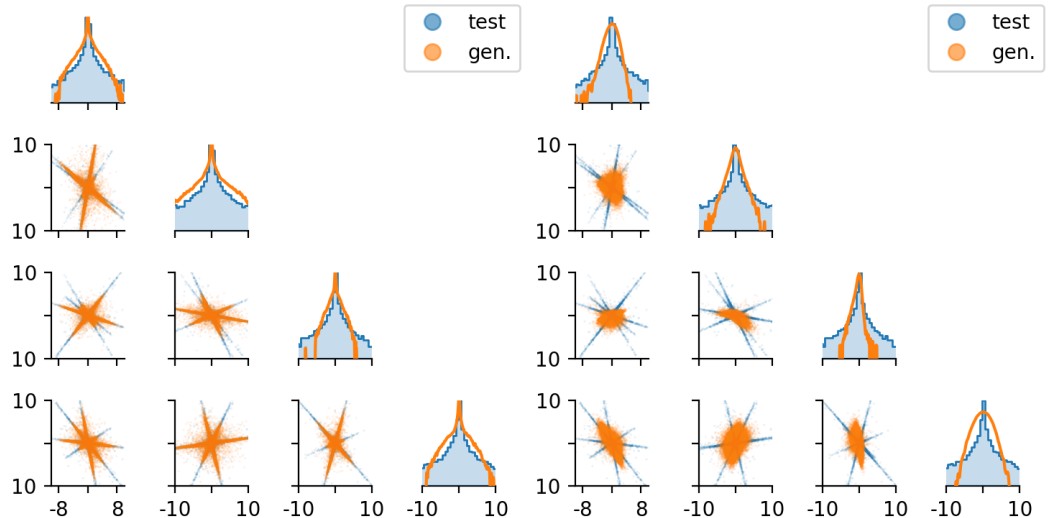

Figure 30: Generated data (orange lines and dots) compared to ground truth data (blue lines and dots) with the SGM for a vector of $4$ correlated Cauchy variables with a fast scheduling: $\beta_m = 0.01$, $\beta_M = 1$. The left plot corresponds to our neural network architecture based on spherical decomposition equation L.33 whereas the right plot corresponds to default neural network architecture. On the diagonal, log-histogram of ground truth data (continuous blue line), and logarithm of the pdf KDE estimation of generated data (orange line) are superimposed.

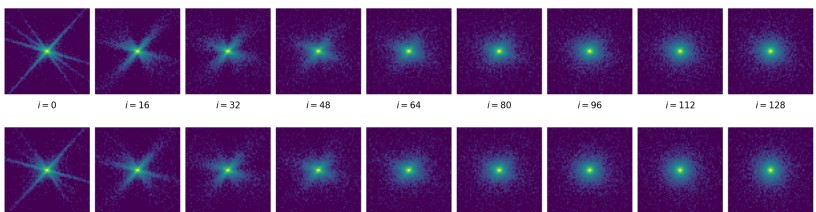

Figure 31: Evolution of the solution log-pdf $\log(p_s(x_1, x_3))$ of MSGM forward SDE (top) and backward SDE (bottom) for a vector of $4$ correlated Cauchy variables, with fast scheduling: $\beta_m = 0.01$, $\beta_M = 1$ and our neural network architecture based on spherical decomposition equation L.33.

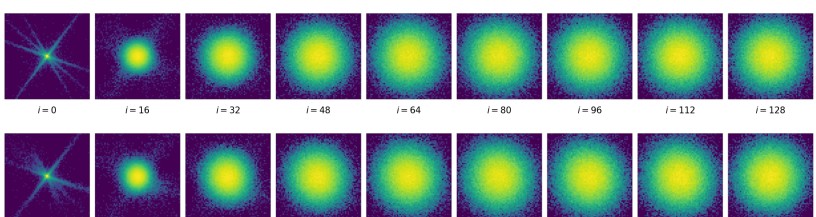

Figure 32: Evolution of the solution log-pdf $\log(p_s(x_1, x_3))$ of SGM forward SDE (top) and backward SDE (bottom) for a vector of $4$ correlated Cauchy variables, with default scheduling: $\beta_m = 0.1$, $\beta_M = 20$ and default neural network architecture.

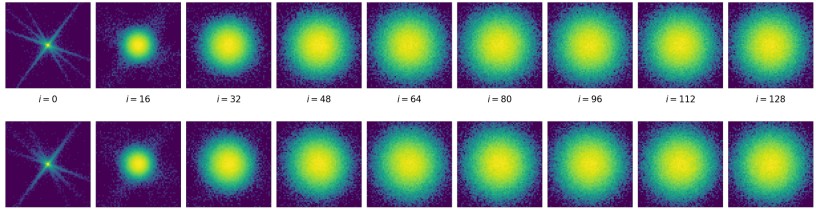

Figure 33: Evolution of the solution log-pdf $\log(p_s(x_1, x_3))$ of SGM forward SDE (top) and backward SDE (bottom) for a vector of $4$ correlated Cauchy variables, with default scheduling: $\beta_m = 0.1$, $\beta_M = 20$ and our neural network architecture based on spherical decomposition equation L.33.

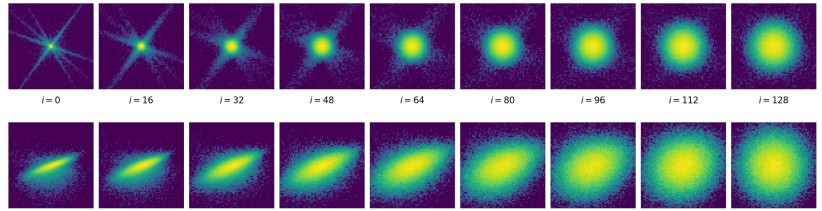

Figure 34: Evolution of the solution log-pdf $\log(p_s(x_1, x_3))$ of SGM forward SDE (top) and backward SDE (bottom) for a vector of $4$ correlated Cauchy variables, with fast scheduling: $\beta_m = 0.01$, $\beta_M = 1$ and default neural network architecture.

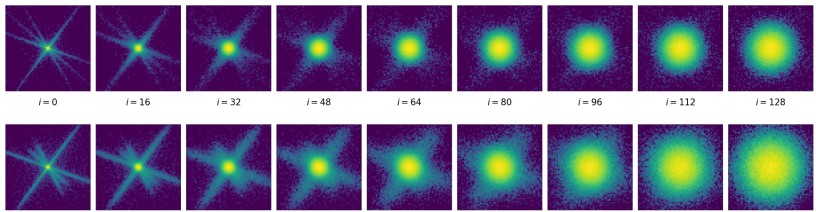

Figure 35: Evolution of the solution log-pdf $\log(p_s(x_1, x_3))$ of SGM forward SDE (top) and backward SDE (bottom) for a vector of $4$ correlated Cauchy variables, with fast scheduling: $\beta_m = 0.01$, $\beta_M = 1$ and our neural network architecture based on spherical decomposition equation L.33.

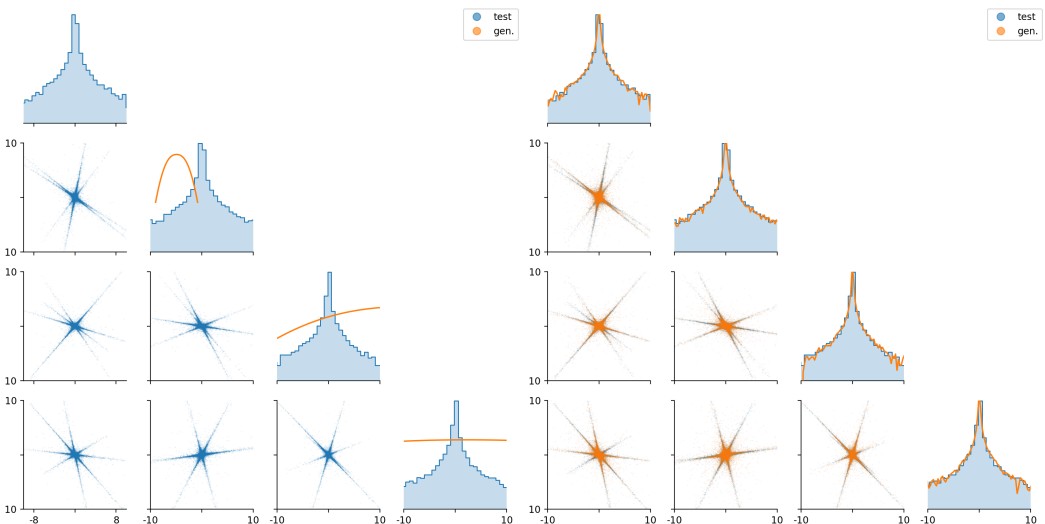

Figure 36: Pair plots of generated data (orange dots) compared to ground truth data (blue dots) with the SGM (left) and MSGM (right) with $2^{24} = 16777216$ ADAMS iterations for a vector of $4$ correlated Cauchy variables. On the diagonal, log-histogram of ground truth data (continuous blue line), and logarithm of the pdf KDE estimation of generated data (orange line) are superimposed.

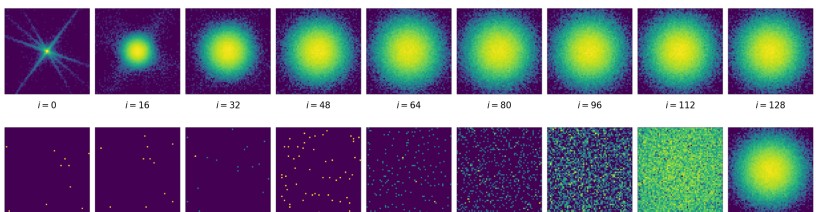

Figure 37: Evolution of the solution log-pdf $\log(p_s(x_1, x_3))$ of SGM forward SDE (top) and backward SDE (bottom, $2^{24} = 16777216$ ADAMS iterations and $2^9 = 512$ time steps) for a vector of $4$ correlated Cauchy variables, with default scheduling: $\beta_m = 0.1$, $\beta_M = 20$ and default neural network architecture.

the number of ADAMS iterations for MSGM in order to make the CPU training time of SGM and MSGM similar (see Section M.3 for details). For SGM with very large number of iterations ($2^{24} = 16777216$), we use a larger number of time steps ($2^9 = 512$) for the backward SDE to prevent all samples generated by SGM to diverge.

## M.7    VORTICITY FIELD FROM PARTICLE IMAGE VELOCIMETRY MEASUREMENTS

Particle Image Velocimetry (PIV) is an experimental technique to measure velocity fields in fluids by tracking the displacement of tracer particles between consecutive images illuminated with lasers (Adrian & Westerweel, 2011). We used two-dimensional, two-component (2D2C) PIV data of Figure 38, which provide both in-plane velocity components. Here PIV is not time-resolved, i.e. each velocity image is uncorrelated to the next. The flow observed is a benchmark configuration : a wake flow at Reynolds number $Re = 3900$ created by a circular cylinder embedded in a mean stream (Parnaudeau et al., 2008). We compute the two-dimensional curl of the velocity. Named vorticity, it is presented in Figure 39.

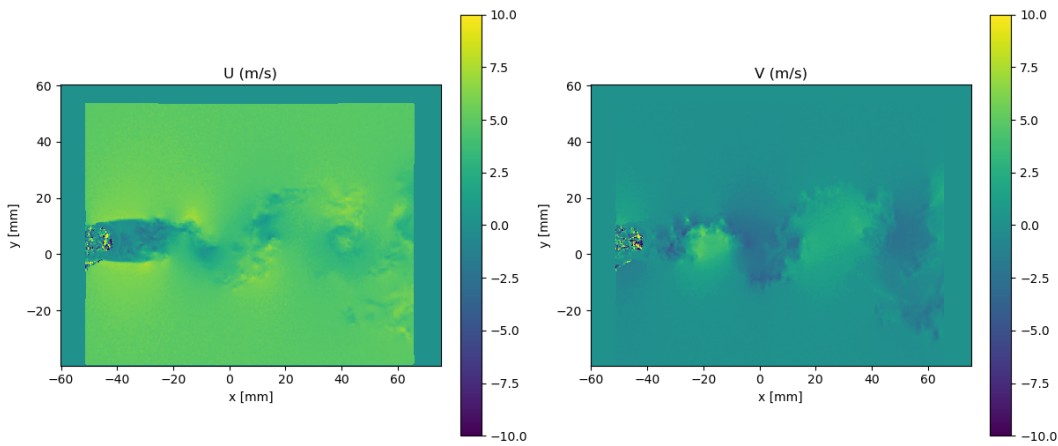

Figure 38: 2D2C PIV velocity field: velocity component along $x$ (left) and velocity component along $y$ (right).

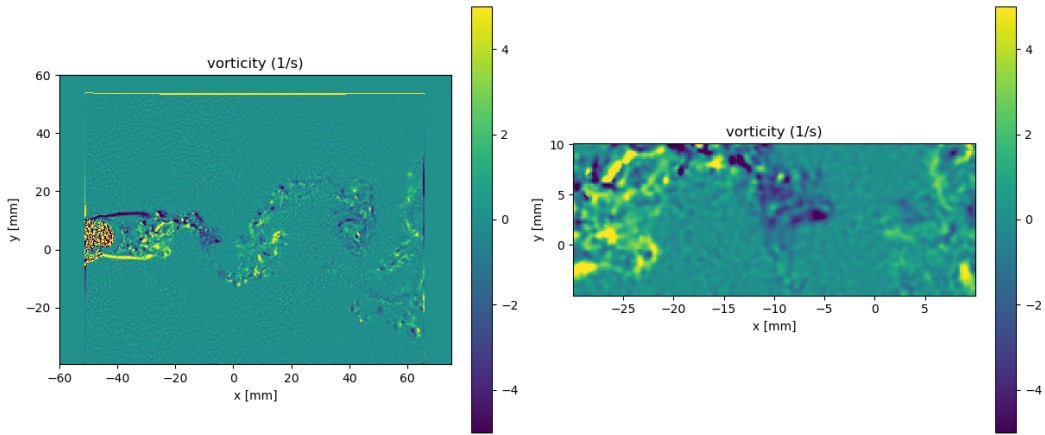

Figure 39: The full two-dimensional vorticity (left) and a zoom (right) of a PIV field

Table 7: Low-dimensional vorticity test case: parameters of the nominal numerical experiments.

| Parameter | SGM | MSGM |
|---|---|---|
| Dimension $d$ | 16 | 16 |
| Number of used training data points ($M$) | $2^{10} = 1024$ | 1024 |
| Number of test data points | 6476 | 6476 |
| Reference number of ADAMS steps | $2^{20}$ | $2^{20}$ |
| Number of ADAMS steps ($N_{\text{iter}}$) | $2^{20} = 1048576$ | 262144 |
| CPU time / ADAMS steps (in ms) | 4 | 32 |
| Batch size | 256 | 256 |
| Number of time steps (forward) $N_T^f$ | 1 | 16 |
| Number of time steps (backward) $N_T^b$ | 8 | 8 |
| $\beta_{\min}$ | 0.025 | 0.025 |
| $\beta_{\max}$ | 5 | 5 |
| $t_\varepsilon$ | $10^{-4}$ | $2.5 \times 10^{-5}$ |
| Learning rate | $10^{-3}$ | $10^{-3}$ |
| Neural network architecture | default | spherical (equation L.33) |
| MMD (MMD(train)=$0.9 \times 10^{-2}$) | $1.5 \times 10^{-2}$ | $1.3 \times 10^{-2}$ |

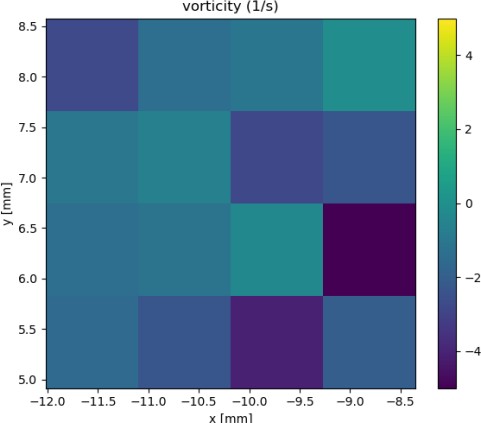

Figure 40: Spatial cropping and spatial subsampling of a vorticity field to obtain a data sample at low dimension $d = 16$.

### M.7.1 Low-dimensional test case: vorticity evaluated on several spatial points

To reduce the dimension $d$ of the data, we severely crop the vorticity images and subsample them spatially, keeping only $4 \times 4$ pixels by images as illustrated by Figure 40. Once reshaped as a vector, each small image represents a data point of dimension 16. If we choose a dimension $d \leqslant 16$, we just keep the first $d$ coefficients of the vector. For this experimental dataset, we investigate the influence of the amount of data available for learning. Our default experiments will train the models with $2^{10} = 1024$ data points only.

As seen previously in Section 6.2, MSGM is more robust in low-data mode and better represents rare events, as also confirmed by the survival function Figure 41. We explain it by a latent distribution close to the data distribution as illustrated in Figure 42.

For a fair numerical comparison, we also test SGM with and without our neural network architecture based on spherical decomposition equation L.33 in Figures 43, 44, and 45. This architecture improves the quality of the generated samples. However, tails are still underestimated and some regions of the space remain clearly badly sampled. In contrast, MSGM samples fit well the data distribution both with dense and with sparse tensor, **G**.

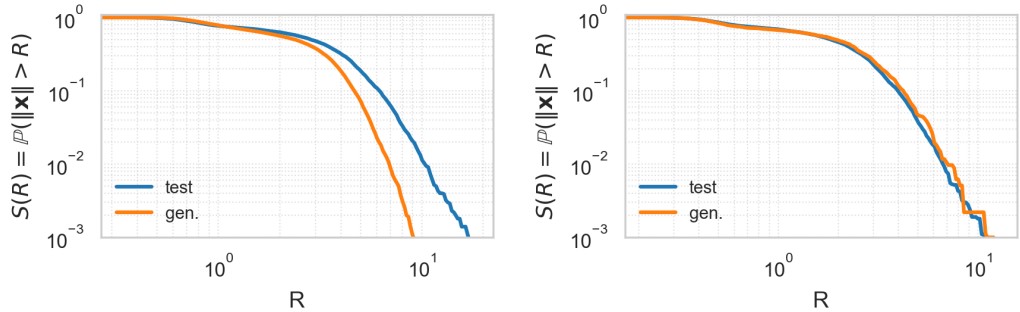

Figure 41: Survival function of generated data (orange line) compared to ground truth data (blue line) with the SGM (left) and MSGM (right) trained on $1024$ 16-dimensional data points representing PIV-based vorticity fields.

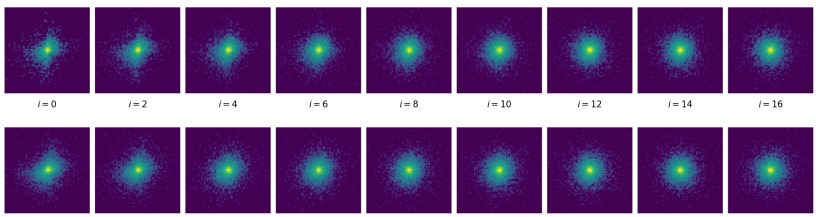

Figure 42: Evolution of the solution log-pdf $\log(p_s(x_1, x_3))$ of MSGM forward SDE (top) and backward SDE (bottom) for the vorticity images distribution, with nominal scheduling ($\beta_m = 0.025$, $\beta_M = 5.0$) and our neural network architecture based on spherical decomposition equation L.33.

To complete the numerical study, we evaluated the MMD between generated samples and test samples for different values of the reference number of ADAMS iterations, different number of time steps to integrate the backward SDE, different dimension $d$, and different numbers of training data. The convergence plots are visible in Figures 46 and 47. Again, MMD may not be the best tool for studying rare events. We can observe some tendencies, but definite conclusions may not be obtained from those convergence plots. For a very small training set ($2^6 = 64$ data points), both SGM and MSGM fail and MMDs are similarly large. The biggest MMD gap between SGM and MSGM appears to be in the intermediate region: $2^{10} = 1024$ data points. As expected, this gap seems to increase with dimension, even though this tendency is not fully clear for the plot. For small numbers of ADAMS iterations or small numbers of time steps, MSGM seems much better than SGM. This is expected since the MSGM latent space is already close to the data distribution. Without enough ADAMS iterations, neither the MSGM nor the SGM samples accurately mimic the data distribution, and in any case, it is better to let the optimization procedure run for a long enough time.

### M.7.2  HIGH-DIMENSIONAL TEST CASE : VORTICITY IMAGE PROCESSING

To demonstrate that MSGM can address high-dimensional problems, we propose here an image generator based on the sparse local tensor of Equation (K.8) and the Unet detailed in Section L.4.3. From the original high-resolution PIV-based vorticity images of Figure 39, we crop, subsample at resolution $64 \times 64$, smooth and subsample again images them spatially, keeping $32 \times 32$ pixels by images as illustrated by Figure 48. Once reshaped as a vector, each small image represents a data point of dimension $1024$.

Figures 49 and 50 present generated images with MSGM and SGM respectively. Table 8 summarizes the parameters of our numerical experiment. The numerical evaluation of image generation skills of MSGM is beyond the scope of this paper and we postpone this study to future work.

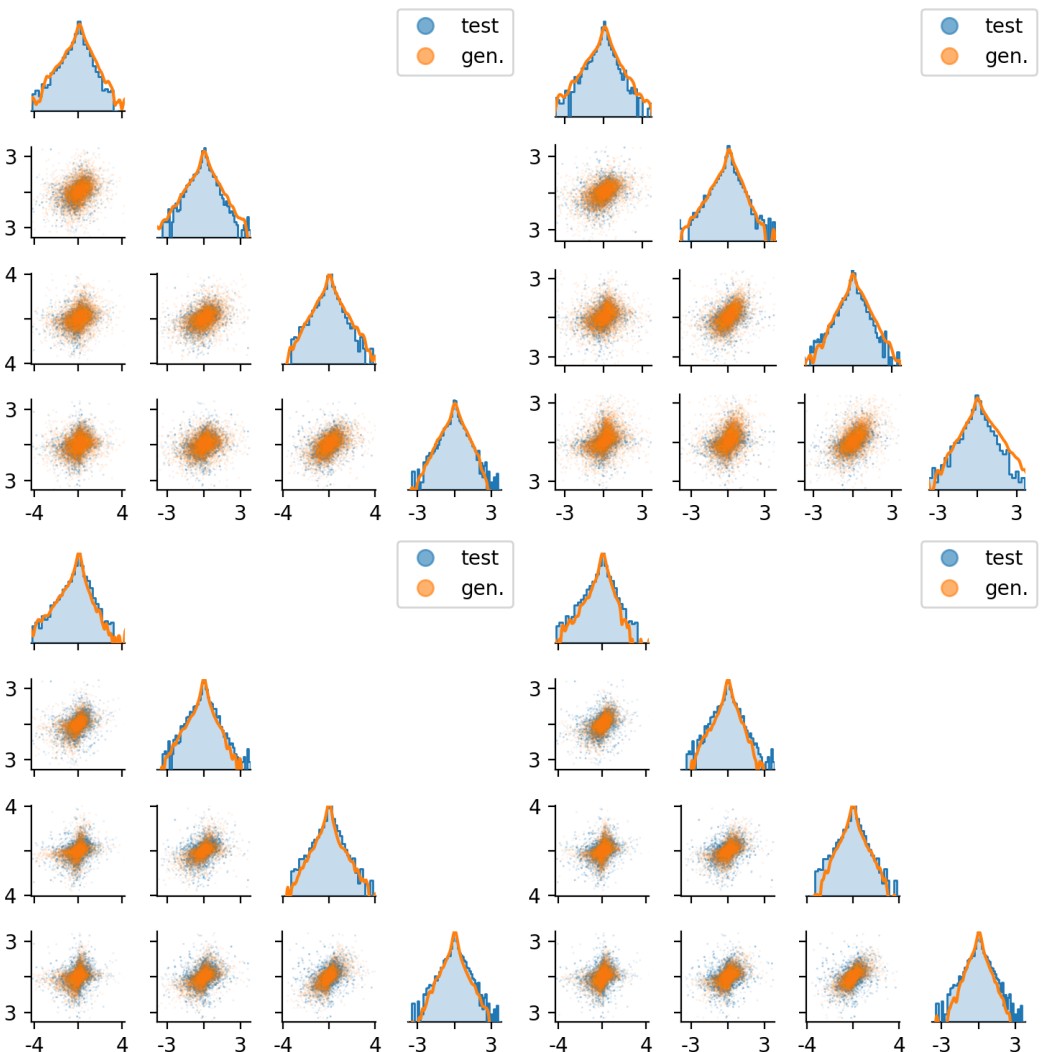

Figure 43: Pair plots of generated data (orange dots) compared to ground truth data (blue dots) with the MSGM with dense tensor **G** (top left), sparse local tensor **G** (top right), and SGM (bottom) trained on $1024$ 16-dimensional data points representing PIV-based vorticity fields. Left and top plots correspond to our neural network architecture based on spherical decomposition equation L.33 whereas the right bottom plot correspond to default neural network architecture. On the diagonal log-histogram of ground truth data (blue line) and logarithm of the pdf KDE estimation of generated data (orange line) are superimposed.

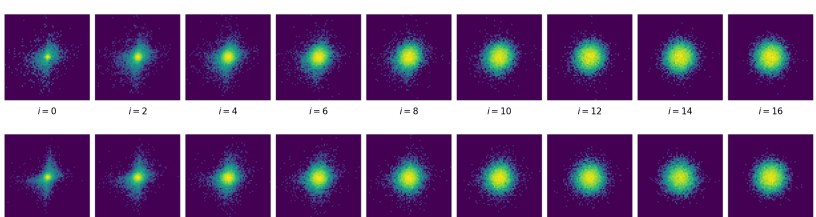

Figure 44: Evolution of the solution log-pdf $\log(p_s(x_1, x_3))$ of SGM forward SDE (top) and backward SDE (bottom) for the vorticity images distribution, with nominal scheduling ($\beta_m = 0.025$, $\beta_M = 5.0$) and default neural network architecture.

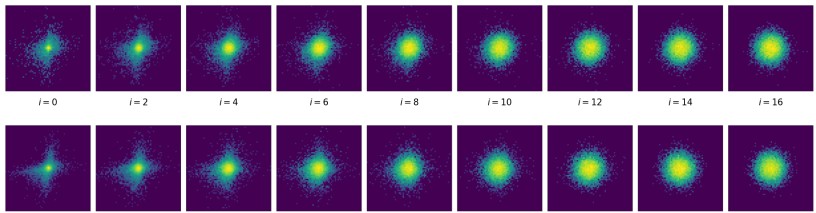

Figure 45: Evolution of the solution log-pdf $\log(p_s(x_1, x_3))$ of SGM forward SDE (top) and backward SDE (bottom) for the vorticity images distribution, with nominal scheduling ($\beta_m = 0.025$, $\beta_M = 5.0$) and our neural network architecture based on spherical decomposition equation L.33.

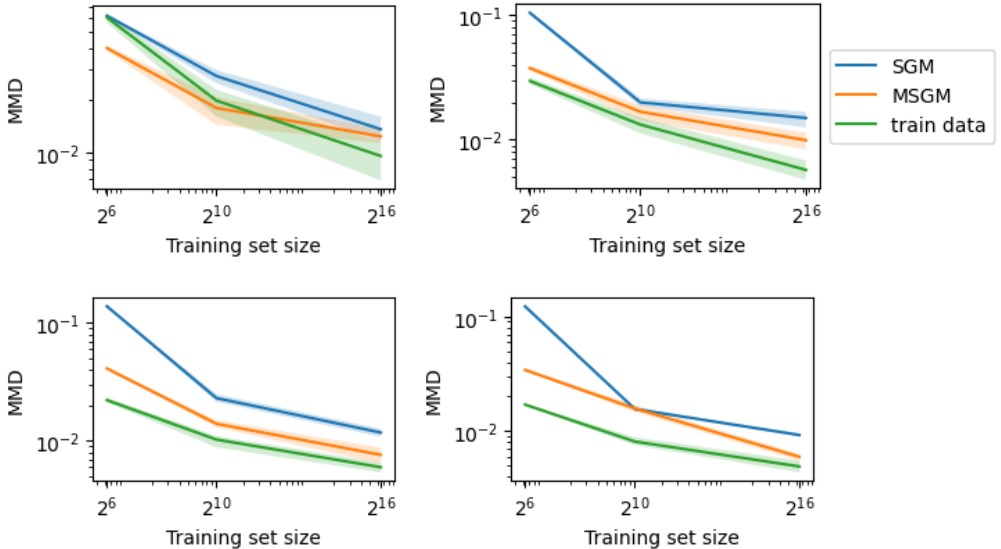

Figure 46: Convergence of MMD (mean and 80% confidence interval) for the vorticity images distribution as a function of number of training data for (from left to right and from top to bottom) dimension $d = 2, 4, 8$, and 16.

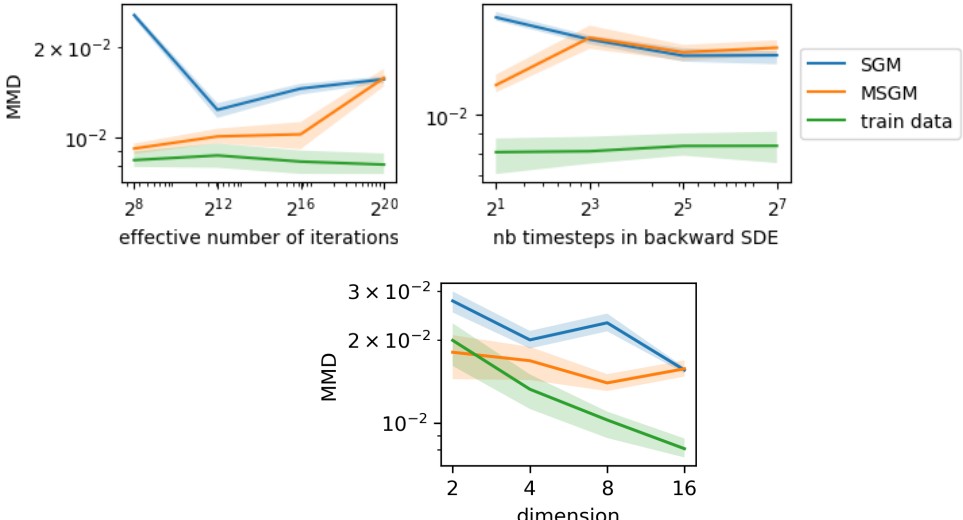

Figure 47: Convergence of MMD (mean and 80% confidence interval) for the vorticity images distribution as a function of reference number of ADAMS iterations (top left), as a function of number of time steps for integrating the backward SDE (top right), and as a function of dimension (bottom).

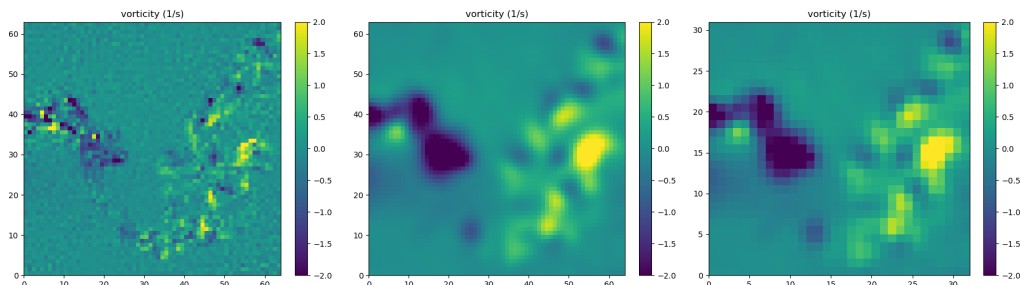

Figure 48: Spatial cropping and subsampling (left), spatial smoothing (middle), and spatial subsampling again (right) of a vorticity field to obtain a data sample at lower but still high dimension $d = 1024$.

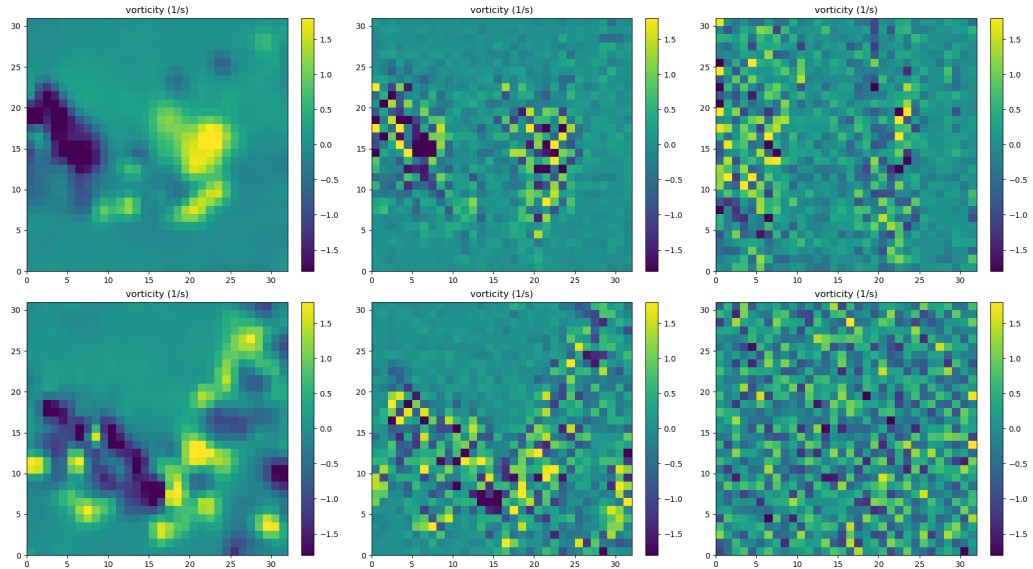

Figure 49: $32 \times 32$ image generation from MSGM with forward (top) and backward diffusion (bottom) at time (from left to right) $s = T - t = 0, 0.25$, and $T = 1$.

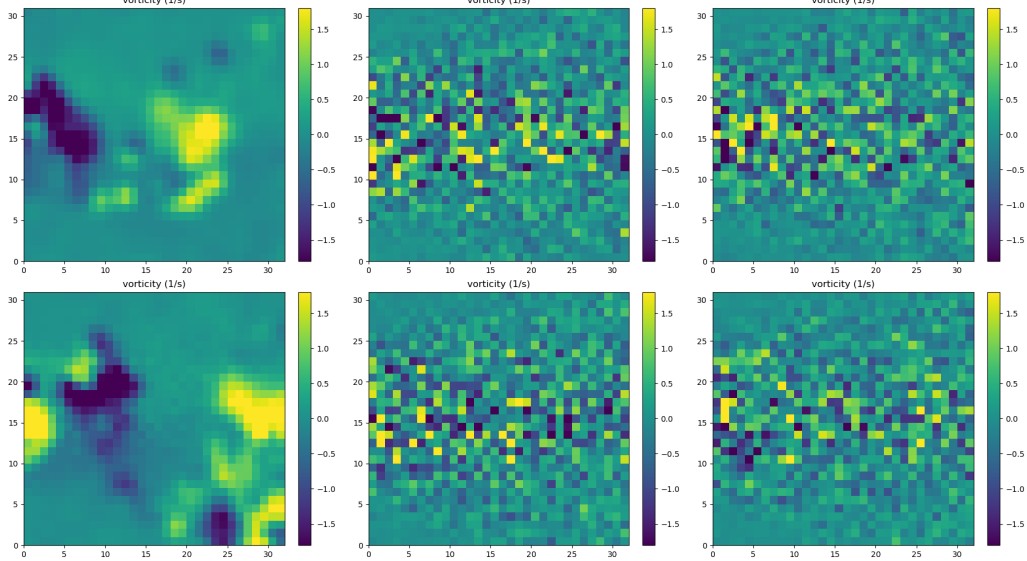

Figure 50: $32 \times 32$ image generation from SGM with forward (top) and backward diffusion (bottom) at time (from left to right) $s = T - t = 0, 0.25$, and $T = 1$. The apparent heteroskedasticity in the diffusion is due to the data normalization (pixel-wise variance is larger on top and bottom boundaries).

Table 8: High-dimensional vorticity test case: parameters of the nominal numerical experiments.

| Parameter | SGM | MSGM |
|---|---|---|
| Dimension $d$ | 1024 | 1024 |
| Number of used training data points ($M$) | 5000 | 5000 |
| Number of test data points | 2500 | 2500 |
| Reference number of ADAMS steps | $10^5$ | $10^5$ |
| Number of ADAMS steps ($N_{\text{iter}}$) | $10^5$ | $10^5$ |
| GPU time / ADAMS steps (in ms) | 410 | 590 |
| Batch size | 128 | 128 |
| Number of time steps (forward) $N_T^f$ | 1 | 128 |
| Number of time steps (backward) $N_T^b$ | 2048 | 2048 |
| $\beta_{\min}$ | 0.8 | 0.8 |
| $\beta_{\max}$ | 160 | 160 |
| $t_\varepsilon$ | $8 \times 10^{-3}$ | $8 \times 10^{-3}$ |
| Learning rate | $10^{-4}$ | $10^{-4}$ |
| Neural network architecture | default | spherical (equation L.33) |
| MMD (MMD(train)=$1.4 \times 10^{-3}$) | $2.4 \times 10^{-3}$ | $3.2 \times 10^{-3}$ |

## N   SUMMARIZED COMPARISON OF MSGM AND SGM

This section is devoted to a brief comparison of these two concepts of generative modeling both from theoretical and empirical point of views.

Each strategy follows its own noising process, leading to different invariant distributions, i.e. Gaussian for SGM and rotational invariant for MSGM. Both latent spaces are tractable, allowing for fast initial sample generation for the reverse process. As a particular added on, the latent distribution of MSGM allows for finite KL divergence when compared to heavy-tail distribution, e.g., as discussed and motivated by Section E.6. From the convergence speed, both dynamics allow for exponential convergence to the invariant distribution, assuming the rank condition A2 is satisfied for $G$. We will conclude this section with a comparison discussion beyond the heavy tail case.

### N.1   THEORETICAL ASPECTS

The latent space of MSGM is data aware, which ensures smaller KL-divergence of target distribution and latent distribution compared to classical SGM, see Section E and Proposition E.5.1. The method allows for inductive bias based on physics in the design of $\mathbf{G}$. For example, in the context of transport noise, making the noising/denoising process more physically relevant. This topic is part of future work by the authors and is briefly discussed in Section 7. Moreover, the conservation of norm in the denoising/backward process of MSGM serves as a stabilization tool, both for training and for sampling stage. In particular, samples cannot diverge.

At first glance, MSGM offers drawbacks compared to SGM. First, we have to rely on SSM and cannot apply DSM since we do not have access to an analytic score solution of the noising process. Second, we have to rely on numerical integration in the training because of no available analytic solution; see also the empirical discussion N.2 below.

When it comes to scalability, as $d \to \infty$, the current theoretical analysis is not yet complete. The current analysis is built on the (strong) rank-condition which can be verified in the case of dense tensors; see Section J. This is a limit in terms of scalability due to the $d^3$ scaling of $\mathbf{G}$. Here, the sparse tensors discussed in Section K will serve as a solution when it comes to scalability. However, in this context the rank condition has to be relaxed and new analysis is required as outlined in Section K.1.1.

### N.2   EMPIRICAL ASPECTS

SGM offers exact integration of the noising process, while MSGM relies on numerical integration. Although this at first glance looks like a drawback in praxis, for most of our test cases, only a few

forward steps were needed in the training process, making the training traceable and comparable to SGM training based on exact integration, while offering the same quality. For a more detailed discussion, we refer to the *fair comparison* discussion in Section M.3. As our current experiments suggest, MSGM requires less data in training. From approximation theory, learning the score reduces to training on a support that is the hyper-sphere in $\mathbb{R}^d$, with a conditioning variable $\log \|\boldsymbol{x}\| \in \mathbb{R}$. In particular, the effective domain for learning a neuronal remains bounded in $d$. It may affect the stability of the approximation using such an approximation class. Finally, the stabilization due to the conservation of norm avoids divergence instabilities of SSM solvers for MSGM, when compared to well known instabilities of SSM solvers for SGM.

