# OpenReview forum: "Multiplicative Diffusion Models: Beyond Gaussian Latents"
_ICLR.cc/2026/Conference — ICLR 2026 Poster_

### Official Review · Reviewer_Cw2o · 2025-10-19

**Soundness:** 2
**Presentation:** 3
**Contribution:** 2
**Rating:** 4
**Confidence:** 4

**Summary:**

This paper proposes multiplicative score-based generative models, replacing the usual additive Gaussian noise with skew-symmetric multiplicative noise that preserves data norms and yields conservative dynamics. The author prove properties of the forward process and show that norm is invariant, leading to a tractable, non-Gaussian latent that is data-aware. The proposed method is tested on heavy-tailed correlated Cauchy vectors and experimental fluid vorticity data.

**Strengths:**

- Using skew-symmetric multiplicative noise preserves data norms, yielding a principled generalization of Gaussian latents.
- Reverse SDE/ODE is derived and the equivalence to ELBO is proved for the proposed method.
- Experiments show meaningful improvements.

**Weaknesses:**

- The dense third-order tensor $G$ has $O(d^3)$ parameterse and should be prohibitive for high-dimensional applications.
- The reported CPU time is also an order of magnitude higher for MSGM than SGM, which may limit practicality.
- While the focus on heavy-tailed distributions is interesting, adding stronger baselines or more metrics showing the proposed method's improvement in this setting would better support the superiority claims.

**Questions:**

- How sensitive are results to the choice and structure of the tensor $G$? Could physics-motivated sparse or low-rank $G$, as discussed, preserve the theory while reducing cost?
- While the multiplicative noise idea is interesting, the derivation of the reverse process and training objective is solid, and the experiments are clear and well-motivated, I suggest the authors include missing related literature [1,2] that has already addressed similar generalizations of diffusion models or the so-called denoising Markov models from a higher level. Especially, Section 5.1 of [2] designs a denoising diffusion model with geometrical Brownian motion, which looks very much a prototype of the multiplicative noise studied in this manuscript.
- It seems that the authors claim that using multiplicative noise could lead to possible steady norm through the forward/backward process. I wonder if the same objective could be achieved by Riemannian manifold-related techniques, e.g., in [3].

[1] Benton, Joe, et al. "From denoising diffusions to denoising markov models." Journal of the Royal Statistical Society Series B: Statistical Methodology 86.2 (2024): 286-301.
[2] Ren, Yinuo, Grant M. Rotskoff, and Lexing Ying. "A unified approach to analysis and design of denoising markov models." arXiv preprint arXiv:2504.01938 (2025).
[3] De Bortoli, Valentin, et al. "Riemannian score-based generative modelling." Advances in neural information processing systems 35 (2022): 2406-2422.

---

> ### Author Response · Authors · 2025-11-25
>
> Dear Reviewer Cw2o, thank you very much for your detailed review. In the following, we address your comments on weaknesses and questions separately. We added several new contributions to the manuscript.
> Consequently, due to the length of the manuscript, we have now added a table of contents in the appendix for easier navigation and access to the related topics. Answers will include references to the appendix of the manuscript, which will be accessible with the table of content from page 16-18. Moreover, extended the analysis to cover scalability using sparse tensors.

---

> ### Author Response · Authors · 2025-11-25
> **Comments on Weaknesses**
>
> > The dense third-order tensor has parameters and should be prohibitive for high-dimensional applications.
>
> While the scalability was not the main focus of research in the first version of the manuscript, we have now added insight into the scalability of MSGM. For this, we introduced section K, where we discuss relaxation of the (strong) rank-condition in subsection K.1. Subsequently, in subsection K.2 we introduce sparse (local and non-local) tensors, which enable scalability, in particular in the simulation of the noising process, i.e., the forward SDE. In addition, we carried out a numerical experiment with application to image processing in section M.7, in particular in M.7.2 we investigate a $d=1024$ dimensional image generation task. The associated neural network structure for this experiment is described in L.4.3.
>
>
> We hope that both the new theoretical and empirical considerations serve as an illustration of the scalability of MSGM.
>
>
> > The reported CPU time is also an order of magnitude higher for MSGM than SGM, which may limit practicality.
>
> The CPU time by ADAMS step is indeed an order of magnitude higher for MSGM than for SGM. Therefore, in all our experiments except for the Cauchy distributions, we reduce the number of ADAMS steps accordingly for a fair comparison. We detailed this point in Appendix "M.3 Comparison Strategy". For Cauchy distribution, the SGM fails to reproduce the distribution and even diverges for a large number of ADAMS iterations, so we choose the same number of ADAMS iterations for MSGM and SGM.

---

> ### Author Response · Authors · 2025-11-25
> **Comments on Question 1**
>
> > How sensitive are results to the choice and structure of the tensor $G$? Could physics-motivated sparse or low-rank $G$, as discussed, preserve the theory while reducing cost?
>
> Regarding the sensitivity of the tensor $G$: If the assumptions on the rank condition and skew-symmetry are met then the invariant measure $p_{\infty}$ and convergence towards it is ensured independent of $\mathbf{G}$ as result of Theorem 3.3.1 (and in more detail in Theorem D.5.1). However, the choice of $\mathbf{G}$ determines the convergence speed of the continuous dynamics, in particular the exponential convergence rate $\alpha = \alpha(\mathbf{G},d)$ explicitly given in Theorem D.5.1.
>
> Regarding other tensors: This is a very interesting question that is also a matter of current active research of the authors for follow-up work. To still put some intuition to the case of more general classes of tensors, we introduce a new section *Appendix K Going beyond the rank condition for MSGM scalability*, where we discuss several options of multiplicative Tensors that are not dense. In particular, we discuss relaxation of the rank condition and its consequences on the latent distribution, and introduce local and non-local sparse tensors. While the latter allows for scalability, they introduce new questions that are slightly discussed in the associated subsection. Moreover, in I.2 we derive a counterexample to the strong rank-deficient case, i.e., the rank condition does not even hold almost surely. In this case, the invariant distribution is shown not to be rotational invariant. For an illustration, we also refer to Figures 5,6,7.
>
> We hope to give some additional insight into this topic. Please let us know if any further clarification would be helpful in this direction.
>
> From the empirical side, we added a new experiment in the high-dimensional case ($d=1024$) for an image generation task, which relies on a tensor $\mathbf{G}$ being of local sparse form as derived in subsection K.2. While such a tensor does not preserve the classic theory developed, the rank condition holds almost surely. The experimental results are promising, where the mechanics behind may be related to the possibility of the MGSM process not being trapped, see section K.1 for a related discussion.

---

> ### Author Response · Authors · 2025-11-25
> **Comments on Question 2**
>
> > While the multiplicative noise idea is interesting, the derivation of the reverse process and training objective is solid, and the experiments are clear and well-motivated, I suggest the authors include missing related literature [1,2] that has already addressed similar generalizations of diffusion models or the so-called denoising Markov models from a higher level. Especially, Section 5.1 of [2] designs a denoising diffusion model with geometrical Brownian motion, which looks very much a prototype of the multiplicative noise studied in this manuscript.
>
> Thanks to pointing out this interesting and related literature about unified and generalized frameworks in [1,2], which we added accordingly in sections 3 and 5 and in Appendix G and H. The ELBO and score matching losses derived in [1,2] could be consider as generalisation of our ELBO. Moreover, SGM based on heavy-tails additive noises, multiplicative Gaussian noises or along manifolds are particular cases treated in [1,2] and are thus relevent references for our work, even though convergence of forward diffusions is not the main focus of [1,2].
>
> While the experiment carried out in section 5.1 *Diffusion Model with Geometrical Brownian Motion* from [2] looks to share similarities, we stress the fact that our dynamics are not geometrical Brownian motion. In fact, our noising SDE is given in Stratonovich form, and we discuss the related Itô form in Appendix B.3.
> Our noise term $\mathbf{Z}_s$ is not diagonal. It includes zeros on the diagonal due to the skew-symmetry of $\mathbf{G}^k$. A geometric Brownian motion would necessitates $\mathbf{Z}_s$ to be diagonal with Brownian motion on this diagonal. From a dynamical system point of view, our multiplicative diffusion is probably closer to a Brownian motion on the $d$-sphere, a case also treated by [1,2].
>
> We hope that our investigation with all its details, including explicit convergence rates, sheds some additional light on the field of generative modeling with diffusion models, including the case of data distributions with heavy tails.

---

> ### Author Response · Authors · 2025-11-25
> **Comments on Question 3**
>
> > It seems that the authors claim that using multiplicative noise could lead to a possible steady norm through the forward/backward process. I wonder if the same objective could be achieved by Riemannian manifold-related techniques, e.g., in [3].
>
> Thank you for this curious question. There is indeed a deep connection to the Riemannian manifold setup and we investigated this in the Appendix *H Comparison with diffusions on Riemannian manifolds*.
> First, we explain that Brownian motions on the $d$-sphere as investigated in [3] are not solutions of our forward process using in case of using a linear tensor $\mathbf{G}$. However, when relaxing the linearity condition, i.e., choosing $\mathbf{G}(x) = \|x\|(I_d- x_n x_n^\intercal)$ (see equation $(\mathrm{H}.20)$) being non-linear, Lipschitz-continuous and sparse, we can match both diffusion processes. In particular, in this case, the diffusion will correspond to a Brownian motion on the scaled sphere $\|x_0\| \mathbb{S}^{d-1}$. Since still $\mathrm{Im}(\mathbf{G}(x)) = x^\bot$, we expect most of our theoretical results to remain valid. The authors already planned to explore this research direction in follow up work. In addition to Appendix H, this discussion is now part in the *conclusion and discussion* section.
>
>
>
>
>
> [1] Benton, Joe, et al. "From denoising diffusions to denoising markov models." Journal of the Royal Statistical Society Series B: Statistical Methodology 86.2 (2024): 286-301.
>
> [2] Ren, Yinuo, Grant M. Rotskoff, and Lexing Ying. "A unified approach to analysis and design of denoising markov models." arXiv preprint arXiv:2504.01938 (2025).
>
> [3] De Bortoli, Valentin, et al. "Riemannian score-based generative modelling." Advances in neural information processing systems 35 (2022): 2406-2422.

---

### Official Review · Reviewer_MUAR · 2025-10-30

**Soundness:** 3
**Presentation:** 3
**Contribution:** 3
**Rating:** 8
**Confidence:** 3

**Summary:**

The paper proposes to replace the additive Ornstein-Uhlenbeck (OU) diffusion process used in score-based generative models (SGM) with a *multiplicative* noising process $d \overrightarrow{\mathbf{x}}_s = \mathbf{G}(\overrightarrow{\mathbf{x}}_s) \circ d \overrightarrow{\mathbf{B}}_s$ where $\mathbf{G}: \mathbb{R} \to \mathbb{R}^{d \times d}$ is linear with $\operatorname{rank}(\mathbf{G}(\mathbf{x})) = d-1$ and each slice $\mathbf{G}^k$ is skew-symmetric, which is inspired by transport noises in fluid dynamics.
The authors provide the corresponding reverse SDE and probability-flow ODE and show that the forward SDE preserves the distribution of data norms, while the distribution of the directions converge exponentially fast to the uniform distribution on the sphere. Hence, the latent distribution is non-Gaussian, which may be more appropriate for heavy-tailed and anisotropic data, while still being easy to sample.
Since an analytic solution for the conditional score is unavailable, popular denoising score matching losses cannot be used, and the authors resort to sliced score matching to train multiplicative SGMs, which is shown to be equivalent to ELBO maximization.
Empirical results on low-dimensional data (correlated Cauchy, fluid dynamics data) suggest that multiplicative SGMs fit the data distribution's tail more accurately than vanilla SGMs.

**Strengths:**

+ **Novel diffusion process in generative modeling**
	+ Modifying the diffusion process which underlies modern SGMs is an important line of research, as it may lead to more meaningful latent spaces, and can incorporate prior knowledge into the corruption process (e.g. physical priors in $\mathbf{G}$). to yield data-dependent latent distributions adapt to the   and the standard forward process
+ **Data-dependent latent distribution**
	+ Typical OU diffusion processes yields a latent distribution independent of the data, while multiplicative SGMs (MSGMs) are equipped with a latent distribution which depends on the data distribution, which may be beneficial in low-data environments, or in distributions where rare events dominate.
	+ While the latent distribution is non-Gaussian in MSGMs, it is still easy to sample by e.g. using the 1D empirical CDF of data norms.
+ **Theoretical results**
	+ The authors provide an extensive theoretical analysis of the proposed technique, which is an important contribution. Future work on multiplicative diffusion processes may benefit from these results.

**Weaknesses:**

+ **Model training & Scalability**
	+ Since the conditional score does not admit a known analytic solution (in finite time), denoising score matching (DSM)---the de-facto standard approach to train modern diffusion models---cannot be used to train MSGM, and techniques like sliced score matching must be used, which are less stable in practice.
	+ The forward SDE must be numerically integrated since no analytical solution is known for $d > 2$. This is computationally expensive, which hinders scalability to larger models/problems.
	+ A dense tensor $\mathbf{G}$ requires $d^3$ coefficients, which prohibits the use of this general method in high-dimensional problems (e.g. images, video, physical problems).

+ **Empirical results**
	+ The empirical results in the main text are purely qualitative. While there are quantitative evaluations in the appendix, I suggest that important results are moved to the main text.
	+ Some of the experimental results are unconvincing: For example, I cannot see any improvement over SGM in Figure 4, and Figure 7 (right) seems to suggest that SGM converges faster to the data distribution (in terms of MMD).
	+ Due to the computational burden, only low-dimensional experiments are conducted. However, as SGMs thrive especially in complex, high-dimensional domains, providing experiments in these settings would be a valuable addition to this work.

+ **Experimental setup**
	+ The main text is missing important details regarding the experimental setup, especially w.r.t. the baseline SGM: Was it trained with DSM or also with SSM?

**Questions:**

+ It seems that in all experiments, $\mathbf{G}$ is randomly constructed (Eq. 6.1) and all $\mathbf{G}^k$ are independent. Would it make sense to "couple" them in a way that makes the forward process more benign?
+ Could $\mathbf{G}$ be learned from data?
+ Since the network learns $a_\theta(\mathbf{x},t)\approx \mathbf{G}(\mathbf{x})^\top\nabla\log p_{T-t}(\mathbf{x})$, it seems that the network is tightly coupled to the choice of $\mathbf{G}$ in the forward process (which is different in regular SGMs). Under what conditions can the score be recovered from $a_\theta$?

---

> ### Author Response · Authors · 2025-11-25
>
> We want to thank reviewer MUAR for pointing out the strengths but also the weaknesses of the manuscript, which help to improve the current state.
> Due to the length of the manuscript, we have now added a table of contents in the appendix for easier access to the related topics. Answers will include references to the appendix of the manuscript, which will be accessible with the table of content from page 16-18. Moreover, we extended the analysis to cover scalability using sparse tensors. First, we would like to discuss aspects regarding the weaknesses. Then respond to the given questions.

---

> ### Author Response · Authors · 2025-11-25
> **Comments on Weakness 1**
>
> **Model training & Scalability**
>
> > Since the conditional score does not admit a known analytic solution (in finite time), denoising score matching (DSM)---the de facto standard approach to train modern diffusion models---cannot be used to train MSGM, and techniques like sliced score matching must be used, which are less stable in practice.
>
> We agree that DSM cannot be applied in the current version of MSGM, and we have to rely on SSM.
>
> Here, we want to discuss aspects of MSGM that possibly explain stabilization effects for SSM and with that, make SSM an attractive approach in the MSGM context:
> - The latent distribution for MSGM is "closer" to the data distribution (see Proposition E.5.1.), possibly adding a stability aspect beyond the scope of normalization of data as in the SGM context.
> - Due to the conservation of norm in the MSGM (see equation (3.7) in section 3.2), there is an additional robustness of the reverse process compared to SGM, since particles cannot diverge (for given score approximation).
> -  The surrogate has to approximate a score that, by design (variation in direction only), requires support only on the unit hypersphere. With that, the domain of interest is bounded compared to the SGM case, where the effective domain size may scale unbounded with $d$. In particular, for OU processes, the dynamics have a norm concentration effect that scales $\sqrt{d}$. We added Section D.3.1 for a related discussion.
>
> With that, while SSM may cause instabilities for SGM, the same is not observed for MSGM, possibly offering enhanced training dynamics.
> We additionally provide an embedded overview of practical differences between MSGM and SGM in the new appendix section N.
>
>
> > The forward SDE must be numerically integrated since no analytical solution is known for. This is computationally expensive, which hinders scalability to larger models/problems.
>
> While this is a noticeable difference to SGM, in practice a limited number of integration steps can be used, leading to a similar workload as forward SDE in SGM while offering better accuracy (in the heavy tail case) or similar accuracy (in the considered non-heavy tail examples). We refer to the *fair comparison strategy* from section M.3 for a detailed discussion on this topic and the updated numerical results in sections M.4-M.7 and in the main text, in particular tables that discuss computation time (CPU in this work).
>
>  From our experience, the scaling $\mathcal{O}(d^3)$ may be the dominating restriction in the forward SDE.
> To address this scalability topic, the work was extended to discuss the case of sparse tensors, allowing the generation of images, a high-dimensional $(d=1024)$ experiment in M.7.2.
>
>
> > A dense tensor requires coefficients, which prohibits the use of this general method in high-dimensional problems (e.g. images, video, physical problems).
>
> This scalability issue is now addressed in the new section K in the appendix. Here we discuss relaxation of the assumption on (strong) rank condition of $\mathbf{G}(\mathbf{x})$ in section K.1, including a discussion on their theoretical consequences. Then, several tensor designs, involving the important cases of sparse tensors, are discussed in section K.2. Ultimately, in a new experiment in section M.7.2 on image processing, such a sparse tensor is used, enabling scalability of the proposed method in high dimensions, $d=1024$. We hope that this example motivates the possible application of MSGM beyond the low-dimensional case.

---

> ### Author Response · Authors · 2025-11-25
> **Comments on Weakness 2 + 3**
>
> **Empirical results**
>
> > The empirical results in the main text are purely qualitative. While there are quantitative evaluations in the appendix, I suggest that important results are moved to the main text.
>
> We updated the presentation of plots, in particular, related to Figure 4. Due to the limited space, a more detailed analysis for several experiments is carried out in the updated appendix section M. We updated the main text accordingly to point to relevant results in the appendix, involving figures that study accuracy in dependence on hyperparameters such as number of integration steps, dimension $d$, and learning steps.
>
> > Some of the experimental results are unconvincing: For example, I cannot see any improvement over SGM in Figure 4, and Figure 7 (right) seems to suggest that SGM converges faster to the data distribution (in terms of MMD).
>
> At this point, we politely want to mention that we do not compete with SGM outside heavy-tail regimes or physically-induced bias setups.
> In short, for many usual low-dimension light-tail test cases, SGM is sometimes slightly better and sometimes slightly worse than MSGM under the *fair comparison setup* defined in section M.3. In particular, no method is superior in terms of accuracy in the light-tail case. However, the picture is opposite in the heavy tail case as discussed in section M.6
>
> > Due to the computational burden, only low-dimensional experiments are conducted. However, as SGMs thrive especially in complex, high-dimensional domains, providing experiments in these settings would be a valuable addition to this work.
>
> We advertise the new section M.7.2, that related to image processing, a high-dimensional case ($d=1024$) based on sparse tensors developed in section K.2.
>
> **Experimental setup**
>
> > The main text is missing important details regarding the experimental setup, especially w.r.t. the baseline SGM: Was it trained with DSM or also with SSM?
>
> Thank you for pointing this out. Section L.3 described the loss, including the baseline SGM case, which is trained with SSM as well.
> While SSM is known to lead to possible unstable training dynamics in the SGM context, the opposite is observed for MSGM, in particular, SSM was stable in all numerical investigations considered. We added the baseline information to the main text.

---

> ### Author Response · Authors · 2025-11-25
> **Comments on Question 1 + 2**
>
> > It seems that in all experiments, is randomly constructed (Eq. 6.1) and all are independent. Would it make sense to "couple" them in a way that makes the forward process more benign?
> --physical?
>
> From the given analysis, the convergence speed encoded in the exponential rate $\alpha=\alpha(\mathbf{G})$ from Theorem 3.3.1 given as
> $$    \alpha(\mathbf{G},d) = (d-1) \min_{(x,y)\in \mathbb{S}^{d-1}} \| \mathbf{G}^{\intercal}(x)y\|^2, \quad S=\{ (x,y) \in \mathbb{S}^{d-1} \times \mathbb{S}^{d-1} | x \bot y \}.
> $$
> is defined up-on the choice of $\mathbf{G}$ and convergence to the latent space distribution is ensured if $\mathbf{G}$ satisfies the rank-condition (A2).
>
> We added a new section *K Going beyond the rank condition for MSGM scalability* which includes the discussion of other constructions of tensors $\mathbf{G}$. It turns out that for particular cases of "correlated/coupled" structures $\mathbf{G}^k$, all being the same or diagonalizable in the same basis, there is an analytic solution. This is particularly attractive as it would enable denoising score matching (instead of sliced score matching) and bypass the numerical integration of the forward diffusion. However, for these types of solution there is no convergence to a rotation invariant latent space. For this counter example, we refer to the newly added subsection I.2 and the new Figures 5,6,7 illustrating the lack of tractable latent distribution.
> Moreover, we derived a necessary (but not sufficient) condition for convergence in Proposition K.1.2, which requires full rank of the matrix
> $$\mathbf{S} = \frac 12 \sum\limits_{k=1}^d  (\mathbf{G}^k)(\mathbf{G}^k)^\intercal.$$
> We hope that this new analysis may shed some light on constraints when constructing more general $\mathbf{G}$.
>
>
> > Could $\mathbf{G}$ be learned from data?
>
> Thank you for this interesting question that we have not addressed yet.
>
> While the question turns out to be quite general, the desired objective may be relevant to answer in more detail.
> We stress the fact that the latent space is the same independently of $\mathbf{G}$ as long as Assumptions (1) and (2) are satisfied, but the dynamics change under different choices of $\mathbf{G}$.
>
> If one is only interested in the fast convergence speed of the continuous dynamics, one may wish to optimize $\alpha$ from Theorem 3.3.1 (Theorem D.5.1) from data or independent of the data.
>
> If the data to mimic are some time stamps of a dynamical system, we may want to adapt the diffusion to those dynamics.
> Then, the diffusion process could lie -- at least partially -- along the manifold generated from path solutions of the dynamical system.
> We expect that such a physical inductive bias could help both the training and the inference.
> For multiscale nonlinear dynamical systems, dynamic representations with multiplicative noise, as in MSGM forward SDE are highly relevant [1].
> Therefore, we may want to adapt the tensor $\mathbf{G}$ to those dynamics.
> If the dynamical system is not fully known or intractable, we may estimate the tensor $\mathbf{G}$ through data [2,3], typically through third-order moment estimations.
> One of the authors plans to elaborate on this idea in another project, but this is beyond the scope of this paper.
>
> [1] Majda, Andrew J., Ilya Timofeyev, and Eric Vanden Eijnden. "A mathematical framework for stochastic climate models." Communications on Pure and Applied Mathematics: A Journal Issued by the Courant Institute of Mathematical Sciences 54.8 (2001): 891-974.
>
> [2] Peavoy, Daniel, Christian LE Franzke, and Gareth O. Roberts. "Systematic physics constrained parameter estimation of stochastic differential equations." Computational Statistics & Data Analysis 83 (2015): 182-199.
>
> [3] Resseguier, Valentin, et al. "Quantifying truncation-related uncertainties in unsteady fluid dynamics reduced order models." SIAM/ASA Journal on Uncertainty Quantification 9.3 (2021): 1152-1183.

---

> ### Author Response · Authors · 2025-11-25
> **Comment on Question 3**
>
> > Since the network learns $a_{\theta}(\mathbf{x}, t) \approx G(\mathbf{x})^\top \nabla \log p_{T-t}(\mathbf{x})$, it seems that the network is tightly coupled to the choice of G in the forward process (which is different in regular SGMs). Under what conditions can the score be recovered from $a_{\theta}$?
>
> The. reviewer is right.
> The network learns how to denoise the forward diffusion, which itself depends on the tensor $\mathbf{G}$. So, the network is indeed tightly coupled to the choice of $\mathbf{G}$.
>
> The learning of $a_{\theta}(\mathbf{x}, t) \approx G(\mathbf{x})^\top \nabla \log p_{T-t}(\mathbf{x})$ is motivated by the simplicity of the derived loss $L_{SSM}^{SGM}(\theta)$ in equation (2.3).
> To partially answer the question, we added a new section in Appendix G, i.e. *G.8 Remark on the score parametrization*. There we show that under the rank condition on $\mathbf{G}$, we can recover parts of the score, in particular the projected score $\nabla_{\bot}\log p_s$. Up to our knowledge, access to the full score is not as straightforward.
>
> In Appendix K Going beyond the rank condition for MSGM scalability, we propose new sparse tensors to be used in MSGM context. Some have a local structure and other don't. It seems that this local/nonlocal structure should be inline with the local/nonlocal structure of the neural network used. The new Appendix "K.3 Discussion about local and non local structure" discuss this point.

---

### Official Review · Reviewer_QLpk · 2025-10-31

**Soundness:** 3
**Presentation:** 3
**Contribution:** 3
**Rating:** 6
**Confidence:** 4

**Summary:**

The authors propose a "Multiplicative Score-based Generative Model" (MSGM), a new diffusion framework using skew-symmetric multiplicative noise instead of additive Gaussian noise. This construction preserves the $L_2$-norm of the data throughout the forward process. The resulting latent distribution is non-Gaussian; its norm distribution is identical to the data's norm distribution, and its direction is uniform on the d-sphere. The authors derive the reverse dynamics and an ELBO-equivalent score-matching objective. They demonstrate superior performance on low-dimensional, heavy-tailed data (Cauchy and fluid vorticity) compared to standard SGMs.

**Strengths:**

* **Novel Process:** The paper introduces a genuinely new forward diffusion process based on conservative, multiplicative noise. This is a clear theoretical departure from the standard additive noise paradigm.
* **Theoretical Grounding:** The model is well-supported by theory. The authors provide the Fokker-Planck equationm prove exponential convergence to the latent distribution, and formally link the Sliced Score Matching loss to an ELBO , justifying the training procedure.
* **Tractable Non-Gaussian Latent:** The key result—that the data norm is preserved—leads to a tractable, "data-aware" latent space. The 1D norm distribution can be modeled simply (e.g., with eCDF), and the direction is uniform. This avoids the prior mismatch (e.g., infinite KL divergence) that plagues standard SGMs when applied to heavy-tailed data.
* **Empirical Niche:** The model shows strong performance in a specific, known-weakness area for SGMs: heavy-tailed distributions. The results on Cauchy data, while low-dimensional, are compelling and demonstrate a clear advantage in modeling both tail decay and directionality.

**Weaknesses:**

* **Prohibitive Scalability:** The model, as presented, is fundamentally unscalable. The multiplicative noise operator $G$ is a third-order tensor with $O(d^3)$ coefficients. The authors' experiments are restricted to trivial dimensions (e.g., $d=4, d=16$). The authors concede this $d^3$ cost is "prohibitive" for any high-dimensional application, such as image processing. This limitation confines the method to being a theoretical curiosity, not a practical tool.
* **Computational Bottleneck:** The forward process

$d\vec{x}_{s}=G(\vec{x}_{s})\circ d\vec{B}_{s} $

has no analytic solution for $d>2$. This forces the use of numerical SDE integration (SRK4) during *every single training step* to obtain $\vec{x}_s$. This adds a massive computational burden and introduces discretization error directly into the loss calculation, a problem SGMs with analytic forward processes do not have.
* **Training Objective:** The lack of an analytic forward score compels the use of Sliced Score Matching (SSM). The authors admit this is a "less stable approach" than Denoising Score Matching (DSM). This instability is visible in the experiments (Figure 32), where the standard SGM (also trained with SSM) diverges completely, suggesting the training dynamic is brittle. MSGM's success here may be less about the superiority of its process and more about its "closer" prior simply making the unstable SSM objective *manageable*.
* **Arbitrary Noise Operator:** The tensor $G$ is constructed by sampling random skew-symmetric matrices. This choice is arbitrary, lacks any physical or data-driven justification, and serves only to satisfy the theoretical assumptions (A1, A2) "almost surely". The authors' own "future work" suggestion to use sparse, physics-based tensors essentially acknowledges the non-viability of the operator used in the paper.
* **Limited Scope:** The experiments are hyper-focused on a niche (low-d, heavy-tails) where the model is *designed* to win. There is no evidence it offers any advantage—and significant disadvantages in cost and complexity—for standard, high-dimensional benchmarks where SGMs are state-of-the-art.

**Questions:**

* The $O(d^3)$ complexity of the dense tensor $G$ is the primary barrier to practical use. You mention sparse tensors as future work. Have you investigated what degree of sparsity is permissible while still satisfying the rank condition (A2) and ensuring the empirical convergence to a uniform spherical distribution?
*  The numerical integration of the forward SDE (Algorithm 1, line 5) is a major bottleneck. How sensitive is the final model quality to the number of steps ($N_T$) used in this integration? Using a small $N_T$ would be faster but would also increase the error in the $\vec{x}_s$ samples used to approximate the score matching loss.

*  In the reverse process, the score for the direction is conditioned on the norm $||x_0||$. This implies the direction and magnitude are re-coupled during generation. Can you provide more intuition on how the learned score  successfully models this complex conditional distribution to reverse the "whitening" of the forward process?

* The paper's premise is that preserving the data norm is a significant advantage over SGMs, whose latent norm converges to a $\chi_d$ distribution. However, in the high-dimensional regime ($d \to \infty$), the norm of a standard Gaussian vector $\vec{x} \sim \mathcal{N}(0, I_d)$ strongly concentrates around $\sqrt{d}$. If the input data is similarly normalized (a standard practice), the norm is effectively stable in the SGM forward process, just converging to a different constant. Does the exact preservation of the norm distribution offer a fundamental advantage over the concentration of the norm, especially given the $O(d^3)$ computational cost?

* Relatedly, your argument for MSGM's superiority rests on modeling heavy-tailed data. In high dimensions, heavy-tailed distributions are often characterized by anisotropic, sparse, or low-dimensional structures, not just by an extreme radial (norm) component. Are you sure that this 1D norm-preservation mechanism is the correct inductive bias for high-dimensional heavy tails, or is the advantage you observe in low-d experiments an artifact that will not scale?

---

> ### Author Response · Authors · 2025-11-24
>
> Dear Reviewer QLpk. Thank you very much for your detailed review. We greatly appreciate your careful reading and are glad to have received many concrete suggestions. We are happy that you value our problem as significant and appreciate that we introduce our method in detail and with mathematical rigor.
> Due to the length of the manuscript, we have now added a table of contents in the appendix for easier access to the related topics. Answers will include references to the appendix of the manuscript, which will be accessible with the table of content from page 16-18. Moreover, we extended the analysis to cover scalability using sparse tensors. We address your comments and questions in the following. We address your comments in the following.

---

> ### Author Response · Authors · 2025-11-25
> **Comments on Weaknesses 1 + 2 + 3**
>
> > *Prohibitive Scalability:* The model, as presented, is fundamentally unscalable. The multiplicative noise operator is a third-order tensor with coefficients. The authors' experiments are restricted to trivial dimensions (e.g., ). The authors concede this cost is "prohibitive" for any high-dimensional application, such as image processing. This limitation confines the method to being a theoretical curiosity, not a practical tool.
>
> In order to address the scalability of MSGM in terms of $d$, we added a completely new section K, in particular on sparse tensors. From the theoretical side, these sparse tensors may require new aspects of theoretical analysis when it comes to relaxation of the (strong) rank condition. We discuss aspects of weaker assumptions in section K.1. Moreover, we extended our experiment to an image processing application in $d=1014$ added in section M.7.2, and discussed in the Conclusion section.
> Here, empirically sparse tensors serve as a promising tool for scalability.
>
>
>
> > *Computational Bottleneck:* The forward process
> $dx_{s}=G(x_{s})\circ dB_{s}$
> has no analytic solution for. This forces the use of numerical SDE integration (SRK4) during every single training step to obtain. This adds a massive computational burden and introduces discretization error directly into the loss calculation, a problem SGMs with analytic forward processes do not have.
>
>
> We agree with the reviewer that, from a theoretical perspective, the absence of a closed-form solution for
> $$
> dx_{s} = G(x_{s}) \circ dB_{s}
> $$
> requires numerical SDE integration (SRK4) at each training step, which introduces additional computational cost and discretization error.
>
> However, in our empirical studies, this overhead remains modest: only a small number of discretization steps $N_t$ was required. We provide a detailed discussion of the runtime implications—including CPU time per Adam iteration, cost scaling, and our fair comparison strategy—in Appendix M. The notion of fair comparison (see M.3) refers to matching the computational workload between MSGM and SGM and then comparing their accuracies under equalized conditions. Under these matched settings, MSGM achieves superior accuracy in the heavy-tailed case, while in non-heavy-tailed settings the two methods attain comparable performance (see experimental sections M.4-M7 and the experiments in the main text).
>
> Regarding cost scaling under refinement of the time discretization, we observe an $\mathcal{O}(\sqrt{d})$ dependence for the dense-tensor scenario, as discussed at the end of Section M.3. When using sparse tensors, the scaling improves significantly, and runtime becomes may be dominated by the neural-network training rather than by the forward SDE integration. For qualitative timing comparisons, including CPU times, we refer the reviewer to the tables in the numerical section M.
>
>
>
>
> > *Training Objective*: The lack of an analytic forward score compels the use of Sliced Score Matching (SSM). The authors admit this is a "less stable approach" than Denoising Score Matching (DSM). This instability is visible in the experiments (Figure 32), where the standard SGM (also trained with SSM) diverges completely, suggesting the training dynamic is brittle. MSGM's success here may be less about the superiority of its process and more about its "closer" prior, simply making the unstable SSM objective manageable.
>
> We partially agree here with that weakness from a theoretical point of view. Here, we want to discuss aspects of MSGM that possibly explain stabilization effects for SSM. First, the latent distribution for MSGM is "closer" to the data distribution (see Proposition E.5.1.), possibly adding a stability aspect beyond the scope of normalization of data as in the SGM context. Secondly, due to the conservation of norm in the MSGM (see equation (3.7) in section 3.2), there is an additional robustness of the reverse process compared to SGM, since particles cannot diverge (for a given score approximation). Finally, the surrogate has to approximate a score that, by design (variation in direction only) requires support only on the unit hypersphere. With that, the domain of interest is bounded compared to the SGM case, where the effective domain size may scale unbounded with $d$, e.g. for a Gaussian distribution, there is a norm concentration effect around $\sqrt{d}$.
>
> With that, while SSM may cause instabilities for SGM, the same is not observed for MSGM, possible offering enhanced training dynamics.

---

> ### Author Response · Authors · 2025-11-25
> **Comments on Weaknesses 4 + 5**
>
> > *Arbitrary Noise Operator*: The tensor is constructed by sampling random skew-symmetric matrices. This choice is arbitrary, lacks any physical or data-driven justification, and serves only to satisfy the theoretical assumptions (A1, A2) "almost surely". The authors' own "future work" suggestion to use sparse, physics-based tensors essentially acknowledges the non-viability of the operator used in the paper.
>
> To start closing this gap, we added a completely new section K *Going beyond the rank condition for MSGM scalability*. While a complete picture is content of current research out of the scope of this manuscript, we added preliminary analysis and experiments in the context of sparse tensors, which, as empirically seen in M.7.2 allow for scalability with application to image processing.  A motivation of the mathematical justification in the current manuscript is to ensure justified control of a tractable latent space, similar to the non-physical or data-driven SGM. In a new section I.2.2 within the appendix I, we derive examples of non-dense tensors that lead to non-rotational invariant latent distributions. Hence, in general settings, the latent space may not be tractable.
>
> We expect that MSGM is indeed able to include physical justification. For instance, a connection to transport noise in fluid dynamics is now additionally discussed in the introduction, section 3, and the conclusion.
>
>
> > *Limited Scope*: The experiments are hyper-focused on a niche (low-d, heavy-tails) where the model is designed to win. There is no evidence it offers any advantage—and significant disadvantages in cost and complexity—for standard, high-dimensional benchmarks where SGMs are state-of-the-art.
>
> We have now updated and increased the number of experiments, including a new high-dimensional example with $d=1024$. We want to stress the fact that we do not want to compete with state-of-the-art SGMs in their well-established areas, but that MSGM offers similar results.

---

> ### Author Response · Authors · 2025-11-25
> **Comments on Questions 1 + 2 + 3**
>
> >The $\mathcal{O}(d^3)$ complexity of the dense tensor $G$ is the primary barrier to practical use. You mention sparse tensors as future work. Have you investigated what degree of sparsity is permissible while still satisfying the rank condition (A2) and ensuring the empirical convergence to a uniform spherical distribution?
>
> This is indeed a very exciting question and the content of the current research of the authors. For this, we added a preliminary discussion regarding sparse tensors as content of the new section K while providing counterexamples in I.2 and I.3, when such a desired invariant distribution is not met.
>
> In section K.1, we relaxed the strong rank condition to the almost-sure case and discussed consequences from the theoretical side. In particular, classical solution theory does not apply anymore and $\mathbf{G}$ has to be chosen such that MGSM dynamics are not trapped on submanifolds (on which the rank is < d-1).
> In line with this weaker framework, we propose a "local" sparse tensor in section K.2.2.
>
> Moreover, in section K.2, we discuss a generalization to a tensor $\mathbf{G}\in\mathbb{R}^{d,d,D}$ with $D= \tfrac{d(d-1)}{2}$, in which full sparsity ensures (strong) rank conditions. Such non-local, sparse tensors then require a larger number of Brownian motion and adapted surrogate dimensions.
>
> A necessary (but not sufficient) condition for such tensors to allow for rotation invariant distribution was derived in section K.1.2 *Itô term rank condition*, which requires the matrix
> $$
>    \mathbf{S} := \frac 12 \sum\limits_{k=1}^d  (\mathbf{G}^k)(\mathbf{G}^k)^\intercal
> $$
> to be full rank.
>
> We hope that with that preliminary analysis, we could shed some light and intuition on this matter. In our new experiments, we used the local sparse tensors from section K.2.2. with application to image processing.
>
>
>
> >The numerical integration of the forward SDE (Algorithm 1, line 5) is a major bottleneck. How sensitive is the final model quality to the number of steps ($N_T$) used in this integration? Using a small $N_T$  would be faster but would also increase the error in the $\overset{\scriptsize\rightarrow}{x}_s$
>  samples used to approximate the score matching loss.
>
>  For a detailed answer, we refer to the response regarding the reviewer’s comment on weaknesses *Computational Bottleneck: The forward process*.
>
> >In the reverse process, the score for the direction is conditioned on the norm $\|x_0\|$. This implies the direction and magnitude are re-coupled during generation. Can you provide more intuition on how the learned score successfully models this complex conditional distribution to reverse the "whitening" of the forward process?
>
> Thank you for this interesting remark. Indeed, the score for the direction is conditioned on the norm $\|x_0\|$ as explained in Appendix H.2.
> Along the reverse diffusion, the conditional score direction will focus along some orientations, counterbalancing the direction equiprobability of the latent space, i.e., reversing the "whitening" of the forward process.
> On a different scaled $d$-sphere $\|x_0\|\mathbb{S}^{d-1}$, the conditional score direction will be oriented differently, pushing along some orientations on some spheres and along other directions on spheres of larger radius. Accordingly, along the backward diffusion, the directions tend to align differently on different hyperspheres. The distribution of direction becomes more and more radius-dependent. We add this remark in Appendix H.2.

---

> ### Author Response · Authors · 2025-11-25
> **Comments on Question 4**
>
> >The paper's premise is that preserving the data norm is a significant advantage over SGMs, whose latent norm converges to a $\chi_d$ distribution. However, in the high-dimensional regime ($d\to\infty$), the norm of a standard Gaussian vector $\overset{\scriptsize\rightarrow}{x}\sim\mathcal{N}(0,I_d)$ strongly concentrates around $\sqrt{d}$. If the input data is similarly normalized (a standard practice), the norm is effectively stable in the SGM forward process, just converging to a different constant. Does the exact preservation of the norm distribution offer a fundamental advantage over the concentration of the norm, especially given the $\mathcal{O}(d^3)$  computational cost?
>
> We thank the reviewer for raising this point. To better analyze and compare the norm
> dynamics of SGMs and MSGMs, we added Section D.3.1, where the behavior of the SGM norm is discussed in detail.
>
> From a theoretical viewpoint, MSGM offers several advantages beyond mere norm
> concentration:
>
> *1. Latent-space mismatch and KL divergence.*
> In Section E.5 (Proposition E.5.1), we analyze the KL divergence between the latent
> distributions of SGM, MSGM, and the data distribution. We show that in any cases,
> even when the data is normalized so that $\|\vec{x}\|\approx \sqrt{d}$, the MSGM
> latent distribution remains strictly closer (in KL divergence) to the data distribution
> than the SGM latent distribution. This includes the high-dimensional regime where the
> SGM norm concentrates: although the SGM norm concentrates. The *distributional
> mismatch* persists and is quantitatively larger than for MSGM.
>
> *2. Heavy-tailed regime.*
> As discussed in Section E.6, in heavy-tailed settings, the SGM latent distribution does not
> admit a finite KL divergence to the data distribution, while the MSGM latent distribution
> *does*. This creates a fundamental representational gap independent of norm
> concentration effects. Numerically, this manifests in different convergence constants in
> the mean-field analysis, which remain relevant for training stability and resulting accuracy.
>
> *3. Geometric considerations and learning on the correct manifold.*
> MSGM evolves entirely on the unit hypersphere, where any two points have a geodesic
> distance at most $\pi$, independent of $d$. In contrast, SGM latent samples may lie as far
> as $2\sqrt{d}$ apart in Euclidean distance. This has direct implications for learning:
> - the effective domain for the MSGM score is the unit sphere, whose volume *shrinks*
> as $d\to\infty$, concentrating the learning problem on a compact manifold;
> - the SGM score, by contrast, must be learned on a domain whose effective support
> *grows* with $d$.
>
> As discussed in Appendix H.2 (see in particular Eqs.(H.9) and (H.14)), this leads to a fundamentally different approximation setting.
> From an approximation-theoretic perspective, learning on a bounded manifold is known to
> offer more favorable properties, such as better conditioning, improved stability, and reduced
> sample complexity when compared to learning on a domain expanding with dimension.
>
> *4. Computational considerations with sparse tensors.*
> Finally, we emphasize that the $\mathcal{O}(d^3)$ cost arises only in the dense-tensor
> formulation. As discussed in the new Section K, using sparse tensor representations can avoid this
> scaling while preserving the geometric and distributional advantages described above.
>
> In summary, while norm concentration does occur for SGMs in high dimensions, MSGM
> provides advantages that persist beyond this effect---namely, better latent-space
> alignment, finite KL divergence in heavy-tailed regimes, and more favorable geometric
> structure, and improved learning behavior on compact domains.
> We added a related discussion in a new section N *Summarized comparison of MSGM and SGM*, and hope to help the reader understand different aspects of the proposed method.

---

> ### Author Response · Authors · 2025-11-25
> **Comments on Question 5**
>
> > Relatedly, your argument for MSGM's superiority rests on modeling heavy-tailed data. In high dimensions, heavy-tailed distributions are often characterized by anisotropic, sparse, or low-dimensional structures, not just by an extreme radial (norm) component. Are you sure that this 1D norm-preservation mechanism is the correct inductive bias for high-dimensional heavy tails, or is the advantage you observe in low-d experiments an artifact that will not scale?
>
> Please note that in our numerical investigation, we considered a multi-dimensional Cauchy distribution that is highly anisotropic, which is not only heavy-tailed in its radial component. While we do not claim that our Noising/Denoising mechanism is the optimal dynamic for such distributions, it still offers three advantages:
> 1. A provable exponential convergence of the continuous dynamics,  see Theorem 3.3.1.
> 2. A finite KL-divergence of such heavy tail distributions and the accessible latent distribution of our denoising process, in contrast to Latent spaces arising in the context of SGM, see *Appendix E Latent distribution* for the analysis.
> 3. A in general smaller KL-divergence between our Latent distribution (that does not depend on $\mathbf{G}$) and the target distribution, compared to the case of Noising/Denoising based on OU-processes, see Appendix E.5.

---

### Official Review · Reviewer_Bt3f · 2025-11-10

**Soundness:** 3
**Presentation:** 3
**Contribution:** 3
**Rating:** 6
**Confidence:** 3

**Summary:**

This paper introduces a new framework for diffusion models based on the backward prcess corresponding to a forward process which incorporates a geometric brownian motion in its SDE. This geometric term is the result of multiplying a matrix to dB_t. They further assume this term is given by the product of a vector to a three dimensional tensor for which all the sliced matrices are skew symmetric. This skew symmetric property leads to a nice norm-invariant property for the forward and backward process. The authors claim that this norm invariance property enables the diffusion model to sample from rare events in the case where the target distribution is heavy tailed. In this case, the norm invariance implies the latent distributions are also heavy tailed, so one does not end of with a gaussian variable at the end. On the other hand, starting from a gaussian, one cannot transform a gaussian random variable back to the target using the backward process; e.g. the authors show the kl divergence of gaussian with a heavy tailed distribution such as Cauchy is infinite. Hence, this diffusion could cover cases in which the data distribution is heavy tailed.

**Strengths:**

The authors propose a new framework for diffusions which can be useful for heavy tailed targets, and cover more natural data distributions in some cases. The study the fokker plank equation for this process and drive the properties of the SDE such as norm-invariance.

The writing and proofs are written in a clear way and without typos, unfortunately I didn't have time to check the details of most of the proofs.

**Weaknesses:**

Beyond the heavy tail case, they don't really study the effectiveness of their setup compared to traditional diffusion and normalizing flows, hence when would be beneficial to use their framework. their experiments in Figure 3 is vague and I don't understand the catch from the plot on the effectivess of their method compared to gassian diffusion. the amount of experiments is also very limited, and they further don't study this usefulness version traditional non-multiplicative frameworks theoretically.

others:
line 206 possibility of rare events is not clear at this point of the paper. maybe explain more what you mean

**Questions:**

1) is there a benefit of in adding a bias term to the forward process (eg f(x) = -x similar to the OU process) in the multiplicative framework?

2) is there an argument one can make about the effectiveness of your method, either theoretically or empirically, compare to traditional diffusions beyond the heavy tail case?

3) it would be nice to show the derivation of the loss and the backward sde for the new multiplicative diffusion.

---

> ### Author Response · Authors · 2025-11-24
>
> We would like to thank Reviewer Bt3f for the feedback on our work. We made several changes in the manuscript. Due to the length of the manuscript, we have now added a table of contents in the appendix for easier access to the related topics. Answers will include references to the appendix of the manuscript, which will be accessible with the table of content from page 16-18. Moreover, we extended the analysis to cover scalability using sparse tensors. We address your comments and questions in the following.

---

> ### Author Response · Authors · 2025-11-24
> **Comment on Weaknesses**
>
> >  a forward process which incorporates a geometric Brownian motion in its SDE
>
> We want to point out that the underlying process is not a geometric Brownian motion. In fact, it is given in Stratonovich notation and by equation (3.3):
> $$
> dx_s = \circ d Z_s x_s
> $$
> with $\circ dZ_s$ not being diagonal and even containing zeros on the diagonal, because of the skew-symmetric design of the underlying $\mathbf{G}^k$ in the definition of $\mathbf{G}\in\mathbb{R}^{d,d,d}$.
> We added additional notes after the assumptions (A1) and (A2) to avoid confusion for the reader. Please let us know if this clarifies.
>
>
>
> >their experiments in Figure 3 is vague and I don't understand the catch from the plot on the effectiveness of their method compared to Gaussian diffusion.
>
> Thank you for pointing out flaws in the experimental setup. We updated the explanation to understand the test case and what is better.
> About effectiveness: The plot suggests the tail behaviors between (left) and (right) differs. We added a related discussion to guide the reader and understand the qualitative differences. Please let us know if this clarifies the situation better.
>
> >the amount of experiments is also very limited,
>
> We increased the number of experiments in addition to the preliminary experiments in the main text and in the appendix M *Details about our numerical experiments* (pages 65-86), i.e., Swiss roll, multidimensional Gaussian, (un)correlated Cauchy, with analysis in their training speed, number of iterations, and other important quantities. We extended the analysis to the case of sparse tensors. We updated the experiment section 6 in the main text, validated the experiments from section 6.1 and 6.2 also with a particular sparse choice of $\mathbf{G}$, and introduced a high-dimensional image processing experiment best on sparse tensors with $d=1024$ in appendix M.7.
>
> >and they further don't study this usefulness version traditional non-multiplicative frameworks theoretically.
>
> See answer to question (2) below.
>
> >others: line 206 possibility of rare events is not clear at this point in the paper. Maybe explain more what you mean
>
> Yes, we agree with this unclear setup at this point. We updated a short intuition that the property of norm conservation implies that likely large norm events will be conserved along the diffusion. In order to avoid too many technicalities, we choose at this point not to mention the consequences of the latent space (the invariant distribution of our diffusion process) and the data distribution that is discussed in Appendix E.

---

> ### Author Response · Authors · 2025-11-24
> **Comments to Questions 1 + 2**
>
> > (1) is there a benefit of in adding a bias term to the forward process (eg $f(x) = -x$ similar to the OU process) in the multiplicative framework?
>
> We first note that our investigated SDE is written in Stratonovich notation. It is given as
>
> $$
> d\mathbf{x}_s=\mathbf{G}(\mathbf{x}_s)\circ d\mathbf{B}_s.
> $$
>
> with a corresponding Itô form (given by Appendix B.3)
>
> $$
> d\mathbf{x}_s = f(\mathbf{x}_s) ds + \mathbf{G}(\mathbf{x}_s) d\mathbf{B}_s.
> $$
>
> with drift $f$ given as
>
> $$
>  f( \cdot ) = -\frac{1}{2} \sum_{k=1}^{d} \mathbf{G}^k (\mathbf{G}^k)^T\cdot
> $$
>
> With that being said, there is a presence of a drift term in the Itô form.  Similarly to OU processes, since
>
>  $$
> -\frac{1}{2} \sum_{k=1}^d\mathbf{G}^k(\mathbf{G}^k)^\intercal
> $$
>
>  is negative definite, the drift implies that $\mathbb{E}[X_t]\to \mathbf{0}$ as $t\to\infty$ (under suitable conditions).
>
> One straightforward extension of our multiplicative diffusion, in Stratonovich notation, would read as
> $$d\mathbf{x}_s = \mathbf{A}\mathbf{x}_s\mathrm{d}s + \mathbf{G}(\mathbf{x}_s) \circ d\mathbf{B}_s.$$
> with $\mathbf{A}$ being a skew-symmetric matrix.
>
>
> Then, the associated Fokker-Planck equation involves an additional advection term
> $$
> \nabla \cdot (Ax p_s) = (Ax) \cdot \nabla p_s = (Ax) \cdot \nabla_{\bot} p_s
> $$
> in the right hand side, since $\nabla \cdot (Ax)=tr(A)=0$.
>
> We added a new appendix *D.6 Beyond pure Stratonovich noise* and referred to its contained discussion after Theorem 3.1.1.
>
> While adding this drift term showed faster convergence in empirical studies by the authors, more analysis has to be carried out. As this is part of another work by the authors, we decided to exclude this case from the current manuscript.
>
>
>
> >(2) is there an argument one can make about the effectiveness of your method, either theoretically or empirically, compare to traditional diffusions beyond the heavy tail case?
>
> First, we note that we do not aim to compete with SGM outside heavy tail case, and outside physics-based inductive bias. In particular final experimental result shows similar quality.
>
> Regardless, we discuss additional theoretical and empirical aspects separately.
>
> *Theoretically:*
> Pro:
>     - The latent space is data aware, which ensures a smaller KL-divergence of target distribution and latent distribution compared to classical SGM, see Section E Latent Distribution, Proposition E.5.1.
>     - The method allows for physics-based inductive bias in the design of $\mathbf{G}$. e.g., in the context of transport noise, making the noising/denoising process more physically relevant. This topic is part of future work by the authors. We discuss this briefly in the introduction and conclusion in the manuscript.
>     - Stabilization due to the conservation of norm in the denoising/backward process in terms of no possible divergence.
>
>
> Cons:
> - We have to rely on SSM and cannot apply no DSM since we do not have access to the analytic solution of the noising process.
> - We have to rely on numerical integration in the training because of no available analytic solution, see also the empirical discussion.
>
> Regarding Scalability:
> Moreover, the completely understood analysis is built on the rank-condition, which can be verified in the case of dense tensors, see Appendix J. This is a limit in terms of scalability. For this, we now extended the analysis to the sparse tensor case and introduced a new discussion in Section K *Going beyond the rank condition for MSGM scalability*.
>
>
> *Empirically:*
> - SGM offers exact integration of the noising process, while MSGM relies on numerical integration. While this at first glance looks like a drawback in praxis, only few forward steps were needed in the training process, making the training tractable and comparable to SGM training based on exact integration, while offering the same quality. See the *fair comparison* discussion in M.3.
> - As the experiments suggest, MSGM requires less data in the training, see Figure 3, Figure 4(b) or Figure 44.
>  - Learning the score reduces to training on the hypersphere in $\mathbb{R}^d$ only. In particular, the effective domain to learn an ANN remains bounded in $d$, which may positively affect the approximation stability using such an approximation class.
>  - Again, the stabilization due to the conservation of norm avoids divergence instabilities of SSM solvers for MSGM.
>
>
>
> We added a related comparison discussion in the appendix section N *Comparing MSGM and SGM*.

---

> ### Author Response · Authors · 2025-11-24
> **Comment to Question 3**
>
> >(3) it would be nice to show the derivation of the loss and the backward sde for the new multiplicative diffusion.
>
> Thank you for this hint. We updated the main text and appendix accordingly. In order to navigate more easily in the appendix, we added a table of contents.
>
> - For the derivation of the new loss, we added a new subsection *From ELBO to our SSM loss* in Appendix G, to link the connection of Theorem 3.4.1. with the derivation of the loss $\mathcal{L}_{\text{SSM}}(\theta)$ based on Sliced Score Matching.
> - For the derivation of the backward SDE, we modified the main text and refer to the derivation of the backward SDE in Appendix F *Backward diffusion*.
>
> Please let us know if you think that explanations are helpful for the reader or if we can further improve the presentation.

---

### Author Response · Authors · 2025-12-03
**Message to the Area Chair  ( 1 / 2 )**

Dear Area Chair, thank you for taking the time to evaluate our paper. For your convenience, we provide a summary of the key contributions of our work, along with the discussion during the rebuttal phase and the resulting improvements. We appreciate the reviewers' insightful feedback, which allowed us to address important points and significantly strengthen our submission.

### Summary of our contributions


* **New generative model paradigm:**
We introduce a new type of diffusion model where the noising process is multiplicative. We call it a Multiplicative Score-based Generative Model (MSGM).
Involving random rotations around the origin, it greatly differs from previous diffusion models and opens a new research path.


* **Deep theoretical analysis of MSGM:**
Assuming a skew-symmetric structure and a rank condition for this noise, we proved several theoretical results, guiding the use of this new generative tool. These include analysis of the corrsponding Fokker-Planck equations, convergence rates to equilibrium, latent space analyis, data awareness and extensions to scalability to high-dimensionsal cases based on sparse tensor analysis.
In additon we derive analytic solutions, draw connections to Diffusions on Riemmanian manifolds and compare to the differences between MSGM and classical diffusion models based on additive noising processes.


* **General algorithm for MSGM:**
We propose to estimate the scaled diffusion score by a neural network using sliced score matching and derive its equivalence to maximizing the ELBO. Moreover, we adress the efficient sampling from the in general non-gaussian latent space. For the noising process, we propose discretization by high-order norm-preserving stochastic Runge-Kutta integration. For the denoising process, both SDE and ordinary differential equation (ODE) formulations are derived.


* **Application to extremes in moderate dimension:**
We propose a numerical procedure to mimic the heavy-tail distribution with MSGM. A key aspect of MSGM is the latent distribution, which in case of heavy-tail distribution, encompasses the correct data distribution tail decays automatically. Numerical results, show the efficiency with application to highly anistropic heavy tail data distributions.


* **Application in high dimension:**
As a first step, we focus on MSGM scalability and design of sparse underlying tensors in the diffusion component of the noising process. While the latter is not covered completely by the theoretical analysis, our numerical experiments show promising image generation results in dimensions up to $d=1024$.

---

> ### Author Response · Authors · 2025-12-03
> **Message to the Area Chair ( 2 / 2 )**
>
> ### Summary of revisions and main improvements
>
> We thank the reviewers for their constructive feedback and are encouraged that they recognized the mathematical rigor and presentation of our work. Based on the critical points raised, we have significantly updated the manuscript.
>
> In Summary across all reviewer comments, we substantially expanded and strengthened the manuscript both theoretically and empirically, with a particular emphasis on scalability, clarity of the forward/backward SDE, latent-space analysis, and experimental robustness. The most significant upgrade is the introduction of sparse tensor formulations, which meaningfully address scalability, enable a new $d=1024$ experiment, and extend the theoretical framework. We hope that these changes collectively strengthen the manuscript’s clarity, empirical validity, and practical relevance.
>
> The most significant updates are:
>
> **Major Structural Improvements**
>
> • *Added a full Appendix Table of Contents*: The appendix grew considerably (new derivations, new theory sections, new experimental material). To ensure navigability, we added a complete table of contents.
>
> • *Reorganized auxiliary derivations*: We clarified the derivations of the backward SDE (newly reorganized Appendix F), the SSM training loss from ELBO (new Appendix G subsection), norm dynamics of SGM vs MSGM (new Appendix D.3.1), detailed latent-space analysis (Appendix E now substantially expanded).
>
> **Substantial New Theoretical Contributions**
>
> • *New Section K: Scalability via Sparse Tensors*: This is a major addition that directly addresses all scalability concerns:
> 1. It introduces sparse tensor constructions that reduce cost and break the dense $\mathcal{O}(d^3)$ bottleneck.
> 2. It relaxes the strong rank condition (A2) to an “almost-sure” variant and discusses trapping the diffusion on lower-dimensional submanifolds.
> 3. Establishes necessary conditions for invariant spherical distributions via the Itô correction term.
> 4. This section also provides several examples of local sparse tensors, including examples where our MSGM cannot be applied because the latent distribution become untractable (new counterexamples in Appendix I).
>
> • *Extended discussion “Beyond pure Stratonovich noise”* (new Appendix D.6): We now incorporate and analyze drift-augmented multiplicative diffusions.
>
> • *Expanded latent-space and KL-divergence analysis*:
> 1. Strengthened arguments on latent-space alignment between MSGM and data (Proposition E.5.1),
> 2. behavior under heavy tails,
> 3. high-dimensional limit and comparison to SGM’s norm concentration.
>
> • *Clarification of geometric advantages*: We articulate how MSGM learns conditionnal scores only on the unit sphere, yielding:
> 1. bounded domain for approximation,
> 2. potential stability advantages
>
> • *Score parameterization*: Added analysis insight on well-posedness issues with score parameterization (Section G.8)
>
> **Expanded and Strengthened Experimental Section**
>
> • *New high-dimensional experiment:* Image processing ($d=1024$) that illustrates feasibility of MSGM in genuinely high-dimensional settings with practical computational cost using sparse tensors. See Section M.7.2.
> • *Comparison Metrics:* New Figures including improved MMD convergence plots and survival functions convergence plots
> • Better highlight that experiments are based on fair comparison methodology (Appendix M.3)
> • Detailed tables for paramater configurations and CPU/GPU times.
>
>
> **Clarifications to the Forward/Backward SDE**
>
> We improved exposition around:
> • the Stratonovich vs Itô formulation and its implications,
> • the presence of drift in the Itô form,
> • derivation of the backward SDE
> • conceptual differences with “geometric Brownian motion”
>
> **Improved Presentation and Interpretation**
>
> • Explanation of numerical results including what qualitative improvements are visible.
> • Clarified intuition for conditional direction score, geometric intuition and how direction and norm correlate along the reverse diffusion (Appendix H.2).
> • Moved some quantitative (MMD) convergence plots from the Appendix to the main text.
> • Cleaner and more readable MMD convergence plots.
>
> **Addressing SSM and Stability**
>
> We expanded the discussion of why SSM is practically far more stable in MSGM than in SGM:
> • a tighter match between data and latent distribution,
> • the norm conservation along the reverse diffusion prevents divergence,
> • a score learning on a bounded domain (unit sphere).
> We provide empirical evidence that SGM with SSM diverges whereas MSGM with SSM remains stable.
>
> **Consolidated Comparison Between MSGM and SGM**
>
> A new Appendix N summarizes theoretical differences, latent-space geometry, norm dynamics, training stability considerations and scalability under sparse tensors.

---

### Meta-Review · Area_Chair_DaS1 · 2026-01-06

**Summary:**

## Summary of the Paper Contributions

This paper introduces a new class of diffusion-based generative models, termed Multiplicative Score-based Generative Models (MSGM), which depart from classical score-based generative models relying on additive Gaussian noise. Instead, the proposed framework is driven by multiplicative noise with a skew-symmetric structure, inspired by transport noise arising in fluid dynamics and conservative physical systems.

More precisely, the forward diffusion process is defined in Stratonovich form as

$ d \mathbf{x}\_s = \sum_{k=1}^d ( \mathbf{G}^k \circ d B_s^k ), \mathbf{x}_s, $

where $B_s = (B_s^1,\ldots,B_s^d)$ is a $d$-dimensional standard Brownian motion, and $ \circ $ denotes Stratonovich integration. The corresponding Itô formulation reads

$ d \mathbf{x}\_s= \frac{1}{2} \sum\_{k=1}^d (\mathbf{G}^k)^2 \mathbf{x}_s  ds +  \sum\_{k=1}^d \mathbf{G}^k  dB_s^k  \mathbf{x}_s . $

Here, $ \mathbf{G}$ is a third-order tensor whose slices $\mathbf{G}^k $ are assumed to be skew-symmetric, and such that for any $\mathbf{x}$, the matrix obtained by contracting $ \mathbf{G} $ with $ \mathbf{x} $ has rank $ d-1 $. These assumptions lead to the key theoretical properties underlying the proposed approach: the norm of the data is preserved along the forward diffusion**, while the direction converges exponentially fast to the uniform distribution on the unit sphere.

As a consequence, the latent distribution is explicitly characterized and tractable: its radial component coincides with the data’s norm distribution, while its angular component is isotropic. This norm-preserving property is central to the paper’s motivation. In contrast to standard diffusion models, MSGMs naturally retain heavy-tailed behavior in the latent space, making them particularly well suited for modeling heavy-tailed data distributions and potentially enabling more faithful sampling of rare and extreme events.

The time-reversed process associated with the forward diffusion, as well as its practical approximation, are derived using standard diffusion-model theory applied to the Itô formulation.

Empirically, the proposed multiplicative diffusion framework is compared to standard score-based generative models on synthetic toy experiments involving correlated Cauchy distributions, as well as on low- and moderate-dimensional experimental fluid dynamics data. In several of these settings, the model demonstrates an improved ability to capture tail behavior and extreme events relative to classical score-based generative models.


## Summary of the Paper Strengths

The reviewers concur that the paper introduces interesting ideas and provides original insights that lead to a new diffusion framework for generative modeling. In particular, the proposed approach represents a clear departure from the standard additive Ornstein–Uhlenbeck paradigm underlying many score-based generative models. Modifying the forward diffusion process itself is a meaningful and timely research direction.

The main strengths of the paper can be summarized as follows:

-  The paper constructs a noising process whose stationary distribution depends explicitly on the data distribution. In particular, the distribution of the norm of the latent variable exactly matches that of the data, while its angular component is uniform on the sphere. Despite being non-Gaussian, the latent distribution remains tractable and easy to sample from, for instance via a one-dimensional empirical CDF of the norms.

-  The authors derive the associated Fokker–Planck equation, prove exponential convergence of the forward process to the limiting latent distribution, and establish the reverse-time SDE as well as the corresponding probability-flow ODE.

-  The paper provides a theoretical justification for the proposed training procedure.

-  In some modest experimental settings involving heavy-tailed data and rare-event regimes, the proposed model demonstrates qualitative advantages over standard score-based generative models.

## Summary of the Weaknesses

Reviewers pointed out the same weaknesses:

- A major concern raised by the reviewers is the lack of scalability of the proposed approach. As presented, the multiplicative noise operator relies on a dense third-order tensor whose number of parameters grows cubically with the dimension. This makes the method computationally prohibitive for high-dimensional applications such as images, videos, or realistic physical systems. The authors themselves acknowledge that the cost becomes prohibitive beyond very low dimensions, effectively limiting the current framework to small-scale or largely theoretical settings rather than practical generative modeling tasks.

- Closely related to this issue is a significant computational bottleneck during training and inference. Since the forward SDE does not admit an analytic solution, numerical SDE integration is required at every training step to generate corrupted samples. This introduces substantial computational overhead compared to standard score-based generative models, which benefit from closed-form forward processes. Moreover, discretization error is injected directly into the training objective, potentially affecting stability and accuracy.

- The training procedure itself is another point of concern. Because the conditional score is not available in closed form, denoising score matching cannot be used. Instead, the method relies on sliced score matching, which is known to be less stable in practice.

- Another criticism concerns the arbitrary construction of the multiplicative noise operator. The skew-symmetric tensor defining the noise is generated by sampling random matrices, without physical, structural, or data-driven justification.

- From an empirical standpoint, the experimental evaluation is considered limited in both scope and strength. Experiments are restricted to low-dimensional, heavy-tailed settings where the method is specifically designed to perform well. There is no evidence that the approach offers advantages on standard high-dimensional benchmarks where SGMs excel, while its computational cost and complexity are significantly higher. Some reported results are qualitative, and certain quantitative comparisons appear unconvincing, with standard SGMs sometimes converging faster or performing comparably.


## Assessment of the Paper

I largely concur with the reviewers’ feedback, both positive and negative. While the method has clear limitations, many of which are explicitly acknowledged by the authors, and the experimental evaluation is limited and only partially convincing, the reviewers nevertheless express strong interest in the ideas proposed in the paper. This interest is justified by the originality of the approach and by the solid theoretical foundations supporting it.

I share this assessment. Despite its current limitations, the paper introduces a novel and well-motivated diffusion framework that may inspire further research in this direction. For these reasons, I recommend acceptance.

**Reviewer Concerns:**

After reading the authors’ responses to the critical issues and pitfalls raised by the reviewers, I remain largely unconvinced by their replies. Nevertheless, as already stated in my assessment, I believe that the reviewers were overall persuaded by the originality of the approach and its potential for further development. In my view, this constitutes sufficient grounds to recommend acceptance.

**Reviewer Scores:**

Since I remain unconvinced by the authors’ responses, I believe the reviewers are likely to remain unconvinced as well. As a result, I do not think that they would have revised their scores.

---

### Decision · Program_Chairs · 2026-01-26

Accept (Poster)